# ON THE IMPORTANCE OF CONTRASTIVE LOSS IN MULTIMODAL LEARNING

## ABSTRACT

Recently, contrastive learning approaches (e.g., CLIP (Radford et al., 2021)) have received huge success in multimodal learning, where the model tries to minimize the distance between the representations of different views (e.g., image and its caption) of the same data point, while keeping the representations of different data points away from each other. However, from a theoretical perspective, it is unclear how contrastive learning can learn to align the representations from different views efficiently, especially in cases where the data is not isotropic. In this work, we analyze the training dynamics of a simple multimodal contrastive learning model and show that contrastive pairs are important for the model to efficiently balance the learned representations. In particular, we reveal a stage-wise behavior of the learning process: In the first stage, the model aligns the feature representations using positive pairs and the condition number grows in this stage. Then, in the second stage, the model reduces the condition number of the learned representations using negative pairs.

## 1 INTRODUCTION

One of the exceptional abilities of humans is to associate data from different modalities (such as texts and images) together. For example, when we hear the words "white dog", we can immediately align it with the image of a dog with white color. When we hear the loud sound of the engine, we can imagine an expensive sports car passing nearby.

Recently, in machine learning, multimodal learning methods – training the model to align the data from different modules, has become an increasingly popular research direction, especially in deep learning (He & Peng (2017); Stroud et al. (2020); Radford et al. (2021); Ramesh et al. (2021); Xu et al. (2021); Jia et al. (2021); Wang et al. (2022b)). Among them, the recent work CLIP (Radford et al. (2021)) shows remarkable quality results on aligning the features of text and images. The *contrastive learning* based method CLIP empirically outperforms many existing non-contrastive approaches (Grill et al. (2020); Chen & He (2021); He et al. (2020)). The major difference between the contrastive approach and other approaches is that contrastive loss not only requires the learned representations from the same pair of data (i.e. positive pairs) to be positively aligned, but it also requires the data from different pairs (i.e. negative pairs) to be as negatively aligned as possible. In the paper, the authors also identify contrastive loss as the most critical part to the success of CLIP.

Despite the empirical success of this contrastive learning-based method, from a theoretical perspective, the most fundamental questions are still largely open: In particular, how do contrastive pairs help in this new multimodal learning approach? How can the **non-convex** contrastive loss be efficiently minimized to learn features from both modules?

Unlike the prior theoretical works on contrastive learning which mostly focus on extracting features from one module (e.g., Arora et al. (2019); Jing et al. (2022); Pokle et al. (2022); Tian et al. (2021); Wen & Li (2021)), one main technical challenge of analyzing contrastive learning in a multimodal setting is how the model can be trained to align the feature representations $f_A, f_B$ from modules $A$ and $B$ respectively. Due to the existence of negative pairs that emphasize negative correlations of $f_A$ and $f_B$, it is unclear that the model still has incentives to align the features from different modules.

In this paper, we make preliminary theoretical steps towards answering the fundamental theoretical questions of the importance of contrastive loss in multimodal learning. We assume the data from the two modules are of form $x_A = Az_A + A_\xi \xi_A$ and $x_B = Bz_B + B_\xi \xi_B$, respectively, where $z_A, z_B$ are considered as the hidden signals, $A, B$ linear transformations from the signal to the observation and $A_\xi \xi_A, B_\xi \xi_B$ the noises. Similar linear models have also been used in previous works (Tian et al. (2021); Wen & Li (2021)) in the context of single-modal learning ($A = B$). The positive pair of the data shares the same signal $z_A = z_B$, and has different noises $\xi_A, \xi_B$ and transformations $A, B$. The goal is to learn features $f_A, f_B$ that align positive pairs while keeping representations of negative pairs away from each other.

Under this setting, we make the following contributions:

1. We consider the challenging (but more practical) setting where the features in $A$ and $B$ are inhomogeneous, that is, the condition number of $A$ and $B$ can be $\omega(1)$. Prior works (Jing et al. (2022); Tian et al. (2021); Wen & Li (2021)) only consider cases where $A$ and $B$ are exactly column orthonormal matrices even in the simpler single-modal setting ($A = B$).

2. We consider feature learners $f_A, f_B$ with normalization, meaning that $f_A, f_B$ are always normalized to have (expected) norm one during training. Output normalization plays a critical role in the practical success of contrastive learning and is also employed in CLIP, but it is rarely considered in theory due to the additional complexity of the division by norm.

3. We analyze the learning process of stochastic gradient descent from random initialization. We prove that contrastive learning converges efficiently to a nearly optimal solution, which indeed aligns the feature representation $f_A$ and $f_B$.

4. We also demonstrate the importance of negative pairs by comparing with training only over the positive pairs: We prove that although the latter can also learn to align $f_A$ and $f_B$, the features learned by contrastive learning with negative pairs is much more *uniform*, meaning that $f_A, f_B$ can recover all the singular vectors of $A$ and $B$ and normalize them. On the other hand, without negative pairs, the learned representation is close to a rank one solution, meaning that $f_A, f_B$ will only focus on the top singular direction of $A$ and $B$.

5. We also perform simulations and more practical experiments to further support our theory.

## 2 RELATED WORKS

**Multimodal learning**   Despite the empirical success of multimodal learning, there are very few theoretical results on this topic. The one most related to ours is Huang et al. (2021), in which the authors show that, in certain cases, multimodal methods can provably perform better than single-modal models. However, the authors consider neither contrastive pairs nor the training dynamics.

**Contrastive/Non-contrastive learning theory**   Another much richer line of research is about contrastive and non-contrastive methods in the context of single-modal self-supervised learning. Starting from Arora et al. (2019), many recent works have provided various explanations on why the representations learned with contrastive learning are useful in downstream tasks (Chuang et al. (2020); Tosh et al. (2021); Nozawa & Sato (2021); Wang et al. (2022a); HaoChen et al. (2021); Lee et al. (2021); Wang & Isola (2020)). These works mostly focus on the generalization aspect of the problem and do not consider training. Among them, Wang & Isola (2020) also study the problem using the notions of alignment and uniformity, and demonstrate that balanced representations benefit downstream tasks. However, they do not provide guarantees on training. Another related line of research is about non-contrastive learning, where the necessity of negative examples is questioned. In this line of research, the optimization problem does get considered as non-contrastive losses have trivial collapsed solutions. Tian et al. (2021) show that, under certain conditions, non-contrastive learning methods can learn non-collapsed solutions. Jing et al. (2022) show that, even with negative examples, contrastive learning can still suffer from another type of collapse, where the learned representations only span a low-dimensional subspace of the embedding space. In Pokle et al. (2022), the authors show that non-contrastive losses have many non-collapsed bad minima that the training algorithm does not avoid. Another related work that takes optimization into consideration is Wen & Li (2021), in which the authors analyze the training dynamics of contrastive learning and show that,

with random augmentation, neural networks can learn features that are suppressed by noises when no augmentations are used. Though these works do consider the optimization problem, they focus on the case where the features are uniform, and only Wen & Li (2021) considers output normalization. We compare our results with the most relevant works in the next paragraph.

**Comparison with Jing et al. (2022); Tian et al. (2021); Wen & Li (2021)** The dimensional collapse problem reported in Jing et al. (2022) is not a real issue in our setting, since, in our case, the best the model can do is to recover the latent vector $z$, up to some rotation. As a result, it is natural for the learned representations to span only a low-dimensional subspace of the embedding space $\mathbb{R}^m$. Here, the point of choosing $m \gg d$ is to make the optimization dynamics more regular, which is a common strategy in the study of over-parameterized neural networks. The main difference between our work and the analysis in Tian et al. (2021) and Wen & Li (2021) is we do not assume the inputs are isotropic. In our setting, the condition number can be as large as $\Theta(\log d)$. When the condition number is 1, one can imagine that thanks to the symmetry, all directions will be learned simultaneously, and therefore, we do not need negative examples to prevent collapse (Tian et al. (2021)) or the negative examples do not play an important role in analysis (Wen & Li (2021)). On the other hand, when the condition number is larger than 1, we do need to use the negative examples to shrink the condition number, which corresponds to the second stage of our analysis.

## 3 PROBLEM SETUP

Similar to previous theoretical works (e.g., Lee et al. (2021); Wen & Li (2021)), we consider a linear data-generating model. Formally, we assume that the contrastive pairs $(x_A^+, x_B^-)$ are constructed as

$$x_A^+ = Az^+ + A_\xi \xi_A^+, \quad x_B^- = Bz^- + B_\xi \xi_B^-, \tag{1}$$

where $z^\pm$, $\xi_A^-$, $\xi_B^-$ are independent random variables following the uniforms distributions over $\{\pm 1/\sqrt{d}\}^r, \{\pm 1/\sqrt{d}\}^{d-r}$ and $\{\pm 1/\sqrt{d}\}^{d-r}$, respectively, and $A, B \in \mathbb{R}^{d \times r}, A_\xi, B_\xi \in \mathbb{R}^{d \times (d-r)}$ are matrices with $A^\top A = B^\top B = \mathrm{diag}(\sigma^2)$ and $A_\xi^\top A_\xi = B_\xi^\top B_\xi = \sigma_\xi^2 I_{d-r}$ for some $\sigma \in \mathbb{R}_+^r$ and $\sigma_\xi \in \mathbb{R}_+$. In words, we first sample the latent vectors $z^\pm \in \mathbb{R}^r$ independently and encode them with $A, B$ to form the signal part of the input. Then, we sample the latent noises $\xi_A^+, \xi_B^- \in \mathbb{R}^{d-r}$, and encode them with $A_\xi, B_\xi$ to form the noise part of the input. Finally, we add the signal and noise parts together to obtain $(x_A^+, x_B^-)$. To generate a positive pair $(x_A^+, x_B^+)$, we use the same latent vector. That is, $x_A^+ = Az^+ + A_\xi \xi_A^+$ and $x_B^+ = Bz^+ + B_\xi \xi_B^+$. Note that the latent noises here are still independent. We use $\sigma_{\max}^2$ and $\sigma_{\min}^2$ to denote the maximum and minimum of $\sigma_1^2, \ldots, \sigma_r^2, \sigma_\xi^2$, respectively. We assume that $(\sigma_{\max}^2/\sigma_{\min}^2) \max\left\{1, (d-r)\sigma_\xi^2/(r\sigma_{\min}^2)\right\} \leq c \log d$ for some small constant $c > 0$. Our results can easily be generalized to settings where the dimensions of $x_A$ and $x_B$ are not the same, since one can simply pad zeros at the end of each column of $A$ and $B$.

One way of interpreting this model is to view each coordinate of the latent vector $z$ as an indicator for the presence/absence of a certain object, and the corresponding columns in $A$ and $B$ as the visual and word embeddings of this object, respectively.

Now, we describe our learner model. We consider (normalized) linear feature learners. Define

$$f_A(x_A) := \frac{W_A^\top x_A}{\sqrt{\mathbb{E}_{x_A} \left\| W_A^\top x_A \right\|^2}}, \quad f_B(x_B) := \frac{W_B^\top x_B}{\sqrt{\mathbb{E}_{x_B} \left\| W_B^\top x_A \right\|^2}},$$

where $W_A, W_B \in \mathbb{R}^{d \times m}$ are the trainable parameters. In words, we first map the inputs $(x_A, x_B)$ into the embedding space $\mathbb{R}^m$ using $W_A$ and $W_B$, and then apply batch normalization to the outputs. By saying the learned representations are aligned, we mean that $f_A$ and $f_B$ are close for positive pairs and far away from each other for negative pairs. Meanwhile, we say the learned representations are balanced if changing a small fraction of coordinates of $z$ does not change the representation dramatically. See Section 4 for formal definitions.

One can easily verify that, in the population case, we have $\mathbb{E}_{x_A} \left\| W_A^\top x_A \right\|^2 = \left\| W_A^\top A \right\|_F^2 / d + \left\| W_A^\top A_\xi \right\|_F^2 / d$. For notational simplicity, we write $K_A = W_A^\top A$, $K_B = W_B^\top B$, $K_{A,\xi} = $

$\boldsymbol{W}_{\boldsymbol{A}}^{\top}\boldsymbol{A}_{\boldsymbol{\xi}}$, and $\boldsymbol{K}_{B,\xi} = \boldsymbol{W}_{\boldsymbol{B}}^{\top}\boldsymbol{B}_{\boldsymbol{\xi}}$. These are the matrices that directly map latent vectors to their final representations. We also define $N_{\boldsymbol{A}}^2 = (\|\boldsymbol{K}_{\boldsymbol{A}}\|_F^2 + \|\boldsymbol{K}_{\boldsymbol{A},\xi}\|_F^2)/d$ and $N_{\boldsymbol{B}}^2 = (\|\boldsymbol{K}_{\boldsymbol{B}}\|_F^2 + \|\boldsymbol{K}_{\boldsymbol{B},\xi}\|_F^2)/d$. Then our model can be rewritten as[1]

$$\boldsymbol{f}_{\boldsymbol{A}}(\boldsymbol{x}_{\boldsymbol{A}}) = \frac{\boldsymbol{K}_{\boldsymbol{A}} z_{\boldsymbol{A}} + \boldsymbol{K}_{\boldsymbol{A},\xi}\boldsymbol{\xi}_{\boldsymbol{A}}}{N_{\boldsymbol{A}}}, \quad \boldsymbol{f}_{\boldsymbol{B}}(\boldsymbol{x}_{\boldsymbol{B}}) = \frac{\boldsymbol{K}_{\boldsymbol{B}} z_{\boldsymbol{B}} + \boldsymbol{K}_{\boldsymbol{B},\xi}\boldsymbol{\xi}_{\boldsymbol{B}}}{N_{\boldsymbol{B}}}. \tag{2}$$

We initialize each entry of $\boldsymbol{W}_{\boldsymbol{A}}$ and $\boldsymbol{W}_{\boldsymbol{B}}$ using iid Gaussian $\mathcal{N}(0, 1/m)$. This scaling ensures the norm of outputs before initialization does not blow up as $m \to \infty$. We train our model using gradient descent over the following contrastive loss $\mathcal{L}$. First, we define[2]

$$S_{\boldsymbol{A}}(\boldsymbol{x}_{\boldsymbol{A}}^+, \boldsymbol{x}_{\boldsymbol{B}}^+) = \frac{\exp(\tau_t^2 \boldsymbol{f}_{\boldsymbol{A}}^+ \cdot \boldsymbol{f}_{\boldsymbol{B}}^+)}{\exp(\tau_t^2 \boldsymbol{f}_{\boldsymbol{A}}^+ \cdot \boldsymbol{f}_{\boldsymbol{B}}^+) + K \mathbb{E}_{\boldsymbol{x}_{\boldsymbol{B}}^-} \exp(\tau_t^2 \boldsymbol{f}_{\boldsymbol{A}}^+ \cdot \boldsymbol{f}_{\boldsymbol{B}}^-)},$$

$$S_{\boldsymbol{B}}(\boldsymbol{x}_{\boldsymbol{A}}^+, \boldsymbol{x}_{\boldsymbol{B}}^+) = \frac{\exp(\tau_t^2 \boldsymbol{f}_{\boldsymbol{A}}^+ \cdot \boldsymbol{f}_{\boldsymbol{B}}^+)}{\exp(\tau_t^2 \boldsymbol{f}_{\boldsymbol{A}}^+ \cdot \boldsymbol{f}_{\boldsymbol{B}}^+) + K \mathbb{E}_{\boldsymbol{x}_{\boldsymbol{A}}^-} \exp(\tau_t^2 \boldsymbol{f}_{\boldsymbol{A}}^- \cdot \boldsymbol{f}_{\boldsymbol{B}}^+)},$$

where $K$ is a positive constant controlling the strength of negative samples and $\tau_t \in (0, 1]$ is the inverse temperature. Then, we define the contrastive loss as

$$\mathcal{L} := \mathcal{L}_{\boldsymbol{A}} + \mathcal{L}_{\boldsymbol{B}} := -\mathbb{E}\log S_{\boldsymbol{A}}(\boldsymbol{x}_{\boldsymbol{A}}^+, \boldsymbol{x}_{\boldsymbol{B}}^+) - \mathbb{E}\log S_{\boldsymbol{B}}(\boldsymbol{x}_{\boldsymbol{A}}^+, \boldsymbol{x}_{\boldsymbol{B}}^+). \tag{3}$$

By the non-contrastive loss, we mean

$$\hat{\mathcal{L}} := -\mathbb{E}\left\langle \boldsymbol{f}_{\boldsymbol{A}}^+, \boldsymbol{f}_{\boldsymbol{B}}^+ \right\rangle. \tag{4}$$

Formally, the training algorithm is defined as follows.

**Algorithm 3.1** (Training algorithm). *Let $\tilde{\mathcal{L}}$ be $\mathcal{L}$ in the contrastive case and $\hat{\mathcal{L}}$ in the non-contrastive case. At each step, we first sample a batch of positive/negative pairs $\{(\boldsymbol{x}_{\boldsymbol{A}}^+, \boldsymbol{x}_{\boldsymbol{B}}^+, \boldsymbol{x}_{\boldsymbol{A}}^-, \boldsymbol{x}_{\boldsymbol{B}}^-)\}_{i=1}^N$, use them to compute the empirical version of $\tilde{\mathcal{L}}$, and update the weight matrices using $\boldsymbol{W}_{\boldsymbol{A}} \leftarrow \boldsymbol{W}_{\boldsymbol{A}} - \tau_t^{-2}\eta\nabla_{\boldsymbol{W}_{\boldsymbol{A}}}\tilde{\mathcal{L}}$ and $\boldsymbol{W}_{\boldsymbol{B}} \leftarrow \boldsymbol{W}_{\boldsymbol{B}} - \tau_t^{-2}\eta\nabla_{\boldsymbol{W}_{\boldsymbol{B}}}\tilde{\mathcal{L}}$.*

*In the non-contrastive case, we always use $\tau_t = 1$[3], and we repeat the above update until gradient descent converges to an approximate stationary point. In the contrastive case, we first use a small $\tau_t = 1/\text{poly}(d)$, run the process for $T_1 = \text{poly}(d)$ iterations, switch to $\tau_t = 1$, and run the process for another $T_2 - T_1 = \text{poly}(d)$ iterations.*

## 4 MAIN RESULTS

In words, our results say that though both contrastive and non-contrastive methods can align the representations, the representations learned by contrastive methods are more balanced. First, we need to define what do "alignment" and "balance" mean here. We still use $\boldsymbol{f}_{\boldsymbol{A}}$ and $\boldsymbol{f}_{\boldsymbol{B}}$ to denote the embeddings, but our definitions here will be architecture-agnostic. After some general discussion, we also discuss these definitions in our specific setting.

**Definition 4.1** (Alignment). *We define the alignment score as*

$$\Gamma_{\texttt{Align}} := \frac{1}{2} \mathop{\mathbb{E}}_{\boldsymbol{x}_{\boldsymbol{A}}^{\pm}, \boldsymbol{x}_{\boldsymbol{B}}^{\pm}} \left\{ \mathbb{1}\left\{ \left\| \boldsymbol{f}_{\boldsymbol{A}}^+ - \boldsymbol{f}_{\boldsymbol{B}}^+ \right\| < \left\| \boldsymbol{f}_{\boldsymbol{A}}^+ - \boldsymbol{f}_{\boldsymbol{B}}^- \right\| \right\} + \mathbb{1}\left\{ \left\| \boldsymbol{f}_{\boldsymbol{A}}^+ - \boldsymbol{f}_{\boldsymbol{B}}^+ \right\| < \left\| \boldsymbol{f}_{\boldsymbol{A}}^- - \boldsymbol{f}_{\boldsymbol{B}}^+ \right\| \right\} \right\}.$$

*Namely, $\Gamma_{\texttt{Align}}$ is the accuracy of classifying whether the input pair $(\boldsymbol{x}_{\boldsymbol{A}}, \boldsymbol{x}_{\boldsymbol{B}})$ is a positive pair. We say that the learned representations are aligned if $\Gamma_{\texttt{Align}} \approx 1$.*

Note that the notion of alignment introduced here is stronger than matching the positive pairs, which can be achieved by simply mapping all inputs to one single embedding. In that case, $\Gamma_{\texttt{Align}}$ will be $0$ (or $0.5$ if we choose to break ties randomly instead of using strict inequality).

---

[1]See Section D for discussions on the sample complexity.

[2]We use $\boldsymbol{f}_{\boldsymbol{A}}^+$ as a shorthand for $\boldsymbol{f}_{\boldsymbol{A}}(\boldsymbol{x}_{\boldsymbol{A}}^+)$, similarly for other combinations of $\boldsymbol{A}, \boldsymbol{B}$ and $\pm$.

[3]Note that, in the non-contrastive case, changing $\tau_t$ only changes the learning rate.

**Definition 4.2** (Balance). *We define the balance score as*

$$\Gamma_{\texttt{Balance}} := \|\mathbf{\Sigma}_f\|_F^2 / \|\mathbf{\Sigma}\|_2^2 \quad where \quad \mathbf{\Sigma}_f := \mathop{\mathbb{E}}_{\boldsymbol{x}_A} \left\{ \boldsymbol{f}_A \boldsymbol{f}_A^\top \right\}.$$

*Namely, $\Gamma_{\texttt{Balance}}$ is the stable rank of the covariance matrix of the output embeddings. We say that the learned representations are balanced if $\Gamma_{\texttt{Balance}} \geq \alpha r$ for some $\alpha \approx 1$.*

We make some short remarks on Definition 4.2. First, we only use $\boldsymbol{f}_A$ in this definition because if the embeddings are well-aligned, $\boldsymbol{f}_A \boldsymbol{f}_A^\top$ and $\boldsymbol{f}_B \boldsymbol{f}_B^\top$ should be approximately the same. Second, the stable rank is usually used as an alternative to the actual rank because it is less sensitive to small singular values (Rudelson & Vershynin (2007)). To see why this makes sense, note that $\|\mathbf{\Sigma}_f\|_F^2 / \|\mathbf{\Sigma}\|_2^2 = \sum_{k=1}^m \kappa_k^2 / \max_{k\in[m]} \kappa_k^2$, where $\kappa_1, \ldots, \kappa_m$ are the singular values of $\mathbf{\Sigma}_f$. If all nonzero $\kappa_k$ are the same, then this recovers the actual rank. The intuition behind the use of rank is that the more independent latent variables the model learns, the higher the rank of the representations needs to be.

In Jing et al. (2022), the authors also use rank to measure the degree of "dimensional collapse". Unlike their argument, here we only require $\Gamma_{\texttt{Balance}}$ to be at least $\alpha r$, instead of $\alpha m$, for the representations to be called balanced because even if we can recover the underlying latent vectors, the rank is still at most $r$. Hence, it does not make much sense to expect the learned representations to span the entire embedding space. Finally, note that a sufficient condition for $\Gamma_{\texttt{Balance}} \geq \alpha r$ is that the ratio of the largest and $r$-largest singular values is at most $\sqrt{\alpha}$. In other words, after excluding those singular values that should be 0, the condition number is approximately 1.

**Why balance representations are important?** In our setting, one simple example of aligned but unbalanced representations is $\boldsymbol{W}_A^\top = \operatorname{diag}(\boldsymbol{\nu})\boldsymbol{A}^{-1}$ and $\boldsymbol{W}_B^\top = \operatorname{diag}(\boldsymbol{\nu})\boldsymbol{B}^{-1}$ with $\nu_1 = 1$ and $\nu_2 = \cdots = \nu_r = \exp(-d)$. This model maps inputs whose latent vector is $\boldsymbol{z}$ to $\operatorname{diag}(\boldsymbol{\mu})\boldsymbol{z}$, up to some normalization, for both modules, whence it has $\Gamma_{\texttt{Align}} = 1$. Meanwhile, one can verify that this model has $\Gamma_{\texttt{Balance}}/r \leq 2/r \approx 0$. The problem of this model is that it overly emphasizes $z_1$ and is sensitive to small changes in $z_1$. This model can be made balanced by replacing $\operatorname{diag}(\boldsymbol{\mu})$ with $\boldsymbol{I}_r$, in which case the model directly recovers the latent vector $\boldsymbol{z}$. As a result, all changes in $\boldsymbol{z}$ will be reflected in the final representation in a faithful way. Similar notions have also been studied in Wang & Isola (2020) under the name "uniformity" in the context of self-supervised learning, and they also report that balanced representations lead to better performance in downstream tasks, though, unlike our result, they do not provide guarantees on training. Still, this further suggests that learning balanced representations is a reasonable and important goal.

With these two definitions, we can now state our main results.

**Theorem 4.3.** *Suppose that the network width $m = \operatorname{poly}(d)$ is sufficiently large, the learning rate $\eta = 1/\operatorname{poly}(d)$ is sufficiently small, and we generate sufficiently, but still polynomially, many samples at each step to compute the loss[4]. Suppose that, for some small constant $c > 0$,*

$$\frac{\sigma_{\max}^2}{\sigma_{\min}^2} \max \left\{ 1, \frac{(d-r)\sigma_\xi^2}{r\sigma_{\min}^2} \right\} \leq c \log d.$$

(a) ***Non-contrastive loss.*** *There exists a $\boldsymbol{\sigma} \in \mathbb{R}^r$ satisfying the above assumptions such that, after $\operatorname{poly}(d)$ iterations, Algorithm 3.1 will converge to an approximate stationary point, at which the learned representations are aligned but not balanced, that is, $\Gamma_{\texttt{Align}} \approx 1$ but $\Gamma_{\texttt{Balance}}/r \approx 0$.*

(b) ***Contrastive loss.*** *There exists $\tau_0^2 = 1/\operatorname{poly}(d)$ and $T_1 = \operatorname{poly}(d)$ such that, for any valid $\boldsymbol{\sigma}$, Algorithm 3.1 will reach a point after $\operatorname{poly}(d)$ iterations at which the learned representations are aligned and balanced, that is, $\Gamma_{\texttt{Align}} \approx 1$ and $\Gamma_{\texttt{Balance}}/r \approx 1$.*

We close this section with sufficient conditions for the learned representations to be aligned and balanced in our setting. First, in our specific setting, the most natural way for a model to achieve $\Gamma_{\texttt{Align}} \approx 1$ is to learn $\|\boldsymbol{f}_A - \boldsymbol{f}_B\| \approx \|\boldsymbol{K}_0(\boldsymbol{z}_A - \boldsymbol{z}_B)\|$ for some $\boldsymbol{K}_0 \in \mathbb{R}^{m \times r}$ with $\boldsymbol{K}_0^\top \boldsymbol{K}_0 \succ 0$.

---

[4]To make the proof cleaner, we write it in terms of gradient flow over population loss. See Section D for discussions on the discretization of gradient flow.

In this case, the distance between positive pairs is always close to $0$ while the distance between negative pairs is positive. Recall that the output of our model is $\boldsymbol{f_A} = (\boldsymbol{K_A z_A} + \boldsymbol{K_{A,\xi} \xi_A})/N_A$ and $\boldsymbol{f_B} = (\boldsymbol{K_B z_B} + \boldsymbol{K_{B,\xi} \xi_B})/N_B$. Hence, we have the following sufficient condition.

**Lemma 4.4** (Suffcient condition for aligned representations)**.** *If $\|\boldsymbol{K_A}\|_F \gg \|\boldsymbol{K_{A,\xi}}\|_F$, $\|\boldsymbol{K_B}\|_F \gg \|\boldsymbol{K_{B,\xi}}\|_F$, $\boldsymbol{K_A} \approx \boldsymbol{K_B}$, and the condition number of $\boldsymbol{K_A}$ is upper bounded by $\mathrm{poly}(d)$, then the learned representations are aligned.*

The first two conditions imply the signal parts dominate the outputted representations, the third condition gives the existence of $\boldsymbol{K_0}$, and the final condition makes sure the smallest singular value $\boldsymbol{K_0^\top K_0}$ of is still at least $1/\mathrm{poly}(d)$ after normalization.

Now we consider the balance of the learned representations. Suppose that our representations are already aligned, in the sense of Lemma 4.4. Then we have

$$\boldsymbol{\Sigma}_f = \mathbb{E}_{\boldsymbol{x_A}} \left\{ \boldsymbol{f_A f_A^\top} \right\} = \mathbb{E}_{\boldsymbol{z}} \left\{ \frac{\boldsymbol{K_A z z^\top K_A^\top}}{\|\boldsymbol{K_A}\|_F^2 / d} \right\} = \frac{\boldsymbol{K_A K_A^\top}}{\|\boldsymbol{K_A}\|_F^2}.$$

Recall the relationship between $\Gamma_{\texttt{Balance}}$ and the effective condition number of $\boldsymbol{\Sigma}_f$. We have the following sufficient condition.

**Lemma 4.5** (Sufficient condition for balanced representations)**.** *Suppose that our model is aligned, in the sense of Lemma 4.4. If the condition number of $\boldsymbol{K_A^\top K_A}$ is close to $1$, then the learned representations are balanced.*

Note that, since the columns of $\boldsymbol{A}$ are not orthonormal, even at initialization, the condition number of $\boldsymbol{K_A^\top K_A}$ is not close to $1$. In other words, a condition-number-reducing stage is necessary for the model to learn balanced representations.

## 5 TRAINING DYNAMICS AND PROOF OUTLINE

Our proof is based on characterizing the dynamics of gradient descent over the contrastive loss. We first choose a small inverse temperature $\tau_t^2 = 1/\mathrm{poly}(d)$ and run gradient descent for $\mathrm{poly}(d)$ iterations. This stage is called Stage 1. Then, we set $\tau_t^2 = 1$ and run gradient descent for another $\mathrm{poly}(d)$ iterations. This stage is called Stage 2. The use of different $\tau_t$ is mainly for technical purposes since this gives cleaner separation between stages. We observe similar stage-wise behavior when using a uniform $\tau_t^2$. See Figure 1 for simulation results. We also report results for the non-contrastive loss there.

### 5.1 THE TRAINING DYNAMICS

Instead of tracking the parameters $\boldsymbol{W_A}$ and $\boldsymbol{W_B}$ directly, we will track $\boldsymbol{K_A}, \boldsymbol{K_{A,\xi}}, \boldsymbol{K_B}, \boldsymbol{K_{B,\xi}}$, the matrices that directly map latent signals and noises to the final representations. For the case with contrastive pairs, one can show that the dynamics of $\boldsymbol{K_A}$ are governed by the following equation

$$
\begin{aligned}
\dot{\boldsymbol{K}}_{\boldsymbol{A}} = & \mathbb{E}_{\boldsymbol{x_A^+}, \boldsymbol{x_B^+}} \left\{ \left(2 - S_A(\boldsymbol{x_A^+}, \boldsymbol{x_B^+}) - S_B(\boldsymbol{x_A^+}, \boldsymbol{x_B^+})\right) \left( \frac{\boldsymbol{f_B^+}(\boldsymbol{z^+})^\top}{N_A} - \left\langle \boldsymbol{f_A^+}, \boldsymbol{f_B^+} \right\rangle \frac{\boldsymbol{K_A}}{N_A^2 d} \right) \right\} \mathrm{diag}(\boldsymbol{\sigma^2}) \\
& - K \mathbb{E}_{\boldsymbol{x_A^+}, \boldsymbol{x_B^\pm}} \left\{ \frac{S_A(\boldsymbol{x_A^+}, \boldsymbol{x_B^+}) \exp(\tau_t^2 \boldsymbol{f_A^+} \cdot \boldsymbol{f_B^-})}{\exp(\tau_t^2 \boldsymbol{f_A^+} \cdot \boldsymbol{f_B^+})} \left( \frac{\boldsymbol{f_B^-}(\boldsymbol{z^+})^\top}{N_A} - \left\langle \boldsymbol{f_A^+}, \boldsymbol{f_B^-} \right\rangle \frac{\boldsymbol{K_A}}{N_A^2 d} \right) \right\} \mathrm{diag}(\boldsymbol{\sigma^2}) \\
& - K \mathbb{E}_{\boldsymbol{x_A^\pm}, \boldsymbol{x_B^+}} \left\{ \frac{S_B(\boldsymbol{x_A^+}, \boldsymbol{x_B^+}) \exp(\tau_t^2 \boldsymbol{f_A^-} \cdot \boldsymbol{f_B^+})}{\exp(\tau_t^2 \boldsymbol{f_A^+} \cdot \boldsymbol{f_B^+})} \left( \frac{\boldsymbol{f_B^+}(\boldsymbol{z^-})^\top}{N_A} - \left\langle \boldsymbol{f_A^-}, \boldsymbol{f_B^+} \right\rangle \frac{\boldsymbol{K_A}}{N_A^2 d} \right) \right\} \mathrm{diag}(\boldsymbol{\sigma^2}).
\end{aligned}
\tag{5}
$$

See Lemma A.3 for the calculation. The first term comes from the positive pairs and the second and third terms from the negative pairs. Within each term, the second part comes from the normalization. The equations for the other $\boldsymbol{K}$-matrices can be derived similarly. We rescale the gradients by $1/\tau_t^2$ so that $\frac{\mathrm{d}}{\mathrm{d}t} \boldsymbol{K_A}$ does not shrink with $\tau_t^2$. For the non-contrastive case, the equation is

$$
\frac{\mathrm{d}}{\mathrm{d}t} \boldsymbol{K_A} = \mathbb{E}_{\boldsymbol{x_A^+}, \boldsymbol{x_B^+}} \left\{ \frac{\boldsymbol{f_B^+}(\boldsymbol{z^+})^\top}{N_A} - \left\langle \boldsymbol{f_A^+}, \boldsymbol{f_B^+} \right\rangle \frac{\boldsymbol{K_A}}{N_A^2 d} \right\} \mathrm{diag}(\boldsymbol{\sigma^2}).
\tag{6}
$$

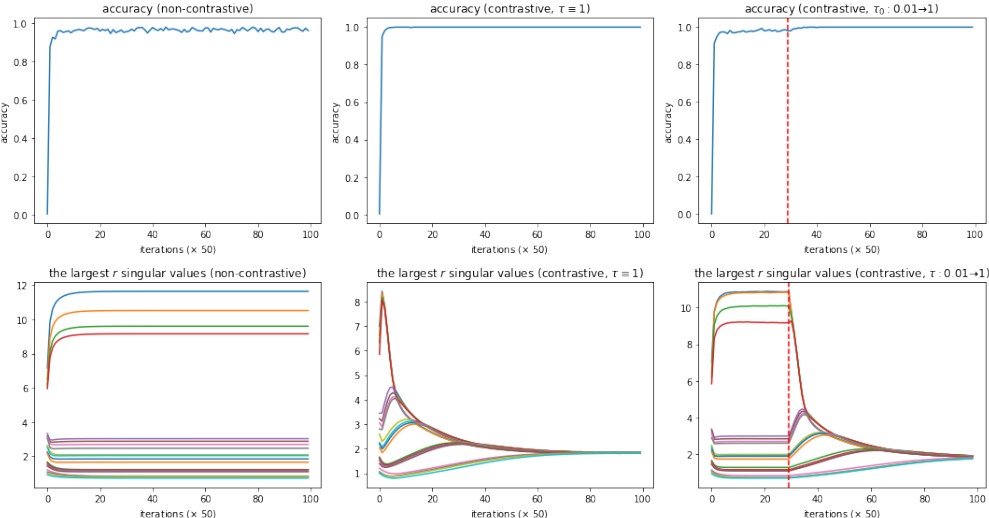

Figure 1: Simulation results. The first row reports the accuracies of different approaches on the classifying positive/negative pairs problem, and the second row reports the values of the largest $r$ singular values. From left to right, the columns correspond to the non-contrastive method, contrastive method with $\tau_t \equiv 1$ throughout the entire process, and contrastive method with $\tau_t$ switch from 0.01 to 1 at the vertical dashed line. One can make several observations here. (a) All methods can quickly attain near 100% accuracy. (b) Only contrastive methods will reduce the condition number to approximately 1. (c) Even when $\tau_t \equiv 1$, we still have the stage-wise behavior, where the models first align the representations in Stage 1, and then balance the representations in Stage 2.

Note that the RHS resembles the first term of (5). This is not a coincidence. We will establish the approximate equivalence between the non-contrastive approach and the contrastive approach with a small inverse temperature $\tau_t^2$ (cf. Section 5.3 and Lemma B.9).

## 5.2 THE INFINITE-WIDTH DYNAMICS

The overall proof strategy is to first characterize the dynamics of the infinite-width limit, which is much simpler compared to (5), and then control the error introduced by discretizing the infinite-width network using polynomially many neurons. This discretization is one of the main technical challenges of the proof. In general, in order to track the infinite-width dynamics, an exponentially large network is needed (Mei et al. (2018)).

The basic idea is to use Taylor expansion around the infinite-width trajectory to factor out the first-order error terms and show that, either they drive the process towards the infinite-width trajectory or the error growth introduced by them is slower than the convergence rate.

Here, for ease of presentation, we will focus on the noiseless infinite-width dynamics and, in particular, the evolution of the condition number. Recall that we use iid Gaussian to initialize the entries of $\boldsymbol{W_A}$ and $\boldsymbol{W_B}$. Hence, in the $m \to \infty$ limit, different columns of $\boldsymbol{K_A}$ and $\boldsymbol{K_B}$ are orthogonal to each other, at least at initialization. Moreover, one can verify that thanks to the symmetry, this holds throughout the entire training procedure. Meanwhile, by symmetry, the corresponding quantities in modules $\boldsymbol{A}$ and $\boldsymbol{B}$ are always the same. Namely, when $m \to \infty$, we have $\boldsymbol{K_A^\top K_A} = \boldsymbol{K_B^\top K_B} = \mathrm{diag}(\boldsymbol{\kappa}^2)$ and $\boldsymbol{K_A^\top K_B} = \mathrm{diag}(\hat{\boldsymbol{\kappa}}^2)$ for some $\boldsymbol{\kappa}, \hat{\boldsymbol{\kappa}} \in \mathbb{R}^r$. As a result, in order to characterize the dynamics of $\boldsymbol{K_A}, \boldsymbol{K_B}$, it suffices to look at $\boldsymbol{\kappa}^2$ and $\hat{\boldsymbol{\kappa}}^2$. One can show that, in this noiseless infinite-width limit, we have (cf. Lemma A.7)

$$\frac{\mathrm{d}}{\mathrm{d}t}\kappa_p^2 = 4\left(1 - \tilde{S}\right)\left(\frac{\hat{\kappa}_p^2}{\|\boldsymbol{\kappa}\|^2} - \frac{\|\hat{\boldsymbol{\kappa}}\|^2}{\|\boldsymbol{\kappa}\|^2}\frac{\kappa_p^2}{\|\boldsymbol{\kappa}\|^2}\right)\sigma_p^2 - 4\left(1 - \tilde{S}\right)\left(\frac{\hat{\kappa}_p^2}{\|\boldsymbol{\kappa}\|^2}T_p - \frac{\kappa_p^2}{\|\boldsymbol{\kappa}\|^2}\tilde{T}\right)\sigma_p^2,$$

$$\frac{\mathrm{d}}{\mathrm{d}t}\hat{\kappa}_p^2 = 4\left(1 - \tilde{S}\right)\left(\frac{\kappa_p^2}{\|\boldsymbol{\kappa}\|^2} - \frac{\|\hat{\boldsymbol{\kappa}}\|^2}{\|\boldsymbol{\kappa}\|^2}\frac{\hat{\kappa}_p^2}{\|\boldsymbol{\kappa}\|^2}\right)\sigma_p^2 - 4\left(1 - \tilde{S}\right)\left(\frac{\kappa_p^2}{\|\boldsymbol{\kappa}\|^2}T_p - \frac{\hat{\kappa}_p^2}{\|\boldsymbol{\kappa}\|^2}\tilde{T}\right)\sigma_p^2,$$

$$(7)$$

where $\tilde{S}$ is a $\Theta(1)$ quantity depending on $\hat{\kappa}$ and $\kappa$, $T_p = \tanh(\tau_t^2 \hat{\kappa}_p^2 / \|\kappa\|^2)$, and $\tilde{T} = \sum_{k=1}^r (\hat{\kappa}_p^2 / \|\kappa\|^2) T_p$. The first term comes from the first term of (5) and the second term from the second and third terms of (5). When $\kappa_p^2 \approx \hat{\kappa}_p^2$ for all $p \in [r]$, the model is aligned. When all $\kappa_p^2$ are roughly the same, the model is balanced.

## 5.3 STAGE 1

In this subsection, we describe the dynamics of gradient descent in Stage 1 and explain how we control the growth of the errors and condition number. For ease of presentation, we will mostly use the infinite-width dynamics (7) instead of the finite-width one (5).

**Equivalence of Stage 1 and non-contrastive methods**  Recall that we use a small $\tau_t^2$ in Stage 1, $T_p \approx 0$ for all $p \in [r]$, whence $\tilde{T}$ is also approximately 0. As a result, the second terms of (7) are approximately 0. In other words, only the positive pairs matter. Meanwhile, one can check that, in the infinite-width limit, (6) corresponds to the first term of (7), up to some multiplicative factor. This gives the equivalence of the dynamics of Stage 1 and the non-contrastive method. See Lemma B.9 for a more formal proof in the finite-width setting.

Now, we consider the contrastive loss. The main result of Stage 1 is as follows.

**Lemma 5.1** (Informal version of Lemma B.1). *Under the assumptions of Theorem 4.3, the finite-width dynamics closely track the infinite-width ones throughout Stage 1, which takes at most $\mathrm{poly}(d)$ iterations, and, at the end of Stage 1, we have, for any $p, q \in [r]$ and $s \in [d-r]$,*

$$\hat{\kappa}_p^2/\kappa_p^2 \approx 1, \quad \|[\boldsymbol{K}_{\boldsymbol{A},\boldsymbol{\xi}}]_s\|^2 / \kappa_p^2 \approx 0, \quad \kappa_p^2/\kappa_q^2 \le O\left(\sqrt{d}\right). \tag{8}$$

*Moreover, there exists a $\boldsymbol{\sigma} \in \mathbb{R}^r$ such that $\max_{p,q} \kappa_p^2/\kappa_q^2 = \Omega(\sqrt{d})$ at the end of Stage 1.*

In words, (8) says, at the end of Stage 1, $\boldsymbol{K}_{\boldsymbol{A}} \approx \boldsymbol{K}_{\boldsymbol{B}}$ in the relative sense, the noise-signal ratio is small, and the condition number is bounded by $O(\sqrt{d})$. By Lemma 4.4, the first two conditions of (8) imply that the learned representations are aligned. The proof of this lemma can be found in Section B. Basically, we couple the convergence of $\tilde{\kappa}_p^2/\kappa_p^2$ and the noise-signal ratio with the growth of discretization error and condition number. The main tool we use is the following nonlinear version of Gronwall's lemma.

**Lemma 5.2.** *Let $A_t$ be a positive process. Let $X_t$ and $Y_t$ be defined as $\dot{X}_t \le -A_t X_t, \dot{Y}_t \le \beta A_t X_t Y_t$, with $X_0, Y_0, \beta$ being positive. Then, for any $T \ge 0$, we have $Y_T \le Y_0 \exp(\beta X_0)$.*

Here, $X_t$ represents the progress we have made and $Y_t$ the error we wish to control. In our case, $X_t$ is the maximum between $1 - \hat{\kappa}_p^2/\kappa_p^2$ and the noise-signal ratio, and $Y_t$ the discretization error and condition number. This lemma says that, if the error growth rate decreases as we make more progress, then by coupling these two processes, we can make sure the error does not blow up.

## 5.4 STAGE 2

In Stage 2, $\tau_t^2$ is no longer $o(1)$, and now the second term of (7) comes into play. We show that, in this stage, the model will reduce the condition number of $\boldsymbol{K}_{\boldsymbol{A}}$ to approximately 1. By Lemma 4.5, this implies that the learned representations are balanced. Formally, we have the following lemma. The proof can be found in Section C.

**Lemma 5.3** (Informal version of Lemma C.1). *Under the assumptions of Theorem 4.3, the finite-width dynamics still closely track the infinite-width one throughout Stage 2, which again takes at most $\mathrm{poly}(d)$ iterations, and, at the end of Stage 2, we have, for any $p, q \in [r]$ and $s \in [d-r]$,*

$$\hat{\kappa}_p^2/\kappa_p^2 \approx 1, \quad \|[\boldsymbol{K}_{\boldsymbol{A},\boldsymbol{\xi}}]_s\|^2 / \kappa_p^2 \approx 0, \quad \kappa_p^2/\kappa_q^2 \approx 1. \tag{9}$$

The way to control $\hat{\kappa}_p^2/\kappa_p^2$ and the noise-signal ratio is similar to Stage 1. For the condition number, note that when $\tau_t^2 = 1$ and $\hat{\kappa} \approx \kappa$, the first term of (7) becomes close to 0 while the second term becomes $-4(1 - \tilde{S})\kappa_p^2/\|\kappa\|^2 (T_p - \tilde{T})\sigma_p^2$. Since $T_p = \tanh(\tau_t^2 \hat{\kappa}_p^2 / \|\kappa\|^2)$ is positively correlated with $\hat{\kappa}_p^2/\|\kappa\|^2$ and $\tilde{T}$ is a weight average of these $T_p$'s, this term will push $\hat{\kappa}_p^2/\|\kappa\|^2$ towards their average and, consequently, reduces the condition number. To obtain a satisfactory convergence rate, it suffices to consider the ratio $\kappa_p^2/\kappa_q^2$ directly. See Appendix C.5 for details. The discretization error is handled in Appendix C.2.

# 6 EXPERIMENTAL RESULTS

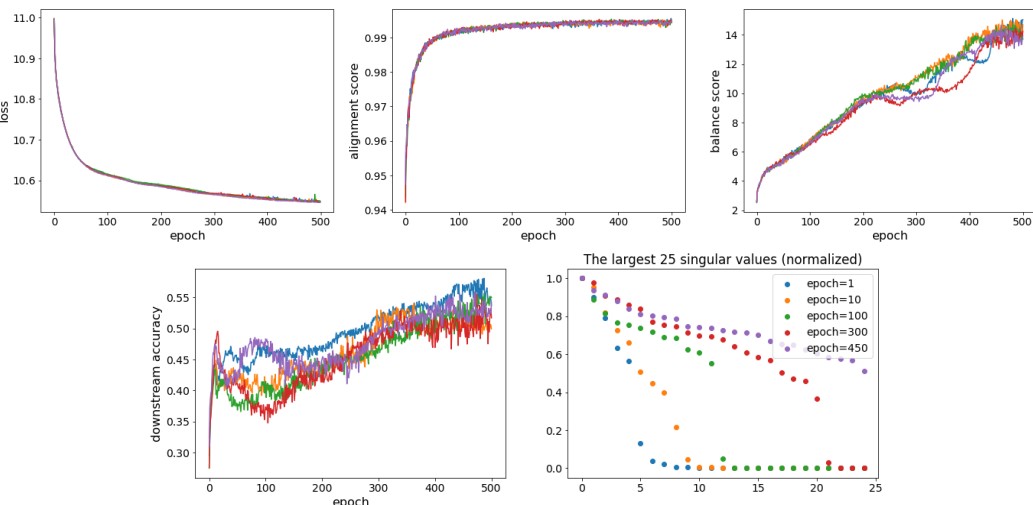

Figure 2: Results of the MSCOCO experiments. The top row plots report the training loss, alignment scores, and balance scores during training, respectively. The bottom row figures plot the downstream accuracies and the largest 25 singular value of $\Sigma_f$ at different epochs, normalized so that the largest one has value 1. One can see that the alignment score quickly reaches near $100\%$, and the balance score, as well as the downstream accuracy, increases gradually during training, which matches our theoretical analysis.

Besides the simulation results reported in Figure 1, we also conduct experiments on the MSCOCO-2014 dataset (Lin et al., 2014) using more practical models. See Figure 2 for the results. For the text part, we use a pre-trained RoBERTa (Liu et al., 2019), followed by a 3-layer fully-connected network with batch norm between layers. For the image part, we use a pre-trained ResNet101 (He et al., 2015), followed by the same layers. In both parts, the width of the fully-connected layers and the output dimension are 768. We freeze the pre-trained parts of the model and only train the fully-connected parts.

We measure the quality of the learned representation using its zero-shot performance on the MSCOCO-2014 validation set. Unlike common image classification datasets, images in the MSCOCO dataset usually have multiple labels, each corresponding to an object that appears in the image, and there are 80 categories in total. We regard a prediction to be correct if it matches one label. The zero-shot classification is done in the same way as in Radford et al. (2021). Namely, we compute the image embedding and the embeddings of prompts "This is a [LABEL_NAME]", and use the prompt with the highest correlation with the image embedding as the prediction.

# 7 CONCLUSION AND DISCUSSION

In this work, we study the role of contrastive pairs in multimodal learning, and show that contrastive pairs are important for the model to learn representations that are both **aligned and balanced**. Our work extends previous results in several directions: First, we consider the more complicated multimodal learning problem. Meanwhile, our data generating model is inhomogeneous, and we show that in this case, non-contrastive method can collapse to an approximately rank-1 solution while contrastive method can learn all features. We also include output normalization in our analysis, a technique that is widely used in practice but is still under-studied in theory.

However, despite the complexity of the analysis, our model is still linear, which is very different from the models used in practice. Also, for the results on non-contrastive methods, we do not consider more advanced training techniques such as Grill et al. (2020) and Chen & He (2021). We leave the analysis of these more practical techniques for future work.

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

# A  GRADIENT CALCULATION

In this section, we compute the gradients and the equations governing some related quantities. We first consider the general finite-width case, and then the infinite-width case, in which we have simple formulas for many quantities of interests.

## A.1  THE FINITE-WIDTH CASE

The results in this subsection are mostly brute-force calculations. First, we prove the following auxiliary lemma.

**Lemma A.1.** *For any $x_A^+$ and $x_B^-$, we have*

$$\nabla_{W_A} \exp(\tau_t^2 f_A^+ \cdot f_B^-) = \frac{\tau_t^2 \exp(\tau_t^2 f_A^+ \cdot f_B^-)}{N_A} \left( x_A^+ (f_B^-)^\top - \langle f_A^+, f_B^- \rangle \frac{\left( AK_A^\top + A_\xi K_{A,\xi}^\top \right)/d}{N_A} \right).$$

Then we compute the gradients.

**Lemma A.2.** *We have*

$$
\begin{aligned}
\nabla_{W_A} \mathcal{L} = &-\tau_t^2 \mathop{\mathbb{E}}_{x_A^+, x_B^+} \left\{ (2 - S_A(x_A^+, x_B^+) - S_B(x_A^+, x_B^+)) \left( \frac{x_A^+ (f_B^+)^\top}{N_A} - \langle f_A^+, f_B^+ \rangle \frac{\left( AK_A^\top + A_\xi K_{A,\xi}^\top \right)}{N_A^2 d} \right) \right\} \\
&+ \frac{K\tau_t^2}{N_A} \mathop{\mathbb{E}}_{x_A^+, x_B^\pm} \left\{ \frac{S_A(x_A^+, x_B^+) \exp(\tau_t^2 f_A^+ \cdot f_B^-)}{\exp(\tau_t^2 f_A^+ \cdot f_B^+)} \left( x_A^+ (f_B^-)^\top - \langle f_A^+, f_B^- \rangle \frac{\left( AK_A^\top + A_\xi K_{A,\xi}^\top \right)/\sqrt{d}}{\sqrt{\|K_A\|_F^2 + \|K_{A,\xi}\|_F^2}} \right) \right\} \\
&+ \frac{K\tau_t^2}{N_A} \mathop{\mathbb{E}}_{x_A^\pm, x_B^+} \left\{ \frac{S_B(x_A^+, x_B^+) \exp(\tau_t^2 f_A^- \cdot f_B^+)}{\exp(\tau_t^2 f_A^+ \cdot f_B^+)} \left( x_A^- (f_B^+)^\top - \langle f_A^-, f_B^+ \rangle \frac{\left( AK_A^\top + A_\xi K_{A,\xi}^\top \right)/\sqrt{d}}{\sqrt{\|K_A\|_F^2 + \|K_{A,\xi}\|_F^2}} \right) \right\}.
\end{aligned}
$$

*The formula for $\nabla_{W_B} \mathcal{L}$ can be obtained by interchanging the roles of $A$ and $B$.*

Instead of tracking $W_A$ and $W_B$ directly, we are going to track $K_A$, $K_B$, $K_{A,\xi}$ and $K_{B,\xi}$. The dynamics of them is governed by the following equation. As a direct corollary of Lemma A.2, we have the following.

**Lemma A.3.** *We have*

$$
\frac{\mathrm{d}}{\mathrm{d}t} K_A
$$

$$
\begin{aligned}
= &\tau_t^2 \mathop{\mathbb{E}}_{x_A^+, x_B^+} \left\{ (2 - S_A(x_A^+, x_B^+) - S_B(x_A^+, x_B^+)) \left( \frac{(K_B z^+ + K_{B,\xi} \xi_B)(z^+)^\top}{N_A N_B} - \langle f_A^+, f_B^+ \rangle \frac{K_A}{N_A^2 d} \right) \right\} \operatorname{diag}(\sigma^2) \\
&- K\tau_t^2 \mathop{\mathbb{E}}_{x_A^+, x_B^\pm} \left\{ \frac{S_A(x_A^+, x_B^+) \exp(\tau_t^2 f_A^+ \cdot f_B^-)}{\exp(\tau_t^2 f_A^+ \cdot f_B^+)} \left( \frac{f_B^-(z^+)^\top}{N_A} - \langle f_A^+, f_B^- \rangle \frac{K_A}{N_A^2 d} \right) \right\} \operatorname{diag}(\sigma^2) \\
&- K\tau_t^2 \mathop{\mathbb{E}}_{x_A^\pm, x_B^+} \left\{ \frac{S_B(x_A^+, x_B^+) \exp(\tau_t^2 f_A^- \cdot f_B^+)}{\exp(\tau_t^2 f_A^+ \cdot f_B^+)} \left( \frac{f_B^+(z^-)^\top}{N_A} - \langle f_A^-, f_B^+ \rangle \frac{K_A}{N_A^2 d} \right) \right\} \operatorname{diag}(\sigma^2),
\end{aligned}
$$

*and*

$$
\begin{aligned}
\frac{\mathrm{d}}{\mathrm{d}t} K_{A,\xi} = &\frac{\tau_t^2}{N_A} \mathop{\mathbb{E}}_{x_A^+, x_B^\pm} \left\{ (2 - S_A(x_A^+, x_B^+) - S_B(x_A^+, x_B^+)) \left( f_B^+ (\xi_A^+)^\top - \frac{\langle f_A^+, f_B^+ \rangle K_{A,\xi}/\sqrt{d}}{\sqrt{\|K_A\|_F^2 + \|K_{A,\xi}\|_F^2}} \right) \right\} \sigma_\xi^2 \\
&- \frac{K\tau_t^2}{N_A} \mathop{\mathbb{E}}_{x_A^+, x_B^\pm} \left\{ \frac{S_A(x_A^+, x_B^+) \exp(\tau_t^2 f_A^+ \cdot f_B^-)}{\exp(\tau_t^2 f_A^+ \cdot f_B^+)} \left( f_B^- (\xi_A^+)^\top - \frac{\langle f_A^+, f_B^- \rangle K_{A,\xi}/\sqrt{d}}{\sqrt{\|K_A\|_F^2 + \|K_{A,\xi}\|_F^2}} \right) \right\} \sigma_\xi^2 \\
&- \frac{K\tau_t^2}{N_A} \mathop{\mathbb{E}}_{x_A^\pm, x_B^+} \left\{ \frac{S_B(x_A^+, x_B^+) \exp(\tau_t^2 f_A^- \cdot f_B^+)}{\exp(\tau_t^2 f_A^+ \cdot f_B^+)} \left( f_B^+ (\xi_A^-)^\top - \frac{\langle f_A^-, f_B^+ \rangle K_{A,\xi}/\sqrt{d}}{\sqrt{\|K_A\|_F^2 + \|K_{A,\xi}\|_F^2}} \right) \right\} \sigma_\xi^2.
\end{aligned}
$$

We can rewrite the above result as follows.

**Corollary A.4.** *Define*

$$Q_0 := - \mathop{\mathbb{E}}_{\boldsymbol{x}_A^+, \boldsymbol{x}_B^+} \left\{ \left(2 - S_A(\boldsymbol{x}_A^+, \boldsymbol{x}_B^+) - S_B(\boldsymbol{x}_A^+, \boldsymbol{x}_B^+)\right) \langle \boldsymbol{f}_A^+, \boldsymbol{f}_B^+ \rangle \right\}$$

$$+ K \mathop{\mathbb{E}}_{\boldsymbol{x}_A^\pm, \boldsymbol{x}_B^\pm} \left\{ \frac{S_A(\boldsymbol{x}_A^+, \boldsymbol{x}_B^+) \exp(\tau_t^2 \boldsymbol{f}_A^+ \cdot \boldsymbol{f}_B^-)}{\exp(\tau_t^2 \boldsymbol{f}_A^+ \cdot \boldsymbol{f}_B^+)} \langle \boldsymbol{f}_A^+, \boldsymbol{f}_B^- \rangle \right\}$$

$$+ K \mathop{\mathbb{E}}_{\boldsymbol{x}_A^\pm, \boldsymbol{x}_B^+} \left\{ \frac{S_B(\boldsymbol{x}_A^+, \boldsymbol{x}_B^+) \exp(\tau_t^2 \boldsymbol{f}_A^- \cdot \boldsymbol{f}_B^+)}{\exp(\tau_t^2 \boldsymbol{f}_A^+ \cdot \boldsymbol{f}_B^+)} \langle \boldsymbol{f}_A^-, \boldsymbol{f}_B^+ \rangle \right\},$$

*and*

$$\boldsymbol{Q}_1 := \mathbb{E}\left\{ \left(2 - S_A(\boldsymbol{x}_A^+, \boldsymbol{x}_B^+) - S_B(\boldsymbol{x}_A^+, \boldsymbol{x}_B^+)\right) \boldsymbol{z}^+ (\boldsymbol{z}^+)^\top d \right\}$$

$$- K \mathbb{E}\left\{ \frac{S_A(\boldsymbol{x}_A^+, \boldsymbol{x}_B^+) \exp(\tau_t^2 \boldsymbol{f}_A^+ \cdot \boldsymbol{f}_B^-)}{\exp(\tau_t^2 \boldsymbol{f}_A^+ \cdot \boldsymbol{f}_B^+)} \boldsymbol{z}^- (\boldsymbol{z}^+)^\top d \right\}$$

$$- K \mathbb{E}\left\{ \frac{S_B(\boldsymbol{x}_A^+, \boldsymbol{x}_B^+) \exp(\tau_t^2 \boldsymbol{f}_A^- \cdot \boldsymbol{f}_B^+)}{\exp(\tau_t^2 \boldsymbol{f}_A^+ \cdot \boldsymbol{f}_B^+)} \boldsymbol{z}^+ (\boldsymbol{z}^-)^\top d \right\},$$

$$\boldsymbol{Q}_{1,\xi_B} := \mathbb{E}\left\{ \left(2 - S_A(\boldsymbol{x}_A^+, \boldsymbol{x}_B^+) - S_B(\boldsymbol{x}_A^+, \boldsymbol{x}_B^+)\right) \boldsymbol{\xi}_B^+ (\boldsymbol{z}^+)^\top d \right\}$$

$$- K \mathbb{E}\left\{ \frac{S_A(\boldsymbol{x}_A^+, \boldsymbol{x}_B^+) \exp(\tau_t^2 \boldsymbol{f}_A^+ \cdot \boldsymbol{f}_B^-)}{\exp(\tau_t^2 \boldsymbol{f}_A^+ \cdot \boldsymbol{f}_B^+)} \boldsymbol{\xi}_B^- (\boldsymbol{z}^+)^\top d \right\}$$

$$- K \mathbb{E}\left\{ \frac{S_B(\boldsymbol{x}_A^+, \boldsymbol{x}_B^+) \exp(\tau_t^2 \boldsymbol{f}_A^- \cdot \boldsymbol{f}_B^+)}{\exp(\tau_t^2 \boldsymbol{f}_A^+ \cdot \boldsymbol{f}_B^+)} \boldsymbol{\xi}_B^+ (\boldsymbol{z}^-)^\top d \right\},$$

$$\boldsymbol{Q}_{1,\xi_A} := \mathbb{E}\left\{ \left(2 - S_A(\boldsymbol{x}_A^+, \boldsymbol{x}_B^+) - S_B(\boldsymbol{x}_A^+, \boldsymbol{x}_B^+)\right) \boldsymbol{\xi}_A^+ (\boldsymbol{z}^+)^\top d \right\}$$

$$- K \mathbb{E}\left\{ \frac{S_A(\boldsymbol{x}_A^+, \boldsymbol{x}_B^+) \exp(\tau_t^2 \boldsymbol{f}_A^+ \cdot \boldsymbol{f}_B^-)}{\exp(\tau_t^2 \boldsymbol{f}_A^+ \cdot \boldsymbol{f}_B^+)} \boldsymbol{\xi}_A^+ (\boldsymbol{z}^-)^\top d \right\}$$

$$- K \mathbb{E}\left\{ \frac{S_B(\boldsymbol{x}_A^+, \boldsymbol{x}_B^+) \exp(\tau_t^2 \boldsymbol{f}_A^- \cdot \boldsymbol{f}_B^+)}{\exp(\tau_t^2 \boldsymbol{f}_A^+ \cdot \boldsymbol{f}_B^+)} \boldsymbol{\xi}_A^- (\boldsymbol{z}^+)^\top d \right\},$$

$$\boldsymbol{Q}_2 := \mathbb{E}\left\{ \left(2 - S_A(\boldsymbol{x}_A^+, \boldsymbol{x}_B^+) - S_B(\boldsymbol{x}_A^+, \boldsymbol{x}_B^+)\right) \boldsymbol{\xi}_B^+ (\boldsymbol{\xi}_A^+)^\top d \right\}$$

$$- K \mathbb{E}\left\{ \frac{S_A(\boldsymbol{x}_A^+, \boldsymbol{x}_B^+) \exp(\tau_t^2 \boldsymbol{f}_A^+ \cdot \boldsymbol{f}_B^-)}{\exp(\tau_t^2 \boldsymbol{f}_A^+ \cdot \boldsymbol{f}_B^+)} \boldsymbol{\xi}_B^- (\boldsymbol{\xi}_A^+)^\top d \right\}$$

$$- K \mathbb{E}\left\{ \frac{S_B(\boldsymbol{x}_A^+, \boldsymbol{x}_B^+) \exp(\tau_t^2 \boldsymbol{f}_A^- \cdot \boldsymbol{f}_B^+)}{\exp(\tau_t^2 \boldsymbol{f}_A^+ \cdot \boldsymbol{f}_B^+)} \boldsymbol{\xi}_B^+ (\boldsymbol{\xi}_A^-)^\top d \right\}.$$

*We have*

$$\frac{\mathrm{d}}{\mathrm{d}t} \boldsymbol{K}_A = \frac{\boldsymbol{K}_B}{N_A N_B d} \boldsymbol{Q}_1 \operatorname{diag}(\boldsymbol{\sigma}^2) + \frac{\boldsymbol{K}_{B,\xi}}{N_A N_B d} \boldsymbol{Q}_{1,\xi_B} \operatorname{diag}(\boldsymbol{\sigma}^2) + \frac{\boldsymbol{K}_A}{N_A^2 d} Q_0 \operatorname{diag}(\boldsymbol{\sigma}^2),$$

$$\frac{\mathrm{d}}{\mathrm{d}t} \boldsymbol{K}_B = \frac{\boldsymbol{K}_A}{N_A N_B d} \boldsymbol{Q}_1^\top \operatorname{diag}(\boldsymbol{\sigma}^2) + \frac{\boldsymbol{K}_{A,\xi}}{N_A N_B d} \boldsymbol{Q}_{1,\xi_A} \operatorname{diag}(\boldsymbol{\sigma}^2) + \frac{\boldsymbol{K}_B}{N_B^2 d} Q_0 \operatorname{diag}(\boldsymbol{\sigma}^2),$$

$$\frac{\mathrm{d}}{\mathrm{d}t} \boldsymbol{K}_{A,\xi} = \frac{\boldsymbol{K}_B}{N_A N_B d} \boldsymbol{Q}_{1,\xi_A}^\top \sigma_\xi^2 + \frac{\boldsymbol{K}_{B,\xi}}{N_A N_B d} \boldsymbol{Q}_2 \sigma_\xi^2 + \frac{\boldsymbol{K}_{A,\xi}}{N_A^2 d} Q_0 \sigma_\xi^2,$$

$$\frac{\mathrm{d}}{\mathrm{d}t} \boldsymbol{K}_{B,\xi} = \frac{\boldsymbol{K}_A}{N_A N_B d} \boldsymbol{Q}_{1,\xi_B}^\top \sigma_\xi^2 + \frac{\boldsymbol{K}_{A,\xi}}{N_A N_B d} \boldsymbol{Q}_2^\top \sigma_\xi^2 + \frac{\boldsymbol{K}_{B,\xi}}{N_B^2 d} Q_0 \sigma_\xi^2.$$

We are interested in each column of $\boldsymbol{K}_A$ and $\boldsymbol{K}_B$, whose dynamics is given by the next lemma. The next lemma also decompose the dynamics along the radial and tangent directions.

**Lemma A.5.** *For any $p \in [r]$ and $q \in [d-r]$, we have*

$$\frac{\mathrm{d}}{\mathrm{d}t}\|[K_A]_p\|^2 = 2\frac{\langle [K_A]_p, [K_B Q_1]_p\rangle}{N_A N_B d}\sigma_p^2 + 2\frac{\langle [K_A]_p, [K_{B,\xi}Q_{1,\xi_B}]_p\rangle}{N_A N_B d}\sigma_p^2 + 2\frac{\|[K_A]_p\|^2}{N_A^2 d}Q_0\sigma_p^2,$$

$$\frac{\mathrm{d}}{\mathrm{d}t}\|[K_B]_p\|^2 = 2\frac{\langle [K_B]_p, [K_A Q_1^\top]_p\rangle}{N_A N_B d}\sigma_p^2 + 2\frac{\langle [K_B]_p, [K_{A,\xi}Q_{1,\xi_A}]_p\rangle}{N_A N_B d}\sigma_p^2 + 2\frac{\|[K_B]_p\|^2}{N_B^2 d}Q_0\sigma_p^2,$$

$$\frac{\mathrm{d}}{\mathrm{d}t}\|[K_{A,\xi}]_q\|^2 = 2\frac{\left\langle [K_{A,\xi}]_q, [K_B Q_{1,\xi_A}^\top]_q\right\rangle}{N_A N_B d}\sigma_\xi^2 + 2\frac{\langle [K_{A,\xi}]_q, [K_{B,\xi}Q_2]_q\rangle}{N_A N_B d}\sigma_\xi^2 + 2\frac{\|[K_{A,\xi}]_q\|_F^2}{N_A^2 d}Q_0\sigma_\xi^2,$$

$$\frac{\mathrm{d}}{\mathrm{d}t}\|[K_{B,\xi}]_q\|^2 = 2\frac{\left\langle [K_{B,\xi}]_q, [K_A Q_{1,\xi_B}^\top]_q\right\rangle}{N_A N_B d}\sigma_\xi^2 + 2\frac{\langle [K_{B,\xi}]_q, [K_{A,\xi}Q_2^\top]_q\rangle}{N_A N_B d}\sigma_\xi^2 + 2\frac{\|[K_{B,\xi}]_q\|^2}{N_A^2 d}Q_0\sigma_\xi^2,$$

*and*

$$\frac{\mathrm{d}}{\mathrm{d}t}\overline{[K_A]_p} = \left(I - \overline{[K_A]_p}\left(\overline{[K_A]_p}\right)^\top\right)\left(\frac{[K_B Q_1]_p}{\|[K_A]_p\|} + \frac{[K_{B,\xi}Q_{1,\xi_B}]_p}{\|[K_A]_p\|}\right)\frac{\sigma_p^2}{N_A N_B d},$$

$$\frac{\mathrm{d}}{\mathrm{d}t}\overline{[K_B]_p} = \left(I - \overline{[K_B]_p}\left(\overline{[K_B]_p}\right)^\top\right)\left(\frac{[K_A Q_1^\top]_p}{\|[K_B]_p\|} + \frac{[K_{A,\xi}Q_{1,\xi_A}]_p}{\|[K_B]_p\|}\right)\frac{\sigma_p^2}{N_A N_B d},$$

$$\frac{\mathrm{d}}{\mathrm{d}t}\overline{[K_{A,\xi}]_q} = \left(I - \overline{[K_{A,\xi}]_q}\left(\overline{[K_{A,\xi}]_q}\right)^\top\right)\left(\frac{[K_B Q_{1,\xi_A}^\top]_q}{\|[K_{A,\xi}]_q\|} + \frac{[K_{B,\xi}Q_2]_q}{\|[K_{A,\xi}]_q\|}\right)\frac{\sigma_\xi^2}{N_A N_B d},$$

$$\frac{\mathrm{d}}{\mathrm{d}t}\overline{[K_{B,\xi}]_q} = \left(I - \overline{[K_{B,\xi}]_q}\left(\overline{[K_{B,\xi}]_q}\right)^\top\right)\left(\frac{[K_A Q_{1,\xi_B}^\top]_q}{\|[K_{B,\xi}]_q\|} + \frac{[K_{A,\xi}Q_2^\top]_q}{\|[K_{B,\xi}]_q\|}\right)\frac{\sigma_\xi^2}{N_A N_B d}.$$

Finally, as a simple corollary of Lemma A.1, we have the following result on the gradients of the non-contrastive loss.

**Lemma A.6.** *For the non-contrastive loss* (4)*, we have*

$$\nabla_{W_A}\hat{\mathcal{L}} = \mathbb{E}\left\{\frac{x_A^+(f_B^+)^\top}{N_A} - \langle f_A^+, f_B^+\rangle\frac{\left(A K_A^\top + A_\xi K_{A,\xi}^\top\right)}{N_A^2 d}\right\}$$

*As a corollary, we have, in this case,*

$$\frac{\mathrm{d}}{\mathrm{d}t}K_A = \mathbb{E}\left\{\frac{f_B^+(z^+)^\top}{N_A} - \langle f_A^+, f_B^+\rangle\frac{K_A}{N_A^2 d}\right\}\mathrm{diag}(\sigma^2),$$

$$\frac{\mathrm{d}}{\mathrm{d}t}K_{A,\xi} = \mathbb{E}\left\{\frac{f_B^+(\xi_A^+)^\top}{N_A} - \langle f_A^+, f_B^+\rangle\frac{K_{A,\xi}}{N_A^2 d}\right\}\sigma_\xi^2.$$

OMITTED PROOF OF THIS SUBSECTION

*Proof of Lemma A.1.* We compute

$$\nabla_{W_A}\exp(\tau_t^2 f_A^+ \cdot f_B^-) = \tau_t^2\exp(\tau_t^2 f_A^+ \cdot f_B^-)\nabla_{W_A}\langle f_A^+, f_B^-\rangle$$

$$= \tau_t^2\exp(\tau_t^2 f_A^+ \cdot f_B^-)\nabla_{W_A}\frac{\langle W_A^\top x_A^+, f_B^-\rangle}{\sqrt{\|K_A\|_F^2 + \|K_{A,\xi}\|_F^2}/\sqrt{d}}$$

$$= \tau_t^2\exp(\tau_t^2 f_A^+ \cdot f_B^-)\frac{\nabla_{W_A}\langle W_A^\top x_A^+, f_B^-\rangle}{\sqrt{\|K_A\|_F^2 + \|K_{A,\xi}\|_F^2}/\sqrt{d}}$$

$$- \tau_t^2\exp(\tau_t^2 f_A^+ \cdot f_B^-)\langle f_A^+, f_B^-\rangle\frac{\nabla_{W_A}\sqrt{\|K_A\|_F^2 + \|K_{A,\xi}\|_F^2}}{\sqrt{\|K_A\|_F^2 + \|K_{A,\xi}\|_F^2}}.$$

For the first term, we have $\nabla_{\boldsymbol{W_A}} \left\langle \boldsymbol{W_A}^\top \boldsymbol{x_A^+}, \boldsymbol{f_B^-} \right\rangle = \boldsymbol{x_A^+}(\boldsymbol{f_B^-})^\top$. For the second term, we have

$$\frac{\nabla_{\boldsymbol{W_A}} \sqrt{\|\boldsymbol{K_A}\|_F^2 + \|\boldsymbol{K_{A,\xi}}\|_F^2}}{\sqrt{\|\boldsymbol{K_A}\|_F^2 + \|\boldsymbol{K_{A,\xi}}\|_F^2}} = \frac{\nabla_{\boldsymbol{W_A}} \|\boldsymbol{K_A}\|_F^2 + \nabla_{\boldsymbol{W_A}} \|\boldsymbol{K_{A,\xi}}\|_F^2}{2(\|\boldsymbol{K_A}\|_F^2 + \|\boldsymbol{K_{A,\xi}}\|_F^2)} = \frac{\boldsymbol{A}\boldsymbol{K_A}^\top + \boldsymbol{A_\xi}\boldsymbol{K_{A,\xi}}^\top}{\|\boldsymbol{K_A}\|_F^2 + \|\boldsymbol{K_{A,\xi}}\|_F^2}.$$

Hence,

$$\nabla_{\boldsymbol{W_A}} \exp(\tau_t^2 \boldsymbol{f_A^+} \cdot \boldsymbol{f_B^-}) = \tau_t^2 \exp(\tau_t^2 \boldsymbol{f_A^+} \cdot \boldsymbol{f_B^-}) \frac{\boldsymbol{x_A^+}(\boldsymbol{f_B^-})^\top}{\sqrt{\|\boldsymbol{K_A}\|_F^2 + \|\boldsymbol{K_{A,\xi}}\|_F^2}/\sqrt{d}}$$

$$- \tau_t^2 \exp(\tau_t^2 \boldsymbol{f_A^+} \cdot \boldsymbol{f_B^-}) \left\langle \boldsymbol{f_A^+}, \boldsymbol{f_B^-} \right\rangle \frac{\boldsymbol{A}\boldsymbol{K_A}^\top + \boldsymbol{A_\xi}\boldsymbol{K_{A,\xi}}^\top}{\|\boldsymbol{K_A}\|_F^2 + \|\boldsymbol{K_{A,\xi}}\|_F^2}$$

$$= \frac{\tau_t^2 \exp(\tau_t^2 \boldsymbol{f_A^+} \cdot \boldsymbol{f_B^-})}{\sqrt{\|\boldsymbol{K_A}\|_F^2 + \|\boldsymbol{K_{A,\xi}}\|_F^2}/\sqrt{d}} \left( \boldsymbol{x_A^+}(\boldsymbol{f_B^-})^\top - \left\langle \boldsymbol{f_A^+}, \boldsymbol{f_B^-} \right\rangle \frac{\left(\boldsymbol{A}\boldsymbol{K_A}^\top + \boldsymbol{A_\xi}\boldsymbol{K_{A,\xi}}^\top\right)/\sqrt{d}}{\sqrt{\|\boldsymbol{K_A}\|_F^2 + \|\boldsymbol{K_{A,\xi}}\|_F^2}} \right).$$

$\square$

*Proof of Lemma A.2.* We compute

$$\nabla_{\boldsymbol{W_A}} \mathcal{L}_A = - \mathbb{E} \frac{\nabla_{\boldsymbol{W_A}} S_A(\boldsymbol{x_A^+}, \boldsymbol{x_B^+})}{S_A(\boldsymbol{x_A^+}, \boldsymbol{x_B^+})}$$

$$= - \mathbb{E} \left\{ \frac{1}{S_A(\boldsymbol{x_A^+}, \boldsymbol{x_B^+})} \nabla_{\boldsymbol{W_A}} \frac{\exp(\tau_t^2 \boldsymbol{f_A^+} \cdot \boldsymbol{f_B^+})}{\exp(\tau_t^2 \boldsymbol{f_A^+} \cdot \boldsymbol{f_B^+}) + K \mathbb{E}_{\boldsymbol{z^-}} \exp(\tau_t^2 \boldsymbol{f_A^+} \cdot \boldsymbol{f_B^-})} \right\}$$

$$= - \mathbb{E} \left\{ \frac{1}{S_A(\boldsymbol{x_A^+}, \boldsymbol{x_B^+})} \frac{\nabla_{\boldsymbol{W_A}} \exp(\tau_t^2 \boldsymbol{f_A^+} \cdot \boldsymbol{f_B^+})}{\exp(\tau_t^2 \boldsymbol{f_A^+} \cdot \boldsymbol{f_B^+}) + K \mathbb{E}_{\boldsymbol{z^-}} \exp(\tau_t^2 \boldsymbol{f_A^+} \cdot \boldsymbol{f_B^-})} \right\}$$

$$+ \mathbb{E} \left\{ \frac{\nabla_{\boldsymbol{W_A}} \exp(\tau_t^2 \boldsymbol{f_A^+} \cdot \boldsymbol{f_B^+}) + K \mathbb{E}_{\boldsymbol{z^-}} \nabla_{\boldsymbol{W_A}} \exp(\tau_t^2 \boldsymbol{f_A^+} \cdot \boldsymbol{f_B^-})}{\exp(\tau_t^2 \boldsymbol{f_A^+} \cdot \boldsymbol{f_B^+}) + K \mathbb{E}_{\boldsymbol{z^-}} \exp(\tau_t^2 \boldsymbol{f_A^+} \cdot \boldsymbol{f_B^-})} \right\}.$$

By Lemma A.1, the first term is

$$- \mathbb{E} \left\{ \frac{1}{S_A(\boldsymbol{x_A^+}, \boldsymbol{x_B^+})} \frac{\nabla_{\boldsymbol{W_A}} \exp(\tau_t^2 \boldsymbol{f_A^+} \cdot \boldsymbol{f_B^+})}{\exp(\tau_t^2 \boldsymbol{f_A^+} \cdot \boldsymbol{f_B^+}) + K \mathbb{E}_{\boldsymbol{z^-}} \exp(\tau_t^2 \boldsymbol{f_A^+} \cdot \boldsymbol{f_B^-})} \right\}$$

$$= - \frac{\tau_t^2}{N_A} \mathop{\mathbb{E}}_{\boldsymbol{x_A^+}, \boldsymbol{x_B^+}} \left\{ \boldsymbol{x_A^+}(\boldsymbol{f_B^+})^\top - \left\langle \boldsymbol{f_A^+}, \boldsymbol{f_B^+} \right\rangle \frac{\left(\boldsymbol{A}\boldsymbol{K_A}^\top + \boldsymbol{A_\xi}\boldsymbol{K_{A,\xi}}^\top\right)/\sqrt{d}}{\sqrt{\|\boldsymbol{K_A}\|_F^2 + \|\boldsymbol{K_{A,\xi}}\|_F^2}} \right\},$$

and the second term is

$$\mathbb{E} \left\{ \frac{\nabla_{\boldsymbol{W_A}} \exp(\tau_t^2 \boldsymbol{f_A^+} \cdot \boldsymbol{f_B^+}) + K \mathbb{E}_{\boldsymbol{x_B^-}} \nabla_{\boldsymbol{W_A}} \exp(\tau_t^2 \boldsymbol{f_A^+} \cdot \boldsymbol{f_B^-})}{\exp(\tau_t^2 \boldsymbol{f_A^+} \cdot \boldsymbol{f_B^+}) + K \mathbb{E}_{\boldsymbol{z^-}} \exp(\tau_t^2 \boldsymbol{f_A^+} \cdot \boldsymbol{f_B^-})} \right\}$$

$$= \frac{\tau_t^2}{N_A} \mathbb{E} \left\{ S_A(\boldsymbol{x_A^+}, \boldsymbol{x_B^+}) \left( \boldsymbol{x_A^+}(\boldsymbol{f_B^+})^\top - \left\langle \boldsymbol{f_A^+}, \boldsymbol{f_B^+} \right\rangle \frac{\left(\boldsymbol{A}\boldsymbol{K_A}^\top + \boldsymbol{A_\xi}\boldsymbol{K_{A,\xi}}^\top\right)/\sqrt{d}}{\sqrt{\|\boldsymbol{K_A}\|_F^2 + \|\boldsymbol{K_{A,\xi}}\|_F^2}} \right) \right\}$$

$$+ \frac{K\tau_t^2}{N_A} \mathbb{E} \left\{ \frac{S_A(\boldsymbol{x_A^+}, \boldsymbol{x_B^+}) \exp(\tau_t^2 \boldsymbol{f_A^+} \cdot \boldsymbol{f_B^-})}{\exp(\tau_t^2 \boldsymbol{f_A^+} \cdot \boldsymbol{f_B^+})} \left( \boldsymbol{x_A^+}(\boldsymbol{f_B^-})^\top - \left\langle \boldsymbol{f_A^+}, \boldsymbol{f_B^-} \right\rangle \frac{\left(\boldsymbol{A}\boldsymbol{K_A}^\top + \boldsymbol{A_\xi}\boldsymbol{K_{A,\xi}}^\top\right)/\sqrt{d}}{\sqrt{\|\boldsymbol{K_A}\|_F^2 + \|\boldsymbol{K_{A,\xi}}\|_F^2}} \right) \right\}.$$

Thus,

$$\nabla_{\boldsymbol{W_A}} \mathcal{L}_A = - \frac{\tau_t^2}{N_A} \mathop{\mathbb{E}}_{\boldsymbol{x_A^+}, \boldsymbol{x_B^+}} \left\{ (1 - S_A(\boldsymbol{x_A^+}, \boldsymbol{x_B^+})) \left( \boldsymbol{x_A^+}(\boldsymbol{f_B^+})^\top - \left\langle \boldsymbol{f_A^+}, \boldsymbol{f_B^+} \right\rangle \frac{\left(\boldsymbol{A}\boldsymbol{K_A}^\top + \boldsymbol{A_\xi}\boldsymbol{K_{A,\xi}}^\top\right)/\sqrt{d}}{\sqrt{\|\boldsymbol{K_A}\|_F^2 + \|\boldsymbol{K_{A,\xi}}\|_F^2}} \right) \right\}$$

$$+ \frac{K\tau_t^2}{N_A} \mathbb{E} \left\{ \frac{S_A(\boldsymbol{x_A^+}, \boldsymbol{x_B^+}) \exp(\tau_t^2 \boldsymbol{f_A^+} \cdot \boldsymbol{f_B^-})}{\exp(\tau_t^2 \boldsymbol{f_A^+} \cdot \boldsymbol{f_B^+})} \left( \boldsymbol{x_A^+}(\boldsymbol{f_B^-})^\top - \left\langle \boldsymbol{f_A^+}, \boldsymbol{f_B^-} \right\rangle \frac{\left(\boldsymbol{A}\boldsymbol{K_A}^\top + \boldsymbol{A_\xi}\boldsymbol{K_{A,\xi}}^\top\right)/\sqrt{d}}{\sqrt{\|\boldsymbol{K_A}\|_F^2 + \|\boldsymbol{K_{A,\xi}}\|_F^2}} \right) \right\}.$$

Then, for $\nabla_{\boldsymbol{W_A}} \mathcal{L}_B$, we compute

$$\nabla_{\boldsymbol{W_A}} \mathcal{L}_B = -\mathbb{E} \frac{\nabla_{\boldsymbol{W_A}} S_B(\boldsymbol{x}_A^+, \boldsymbol{x}_B^+)}{S_B(\boldsymbol{x}_A^+, \boldsymbol{x}_B^+)}$$

$$= -\mathbb{E} \left\{ \frac{1}{S_B(\boldsymbol{x}_A^+, \boldsymbol{x}_B^+)} \frac{\nabla_{\boldsymbol{W_A}} \exp(\tau_t^2 \boldsymbol{f}_A^+ \cdot \boldsymbol{f}_B^+)}{\exp(\tau_t^2 \boldsymbol{f}_A^+ \cdot \boldsymbol{f}_B^+) + K \mathbb{E}_{\boldsymbol{x}_A^-} \exp(\tau_t^2 \boldsymbol{f}_A^- \cdot \boldsymbol{f}_B^+)} \right\}$$

$$+ \mathbb{E} \left\{ \frac{\nabla_{\boldsymbol{W_A}} \exp(\tau_t^2 \boldsymbol{f}_A^+ \cdot \boldsymbol{f}_B^+) + K \nabla_{\boldsymbol{W_A}} \mathbb{E}_{\boldsymbol{x}_A^-} \exp(\tau_t^2 \boldsymbol{f}_A^- \cdot \boldsymbol{f}_B^+)}{\exp(\tau_t^2 \boldsymbol{f}_A^+ \cdot \boldsymbol{f}_B^+) + K \mathbb{E}_{\boldsymbol{x}_A^-} \exp(\tau_t^2 \boldsymbol{f}_A^- \cdot \boldsymbol{f}_B^+)} \right\}.$$

Again, by Lemma A.1, the first term is

$$-\mathbb{E} \left\{ \frac{1}{S_B(\boldsymbol{x}_A^+, \boldsymbol{x}_B^+)} \frac{\nabla_{\boldsymbol{W_A}} \exp(\tau_t^2 \boldsymbol{f}_A^+ \cdot \boldsymbol{f}_B^+)}{\exp(\tau_t^2 \boldsymbol{f}_A^+ \cdot \boldsymbol{f}_B^+) + K \mathbb{E}_{\boldsymbol{x}_A^-} \exp(\tau_t^2 \boldsymbol{f}_A^- \cdot \boldsymbol{f}_B^+)} \right\}$$

$$= -\frac{\tau_t^2}{N_A} \mathbb{E} \left\{ \boldsymbol{x}_A^+ (\boldsymbol{f}_B^+)^\top - \langle \boldsymbol{f}_A^+, \boldsymbol{f}_B^+ \rangle \frac{\left( \boldsymbol{A} \boldsymbol{K}_A^\top + \boldsymbol{A}_\xi \boldsymbol{K}_{A,\xi}^\top \right) / \sqrt{d}}{\sqrt{\|\boldsymbol{K}_A\|_F^2 + \|\boldsymbol{K}_{A,\xi}\|_F^2}} \right\},$$

and the second term is

$$\mathbb{E} \left\{ \frac{\nabla_{\boldsymbol{W_A}} \exp(\tau_t^2 \boldsymbol{f}_A^+ \cdot \boldsymbol{f}_B^+) + K \nabla_{\boldsymbol{W_A}} \mathbb{E}_{\boldsymbol{x}_A^-} \exp(\tau_t^2 \boldsymbol{f}_A^- \cdot \boldsymbol{f}_B^+)}{\exp(\tau_t^2 \boldsymbol{f}_A^+ \cdot \boldsymbol{f}_B^+) + K \mathbb{E}_{\boldsymbol{x}_A^-} \exp(\tau_t^2 \boldsymbol{f}_A^- \cdot \boldsymbol{f}_B^+)} \right\}$$

$$= \frac{\tau_t^2}{N_A} \mathbb{E} \left\{ S_B(\boldsymbol{x}_A^+, \boldsymbol{x}_B^+) \left( \boldsymbol{x}_A^+ (\boldsymbol{f}_B^+)^\top - \langle \boldsymbol{f}_A^+, \boldsymbol{f}_B^+ \rangle \frac{\left( \boldsymbol{A} \boldsymbol{K}_A^\top + \boldsymbol{A}_\xi \boldsymbol{K}_{A,\xi}^\top \right) / \sqrt{d}}{\sqrt{\|\boldsymbol{K}_A\|_F^2 + \|\boldsymbol{K}_{A,\xi}\|_F^2}} \right) \right\}$$

$$+ \frac{K \tau_t^2}{N_A} \mathbb{E} \left\{ \frac{S_B(\boldsymbol{x}_A^+, \boldsymbol{x}_B^+) \exp(\tau_t^2 \boldsymbol{f}_A^- \cdot \boldsymbol{f}_B^+)}{\exp(\tau_t^2 \boldsymbol{f}_A^+ \cdot \boldsymbol{f}_B^+)} \left( \boldsymbol{x}_A^- (\boldsymbol{f}_B^+)^\top - \langle \boldsymbol{f}_A^-, \boldsymbol{f}_B^+ \rangle \frac{\left( \boldsymbol{A} \boldsymbol{K}_A^\top + \boldsymbol{A}_\xi \boldsymbol{K}_{A,\xi}^\top \right) / \sqrt{d}}{\sqrt{\|\boldsymbol{K}_A\|_F^2 + \|\boldsymbol{K}_{A,\xi}\|_F^2}} \right) \right\}.$$

Thus,

$$\nabla_{\boldsymbol{W_A}} \mathcal{L}_B = -\frac{\tau_t^2}{N_A} \mathbb{E} \left\{ (1 - S_B(\boldsymbol{x}_A^+, \boldsymbol{x}_B^+)) \left( \boldsymbol{x}_A^+ (\boldsymbol{f}_B^+)^\top - \langle \boldsymbol{f}_A^+, \boldsymbol{f}_B^+ \rangle \frac{\left( \boldsymbol{A} \boldsymbol{K}_A^\top + \boldsymbol{A}_\xi \boldsymbol{K}_{A,\xi}^\top \right) / \sqrt{d}}{\sqrt{\|\boldsymbol{K}_A\|_F^2 + \|\boldsymbol{K}_{A,\xi}\|_F^2}} \right) \right\}$$

$$+ \frac{K \tau_t^2}{N_A} \mathbb{E} \left\{ \frac{S_B(\boldsymbol{x}_A^+, \boldsymbol{x}_B^+) \exp(\tau_t^2 \boldsymbol{f}_A^- \cdot \boldsymbol{f}_B^+)}{\exp(\tau_t^2 \boldsymbol{f}_A^+ \cdot \boldsymbol{f}_B^+)} \left( \boldsymbol{x}_A^- (\boldsymbol{f}_B^+)^\top - \langle \boldsymbol{f}_A^-, \boldsymbol{f}_B^+ \rangle \frac{\left( \boldsymbol{A} \boldsymbol{K}_A^\top + \boldsymbol{A}_\xi \boldsymbol{K}_{A,\xi}^\top \right) / \sqrt{d}}{\sqrt{\|\boldsymbol{K}_A\|_F^2 + \|\boldsymbol{K}_{A,\xi}\|_F^2}} \right) \right\}.$$

Combine these formulas together and we complete the proof. □

## A.2 THE INFINITE-WIDTH CASE

Now we consider the noiseless infinite-width dynamics. The results of this subsection will not be used in the proof. It mainly serves as a way to give intuitions on how the dynamics looks. As we have discussed in the main text, in this noiseless infinite-width case, it suffices to track $\kappa_p^2 = \|[\boldsymbol{K}_A]_p\|^2$ and $\hat{\kappa}_p^2 = \langle [\boldsymbol{K}_A]_p, [\boldsymbol{K}_B]_p \rangle$.

**Lemma A.7.** *In the noiseless infinite-width case, we have*

$$\frac{\mathrm{d}}{\mathrm{d}t} \kappa_p^2 = 4 \left( 1 - \tilde{S} \right) \left( \frac{\hat{\kappa}_p^2}{\|\boldsymbol{\kappa}\|^2} - \frac{\|\hat{\boldsymbol{\kappa}}\|^2}{\|\boldsymbol{\kappa}\|^2} \frac{\kappa_p^2}{\|\boldsymbol{\kappa}\|^2} \right) \sigma_p^2 - 4 \left( 1 - \tilde{S} \right) \left( \frac{\hat{\kappa}_p^2}{\|\boldsymbol{\kappa}\|^2} T_p - \frac{\kappa_p^2}{\|\boldsymbol{\kappa}\|^2} \tilde{T} \right) \sigma_p^2,$$

$$\frac{\mathrm{d}}{\mathrm{d}t} \hat{\kappa}_p^2 = 4 \left( 1 - \tilde{S} \right) \left( \frac{\kappa_p^2}{\|\boldsymbol{\kappa}\|^2} - \frac{\|\hat{\boldsymbol{\kappa}}\|^2}{\|\boldsymbol{\kappa}\|^2} \frac{\hat{\kappa}_p^2}{\|\boldsymbol{\kappa}\|^2} \right) \sigma_p^2 - 4 \left( 1 - \tilde{S} \right) \left( \frac{\kappa_p^2}{\|\boldsymbol{\kappa}\|^2} T_p - \frac{\hat{\kappa}_p^2}{\|\boldsymbol{\kappa}\|^2} \tilde{T} \right) \sigma_p^2,$$

*Proof.* First, note hat

$$\langle \boldsymbol{f}_A^+, \boldsymbol{f}_B^- \rangle = \frac{\langle \boldsymbol{K}_{BA} \boldsymbol{z}^+, \boldsymbol{z}^- \rangle}{\|\boldsymbol{\kappa}\|^2 / d} = \frac{\langle \boldsymbol{z}^+, \boldsymbol{z}^- \rangle_{\hat{\boldsymbol{\kappa}}^2}}{\|\boldsymbol{\kappa}\|^2 / d}.$$

This implies that (a) $\langle \boldsymbol{f}_A^+, \boldsymbol{f}_B^+ \rangle$ does not depend on the actual value of $\boldsymbol{z}^+$, and (b) if we flip the signs of $z_p^\pm$ simultaneously, then the value of $\langle \boldsymbol{f}_A^+, \boldsymbol{f}_B^- \rangle$ remain unchanged. To compute $S_A$, we then need to take expectation over $\boldsymbol{z}^-$. We compute

$$\mathbb{E}_{\boldsymbol{z}^-} \exp\left(\tau_t^2 \boldsymbol{f}_A^+ \cdot \boldsymbol{f}_B^-\right) = \prod_{k=1}^r \mathbb{E}_{z_k^-} \exp\left(\tau_t^2 \frac{\hat{\kappa}_k^2 z_k^+ z_k^-}{\|\boldsymbol{\kappa}\|^2 / d}\right) = \prod_{k=1}^r \cosh\left(\frac{\tau_t^2 \hat{\kappa}_k^2}{\|\boldsymbol{\kappa}\|^2}\right) =: Z_c.$$

Again, it does not depend on the actual value of $\boldsymbol{z}^+$. One can conduct similar calculation for $S_B$ and, consequently, we have $S_A \equiv S_B \equiv \tilde{S}$ for some $\tilde{S}$ that depends on $\boldsymbol{\kappa}$ and $\hat{\boldsymbol{\kappa}}$ but not on $\boldsymbol{z}^+$. Then, we can rewrite (5) as

$$\begin{aligned}
\frac{\mathrm{d}}{\mathrm{d}t} \boldsymbol{K}_A &= 2\left(1 - \tilde{S}\right) \mathbb{E}_{\boldsymbol{x}_A^+, \boldsymbol{x}_B^+} \left\{ \frac{\boldsymbol{f}_B^+(\boldsymbol{z}^+)^\top}{N_A} - \langle \boldsymbol{f}_A^+, \boldsymbol{f}_B^+ \rangle \frac{\boldsymbol{K}_A}{N_A^2 d} \right\} \mathrm{diag}(\boldsymbol{\sigma}^2) \\
&\quad - 2K\tilde{S} \mathbb{E}_{\boldsymbol{x}_A^+, \boldsymbol{x}_B^\pm} \left\{ \frac{\exp(\tau_t^2 \boldsymbol{f}_A^+ \cdot \boldsymbol{f}_B^-)}{\exp(\tau_t^2 \boldsymbol{f}_A^+ \cdot \boldsymbol{f}_B^+)} \left( \frac{\boldsymbol{f}_B^-(\boldsymbol{z}^+)^\top}{N_A} - \langle \boldsymbol{f}_A^+, \boldsymbol{f}_B^- \rangle \frac{\boldsymbol{K}_A}{N_A^2 d} \right) \right\} \mathrm{diag}(\boldsymbol{\sigma}^2) \\
&= 2\left(1 - \tilde{S}\right) \left( \frac{\boldsymbol{K}_B}{\|\boldsymbol{\kappa}\|^2} - \frac{\|\hat{\boldsymbol{\kappa}}\|^2}{\|\boldsymbol{\kappa}\|^2} \frac{\boldsymbol{K}_A}{\|\boldsymbol{\kappa}\|^2} \right) \mathrm{diag}(\boldsymbol{\sigma}^2) \\
&\quad - \frac{2K\tilde{S}}{\exp\left(\frac{\tau_t^2 \|\hat{\boldsymbol{\kappa}}\|^2}{\|\boldsymbol{\kappa}\|^2}\right)} \mathbb{E}_{\boldsymbol{x}_A^+, \boldsymbol{x}_B^\pm} \left\{ \exp\left(\frac{\tau_t^2 \langle \boldsymbol{z}^+, \boldsymbol{z}^- \rangle_{\hat{\boldsymbol{\kappa}}^2} d}{\|\boldsymbol{\kappa}\|^2}\right) \left( \frac{\boldsymbol{K}_B \boldsymbol{z}^-(\boldsymbol{z}^+)^\top}{\|\boldsymbol{\kappa}\|^2 / d} - \frac{\langle \boldsymbol{z}^+, \boldsymbol{z}^- \rangle_{\hat{\boldsymbol{\kappa}}^2}}{\|\boldsymbol{\kappa}\|^2 / d} \frac{\boldsymbol{K}_A}{\|\boldsymbol{\kappa}\|^2} \right) \right\} \mathrm{diag}(\boldsymbol{\sigma}^2).
\end{aligned}$$

Note that

$$\begin{aligned}
&\mathbb{E}_{\boldsymbol{z}^\pm} \left\{ \exp\left(\frac{\tau_t^2 \langle \boldsymbol{z}^+, \boldsymbol{z}^- \rangle_{\hat{\boldsymbol{\kappa}}^2} d}{\|\boldsymbol{\kappa}\|^2}\right) z_p^- z_q^+ d \right\} \\
&= \mathbb{1}\{p = q\} \mathbb{E}_{z_p^\pm} \left\{ \exp\left(\frac{\tau_t^2 \hat{\kappa}_k^2 z_k^+ z_k^- d}{\|\boldsymbol{\kappa}\|^2}\right) z_p^- z_p^+ d \right\} \prod_{k \neq p} \mathbb{E}_{z_k^\pm} \left\{ \exp\left(\frac{\tau_t^2 \hat{\kappa}_k^2 z_k^+ z_k^- d}{\|\boldsymbol{\kappa}\|^2}\right) \right\} \\
&= \mathbb{1}\{p = q\} \sinh\left(\frac{\tau_t^2 \hat{\kappa}_k^2}{\|\boldsymbol{\kappa}\|^2}\right) \prod_{k \neq p} \cosh\left(\frac{\tau_t^2 \hat{\kappa}_k^2}{\|\boldsymbol{\kappa}\|^2}\right) \\
&= \mathbb{1}\{p = q\} Z_c T_p.
\end{aligned}$$

Meanwhile, note that

$$\begin{aligned}
\mathbb{E}_{\boldsymbol{x}_A^+, \boldsymbol{x}_B^\pm} \left\{ \exp\left(\frac{\tau_t^2 \langle \boldsymbol{z}^+, \boldsymbol{z}^- \rangle_{\hat{\boldsymbol{\kappa}}^2} d}{\|\boldsymbol{\kappa}\|^2}\right) \frac{\langle \boldsymbol{z}^+, \boldsymbol{z}^- \rangle_{\hat{\boldsymbol{\kappa}}^2}}{\|\boldsymbol{\kappa}\|^2 / d} \right\} &= \sum_{k=1}^r \frac{\hat{\kappa}_k^2}{\|\boldsymbol{\kappa}\|^2 / d} \mathbb{E}_{\boldsymbol{z}^\pm} \left\{ \exp\left(\frac{\tau_t^2 \langle \boldsymbol{z}^+, \boldsymbol{z}^- \rangle_{\hat{\boldsymbol{\kappa}}^2} d}{\|\boldsymbol{\kappa}\|^2}\right) z_k^+ z_k^- \right\} \\
&= Z_c \sum_{k=1}^r \frac{\hat{\kappa}_k^2}{\|\boldsymbol{\kappa}\|^2} T_k \\
&=: Z_c \tilde{T}.
\end{aligned}$$

Thus,

$$
\frac{\mathrm{d}}{\mathrm{d}t}\boldsymbol{K_A} = 2\left(1 - \tilde{S}\right)\left(\frac{\boldsymbol{K_B}}{\|\boldsymbol{\kappa}\|^2} - \frac{\|\hat{\boldsymbol{\kappa}}\|^2}{\|\boldsymbol{\kappa}\|^2}\frac{\boldsymbol{K_A}}{\|\boldsymbol{\kappa}\|^2}\right)\mathrm{diag}(\boldsymbol{\sigma}^2)
$$

$$
- \frac{2K\tilde{S}Z_c}{\exp\left(\frac{\tau_t^2\|\hat{\boldsymbol{\kappa}}\|^2}{\|\boldsymbol{\kappa}\|^2}\right)}\left(\frac{\boldsymbol{K_B}}{\|\boldsymbol{\kappa}\|^2}\mathrm{diag}\left([T_k]_{k\in[r]}\right) - \frac{\boldsymbol{K_A}}{\|\boldsymbol{\kappa}\|^2}\tilde{T}\right)\mathrm{diag}(\boldsymbol{\sigma}^2)
$$

$$
= 2\left(1 - \tilde{S}\right)\left(\frac{\boldsymbol{K_B}}{\|\boldsymbol{\kappa}\|^2} - \frac{\|\hat{\boldsymbol{\kappa}}\|^2}{\|\boldsymbol{\kappa}\|^2}\frac{\boldsymbol{K_A}}{\|\boldsymbol{\kappa}\|^2}\right)\mathrm{diag}(\boldsymbol{\sigma}^2)
$$

$$
- 2\left(1 - \tilde{S}\right)\left(\frac{\boldsymbol{K_B}}{\|\boldsymbol{\kappa}\|^2}\mathrm{diag}\left([T_k]_{k\in[r]}\right) - \frac{\boldsymbol{K_A}}{\|\boldsymbol{\kappa}\|^2}\tilde{T}\right)\mathrm{diag}(\boldsymbol{\sigma}^2).
$$

As a corollary, we have

$$
\frac{\mathrm{d}}{\mathrm{d}t}[\boldsymbol{K_A}]_p = 2\left(1 - \tilde{S}\right)\left(\frac{[\boldsymbol{K_B}]_p}{\|\boldsymbol{\kappa}\|^2} - \frac{\|\hat{\boldsymbol{\kappa}}\|^2}{\|\boldsymbol{\kappa}\|^2}\frac{[\boldsymbol{K_A}]_p}{\|\boldsymbol{\kappa}\|^2}\right)\sigma_p^2 - 2\left(1 - \tilde{S}\right)\left(\frac{[\boldsymbol{K_B}]_p}{\|\boldsymbol{\kappa}\|^2}T_p - \frac{[\boldsymbol{K_A}]_p}{\|\boldsymbol{\kappa}\|^2}\tilde{T}\right)\sigma_p^2.
$$

Hence,

$$
\frac{\mathrm{d}}{\mathrm{d}t}\kappa_p^2 = 2\left\langle[\boldsymbol{K_A}]_p, \frac{\mathrm{d}}{\mathrm{d}t}[\boldsymbol{K_A}]_p\right\rangle
$$

$$
= 4\left(1 - \tilde{S}\right)\left(\frac{\hat{\kappa}_p^2}{\|\boldsymbol{\kappa}\|^2} - \frac{\|\hat{\boldsymbol{\kappa}}\|^2}{\|\boldsymbol{\kappa}\|^2}\frac{\kappa_p^2}{\|\boldsymbol{\kappa}\|^2}\right)\sigma_p^2 - 2\left(1 - \tilde{S}\right)\left(\frac{\hat{\kappa}_p^2}{\|\boldsymbol{\kappa}\|^2}T_p - \frac{\kappa_p^2}{\|\boldsymbol{\kappa}\|^2}\tilde{T}\right)\sigma_p^2.
$$

By symmetry, for $\boldsymbol{K_B}$, we have

$$
\frac{\mathrm{d}}{\mathrm{d}t}[\boldsymbol{K_B}]_p = 2\left(1 - \tilde{S}\right)\left(\frac{[\boldsymbol{K_A}]_p}{\|\boldsymbol{\kappa}\|^2} - \frac{\|\hat{\boldsymbol{\kappa}}\|^2}{\|\boldsymbol{\kappa}\|^2}\frac{[\boldsymbol{K_B}]_p}{\|\boldsymbol{\kappa}\|^2}\right)\sigma_p^2 - 2\left(1 - \tilde{S}\right)\left(\frac{[\boldsymbol{K_A}]_p}{\|\boldsymbol{\kappa}\|^2}T_p - \frac{[\boldsymbol{K_B}]_p}{\|\boldsymbol{\kappa}\|^2}\tilde{T}\right)\sigma_p^2.
$$

Then, we can compute

$$
\frac{\mathrm{d}}{\mathrm{d}t}\hat{\kappa}_p^2 = 4\left(1 - \tilde{S}\right)\left(\frac{\kappa_p^2}{\|\boldsymbol{\kappa}\|^2} - \frac{\|\hat{\boldsymbol{\kappa}}\|^2}{\|\boldsymbol{\kappa}\|^2}\frac{\hat{\kappa}_p^2}{\|\boldsymbol{\kappa}\|^2}\right)\sigma_p^2 - 4\left(1 - \tilde{S}\right)\left(\frac{\kappa_p^2}{\|\boldsymbol{\kappa}\|^2}T_p - \frac{\hat{\kappa}_p^2}{\|\boldsymbol{\kappa}\|^2}\tilde{T}\right)\sigma_p^2.
$$

$\square$

## B  STAGE 1

In this section, we show that the following hold:

(a) $\boldsymbol{K_A} \approx \boldsymbol{K_B}$ after Stage 1.

(b) The noise-signal ratio is small after Stage 1.

(c) The condition number is $O(\sqrt{d})$ in Stage 1.

(d) The trajectory is still close to the infinite-width one in Stage 1.

We formalize the main results of Stage 1 bellow.

**Lemma B.1** (Stage 1). *Under the assumption of Theorem 4.3. We can choose a sufficiently (polynomially) large $m$ and a sufficiently (inverse polynomially) small $\tau_t^2$ that may depend on the $\delta$'s that appear in this lemma so that the following statement holds.*

*Let $T_1$ be the earliest time all the following hold:*

$$
\left\langle\overline{[\boldsymbol{K_A}]_p}, \overline{[\boldsymbol{K_B}]_p}\right\rangle \geq 1 - \delta_-, \qquad\qquad\qquad \forall p \in [r],
$$

$$
\frac{\|[\boldsymbol{K_{A,\xi}}]_q\|}{\|[\boldsymbol{K_A}]_p\|} \leq \delta_{N/S}, \qquad\qquad \forall p \in [r], q \in [d - r],
$$

*where $\delta_-, \delta_{N/S} \in 1/\operatorname{poly}(d)$ are two given parameters. We have $T_1 \leq \operatorname{poly}(d)$. Moreover, at any time $t \in [0, T_1]$, we have $\kappa_0 := \max_{p,q \in [r]} \|[K_A]_p\| / \|[K_A]_q\| \leq O(\sqrt{d})$ and*

$$\|[K_A]_p\|^2 = (1 \pm \delta_{A/B}) \|[K_B]_p\|^2, \quad \|[K_{A,\xi}]_q\|^2 = (1 \pm \delta_{A/B}) \|[K_{B,\xi}]_q\|^2, \qquad \forall p \in [r], q \in [d-r],$$

$$\left| 1 - \frac{\|[K_{A,\xi}]_p\|}{\|[K_{A,\xi}]_q\|} \right| \leq \delta_{\xi,\kappa_0}, \qquad \forall p, q \in [d-r],$$

$$\max \left\{ \left| \left\langle \overline{[K_C]_p}, \overline{[K_D]_q} \right\rangle \right| : C, D \in \{A, B\} \right\} \leq \delta_{AB,\perp}, \qquad \forall p \neq q \in [r],$$

$$\max \left\{ \left| \left\langle \overline{[K_C]_p}, \overline{[K_{D,\xi}]_q} \right\rangle \right|, \left| \left\langle \overline{[K_{A,\xi}]_s}, \overline{[K_{B,\xi}]_q} \right\rangle \right| : C, D \in \{A, B\} \right\} \leq \delta_{\xi,\perp}, \quad \forall p \in [r], q, s \in [d-r],$$

$$(10)$$

*where $\delta_{A/B}, \delta_{\xi,\kappa_0}, \delta_{AB,\perp}, \delta_{\xi,\perp} \in 1/\operatorname{poly}(d)$ are given parameters.*

Basically, the conditions in (10) mean that the norm of the corresponding columns are roughly the same, and the columns of all these matrices are approximately orthogonal to each other. Both of them are true in the infinite-width limit, and by some standard concentration argument, one can make all these errors to be arbitrarily inverse-polynomially small, at least at initialization. Note that, as a simple corollary of (10), we have

$$N_A = N_B \left( 1 \pm \sqrt{\delta_{A/B}} \right).$$

For notational simplicity, we also define

$$\rho_- = \max_{p \in [r]} \left\{ 1 - \left\langle \overline{[K_A]_p}, \overline{[K_B]_p} \right\rangle \right\}, \quad \rho_{N/S} = \max_{\substack{p \in [r] \\ q \in [d-r]}} \frac{\|[K_{A,\xi}]_q\|}{\|[K_A]_p\|}.$$

Characterizing the dynamics in Stage 1 is relatively straightforward. We will see in Section B.1 that all those $Q$-matrices have simple forms. The main tool we use to control the condition number and the discretization error is the following nonlinear version of Gronwall's lemma.

**Lemma B.2.** *Let $A_t$ be a positive process. Let $X_t$ and $Y_t$ be defined as*

$$\dot{X}_t \leq -A_t X_t, \quad \dot{Y}_t \leq \alpha A_t X_t Y_t,$$

*with $X_0, Y_0, \alpha$ being positive. Then, for any $T \geq 0$, we have $Y_T \leq Y_0 \exp(\alpha X_0)$.*

**Remark**. Here, $X_t$ represent the progress we have made and $Y_t$ the error. In our case, $X_t$ is the maximum between $1 - \left\langle \overline{[K_A]_p}, \overline{[K_B]_p} \right\rangle$ and the noise-signal ratio, and $Y_t$ the discretization error. This lemma says that, if the error growth rate depends on the progress, then by coupling these two processes, we can make sure the error does not blow up. The point of this lemma is that, with coupling, we do not need a very tight estimation on the convergence time nor the error growth rate. ♣

*Proof.* The solution of this ODE system is given by

$$X_T = X_0 \exp \left( - \int_0^T A_t \, dt \right), \quad Y_T = Y_0 \exp \left( \alpha X_0 \int_0^T A_t \exp \left( - \int_0^t A_s \, ds \right) dt \right).$$

Note that

$$\int_0^T A_t \exp \left( - \int_0^t A_s \, ds \right) dt = - \int_0^T d \exp \left( - \int_0^t A_s \, ds \right) = 1 - \exp \left( - \int_0^T A_t \, dt \right).$$

Hence,

$$Y_T = Y_0 \exp \left( \alpha X_0 \left( 1 - \exp \left( - \int_0^T A_t \, dt \right) \right) \right) \leq Y_0 \exp \left( \alpha X_0 \right).$$

$\square$

The organization of this section is as follows. In Section B.1, we derive estimations for the $\boldsymbol{Q}$-matrices defined in Corollary A.4 and use them to simplify the equations governing the training dynamics. In Section B.2, we estimate the rate at which $1 - \left\langle \overline{[\boldsymbol{K_A}]_p}, \overline{[\boldsymbol{K_B}]_p} \right\rangle$ and the noise-signal ratio converge to 0 and the growth rate of the condition number. In Section B.3, we estimate the growth rate of the discretization error. Then, in Section B.4, we prove Lemma B.1, the main lemma of Stage 1. Finally, we prove the negative result for non-contrastive learning in Section B.5.

### B.1 ESTIMATIONS FOR $Q$ AND THE DYNAMICS

Thanks to Lemma A.5, in order to analyze the dynamics, it suffices to estimate the $\boldsymbol{Q}$-matrices. In this subsection, we derive estimations for them and use these estimations to simplify the equations in Lemma A.5. Recall that, in Stage 1, $\tau_t^2$ is small. Hence, $\exp(\tau_t^2 \langle \boldsymbol{f_A^+}, \boldsymbol{f_B^-} \rangle) = 1 \pm O(\tau_t^2)$. With this approximation, one can derive the following estimation for $S_A$ and $S_B$.

**Lemma B.3** (Estimations for $S$). *In Stage 1, we have*

$$S_{\boldsymbol{A}} = \frac{1}{1+K} \pm O_z\left(\tau_t^2\right) \pm O\left(\tau_t^2 d \delta_{\xi, \perp} \rho_{N/S}\right),$$

*and the same is also true for $S_B$. Here, $O_z(\tau_t^2)$ means $O(\tau_t^2)$ and this does not depend on $\boldsymbol{\xi_A}$ nor $\boldsymbol{\xi_B}$.*

The proof of this lemma and all following lemma is deferred to the end of this subsection. Note that we derive a slightly finer estimation for the noise-related part. This additional $\rho_{N/S}$ will be used cancel with terms like $\|[\boldsymbol{K_A}]_p\| / \|[\boldsymbol{K_{A,\xi}}]_q\|$, at the cost of a $\kappa_0$ factor, in later analysis. We emphasize here that $O_z$ does not depend on the noises so that later we can argue $\mathbb{E}_{\boldsymbol{\xi}} \left\{ O_z(\tau_t^2) \boldsymbol{\xi} \right\} = 0$. With this lemma, we now derive estimations for $\boldsymbol{Q}_1$, $\boldsymbol{Q}_{1,\boldsymbol{\xi}}$, $\boldsymbol{Q}_2$ and $Q_0$, respectively.

**Lemma B.4** (Estimations for $\boldsymbol{Q}_1$). *In Stage 1, we have*

$$\boldsymbol{Q}_1 = \frac{2K}{1+K} \boldsymbol{I}_d \pm O\left(d\tau_t^2\right).$$

**Lemma B.5** (Estimations for $\boldsymbol{Q}_{1,\boldsymbol{\xi}}$ and $\boldsymbol{Q}_2$). *In Stage 1, we have*

$$\max\left\{ \|\boldsymbol{Q}_{1,\xi_A}\|_F, \|\boldsymbol{Q}_{1,\xi_B}\|_F, \|\boldsymbol{Q}_2\|_F \right\} = \pm O\left(\tau_t^2 d^2 \rho_{N/S} \delta_{\xi,\perp}\right).$$

The above two lemmas say that in Stage 1, $\boldsymbol{Q}_1$ is approximately diagonal, and $\boldsymbol{Q}_{1,\boldsymbol{\xi}}$ and $\boldsymbol{Q}_2$ can more or less be ignored.

**Lemma B.6** (Estimations for $Q_0$). *In Stage 1, we have*

$$Q_0 = -\frac{2K}{1+K} \frac{\langle \boldsymbol{K_A}, \boldsymbol{K_B} \rangle}{N_{\boldsymbol{A}} N_{\boldsymbol{B}} d} \pm O\left(d\tau_t^2\right).$$

The proof of Lemma B.6 is essentially the same as the proof of Lemma B.4 so we omit it. With these three lemmas, we can now simplify Lemma A.5 as follows.

**Corollary B.7.** *In Stage 1, for any $p \in [r]$ and $q \in [d-r]$, we have*

$$\frac{\mathrm{d}}{\mathrm{d}t} \|[\boldsymbol{K_A}]_p\|^2 = \frac{4K}{1+K} \frac{\sigma_p^2}{N_{\boldsymbol{A}} N_{\boldsymbol{B}} d} \left( \frac{\langle [\boldsymbol{K_A}]_p, [\boldsymbol{K_B}]_p \rangle}{\|[\boldsymbol{K_A}]_p\|^2} - \frac{\langle \boldsymbol{K_A}, \boldsymbol{K_B} \rangle}{N_{\boldsymbol{A}} N_{\boldsymbol{B}} d} \right) \|[\boldsymbol{K_A}]_p\|^2$$

$$\pm O\left( \frac{\sigma_p^2}{N_{\boldsymbol{A}} N_{\boldsymbol{B}} d} \left( \sqrt{\delta_{\boldsymbol{A/B}}} + \tau_t^2 d \right) \kappa_p^2 \right),$$

$$\frac{\mathrm{d}}{\mathrm{d}t} \|[\boldsymbol{K_{A,\xi}}]_q\|^2 = -\frac{4K}{1+K} \frac{\sigma_\xi^2}{N_{\boldsymbol{A}} N_{\boldsymbol{B}} d} \frac{\langle \boldsymbol{K_A}, \boldsymbol{K_B} \rangle}{N_{\boldsymbol{A}} N_{\boldsymbol{B}} d} \|[\boldsymbol{K_{A,\xi}}]_q\|^2 \pm O\left( \frac{\sigma_\xi^2}{N_{\boldsymbol{A}} N_{\boldsymbol{B}} d} \left( d\tau_t^2 + \sqrt{\delta_{\boldsymbol{A/B}}} \right) \|[\boldsymbol{K_{A,\xi}}]_q\|^2 \right),$$

*The formulas for $\boldsymbol{K_B}$ and $\boldsymbol{K_{B,\xi}}$ can be obtained by interchanging the roles of $\boldsymbol{A}$ and $\boldsymbol{B}$.*

Basically, the two parts of the first term of $\frac{d}{dt}\left\|[K_A]_p\right\|^2$ correspond to the signal and normalization, respectively. For $\frac{d}{dt}\left\|[K_{A,\xi}]_q\right\|^2$, the signal term is $0$ because the noises in $x_A^+$ and $x_B^+$ are independent and have mean $0$.

**Corollary B.8.** *In Stage 1, we have*

$$\frac{d}{dt}\overline{[K_A]}_p = \frac{2K}{1+K}\frac{\sigma_p^2}{N_A N_B d}\left(I - \overline{[K_A]}_p\left(\overline{[K_A]}_p\right)^\top\right)\overline{[K_B]}_p \pm O\left(\frac{\sigma_p^2}{N_A N_B d}\left(\tau_t^2 d^2 \kappa_0 + d\sqrt{\delta_{A/B}}\right)\right),$$

$$\frac{d}{dt}\overline{[K_{A,\xi}]}_q = \pm O\left(\frac{\sigma_\xi^2}{N_A N_B d}\tau_t^2 d^3 \kappa_0 \delta_{\xi,\perp}\right).$$

*Interchange the roles of $A$ and $B$ and one can obtain the formulas for $K_B$ and $K_{B,\xi}$.*

Note that, after normalization, the normalization terms cancel with each other, and the signal terms drive $\overline{[K_A]}_p$ and $\overline{[K_A]}_q$ toward each other. Moreover, without the normalization terms, different $\overline{[K_A]}_p$ are approximately independent of each other. This allows us to maintain the orthogonality between different columns.

Note that the above results also imply the following lemma.

**Lemma B.9.** *The dynamics of the non-contrastive method and the Stage 1 dynamics are equivalent, up to a multiplicative constant and some higher-order terms.*

OMITTED PROOFS OF THIS SUBSECTION

*Proof of Lemma B.3.* First, we write

$$\left\langle f_A^+, f_B^-\right\rangle = \frac{\left\langle K_A z^+, K_B z^-\right\rangle}{N_A N_B} + \frac{\left\langle K_A z^+, K_{B,\xi}\xi_B^-\right\rangle + \left\langle K_{A,\xi}\xi_A^+, K_B z^-\right\rangle}{N_A N_B} + \frac{\left\langle K_{A,\xi}\xi_A^+, K_{B,\xi}\xi_B^-\right\rangle}{N_A N_B}.$$

For the second term, we write

$$\frac{\left\langle K_A z^+, K_{B,\xi}\xi_B^-\right\rangle}{N_A N_B} = \sum_{i\in[r], j\in[d-r]} \frac{\|[K_A]_i\|^2}{N_A N_B d} \frac{\|[K_{B,\xi}]_j\|}{\|[K_A]_i\|}\left\langle\overline{[K_A]}_i, \overline{[K_{B,\xi}]}_j\right\rangle z_i^+ \xi_{B,j}^- d.$$

For each summand, we have $\|[K_{B,\xi}]_j\|/\|[K_A]_i\| \le O(\rho_{N/S})$, $\left\langle\overline{[K_A]}_i, \overline{[K_{B,\xi}]}_j\right\rangle = \pm O(\delta_{\xi,\perp})$, and $|z_i^+ \xi_{B,j}^- d| \le 1$. Hence, $\frac{\left\langle K_A z^+, K_{B,\xi}\xi_B^-\right\rangle}{N_A N_B} = \pm O(d\delta_{\xi,\perp}\rho_{N/S})$. The same is also true for $\frac{\left\langle K_{A,\xi}\xi_A^+, K_B z^-\right\rangle}{N_A N_B d}$ and the third term. Therefore,

$$\left\langle f_A^+, f_B^-\right\rangle = \frac{\left\langle K_A z^+, K_B z^-\right\rangle}{N_A N_B} \pm O\left(d\delta_{\xi,\perp}\rho_{N/S}\right).$$

Then, we compute

$$\exp\left(\tau_t^2\left\langle f_A^+, f_B^-\right\rangle\right) = \exp\left(\tau_t^2\frac{\left\langle K_A z^+, K_B z^-\right\rangle}{N_A N_B}\right)\left(1 \pm O\left(\tau_t^2 d\delta_{\xi,\perp}\rho_{N/S}\right)\right)$$
$$= 1 \pm O_z\left(\tau_t^2\right) \pm O\left(\tau_t^2 d\delta_{\xi,\perp}\rho_{N/S}\right).$$

Thus,

$$S_A = \frac{1}{1+K} \pm O_z\left(\tau_t^2\right) \pm O\left(\tau_t^2 d\delta_{\xi,\perp}\rho_{N/S}\right).$$

The proof for $S_B$ is essentially the same. □

*Proof of Lemma B.4.* Recall that

$$Q_1 := \mathbb{E}\left\{\left(2 - S_A(x_A^+, x_B^+) - S_B(x_A^+, x_B^+)\right)z^+(z^+)^\top d\right\}$$
$$- K\mathbb{E}\left\{\frac{S_A(x_A^+, x_B^+)\exp(\tau_t^2 f_A^+ \cdot f_B^-)}{\exp(\tau_t^2 f_A^+ \cdot f_B^+)}z^-(z^+)^\top d\right\}$$
$$- K\mathbb{E}\left\{\frac{S_B(x_A^+, x_B^+)\exp(\tau_t^2 f_A^- \cdot f_B^+)}{\exp(\tau_t^2 f_A^+ \cdot f_B^+)}z^+(z^-)^\top d\right\}.$$

Since $S_{\boldsymbol{A}} = (1 + K)^{-1} \pm O(\tau_t^2)$ and $S_{\boldsymbol{B}} = (1 + K)^{-1} \pm O(\tau_t^2)$, we have

$$2 - S_{\boldsymbol{A}}(\boldsymbol{x}_{\boldsymbol{A}}^+, \boldsymbol{x}_{\boldsymbol{B}}^+) - S_{\boldsymbol{B}}(\boldsymbol{x}_{\boldsymbol{A}}^+, \boldsymbol{x}_{\boldsymbol{B}}^+) = 2 - \frac{2}{1 + K} \pm O(\tau_t^2) = \frac{2K}{1 + K} \pm O(\tau_t^2).$$

As a result,

$$\boldsymbol{Q}_1 = \frac{2K}{1 + K} \mathbb{E}\left\{\boldsymbol{z}^+(\boldsymbol{z}^+)^\top d\right\} - \frac{K}{1 + K} \mathbb{E}\left\{\boldsymbol{z}^-(\boldsymbol{z}^+)^\top d\right\} - \frac{K}{1 + K} \mathbb{E}\left\{\boldsymbol{z}^+(\boldsymbol{z}^-)^\top d\right\} \pm O\left(d\tau_t^2\right).$$

Note that $\mathbb{E}\left\{\boldsymbol{z}^-(\boldsymbol{z}^+)^\top\right\} = 0$ and $\mathbb{E}\left\{\boldsymbol{z}^+(\boldsymbol{z}^+)^\top\right\} = \boldsymbol{I}_d/d$. Therefore,

$$\boldsymbol{Q}_1 = \frac{2K}{1 + K} \boldsymbol{I}_d \pm O\left(d\tau_t^2\right).$$

$\square$

*Proof of Lemma B.5.* Recall that

$$\boldsymbol{Q}_{1,\xi_{\boldsymbol{A}}} := \mathbb{E}\left\{\left(2 - S_{\boldsymbol{A}}(\boldsymbol{x}_{\boldsymbol{A}}^+, \boldsymbol{x}_{\boldsymbol{B}}^+) - S_{\boldsymbol{B}}(\boldsymbol{x}_{\boldsymbol{A}}^+, \boldsymbol{x}_{\boldsymbol{B}}^+)\right) \boldsymbol{\xi}_{\boldsymbol{A}}^+(\boldsymbol{z}^+)^\top d\right\}$$
$$- K \,\mathbb{E}\left\{\frac{S_{\boldsymbol{A}}(\boldsymbol{x}_{\boldsymbol{A}}^+, \boldsymbol{x}_{\boldsymbol{B}}^+) \exp(\tau_t^2 \boldsymbol{f}_{\boldsymbol{A}}^+ \cdot \boldsymbol{f}_{\boldsymbol{B}}^-)}{\exp(\tau_t^2 \boldsymbol{f}_{\boldsymbol{A}}^+ \cdot \boldsymbol{f}_{\boldsymbol{B}}^+)} \boldsymbol{\xi}_{\boldsymbol{A}}^+(\boldsymbol{z}^-)^\top d\right\}$$
$$- K \,\mathbb{E}\left\{\frac{S_{\boldsymbol{B}}(\boldsymbol{x}_{\boldsymbol{A}}^+, \boldsymbol{x}_{\boldsymbol{B}}^+) \exp(\tau_t^2 \boldsymbol{f}_{\boldsymbol{A}}^- \cdot \boldsymbol{f}_{\boldsymbol{B}}^+)}{\exp(\tau_t^2 \boldsymbol{f}_{\boldsymbol{A}}^+ \cdot \boldsymbol{f}_{\boldsymbol{B}}^+)} \boldsymbol{\xi}_{\boldsymbol{A}}^-(\boldsymbol{z}^+)^\top d\right\}.$$

Note that if some quantity $X$ does not depend on $\boldsymbol{\xi}$, then $\mathbb{E}\{X\boldsymbol{\xi}\} = 0$. Hence, by Lemma B.3, we have

$$\boldsymbol{Q}_{1,\xi_{\boldsymbol{A}}} = \pm O\left(\tau_t^2 d^2 \rho_{N/S} \delta_{\xi,\perp}\right).$$

The proof for $\boldsymbol{Q}_{1,\xi_{\boldsymbol{A}}}$ and $\boldsymbol{Q}_2$ is essentially the same.

$\square$

*Proof of Corollary B.7.* Recall that

$$\frac{\mathrm{d}}{\mathrm{d}t}\|[\boldsymbol{K}_{\boldsymbol{A}}]_p\|^2 = 2\frac{\langle[\boldsymbol{K}_{\boldsymbol{A}}]_p, [\boldsymbol{K}_{\boldsymbol{B}}\boldsymbol{Q}_1]_p\rangle}{N_{\boldsymbol{A}}N_{\boldsymbol{B}}d}\sigma_p^2 + 2\frac{\langle[\boldsymbol{K}_{\boldsymbol{A}}]_p, [\boldsymbol{K}_{\boldsymbol{B},\xi}\boldsymbol{Q}_{1,\xi_{\boldsymbol{B}}}]_p\rangle}{N_{\boldsymbol{A}}N_{\boldsymbol{B}}d}\sigma_p^2 + 2\frac{\|[\boldsymbol{K}_{\boldsymbol{A}}]_p\|^2}{N_{\boldsymbol{A}}^2 d}Q_0\sigma_p^2$$
$$= \sum_{i=1}^3 \mathrm{T}_i\left(\frac{\mathrm{d}}{\mathrm{d}t}\|[\boldsymbol{K}_{\boldsymbol{A}}]_p\|^2\right).$$

We now estimate these terms one-by-one. For $\mathrm{T}_1$, we have

$$\langle[\boldsymbol{K}_{\boldsymbol{A}}]_p, [\boldsymbol{K}_{\boldsymbol{B}}\boldsymbol{Q}_1]_p\rangle = \langle[\boldsymbol{K}_{\boldsymbol{A}}]_p, [\boldsymbol{K}_{\boldsymbol{B}}]_p\rangle [\boldsymbol{Q}_1]_{p,p} + \sum_{k \neq p} \langle[\boldsymbol{K}_{\boldsymbol{A}}]_p, [\boldsymbol{K}_{\boldsymbol{B}}]_k\rangle [\boldsymbol{Q}_1]_{k,p}$$
$$= \frac{2K}{1 + K} \langle[\boldsymbol{K}_{\boldsymbol{A}}]_p, [\boldsymbol{K}_{\boldsymbol{B}}]_p\rangle \pm O\left(\tau_t^2 d\kappa_p^2\right) \pm O\left(\tau_t^2 d\kappa_p^2 \kappa_0 \delta_{\boldsymbol{AB},\perp}\right)$$
$$= \frac{2K}{1 + K} \langle[\boldsymbol{K}_{\boldsymbol{A}}]_p, [\boldsymbol{K}_{\boldsymbol{B}}]_p\rangle \pm O\left(\tau_t^2 d\kappa_p^2\right).$$

Hence,

$$\mathrm{T}_1\left(\frac{\mathrm{d}}{\mathrm{d}t}\|[\boldsymbol{K}_{\boldsymbol{A}}]_p\|^2\right) = \frac{4K}{1 + K}\frac{\sigma_p^2}{N_{\boldsymbol{A}}N_{\boldsymbol{B}}d}\langle[\boldsymbol{K}_{\boldsymbol{A}}]_p, [\boldsymbol{K}_{\boldsymbol{B}}]_p\rangle \pm O\left(\frac{\sigma_p^2}{N_{\boldsymbol{A}}N_{\boldsymbol{B}}d}\tau_t^2 d\kappa_p^2\right).$$

For $\mathrm{T}_2$, we compute

$$\mathrm{T}_2\left(\frac{\mathrm{d}}{\mathrm{d}t}\|[\boldsymbol{K}_{\boldsymbol{A}}]_p\|^2\right) = \frac{2\sigma_p^2}{N_{\boldsymbol{A}}N_{\boldsymbol{B}}d}\sum_{k=1}^{d-r} \langle[\boldsymbol{K}_{\boldsymbol{A}}]_p, [\boldsymbol{K}_{\boldsymbol{B},\xi}]_k\rangle [\boldsymbol{Q}_{1,\xi_{\boldsymbol{B}}}]_{k,p}$$
$$= \pm O(1)\frac{\sigma_p^2}{N_{\boldsymbol{A}}N_{\boldsymbol{B}}d}\sum_{k=1}^{d-r} \|[\boldsymbol{K}_{\boldsymbol{A}}]_p\|^2 \frac{\|[\boldsymbol{K}_{\boldsymbol{B},\xi}]_k\|}{\|[\boldsymbol{K}_{\boldsymbol{A}}]_p\|}\left\langle\overline{[\boldsymbol{K}_{\boldsymbol{A}}]_p}, \overline{[\boldsymbol{K}_{\boldsymbol{B},\xi}]_k}\right\rangle [\boldsymbol{Q}_{1,\xi_{\boldsymbol{B}}}]_{k,p}$$
$$= \pm O\left(\frac{\sigma_p^2}{N_{\boldsymbol{A}}N_{\boldsymbol{B}}d}\tau_t^2 d^3 \rho_{N/S}^2 \delta_{\xi,\perp}^2 \kappa_p^2\right).$$

Finally, for $T_3$, we compute

$$
\begin{aligned}
T_3\left(\frac{\mathrm{d}}{\mathrm{d}t}\|[\boldsymbol{K_A}]_p\|^2\right) &= 2\frac{\|[\boldsymbol{K_A}]_p\|^2}{N_{\boldsymbol{A}}^2 d}\left(-\frac{2K}{1+K}\frac{\langle\boldsymbol{K_A},\boldsymbol{K_B}\rangle}{N_{\boldsymbol{A}}N_{\boldsymbol{B}}d}\pm O\left(d\tau_t^2\right)\right)\sigma_p^2 \\
&= -\frac{4K}{1+K}\frac{\sigma_p^2}{N_{\boldsymbol{A}}N_{\boldsymbol{B}}d}\left(1\pm\sqrt{\delta_{\boldsymbol{A/B}}}\right)\frac{\langle\boldsymbol{K_A},\boldsymbol{K_B}\rangle}{N_{\boldsymbol{A}}N_{\boldsymbol{B}}d}\|[\boldsymbol{K_A}]_p\|^2\pm O\left(\frac{\kappa_p^2}{N_{\boldsymbol{A}}N_{\boldsymbol{B}}d}\tau_t^2 d\sigma_p^2\right) \\
&= -\frac{4K}{1+K}\frac{\sigma_p^2}{N_{\boldsymbol{A}}N_{\boldsymbol{B}}d}\frac{\langle\boldsymbol{K_A},\boldsymbol{K_B}\rangle}{N_{\boldsymbol{A}}N_{\boldsymbol{B}}d}\pm O\left(\frac{\sigma_p^2}{N_{\boldsymbol{A}}N_{\boldsymbol{B}}d}\left(\sqrt{\delta_{\boldsymbol{A/B}}}+\tau_t^2 d\right)\kappa_p^2\right).
\end{aligned}
$$

Combine these together and we obtain

$$
\begin{aligned}
\frac{\mathrm{d}}{\mathrm{d}t}\|[\boldsymbol{K_A}]_p\|^2 &= \frac{4K}{1+K}\frac{\sigma_p^2}{N_{\boldsymbol{A}}N_{\boldsymbol{B}}d}\langle[\boldsymbol{K_A}]_p,[\boldsymbol{K_B}]_p\rangle - \frac{4K}{1+K}\frac{\sigma_p^2}{N_{\boldsymbol{A}}N_{\boldsymbol{B}}d}\frac{\langle\boldsymbol{K_A},\boldsymbol{K_B}\rangle}{N_{\boldsymbol{A}}N_{\boldsymbol{B}}d}\|[\boldsymbol{K_A}]_p\|^2 \\
&\quad \pm O\left(\frac{\sigma_p^2}{N_{\boldsymbol{A}}N_{\boldsymbol{B}}d}\tau_t^2 d^3\rho_{N/S}^2\delta_{\xi,\perp}^2\kappa_p^2\right)\pm O\left(\frac{\sigma_p^2}{N_{\boldsymbol{A}}N_{\boldsymbol{B}}d}\left(\sqrt{\delta_{\boldsymbol{A/B}}}+\tau_t^2 d\right)\kappa_p^2\right) \\
&= \frac{4K}{1+K}\frac{\sigma_p^2}{N_{\boldsymbol{A}}N_{\boldsymbol{B}}d}\left(\frac{\langle[\boldsymbol{K_A}]_p,[\boldsymbol{K_B}]_p\rangle}{\|[\boldsymbol{K_A}]_p\|^2}-\frac{\langle\boldsymbol{K_A},\boldsymbol{K_B}\rangle}{N_{\boldsymbol{A}}N_{\boldsymbol{B}}d}\right)\|[\boldsymbol{K_A}]_p\|^2 \\
&\quad \pm O\left(\frac{\sigma_p^2}{N_{\boldsymbol{A}}N_{\boldsymbol{B}}d}\left(\sqrt{\delta_{\boldsymbol{A/B}}}+\tau_t^2 d\right)\kappa_p^2\right).
\end{aligned}
$$

Interchange the roles of $\boldsymbol{A}$ and $\boldsymbol{B}$ and we obtain the formula for $\frac{\mathrm{d}}{\mathrm{d}t}\|[\boldsymbol{K_B}]_p\|^2$. Now we consider $\boldsymbol{K_{A,\xi}}$. We write

$$
\begin{aligned}
\frac{\mathrm{d}}{\mathrm{d}t}\|[\boldsymbol{K_{A,\xi}}]_q\|^2 &= 2\frac{\left\langle[\boldsymbol{K_{A,\xi}}]_q,[\boldsymbol{K_B}\boldsymbol{Q}_{1,\boldsymbol{\xi_A}}^\top]_q\right\rangle}{N_{\boldsymbol{A}}N_{\boldsymbol{B}}d}\sigma_\xi^2 + 2\frac{\langle[\boldsymbol{K_{A,\xi}}]_q,[\boldsymbol{K_{B,\xi}}\boldsymbol{Q}_2]_q\rangle}{N_{\boldsymbol{A}}N_{\boldsymbol{B}}d}\sigma_\xi^2 + 2\frac{\|[\boldsymbol{K_{A,\xi}}]_q\|_F^2}{N_{\boldsymbol{A}}^2 d}Q_0\sigma_\xi^2 \\
&= \sum_{i=1}^3 T_i\left(\frac{\mathrm{d}}{\mathrm{d}t}\|[\boldsymbol{K_{A,\xi}}]_q\|^2\right).
\end{aligned}
$$

For $T_1$, we compute

$$
\begin{aligned}
T_1\left(\frac{\mathrm{d}}{\mathrm{d}t}\|[\boldsymbol{K_{A,\xi}}]_q\|^2\right) &= 2\sum_{k=1}^r\frac{\langle[\boldsymbol{K_{A,\xi}}]_q,[\boldsymbol{K_B}]_k\rangle[\boldsymbol{Q}_{1,\boldsymbol{\xi_A}}]_{q,k}}{N_{\boldsymbol{A}}N_{\boldsymbol{B}}d}\sigma_\xi^2 \\
&= \frac{2\sigma_\xi^2}{N_{\boldsymbol{A}}N_{\boldsymbol{B}}d}\sum_{k=1}^r\|[\boldsymbol{K_{A,\xi}}]_q\|^2\frac{\|[\boldsymbol{K_B}]_k\|}{\|[\boldsymbol{K_{A,\xi}}]_q\|}\left\langle\overline{[\boldsymbol{K_{A,\xi}}]_q},\overline{[\boldsymbol{K_B}]_k}\right\rangle[\boldsymbol{Q}_{1,\boldsymbol{\xi_A}}]_{q,k} \\
&= \pm O(1)\frac{2\sigma_\xi^2}{N_{\boldsymbol{A}}N_{\boldsymbol{B}}d}\|[\boldsymbol{K_{A,\xi}}]_q\|^2\sum_{k=1}^r\frac{\kappa_0}{\rho_{N/S}}\delta_{\xi,\perp}\tau_t^2 d^2\rho_{N/S}\delta_{\xi,\perp} \\
&= \pm O\left(\frac{\sigma_\xi^2}{N_{\boldsymbol{A}}N_{\boldsymbol{B}}d}\tau_t^2 d^3\kappa_0\delta_{\xi,\perp}^2\|[\boldsymbol{K_{A,\xi}}]_q\|^2\right).
\end{aligned}
$$

Similarly, one can show that this bound also holds for $T_2$. Finally, for $T_3$, by Lemma B.6, we have

$$
\begin{aligned}
T_3\left(\frac{\mathrm{d}}{\mathrm{d}t}\|[\boldsymbol{K_{A,\xi}}]_q\|^2\right) &= 2\frac{\|[\boldsymbol{K_{A,\xi}}]_q\|_F^2}{N_{\boldsymbol{A}}N_{\boldsymbol{B}}d}\left(1\pm\sqrt{\delta_{\boldsymbol{A/B}}}\right)\left(-\frac{2K}{1+K}\frac{\langle\boldsymbol{K_A},\boldsymbol{K_B}\rangle}{N_{\boldsymbol{A}}N_{\boldsymbol{B}}d}\pm O\left(d\tau_t^2\right)\right)\sigma_\xi^2 \\
&= -\frac{4K}{1+K}\frac{\sigma_\xi^2}{N_{\boldsymbol{A}}N_{\boldsymbol{B}}d}\frac{\langle\boldsymbol{K_A},\boldsymbol{K_B}\rangle}{N_{\boldsymbol{A}}N_{\boldsymbol{B}}d}\|[\boldsymbol{K_{A,\xi}}]_q\|_F^2 \\
&\quad \pm O\left(\frac{\sigma_\xi^2}{N_{\boldsymbol{A}}N_{\boldsymbol{B}}d}\left(d\tau_t^2+\sqrt{\delta_{\boldsymbol{A/B}}}\right)\|[\boldsymbol{K_{A,\xi}}]_q\|_F^2\right).
\end{aligned}
$$

Combine these together and we obtain

$$\frac{\mathrm{d}}{\mathrm{d}t}\left\|[\boldsymbol{K}_{\boldsymbol{A},\boldsymbol{\xi}}]_q\right\|^2 = -\frac{4K}{1+K}\frac{\sigma_\xi^2}{N_{\boldsymbol{A}}N_{\boldsymbol{B}}d}\frac{\langle\boldsymbol{K}_{\boldsymbol{A}},\boldsymbol{K}_{\boldsymbol{B}}\rangle}{N_{\boldsymbol{A}}N_{\boldsymbol{B}}d}\left\|[\boldsymbol{K}_{\boldsymbol{A},\boldsymbol{\xi}}]_q\right\|_F^2 \pm O\left(\frac{\sigma_\xi^2}{N_{\boldsymbol{A}}N_{\boldsymbol{B}}d}\left(d\tau_t^2+\sqrt{\delta_{\boldsymbol{A}/\boldsymbol{B}}}\right)\left\|[\boldsymbol{K}_{\boldsymbol{A},\boldsymbol{\xi}}]_q\right\|_F^2\right).$$

Interchange the roles of $\boldsymbol{A}$ and $\boldsymbol{B}$ and we obtain the formula for $\boldsymbol{K}_{\boldsymbol{B},\boldsymbol{\xi}}$. $\qquad\square$

*Proof of Corollary B.8.* We write

$$\frac{\mathrm{d}}{\mathrm{d}t}\overline{[\boldsymbol{K}_{\boldsymbol{A}}]_p} = \left(\boldsymbol{I}-\overline{[\boldsymbol{K}_{\boldsymbol{A}}]_p}\left(\overline{[\boldsymbol{K}_{\boldsymbol{A}}]_p}\right)^\top\right)\frac{[\boldsymbol{K}_{\boldsymbol{B}}\boldsymbol{Q}_1]_p}{\|[\boldsymbol{K}_{\boldsymbol{A}}]_p\|}\frac{\sigma_p^2}{N_{\boldsymbol{A}}N_{\boldsymbol{B}}d}$$
$$+\left(\boldsymbol{I}-\overline{[\boldsymbol{K}_{\boldsymbol{A}}]_p}\left(\overline{[\boldsymbol{K}_{\boldsymbol{A}}]_p}\right)^\top\right)\frac{[\boldsymbol{K}_{\boldsymbol{B},\boldsymbol{\xi}}\boldsymbol{Q}_{1,\xi_B}]_p}{\|[\boldsymbol{K}_{\boldsymbol{A}}]_p\|}\frac{\sigma_p^2}{N_{\boldsymbol{A}}N_{\boldsymbol{B}}d}$$
$$= \mathrm{T}_1\left(\frac{\mathrm{d}}{\mathrm{d}t}\overline{[\boldsymbol{K}_{\boldsymbol{A}}]_p}\right)+\mathrm{T}_2\left(\frac{\mathrm{d}}{\mathrm{d}t}\overline{[\boldsymbol{K}_{\boldsymbol{A}}]_p}\right).$$

For $\mathrm{T}_1$, we have

$$\mathrm{T}_1\left(\frac{\mathrm{d}}{\mathrm{d}t}\overline{[\boldsymbol{K}_{\boldsymbol{A}}]_p}\right) = \sum_{k=1}^{r}\left(\boldsymbol{I}-\overline{[\boldsymbol{K}_{\boldsymbol{A}}]_p}\left(\overline{[\boldsymbol{K}_{\boldsymbol{A}}]_p}\right)^\top\right)\overline{[\boldsymbol{K}_{\boldsymbol{B}}]_k}\frac{\|[\boldsymbol{K}_{\boldsymbol{B}}]_k\|\,[\boldsymbol{Q}_1]_{k,p}}{\|[\boldsymbol{K}_{\boldsymbol{A}}]_p\|}\frac{\sigma_p^2}{N_{\boldsymbol{A}}N_{\boldsymbol{B}}d}$$
$$= \left(\boldsymbol{I}-\overline{[\boldsymbol{K}_{\boldsymbol{A}}]_p}\left(\overline{[\boldsymbol{K}_{\boldsymbol{A}}]_p}\right)^\top\right)\overline{[\boldsymbol{K}_{\boldsymbol{B}}]_p}\frac{\|[\boldsymbol{K}_{\boldsymbol{B}}]_p\|\,[\boldsymbol{Q}_1]_{p,p}}{\|[\boldsymbol{K}_{\boldsymbol{A}}]_p\|}\frac{\sigma_p^2}{N_{\boldsymbol{A}}N_{\boldsymbol{B}}d}$$
$$+\sum_{k\neq p}\left(\boldsymbol{I}-\overline{[\boldsymbol{K}_{\boldsymbol{A}}]_p}\left(\overline{[\boldsymbol{K}_{\boldsymbol{A}}]_p}\right)^\top\right)\overline{[\boldsymbol{K}_{\boldsymbol{B}}]_k}\frac{\|[\boldsymbol{K}_{\boldsymbol{B}}]_k\|\,[\boldsymbol{Q}_1]_{k,p}}{\|[\boldsymbol{K}_{\boldsymbol{A}}]_p\|}\frac{\sigma_p^2}{N_{\boldsymbol{A}}N_{\boldsymbol{B}}d}.$$

For the first term, by Lemma B.4, we have

$$\frac{\|[\boldsymbol{K}_{\boldsymbol{B}}]_p\|\,[\boldsymbol{Q}_1]_{p,p}}{\|[\boldsymbol{K}_{\boldsymbol{A}}]_p\|} = \left(1\pm\sqrt{\delta_{\boldsymbol{A}/\boldsymbol{B}}}\right)\left(\frac{2K}{1+K}\pm O\left(d\tau_t^2\right)\right) = \frac{2K}{1+K}\pm O\left(d\tau_t^2+\sqrt{\delta_{\boldsymbol{A}/\boldsymbol{B}}}\right).$$

Also by Lemma B.4, for each summand in the second term, we have

$$\frac{\|[\boldsymbol{K}_{\boldsymbol{B}}]_k\|\,[\boldsymbol{Q}_1]_{k,p}}{\|[\boldsymbol{K}_{\boldsymbol{A}}]_p\|} = \pm O\left(\tau_t^2 d\kappa_0\right).$$

Therefore,

$$\mathrm{T}_1\left(\frac{\mathrm{d}}{\mathrm{d}t}\overline{[\boldsymbol{K}_{\boldsymbol{A}}]_p}\right) = \frac{2K}{1+K}\frac{\sigma_p^2}{N_{\boldsymbol{A}}N_{\boldsymbol{B}}d}\left(\boldsymbol{I}-\overline{[\boldsymbol{K}_{\boldsymbol{A}}]_p}\left(\overline{[\boldsymbol{K}_{\boldsymbol{A}}]_p}\right)^\top\right)\overline{[\boldsymbol{K}_{\boldsymbol{B}}]_p}$$
$$\pm\frac{\sigma_p^2}{N_{\boldsymbol{A}}N_{\boldsymbol{B}}d}\sum_{k=1}^{r}O\left(\tau_t^2 d\kappa_0+\sqrt{\delta_{\boldsymbol{A}/\boldsymbol{B}}}\right)\overline{[\boldsymbol{K}_{\boldsymbol{B}}]_k}.$$

For $\mathrm{T}_2$, by Lemma B.5, we have

$$\mathrm{T}_2\left(\frac{\mathrm{d}}{\mathrm{d}t}\overline{[\boldsymbol{K}_{\boldsymbol{A}}]_p}\right) = \frac{\sigma_p^2}{N_{\boldsymbol{A}}N_{\boldsymbol{B}}d}\sum_{k=1}^{d-r}\left(\boldsymbol{I}-\overline{[\boldsymbol{K}_{\boldsymbol{A}}]_p}\left(\overline{[\boldsymbol{K}_{\boldsymbol{A}}]_p}\right)^\top\right)\overline{[\boldsymbol{K}_{\boldsymbol{B},\boldsymbol{\xi}}]_k}\frac{\|[\boldsymbol{K}_{\boldsymbol{B},\boldsymbol{\xi}}]_k\|\,[\boldsymbol{Q}_{1,\xi_B}]_{k,p}}{\|[\boldsymbol{K}_{\boldsymbol{A}}]_p\|}$$
$$= \frac{\sigma_p^2}{N_{\boldsymbol{A}}N_{\boldsymbol{B}}d}\sum_{k=1}^{d-r}O\left(\tau_t^2 d^2\rho_{N/S}^2\delta_{\xi,\perp}\right)\overline{[\boldsymbol{K}_{\boldsymbol{B},\boldsymbol{\xi}}]_k}.$$

Combine these together, and we obtain

$$\frac{\mathrm{d}}{\mathrm{d}t}\overline{[\boldsymbol{K}_{\boldsymbol{A}}]_p} = \frac{2K}{1+K}\frac{\sigma_p^2}{N_{\boldsymbol{A}}N_{\boldsymbol{B}}d}\left(\boldsymbol{I}-\overline{[\boldsymbol{K}_{\boldsymbol{A}}]_p}\left(\overline{[\boldsymbol{K}_{\boldsymbol{A}}]_p}\right)^\top\right)\overline{[\boldsymbol{K}_{\boldsymbol{B}}]_p}$$
$$\pm\frac{\sigma_p^2}{N_{\boldsymbol{A}}N_{\boldsymbol{B}}d}\sum_{k=1}^{r}O\left(\tau_t^2 d\kappa_0+\sqrt{\delta_{\boldsymbol{A}/\boldsymbol{B}}}\right)\overline{[\boldsymbol{K}_{\boldsymbol{B}}]_k}\pm\frac{\sigma_p^2}{N_{\boldsymbol{A}}N_{\boldsymbol{B}}d}\sum_{k=1}^{d-r}O\left(\tau_t^2 d^2\rho_{N/S}^2\delta_{\xi,\perp}\right)\overline{[\boldsymbol{K}_{\boldsymbol{B},\boldsymbol{\xi}}]_k}$$
$$= \frac{2K}{1+K}\frac{\sigma_p^2}{N_{\boldsymbol{A}}N_{\boldsymbol{B}}d}\left(\boldsymbol{I}-\overline{[\boldsymbol{K}_{\boldsymbol{A}}]_p}\left(\overline{[\boldsymbol{K}_{\boldsymbol{A}}]_p}\right)^\top\right)\overline{[\boldsymbol{K}_{\boldsymbol{B}}]_p}\pm O\left(\frac{\sigma_p^2}{N_{\boldsymbol{A}}N_{\boldsymbol{B}}d}\left(\tau_t^2 d^2\kappa_0+d\sqrt{\delta_{\boldsymbol{A}/\boldsymbol{B}}}\right)\right).$$

Now, we consider $\boldsymbol{K}_{\boldsymbol{A},\boldsymbol{\xi}}$. Again, we write

$$\frac{\mathrm{d}}{\mathrm{d}t}\overline{[\boldsymbol{K}_{\boldsymbol{A},\boldsymbol{\xi}}]_q} = \left(\boldsymbol{I} - \overline{[\boldsymbol{K}_{\boldsymbol{A},\boldsymbol{\xi}}]_q}\left(\overline{[\boldsymbol{K}_{\boldsymbol{A},\boldsymbol{\xi}}]_q}\right)^\top\right)\frac{[\boldsymbol{K}_{\boldsymbol{B}}\boldsymbol{Q}_{1,\boldsymbol{\xi}_{\boldsymbol{A}}}^\top]_q}{\|[\boldsymbol{K}_{\boldsymbol{A},\boldsymbol{\xi}}]_q\|}\frac{\sigma_\xi^2}{N_A N_B d}$$
$$+ \left(\boldsymbol{I} - \overline{[\boldsymbol{K}_{\boldsymbol{A},\boldsymbol{\xi}}]_q}\left(\overline{[\boldsymbol{K}_{\boldsymbol{A},\boldsymbol{\xi}}]_q}\right)^\top\right)\frac{[\boldsymbol{K}_{\boldsymbol{B},\boldsymbol{\xi}}\boldsymbol{Q}_2]_q}{\|[\boldsymbol{K}_{\boldsymbol{A},\boldsymbol{\xi}}]_q\|}\frac{\sigma_\xi^2}{N_A N_B d}$$
$$=: \mathrm{T}_1\left(\frac{\mathrm{d}}{\mathrm{d}t}\overline{[\boldsymbol{K}_{\boldsymbol{A},\boldsymbol{\xi}}]_q}\right) + \mathrm{T}_2\left(\frac{\mathrm{d}}{\mathrm{d}t}\overline{[\boldsymbol{K}_{\boldsymbol{A},\boldsymbol{\xi}}]_q}\right).$$

For the first term, by Lemma B.5, we have

$$\mathrm{T}_1\left(\frac{\mathrm{d}}{\mathrm{d}t}\overline{[\boldsymbol{K}_{\boldsymbol{A},\boldsymbol{\xi}}]_q}\right) = \sum_{k=1}^r \left(\boldsymbol{I} - \overline{[\boldsymbol{K}_{\boldsymbol{A},\boldsymbol{\xi}}]_q}\left(\overline{[\boldsymbol{K}_{\boldsymbol{A},\boldsymbol{\xi}}]_q}\right)^\top\right)\overline{[\boldsymbol{K}_{\boldsymbol{B}}]_k}\frac{\|[\boldsymbol{K}_{\boldsymbol{B}}]_k\|[\boldsymbol{Q}_{1,\boldsymbol{\xi}_{\boldsymbol{A}}}]_{q,k}}{\|[\boldsymbol{K}_{\boldsymbol{A},\boldsymbol{\xi}}]_q\|}\frac{\sigma_\xi^2}{N_A N_B d}$$
$$= \pm \sum_{k=1}^r O\left(\frac{\sigma_\xi^2}{N_A N_B d}\frac{\kappa_0}{\rho_{N/S}}\tau_t^2 d^2 \rho_{N/S}\delta_{\xi,\perp}\right)$$
$$= \pm O\left(\frac{\sigma_\xi^2}{N_A N_B d}\tau_t^2 d^3 \kappa_0 \delta_{\xi,\perp}\right).$$

The same bound also hold for $\mathrm{T}_2$. In fact, we can have a slightly sharper bound for it because we no longer have $\|[\boldsymbol{K}_{\boldsymbol{B}}]_k\| / \|[\boldsymbol{K}_{\boldsymbol{A},\boldsymbol{\xi}}]_q\|$. Combine these and we obtain

$$\frac{\mathrm{d}}{\mathrm{d}t}\overline{[\boldsymbol{K}_{\boldsymbol{A},\boldsymbol{\xi}}]_q} = \pm O\left(\frac{\sigma_\xi^2}{N_A N_B d}\tau_t^2 d^3 \kappa_0 \delta_{\xi,\perp}\right).$$

$\square$

*Proof of Lemma B.9.* The proof of Corollary B.8 and Corollary B.7, *mutatis mutandis*, yields

$$\frac{\mathrm{d}}{\mathrm{d}t}\boldsymbol{K}_{\boldsymbol{A}} = \frac{2K}{1+K}\frac{\boldsymbol{K}_{\boldsymbol{B}}}{N_A N_B d}\mathrm{diag}(\boldsymbol{\sigma}^2) - \frac{2K}{1+K}\frac{\boldsymbol{K}_{\boldsymbol{A}}}{N_A^2 d}\frac{\langle\boldsymbol{K}_{\boldsymbol{A}},\boldsymbol{K}_{\boldsymbol{B}}\rangle}{N_A N_B d}\mathrm{diag}(\boldsymbol{\sigma}^2)$$
$$\pm O\left(\frac{\sigma_{\max}^2}{N_A N_B d}\left(d\tau_t^2 + \sqrt{\delta_{\boldsymbol{A}/\boldsymbol{B}}}\right)\|\boldsymbol{K}_{\boldsymbol{A}}\|_F\right),$$
$$\frac{\mathrm{d}}{\mathrm{d}t}\boldsymbol{K}_{\boldsymbol{A},\boldsymbol{\xi}} = -\frac{2K}{1+K}\frac{\boldsymbol{K}_{\boldsymbol{A},\boldsymbol{\xi}}}{N_A^2 d}\frac{\langle\boldsymbol{K}_{\boldsymbol{A}},\boldsymbol{K}_{\boldsymbol{B}}\rangle}{N_A N_B d}\sigma_\xi^2 \pm O\left(\frac{\sigma_\xi^2}{N_A^2 d}d\tau_t^2\|\boldsymbol{K}_{\boldsymbol{A},\boldsymbol{\xi}}\|_F\right).$$

Recall from Lemma A.6 that, in the non-contrastive case, we have

$$\frac{\mathrm{d}}{\mathrm{d}t}\boldsymbol{K}_{\boldsymbol{A}} = \mathbb{E}\left\{\frac{\boldsymbol{f}_{\boldsymbol{B}}^+(\boldsymbol{z}^+)^\top}{N_A} - \langle\boldsymbol{f}_{\boldsymbol{A}}^+, \boldsymbol{f}_{\boldsymbol{B}}^+\rangle\frac{\boldsymbol{K}_{\boldsymbol{A}}}{N_A^2 d}\right\}\mathrm{diag}(\boldsymbol{\sigma}^2)$$
$$= \frac{\boldsymbol{K}_{\boldsymbol{B}}}{N_A N_B d}\mathrm{diag}(\boldsymbol{\sigma}^2) - \frac{\langle\boldsymbol{K}_{\boldsymbol{A}},\boldsymbol{K}_{\boldsymbol{B}}\rangle}{N_A N_B d}\frac{\boldsymbol{K}_{\boldsymbol{A}}}{N_A^2 d}\mathrm{diag}(\boldsymbol{\sigma}^2),$$
$$\frac{\mathrm{d}}{\mathrm{d}t}\boldsymbol{K}_{\boldsymbol{A},\boldsymbol{\xi}} = -\frac{\langle\boldsymbol{K}_{\boldsymbol{A}},\boldsymbol{K}_{\boldsymbol{B}}\rangle}{N_A N_B d}\frac{\boldsymbol{K}_{\boldsymbol{A},\boldsymbol{\xi}}}{N_A^2 d}\sigma_\xi^2.$$

Note that they are exactly the same, except for a $2K/(1 + K)$ factor and some higher order error terms. $\square$

## B.2 CONVERGENCE RATE AND THE CONDITION NUMBER

In this subsection, we estimate the rate at which $1 - \rho_-$ and $\rho_{N/S}$ converge to $0$ and the growth rate of the condition number. The basic idea is to use the estimations we have derived in Corollary B.7 and Corollary B.8 to approximate the finite-width dynamics with the infinite-width ones.

**Lemma B.10** (Convergence rate of $\rho_-$). *In Stage 1, we have, for any $p \in [r]$,*

$$\frac{\mathrm{d}}{\mathrm{d}t} \left\langle \overline{[K_A]_p}, \overline{[K_B]_p} \right\rangle = \frac{4K}{1+K} \frac{\sigma_p^2}{N_A N_B d} \left(1 + \left\langle \overline{[K_A]_p}, \overline{[K_B]_p} \right\rangle\right) \left(1 - \left\langle \overline{[K_A]_p}, \overline{[K_B]_p} \right\rangle\right)$$
$$\pm O\left(\frac{\sigma_p^2}{N_A N_B d} \left(\tau_t^2 d^2 \kappa_0 + \sqrt{\delta_{A/B}}\right)\right).$$

Note that $1 + \left\langle \overline{[K_A]_p}, \overline{[K_B]_p} \right\rangle = \Theta(1)$. Hence, $\left\langle \overline{[K_A]_p}, \overline{[K_B]_p} \right\rangle$ will converge to 1 at a linear rate.

Now we consider the noise-signal ratio. For some technical reason, instead of characterizing the dynamics of $\rho_{N/S}$, we consider

$$\hat{\rho}_{N/S} := \frac{\|K_{A,\xi}\|_F}{\|K_A + K_B\|_F}.$$

Note that we always have

$$\hat{\rho}_{N/S}^2 \geq \Theta(1) \frac{\|K_{A,\xi}\|_F^2}{\|K_A\|_F^2} \geq \Theta(1) \frac{(d-r)\|[K_{A,\xi}]_q\|^2}{\kappa_0^2 r \|[K_A]_p\|^2}$$

In other words, $\rho_{N/S}$ can be bounded by $\rho_{N/S} \leq O\left(\frac{d\kappa_0}{r} \hat{\rho}_{N/S}\right)$.

**Lemma B.11** (Convergence rate of $\rho_{N/S}$). *In Stage 1, we have*

$$\frac{\mathrm{d}}{\mathrm{d}t} \hat{\rho}_{N/S}^2 \leq -\frac{4K}{1+K} \frac{\sigma_{\min}^2}{N_A N_B d} \hat{\rho}_{N/S}^2 + O\left(\frac{\sigma_{\max}^2}{N_A N_B d} \left(d\tau_t^2 + \sqrt{\delta_{A/B}}\right) \hat{\rho}_{N/S}^2\right).$$

This lemma says the noise-signal ratio decreases exponentially fast.

**Lemma B.12** (Growth rate of the condition number). *Define $\rho_{p/q} = \|[K_A]_p\|^2 / \|[K_A]_q\|^2$. In Stage 1, we have*

$$\dot{\rho}_{p/q} \leq \frac{16K}{1+K} \frac{\sigma_{\max}^2}{N_A N_B d} \left(\rho_- + \min\left\{\hat{\rho}_{N/S}, 1\right\}\right) \rho_{p/q} \pm O\left(\frac{\sigma_{\max}^2}{N_A N_B d} \left(\sqrt{\delta_{A/B}} + \tau_t^2 d\right) \rho_{p/q}\right).$$

Note that error growth slows down as $\rho_-$ and $\hat{\rho}_{N/S}$ decrease. This allows us to use Lemma B.2 (cf. Section B.4).

OMITTED PROOFS OF THIS SUBSECTION

*Proof of Lemma B.10.* By Corollary B.8, we have

$$\left\langle \frac{\mathrm{d}}{\mathrm{d}t} \overline{[K_A]_p}, \overline{[K_B]_p} \right\rangle = \frac{2K}{1+K} \frac{\sigma_p^2}{N_A N_B d} \left(1 - \left\langle \overline{[K_A]_p}, \overline{[K_B]_p} \right\rangle^2\right) \pm O\left(\frac{\sigma_p^2}{N_A N_B d} \left(\tau_t^2 d^2 \kappa_0 + \sqrt{\delta_{A/B}}\right)\right)$$

$$= \frac{2K}{1+K} \frac{\sigma_p^2}{N_A N_B d} \left(1 + \left\langle \overline{[K_A]_p}, \overline{[K_B]_p} \right\rangle\right) \left(1 - \left\langle \overline{[K_A]_p}, \overline{[K_B]_p} \right\rangle\right)$$
$$\pm O\left(\frac{\sigma_p^2}{N_A N_B d} \left(\tau_t^2 d^2 \kappa_0 + \sqrt{\delta_{A/B}}\right)\right).$$

Hence, by symmetry, we have

$$\frac{\mathrm{d}}{\mathrm{d}t} \left\langle \overline{[K_A]_p}, \overline{[K_B]_p} \right\rangle = \frac{4K}{1+K} \frac{\sigma_p^2}{N_A N_B d} \left(1 + \left\langle \overline{[K_A]_p}, \overline{[K_B]_p} \right\rangle\right) \left(1 - \left\langle \overline{[K_A]_p}, \overline{[K_B]_p} \right\rangle\right)$$
$$\pm O\left(\frac{\sigma_p^2}{N_A N_B d} \left(\tau_t^2 d^2 \kappa_0 + \sqrt{\delta_{A/B}}\right)\right).$$

$\square$

*Proof of Lemma B.11.* Similar to the proof of Corollary B.7 and Corollary B.8, we have

$$\frac{\mathrm{d}}{\mathrm{d}t}\boldsymbol{K_A} = \frac{\boldsymbol{K_B}}{N_A N_B d}\boldsymbol{Q}_1\operatorname{diag}(\boldsymbol{\sigma}^2) + \frac{\boldsymbol{K_{B,\xi}}}{N_A N_B d}\boldsymbol{Q}_{1,\xi_B}\operatorname{diag}(\boldsymbol{\sigma}^2) + \frac{\boldsymbol{K_A}}{N_A^2 d}Q_0\operatorname{diag}(\boldsymbol{\sigma}^2)$$

$$= \frac{\boldsymbol{K_B}}{N_A N_B d}\left(\frac{2K}{1+K}\boldsymbol{I}_d \pm O\left(d\tau_t^2\right)\right)\operatorname{diag}(\boldsymbol{\sigma}^2)$$

$$\pm \frac{\boldsymbol{K_{B,\xi}}}{N_A N_B d}O\left(\tau_t^2 d^2\rho_{N/S}\delta_{\xi,\perp}\right)\operatorname{diag}(\boldsymbol{\sigma}^2)$$

$$+ \frac{\boldsymbol{K_A}}{N_A^2 d}\left(-\frac{2K}{1+K}\frac{\langle\boldsymbol{K_A},\boldsymbol{K_B}\rangle}{N_A N_B d} \pm O\left(d\tau_t^2\right)\right)\operatorname{diag}(\boldsymbol{\sigma}^2)$$

$$= \frac{2K}{1+K}\frac{\boldsymbol{K_B}}{N_A N_B d}\operatorname{diag}(\boldsymbol{\sigma}^2) - \frac{2K}{1+K}\frac{\boldsymbol{K_A}}{N_A N_B d}\frac{\langle\boldsymbol{K_A},\boldsymbol{K_B}\rangle}{N_A N_B d}\operatorname{diag}(\boldsymbol{\sigma}^2)$$

$$\pm O\left(\frac{\sigma_{\max}^2}{N_A N_B d}\left(d\tau_t^2 + \sqrt{\delta_{A/B}}\right)\|\boldsymbol{K_A}\|_F\right).$$

Define $\boldsymbol{K} = \boldsymbol{K_A} + \boldsymbol{K_B}$. Then, by symmetry, we have

$$\frac{\mathrm{d}}{\mathrm{d}t}\boldsymbol{K} = \frac{2K}{1+K}\frac{1}{N_A N_B d}\left(1 - \frac{\langle\boldsymbol{K_A},\boldsymbol{K_B}\rangle}{N_A N_B d}\right)\boldsymbol{K}\operatorname{diag}(\boldsymbol{\sigma}^2) \pm O\left(\frac{\sigma_{\max}^2}{N_A N_B d}\left(d\tau_t^2 + \sqrt{\delta_{A/B}}\right)\|\boldsymbol{K}\|_F\right).$$

Hence,

$$\frac{\mathrm{d}}{\mathrm{d}t}\|\boldsymbol{K}\|_F^2 = \frac{4K}{1+K}\frac{1}{N_A N_B d}\left(1 - \frac{\langle\boldsymbol{K_A},\boldsymbol{K_B}\rangle}{N_A N_B d}\right)\langle\boldsymbol{K},\boldsymbol{K}\operatorname{diag}(\boldsymbol{\sigma}^2)\rangle \pm O\left(\frac{\sigma_{\max}^2}{N_A N_B d}\left(d\tau_t^2 + \sqrt{\delta_{A/B}}\right)\|\boldsymbol{K}\|_F^2\right)$$

$$\geq \frac{4K}{1+K}\frac{\sigma_{\min}^2}{N_A N_B d}\left(1 - \frac{\langle\boldsymbol{K_A},\boldsymbol{K_B}\rangle}{N_A N_B d}\right)\|\boldsymbol{K}\|_F^2 - O\left(\frac{\sigma_{\max}^2}{N_A N_B d}\left(d\tau_t^2 + \sqrt{\delta_{A/B}}\right)\|\boldsymbol{K}\|_F^2\right).$$

Similarly, we also have

$$\frac{\mathrm{d}}{\mathrm{d}t}\boldsymbol{K_{A,\xi}} = \frac{\boldsymbol{K_B}}{N_A N_B d}\boldsymbol{Q}_{1,\xi_A}^\top\sigma_\xi^2 + \frac{\boldsymbol{K_{B,\xi}}}{N_A N_B d}\boldsymbol{Q}_2\sigma_\xi^2 + \frac{\boldsymbol{K_{A,\xi}}}{N_A^2 d}Q_0\sigma_\xi^2$$

$$= \frac{\boldsymbol{K_B}}{N_A N_B d}O\left(\tau_t^2 d^2\rho_{N/S}\delta_{\xi,\perp}\right)\sigma_\xi^2 + \frac{\boldsymbol{K_{B,\xi}}}{N_A N_B d}O\left(\tau_t^2 d^2\rho_{N/S}\delta_{\xi,\perp}\right)\sigma_\xi^2$$

$$+ \frac{\boldsymbol{K_{A,\xi}}}{N_A^2 d}\left(-\frac{2K}{1+K}\frac{\langle\boldsymbol{K_A},\boldsymbol{K_B}\rangle}{N_A N_B d} \pm O\left(d\tau_t^2\right)\right)\sigma_\xi^2$$

$$= -\frac{2K}{1+K}\frac{\boldsymbol{K_{A,\xi}}}{N_A^2 d}\frac{\langle\boldsymbol{K_A},\boldsymbol{K_B}\rangle}{N_A N_B d}\sigma_\xi^2 \pm O\left(\frac{\sigma_\xi^2}{N_A^2 d}d\tau_t^2\|\boldsymbol{K_{A,\xi}}\|_F\right) \pm O\left(\frac{\sigma_\xi^2}{N_A N_B d}\tau_t^2 d^2\delta_{\xi,\perp}\frac{d\kappa_0}{r}\|\boldsymbol{K_{A,\xi}}\|_F\right)$$

$$= -\frac{2K}{1+K}\frac{\boldsymbol{K_{A,\xi}}}{N_A^2 d}\frac{\langle\boldsymbol{K_A},\boldsymbol{K_B}\rangle}{N_A N_B d}\sigma_\xi^2 \pm O\left(\frac{\sigma_\xi^2}{N_A^2 d}d\tau_t^2\|\boldsymbol{K_{A,\xi}}\|_F\right).$$

Therefore,

$$\frac{\mathrm{d}}{\mathrm{d}t}\|\boldsymbol{K_{A,\xi}}\|_F^2 = -\frac{4K}{1+K}\frac{\sigma_\xi^2}{N_A^2 d}\frac{\langle\boldsymbol{K_A},\boldsymbol{K_B}\rangle}{N_A N_B d}\|\boldsymbol{K_{A,\xi}}\|_F^2 \pm O\left(\frac{\sigma_\xi^2}{N_A^2 d}d\tau_t^2\|\boldsymbol{K_{A,\xi}}\|_F^2\right).$$

Then, we compute

$$\frac{\mathrm{d}}{\mathrm{d}t}\hat{\rho}_{N/S}^2 = \frac{\frac{\mathrm{d}}{\mathrm{d}t}\|\boldsymbol{K_{A,\xi}}\|_F^2}{\|\boldsymbol{K}\|_F^2} - \hat{\rho}_{N/S}^2\frac{\frac{\mathrm{d}}{\mathrm{d}t}\|\boldsymbol{K_A}\|_F^2}{\|\boldsymbol{K}\|_F^2}$$

$$\leq -\frac{4K}{1+K}\frac{\sigma_\xi^2}{N_A^2 d}\frac{\langle\boldsymbol{K_A},\boldsymbol{K_B}\rangle}{N_A N_B d}\hat{\rho}_{N/S}^2 \pm O\left(\frac{\sigma_\xi^2}{N_A^2 d}d\tau_t^2\hat{\rho}_{N/S}^2\right)$$

$$- \hat{\rho}_{N/S}^2\left(\frac{4K}{1+K}\frac{\sigma_{\min}^2}{N_A N_B d}\left(1 - \frac{\langle\boldsymbol{K_A},\boldsymbol{K_B}\rangle}{N_A N_B d}\right) - O\left(\frac{\sigma_{\max}^2}{N_A N_B d}\left(d\tau_t^2 + \sqrt{\delta_{A/B}}\right)\right)\right)$$

$$\leq -\frac{4K}{1+K}\frac{\sigma_{\min}^2}{N_A N_B d}\hat{\rho}_{N/S}^2 + O\left(\frac{\sigma_{\max}^2}{N_A N_B d}\left(d\tau_t^2 + \sqrt{\delta_{A/B}}\right)\hat{\rho}_{N/S}^2\right).$$

$\square$

*Proof of Lemma B.12.* For notational simplicity, define $\rho_{p/q} := \|[K_A]_p\|^2 / \|[K_A]_q\|^2$. By Corollary B.7, we have

$$
\dot{\rho}_{p/q} = \frac{\frac{\mathrm{d}}{\mathrm{d}t} \|[K_A]_p\|^2}{\|[K_A]_q\|^2} - \rho_{p/q} \frac{\frac{\mathrm{d}}{\mathrm{d}t} \|[K_A]_q\|^2}{\|[K_A]_q\|^2}
$$

$$
= \frac{4K}{1+K} \frac{\sigma_p^2}{N_A N_B d} \left( \frac{\langle [K_A]_p, [K_B]_p \rangle}{\|[K_A]_p\|^2} - \frac{\langle K_A, K_B \rangle}{N_A N_B d} \right) \rho_{p/q} \pm O\left( \frac{\sigma_p^2}{N_A N_B d} \left( \sqrt{\delta_{A/B}} + \tau_t^2 d \right) \rho_{p/q} \right)
$$

$$
- \rho_{p/q} \left( \frac{4K}{1+K} \frac{\sigma_q^2}{N_A N_B d} \left( \frac{\langle [K_A]_q, [K_B]_q \rangle}{\|[K_A]_q\|^2} - \frac{\langle K_A, K_B \rangle}{N_A N_B d} \right) \pm O\left( \frac{\sigma_q^2}{N_A N_B d} \left( \sqrt{\delta_{A/B}} + \tau_t^2 d \right) \right) \right)
$$

$$
= \frac{4K}{1+K} \frac{\sigma_p^2}{N_A N_B d} \left( \left\langle \overline{[K_A]_p}, \overline{[K_B]_p} \right\rangle - \frac{\langle K_A, K_B \rangle}{N_A N_B d} \right) \rho_{p/q}
$$

$$
- \frac{4K}{1+K} \frac{\sigma_q^2}{N_A N_B d} \left( \left\langle \overline{[K_A]_q}, \overline{[K_B]_q} \right\rangle - \frac{\langle K_A, K_B \rangle}{N_A N_B d} \right) \rho_{p/q}
$$

$$
\pm O\left( \frac{\sigma_{\max}^2}{N_A N_B d} \left( \sqrt{\delta_{A/B}} + \tau_t^2 d \right) \rho_{p/q} \right).
$$

Now we bound $\left\langle \overline{[K_A]_p}, \overline{[K_B]_p} \right\rangle - \langle K_A, K_B \rangle / (N_A N_B d)$. Clear that this term is bounded by 2. Meanwhile, by definition, we have $\left\langle \overline{[K_A]_p}, \overline{[K_B]_p} \right\rangle = 1 \pm \rho_-$. For the second term, we have

$$
\frac{\langle K_A, K_B \rangle}{N_A N_B d} = \sum_{k=1}^{r} \left\langle \overline{[K_A]_k}, \overline{[K_B]_k} \right\rangle \frac{\|K_A\|_k \|K_B\|_k}{\|K_A\|_F \|K_B\|_F} \frac{\|K_A\|_F \|K_B\|_F}{N_A N_B d}
$$

$$
= \sum_{k=1}^{r} (1 \pm \rho_-)(1 \pm \min\{\hat{\rho}_{N/S}, 1\}) \frac{\kappa_k^2}{\|\kappa\|^2} \left( 1 \pm \sqrt{\delta_{A/B}} \right)
$$

$$
= \left( 1 \pm \rho_- \pm \min\{\hat{\rho}_{N/S}, 1\} \right) \left( 1 \pm \sqrt{\delta_{A/B}} \right).
$$

Combine these together and we obtain

$$
\left| \left\langle \overline{[K_A]_p}, \overline{[K_B]_p} \right\rangle - \frac{\langle K_A, K_B \rangle}{N_A N_B d} \right| \le 2\rho_- + 2\min\{\hat{\rho}_{N/S}, 1\} \pm \sqrt{\delta_{A/B}}.
$$

The same is also true for $q$. Thus,

$$
\dot{\rho}_{p/q} \le \frac{16K}{1+K} \frac{\sigma_{\max}^2}{N_A N_B d} \left( \rho_- + \min\{\hat{\rho}_{N/S}, 1\} \right) \rho_{p/q} \pm O\left( \frac{\sigma_{\max}^2}{N_A N_B d} \left( \sqrt{\delta_{A/B}} + \tau_t^2 d \right) \rho_{p/q} \right).
$$

$\square$

## B.3 CONTROLLING THE DISCRETIZATION ERROR

In this subsection, we estimate the growth rate of the errors described in (10). As in the previous subsections, the proofs are deferred to the end of this subsection.

First, we consider the relative difference between $\|[K_A]_p\|^2$ and $\|[K_B]_p\|^2$. Instead of directly control the difference, we define

$$
\rho_{A/B,p} := \frac{\|[K_A]_p\|^2}{\|[K_B]_p\|^2} \quad \text{and} \quad \rho_{B/A,p} := \frac{\|[K_B]_p\|^2}{\|[K_A]_p\|^2}.
$$

Note that $\rho_{A/B,p} + \rho_{B/A,p} \ge 2$, with equality obtained iff $\|[K_A]_p\|^2 = \|[K_B]_p\|^2$. Meanwhile, at initialization, this quantity can be made arbitrarily close to 2. Hence, it suffices to control the growth of this quantity. The reason we consider $\rho_{A/B,p} + \rho_{B/A,p}$ is to leverage the symmetry. Similarly, we also define

$$
\rho_{A/B,\xi,q} := \frac{\|[K_{A,\xi}]_q\|^2}{\|[K_{B,\xi}]_q\|^2} \quad \text{and} \quad \rho_{B/A,\xi,q} := \frac{\|[K_{B,\xi}]_q\|^2}{\|[K_{A,\xi}]_q\|^2},
$$

and analyze $\rho_{A/A,\xi} + \rho_{B/A,\xi}$.

**Lemma B.13** (Difference of diagonal terms). *In Stage 1, we have*

$$\frac{\mathrm{d}}{\mathrm{d}t}\left(\rho_{\boldsymbol{A}/\boldsymbol{B},p} + \rho_{\boldsymbol{B}/\boldsymbol{A},p}\right) \leq O\left(\frac{\sigma_p^2}{N_{\boldsymbol{A}}N_{\boldsymbol{B}}d}\left(\delta_{\boldsymbol{A}/\boldsymbol{B}}^2 + \tau_t^2 d\right)\right),$$

$$\frac{\mathrm{d}}{\mathrm{d}t}\left(\rho_{\boldsymbol{A}/\boldsymbol{B},\xi,q} + \rho_{\boldsymbol{B}/\boldsymbol{A},\xi,q}\right) \leq O\left(\frac{\sigma_{\max}^2}{N_{\boldsymbol{A}}N_{\boldsymbol{B}}d}\left(\delta_{\boldsymbol{A}/\boldsymbol{B}}^2 + \tau_t^2 d\right)\right).$$

Note that the RHS are higher-order terms, which implies the relative difference of the norms barely grows.

Now we consider the condition number of $\boldsymbol{K}_{\boldsymbol{A},\boldsymbol{\xi}}$. Unlike $\boldsymbol{K}_{\boldsymbol{A}}$, for the noises, the $\sigma$'s for different coordinates are the same. Hence, we have the following simple bound on the growth rate of $\delta_{\xi,\kappa_0}$.

**Lemma B.14** (Condition number of $\boldsymbol{K}_{\boldsymbol{A},\boldsymbol{\xi}}$). *Define $\rho_{\xi,p/q} := \|[\boldsymbol{K}_{\boldsymbol{A},\boldsymbol{\xi}}]_p\|^2 / \|[\boldsymbol{K}_{\boldsymbol{A},\boldsymbol{\xi}}]_q\|^2$. In Stage 1, we have*

$$\dot{\rho}_{\xi,p/q} \leq O\left(\frac{\sigma_\xi^2}{N_{\boldsymbol{A}}N_{\boldsymbol{B}}d}\left(d\tau_t^2 + \sqrt{\delta_{\boldsymbol{A}/\boldsymbol{B}}}\right)\right).$$

Again, the RHS is a higher-order term that can be made arbitrarily small by choosing a small $\tau_t^2$ and maintaining a small $\delta_{\boldsymbol{A}/\boldsymbol{B}}$.

Then, we consider the orthogonality conditions.

**Lemma B.15** (Orthogonality between signals). *For any $p \neq q$, define*

$$\hat{\delta}_{\perp,p,q} := \left\langle \overline{[\boldsymbol{K}_{\boldsymbol{A}}]_p}, \overline{[\boldsymbol{K}_{\boldsymbol{B}}]_q} \right\rangle^2 + \left\langle \overline{[\boldsymbol{K}_{\boldsymbol{B}}]_p}, \overline{[\boldsymbol{K}_{\boldsymbol{A}}]_q} \right\rangle^2 + \left\langle \overline{[\boldsymbol{K}_{\boldsymbol{A}}]_p}, \overline{[\boldsymbol{K}_{\boldsymbol{A}}]_q} \right\rangle^2 + \left\langle \overline{[\boldsymbol{K}_{\boldsymbol{B}}]_p}, \overline{[\boldsymbol{K}_{\boldsymbol{B}}]_q} \right\rangle^2.$$

*In Stage 1, we have*

$$\frac{\mathrm{d}}{\mathrm{d}t}\hat{\delta}_{\perp,p,q} \leq O\left(\frac{\sigma_{\max}^2}{N_{\boldsymbol{A}}N_{\boldsymbol{B}}d}\right)\rho_-\hat{\delta}_{\perp,p,q} + O\left(\frac{\sigma_{\max}^2}{N_{\boldsymbol{A}}N_{\boldsymbol{B}}d}\left(\tau_t^2 d^2\kappa_0 + d\sqrt{\delta_{\boldsymbol{A}/\boldsymbol{B}}}\right)\right).$$

Recall that $\rho_-$ converges to 0 at a sufficiently fast rate. As a result, by Lemma B.2, $\hat{\delta}_{\perp,p,q}$ will not blow up. Meanwhile, since $\hat{\delta}_{\perp,p,q}$ can be made arbitrarily small at initialization, this implies that we can make sure it is still small at the end of Stage 1. Finally, we consider the orthogonality conditions between the signals and noises and between noises. The proof follows the same spirit.

**Lemma B.16** (Orthogonality between signals and noises). *For any $p \in [r]$ and $q \in [d - r]$, define*

$$\hat{\delta}_{\perp,\xi_{\boldsymbol{A}},p,q} = \left\langle \overline{[\boldsymbol{K}_{\boldsymbol{A}}]_p}, \overline{[\boldsymbol{K}_{\boldsymbol{A},\boldsymbol{\xi}}]_q} \right\rangle^2 + \left\langle \overline{[\boldsymbol{K}_{\boldsymbol{B}}]_p}, \overline{[\boldsymbol{K}_{\boldsymbol{A},\boldsymbol{\xi}}]_q} \right\rangle^2,$$

*and define $\hat{\delta}_{\perp,\xi_{\boldsymbol{B}},p,q}$ similarly. In Stage 1, we have*

$$\frac{\mathrm{d}}{\mathrm{d}t}\hat{\delta}_{\perp,\xi_{\boldsymbol{A}},p,q} \leq O\left(\frac{\sigma_p^2}{N_{\boldsymbol{A}}N_{\boldsymbol{B}}d}\right)\rho_-\hat{\delta}_{\perp,\xi_{\boldsymbol{A}},p,q} \pm O\left(\frac{\sigma_{\max}^2}{N_{\boldsymbol{A}}N_{\boldsymbol{B}}d}\left(\tau_t^2 d^2\kappa_0 + d\sqrt{\delta_{\boldsymbol{A}/\boldsymbol{B}}}\right)\right).$$

*For any $q, s \in [d - r]$, define*

$$\hat{\delta}_{\perp,\xi,p,q} = \left\langle \overline{[\boldsymbol{K}_{\boldsymbol{A},\boldsymbol{\xi}}]_q}, \overline{[\boldsymbol{K}_{\boldsymbol{B},\boldsymbol{\xi}}]_s} \right\rangle^2 + \left\langle \overline{[\boldsymbol{K}_{\boldsymbol{B},\boldsymbol{\xi}}]_q}, \overline{[\boldsymbol{K}_{\boldsymbol{A},\boldsymbol{\xi}}]_s} \right\rangle^2,$$

*In Stage 1, we have*

$$\frac{\mathrm{d}}{\mathrm{d}t}\hat{\delta}_{\perp,\xi,p,q} \leq O\left(\frac{\sigma_\xi^2}{N_{\boldsymbol{A}}N_{\boldsymbol{B}}d}\tau_t^2 d^3\kappa_0\delta_{\xi,\perp}\sqrt{\hat{\delta}_{\perp,\xi,p,q}}\right).$$

OMITTED PROOFS OF THIS SUBSECTION

*Proof of Lemma B.13 .* Note that we cannot directly use Corollary B.7 as the error term contains $\delta_{A/B}$, the quantity we wish to control. However, by the proof of it, we still have

$$
\frac{\mathrm{d}}{\mathrm{d}t} \left\| [\boldsymbol{K_A}]_p \right\|^2 = \frac{4K}{1+K} \frac{\sigma_p^2}{N_A N_B d} \langle [\boldsymbol{K_A}]_p, [\boldsymbol{K_B}]_p \rangle - \frac{4K}{1+K} \frac{\langle \boldsymbol{K_A}, \boldsymbol{K_B} \rangle \sigma_p^2}{N_A N_B d} \frac{\left\| [\boldsymbol{K_A}]_p \right\|^2}{N_A^2 d}
$$
$$
\pm O \left( \frac{\sigma_p^2}{N_A N_B d} \tau_t^2 d \kappa_p^2 \right).
$$

Interchange the roles of $\boldsymbol{A}$ and $\boldsymbol{B}$ and we obtain

$$
\frac{\mathrm{d}}{\mathrm{d}t} \left\| [\boldsymbol{K_B}]_p \right\|^2 = \frac{4K}{1+K} \frac{\sigma_p^2}{N_A N_B d} \langle [\boldsymbol{K_A}]_p, [\boldsymbol{K_B}]_p \rangle - \frac{4K}{1+K} \frac{\langle \boldsymbol{K_A}, \boldsymbol{K_B} \rangle \sigma_p^2}{N_A N_B d} \frac{\left\| [\boldsymbol{K_B}]_p \right\|^2}{N_B^2 d}
$$
$$
\pm O \left( \frac{\sigma_p^2}{N_A N_B d} \tau_t^2 d \kappa_p^2 \right).
$$

For notational simplicity, define $\rho_{A/B,p} = \left\| [\boldsymbol{K_A}]_p \right\|^2 / \left\| [\boldsymbol{K_B}]_p \right\|^2$. Then, we compute

$$
\dot{\rho}_{A/B,p} = \frac{\frac{\mathrm{d}}{\mathrm{d}t} \left\| [\boldsymbol{K_A}]_p \right\|^2}{\left\| [\boldsymbol{K_B}]_p \right\|^2} - \rho_{A/B,p} \frac{\frac{\mathrm{d}}{\mathrm{d}t} \left\| [\boldsymbol{K_B}]_p \right\|^2}{\left\| [\boldsymbol{K_B}]_p \right\|^2}
$$
$$
= \frac{4K}{1+K} \frac{\sigma_p^2}{N_A N_B d} \frac{\langle [\boldsymbol{K_A}]_p, [\boldsymbol{K_B}]_p \rangle}{\left\| [\boldsymbol{K_A}]_p \right\|^2} \rho_{A/B,p} - \frac{4K}{1+K} \frac{\langle \boldsymbol{K_A}, \boldsymbol{K_B} \rangle \sigma_p^2}{N_A N_B d} \frac{\rho_{A/B,p}}{N_A^2 d}
$$
$$
- \frac{4K}{1+K} \frac{\sigma_p^2}{N_A N_B d} \frac{\langle [\boldsymbol{K_A}]_p, [\boldsymbol{K_B}]_p \rangle}{\left\| [\boldsymbol{K_B}]_p \right\|^2} \rho_{A/B,p} + \frac{4K}{1+K} \frac{\langle \boldsymbol{K_A}, \boldsymbol{K_B} \rangle \sigma_p^2}{N_A N_B d} \frac{\rho_{A/B,p}}{N_B^2 d}
$$
$$
\pm O \left( \frac{\sigma_p^2}{N_A N_B d} \tau_t^2 d \right)
$$
$$
= \frac{4K}{1+K} \frac{\sigma_p^2}{N_A N_B d} \langle [\boldsymbol{K_A}]_p, [\boldsymbol{K_B}]_p \rangle \left( \frac{1}{\left\| [\boldsymbol{K_A}]_p \right\|^2} - \frac{1}{\left\| [\boldsymbol{K_B}]_p \right\|^2} \right) \rho_{A/B,p}
$$
$$
- \frac{4K}{1+K} \frac{\langle \boldsymbol{K_A}, \boldsymbol{K_B} \rangle \sigma_p^2}{N_A N_B d} \left( \frac{1}{N_A^2 d} - \frac{1}{N_B^2 d} \right) \rho_{A/B,p}
$$
$$
\pm O \left( \frac{\sigma_p^2}{N_A N_B d} \tau_t^2 d \right).
$$

Then, by symmetry, we have

$$
\frac{\mathrm{d}}{\mathrm{d}t} \left( \rho_{A/B,p} + \rho_{B/A,p} \right)
$$
$$
= \frac{4K}{1+K} \frac{\sigma_p^2}{N_A N_B d} \langle [\boldsymbol{K_A}]_p, [\boldsymbol{K_B}]_p \rangle \left( \frac{1}{\left\| [\boldsymbol{K_A}]_p \right\|^2} - \frac{1}{\left\| [\boldsymbol{K_B}]_p \right\|^2} \right) \left( \rho_{A/B,p} - \rho_{B/A,p} \right)
$$
$$
- \frac{4K}{1+K} \frac{\langle \boldsymbol{K_A}, \boldsymbol{K_B} \rangle \sigma_p^2}{N_A N_B d} \left( \frac{1}{N_A^2 d} - \frac{1}{N_B^2 d} \right) \left( \rho_{A/B,p} - \rho_{B/A,p} \right)
$$
$$
\pm O \left( \frac{\sigma_p^2}{N_A N_B d} \tau_t^2 d \right)
$$
$$
\leq O \left( \frac{\sigma_p^2}{N_A N_B d} \left( \delta_{A/B}^2 + \tau_t^2 d \right) \right).
$$

The above proof, *mutatis mutandis*, yields the result for $\rho_{A/B,\xi,q} + \rho_{B/A,\xi,q}$. $\qquad \square$

*Proof of Lemma B.14.* By Corollary B.7, we have

$$
\begin{aligned}
\dot{\rho}_{\xi,p/q} &= \frac{\frac{\mathrm{d}}{\mathrm{d}t} \left\| [\boldsymbol{K}_{\boldsymbol{A},\boldsymbol{\xi}}]_p \right\|^2}{\left\| [\boldsymbol{K}_{\boldsymbol{A},\boldsymbol{\xi}}]_q \right\|^2} - \rho_{\xi,p/q} \frac{\frac{\mathrm{d}}{\mathrm{d}t} \left\| [\boldsymbol{K}_{\boldsymbol{A},\boldsymbol{\xi}}]_q \right\|^2}{\left\| [\boldsymbol{K}_{\boldsymbol{A},\boldsymbol{\xi}}]_q \right\|^2} \\
&= -\frac{4K}{1+K} \frac{\sigma_\xi^2}{N_{\boldsymbol{A}} N_{\boldsymbol{B}} d} \frac{\langle \boldsymbol{K}_{\boldsymbol{A}}, \boldsymbol{K}_{\boldsymbol{B}} \rangle}{N_{\boldsymbol{A}} N_{\boldsymbol{B}} d} \rho_{\xi,p/q} \pm O\left( \frac{\sigma_\xi^2}{N_{\boldsymbol{A}} N_{\boldsymbol{B}} d} \left( d\tau_t^2 + \sqrt{\delta_{\boldsymbol{A}/\boldsymbol{B}}} \right) \right) \\
&\quad - \rho_{\xi,p/q} \left( -\frac{4K}{1+K} \frac{\sigma_\xi^2}{N_{\boldsymbol{A}} N_{\boldsymbol{B}} d} \frac{\langle \boldsymbol{K}_{\boldsymbol{A}}, \boldsymbol{K}_{\boldsymbol{B}} \rangle}{N_{\boldsymbol{A}} N_{\boldsymbol{B}} d} \pm O\left( \frac{\sigma_\xi^2}{N_{\boldsymbol{A}} N_{\boldsymbol{B}} d} \left( d\tau_t^2 + \sqrt{\delta_{\boldsymbol{A}/\boldsymbol{B}}} \right) \right) \right) \\
&= \pm O\left( \frac{\sigma_\xi^2}{N_{\boldsymbol{A}} N_{\boldsymbol{B}} d} \left( d\tau_t^2 + \sqrt{\delta_{\boldsymbol{A}/\boldsymbol{B}}} \right) \right).
\end{aligned}
$$

$\square$

*Proof of Lemma B.15.* By Corollary B.8, we have

$$
\begin{aligned}
\left\langle \frac{\mathrm{d}}{\mathrm{d}t} \overline{[\boldsymbol{K}_{\boldsymbol{A}}]_p}, \overline{[\boldsymbol{K}_{\boldsymbol{B}}]_q} \right\rangle &= \frac{2K}{1+K} \frac{\sigma_p^2}{N_{\boldsymbol{A}} N_{\boldsymbol{B}} d} \left( \left\langle \overline{[\boldsymbol{K}_{\boldsymbol{B}}]_p}, \overline{[\boldsymbol{K}_{\boldsymbol{B}}]_q} \right\rangle - \left\langle \overline{[\boldsymbol{K}_{\boldsymbol{A}}]_p}, \overline{[\boldsymbol{K}_{\boldsymbol{B}}]_q} \right\rangle \left\langle \overline{[\boldsymbol{K}_{\boldsymbol{A}}]_p}, \overline{[\boldsymbol{K}_{\boldsymbol{B}}]_p} \right\rangle \right) \\
&\quad \pm O\left( \frac{\sigma_p^2}{N_{\boldsymbol{A}} N_{\boldsymbol{B}} d} \left( \tau_t^2 d^2 \kappa_0 + d\sqrt{\delta_{\boldsymbol{A}/\boldsymbol{B}}} \right) \right) \\
&= \frac{2K}{1+K} \frac{\sigma_p^2}{N_{\boldsymbol{A}} N_{\boldsymbol{B}} d} \left( \left\langle \overline{[\boldsymbol{K}_{\boldsymbol{B}}]_p}, \overline{[\boldsymbol{K}_{\boldsymbol{B}}]_q} \right\rangle - \left\langle \overline{[\boldsymbol{K}_{\boldsymbol{A}}]_p}, \overline{[\boldsymbol{K}_{\boldsymbol{B}}]_q} \right\rangle \right) \\
&\quad + \frac{2K}{1+K} \frac{\sigma_p^2}{N_{\boldsymbol{A}} N_{\boldsymbol{B}} d} \left\langle \overline{[\boldsymbol{K}_{\boldsymbol{A}}]_p}, \overline{[\boldsymbol{K}_{\boldsymbol{B}}]_q} \right\rangle \left( 1 - \left\langle \overline{[\boldsymbol{K}_{\boldsymbol{A}}]_p}, \overline{[\boldsymbol{K}_{\boldsymbol{B}}]_p} \right\rangle \right) \\
&\quad \pm O\left( \frac{\sigma_p^2}{N_{\boldsymbol{A}} N_{\boldsymbol{B}} d} \left( \tau_t^2 d^2 \kappa_0 + d\sqrt{\delta_{\boldsymbol{A}/\boldsymbol{B}}} \right) \right).
\end{aligned}
$$

Interchange the roles of $p, q$ and $\boldsymbol{A}, \boldsymbol{B}$ and we obtain

$$
\begin{aligned}
\left\langle \overline{[\boldsymbol{K}_{\boldsymbol{A}}]_p}, \frac{\mathrm{d}}{\mathrm{d}t} \overline{[\boldsymbol{K}_{\boldsymbol{B}}]_q} \right\rangle &= \frac{2K}{1+K} \frac{\sigma_q^2}{N_{\boldsymbol{A}} N_{\boldsymbol{B}} d} \left( \left\langle \overline{[\boldsymbol{K}_{\boldsymbol{A}}]_p}, \overline{[\boldsymbol{K}_{\boldsymbol{A}}]_q} \right\rangle - \left\langle \overline{[\boldsymbol{K}_{\boldsymbol{A}}]_p}, \overline{[\boldsymbol{K}_{\boldsymbol{B}}]_q} \right\rangle \right) \\
&\quad + \frac{2K}{1+K} \frac{\sigma_q^2}{N_{\boldsymbol{A}} N_{\boldsymbol{B}} d} \left\langle \overline{[\boldsymbol{K}_{\boldsymbol{A}}]_p}, \overline{[\boldsymbol{K}_{\boldsymbol{B}}]_q} \right\rangle \left( 1 - \left\langle \overline{[\boldsymbol{K}_{\boldsymbol{A}}]_q}, \overline{[\boldsymbol{K}_{\boldsymbol{B}}]_q} \right\rangle \right) \\
&\quad \pm O\left( \frac{\sigma_q^2}{N_{\boldsymbol{A}} N_{\boldsymbol{B}} d} \left( \tau_t^2 d^2 \kappa_0 + d\sqrt{\delta_{\boldsymbol{A}/\boldsymbol{B}}} \right) \right).
\end{aligned}
$$

Therefore,

$$
\begin{aligned}
\frac{\mathrm{d}}{\mathrm{d}t} \left\langle \overline{[\boldsymbol{K}_{\boldsymbol{A}}]_p}, \overline{[\boldsymbol{K}_{\boldsymbol{B}}]_q} \right\rangle &= \frac{2K}{1+K} \frac{\sigma_p^2}{N_{\boldsymbol{A}} N_{\boldsymbol{B}} d} \left( \left\langle \overline{[\boldsymbol{K}_{\boldsymbol{B}}]_p}, \overline{[\boldsymbol{K}_{\boldsymbol{B}}]_q} \right\rangle - \left\langle \overline{[\boldsymbol{K}_{\boldsymbol{A}}]_p}, \overline{[\boldsymbol{K}_{\boldsymbol{B}}]_q} \right\rangle \right) \\
&\quad + \frac{2K}{1+K} \frac{\sigma_q^2}{N_{\boldsymbol{A}} N_{\boldsymbol{B}} d} \left( \left\langle \overline{[\boldsymbol{K}_{\boldsymbol{A}}]_p}, \overline{[\boldsymbol{K}_{\boldsymbol{A}}]_q} \right\rangle - \left\langle \overline{[\boldsymbol{K}_{\boldsymbol{A}}]_p}, \overline{[\boldsymbol{K}_{\boldsymbol{B}}]_q} \right\rangle \right) \\
&\quad \pm O\left( \frac{\sigma_{\max}^2}{N_{\boldsymbol{A}} N_{\boldsymbol{B}} d} \right) \rho_- \left\langle \overline{[\boldsymbol{K}_{\boldsymbol{A}}]_p}, \overline{[\boldsymbol{K}_{\boldsymbol{B}}]_q} \right\rangle \\
&\quad \pm O\left( \frac{\sigma_{\max}^2}{N_{\boldsymbol{A}} N_{\boldsymbol{B}} d} \left( \tau_t^2 d^2 \kappa_0 + d\sqrt{\delta_{\boldsymbol{A}/\boldsymbol{B}}} \right) \right).
\end{aligned}
$$

Interchange the roles of $p$ and $q$ and we obtain

$$\frac{\mathrm{d}}{\mathrm{d}t}\left\langle \overline{[K_B]_p}, \overline{[K_A]_q}\right\rangle = \frac{2K}{1+K}\frac{\sigma_q^2}{N_A N_B d}\left(\left\langle \overline{[K_B]_p}, \overline{[K_B]_q}\right\rangle - \left\langle \overline{[K_B]_p}, \overline{[K_A]_q}\right\rangle\right)$$
$$+ \frac{2K}{1+K}\frac{\sigma_p^2}{N_A N_B d}\left(\left\langle \overline{[K_A]_p}, \overline{[K_A]_q}\right\rangle - \left\langle \overline{[K_B]_p}, \overline{[K_A]_q}\right\rangle\right)$$
$$\pm O\left(\frac{\sigma_{\max}^2}{N_A N_B d}\right)\rho_-\left\langle \overline{[K_B]_p}, \overline{[K_A]_q}\right\rangle$$
$$\pm O\left(\frac{\sigma_{\max}^2}{N_A N_B d}\left(\tau_t^2 d^2 \kappa_0 + d\sqrt{\delta_{A/B}}\right)\right).$$

Similarly, we compute

$$\left\langle \frac{\mathrm{d}}{\mathrm{d}t}\overline{[K_A]_p}, \overline{[K_A]_q}\right\rangle = \frac{2K}{1+K}\frac{\sigma_p^2}{N_A N_B d}\left(\left\langle \overline{[K_B]_p}, \overline{[K_A]_q}\right\rangle - \left\langle \overline{[K_A]_p}, \overline{[K_A]_q}\right\rangle\left\langle \overline{[K_A]_p}, \overline{[K_B]_p}\right\rangle\right)$$
$$\pm O\left(\frac{\sigma_p^2}{N_A N_B d}\left(\tau_t^2 d^2 \kappa_0 + d\sqrt{\delta_{A/B}}\right)\right)$$
$$= \frac{2K}{1+K}\frac{\sigma_p^2}{N_A N_B d}\left(\left\langle \overline{[K_B]_p}, \overline{[K_A]_q}\right\rangle - \left\langle \overline{[K_A]_p}, \overline{[K_A]_q}\right\rangle\right)$$
$$+ \frac{2K}{1+K}\frac{\sigma_p^2}{N_A N_B d}\left\langle \overline{[K_A]_p}, \overline{[K_A]_q}\right\rangle\left(1 - \left\langle \overline{[K_A]_p}, \overline{[K_B]_p}\right\rangle\right)$$
$$\pm O\left(\frac{\sigma_p^2}{N_A N_B d}\left(\tau_t^2 d^2 \kappa_0 + d\sqrt{\delta_{A/B}}\right)\right)$$
$$= \frac{2K}{1+K}\frac{\sigma_p^2}{N_A N_B d}\left(\left\langle \overline{[K_B]_p}, \overline{[K_A]_q}\right\rangle - \left\langle \overline{[K_A]_p}, \overline{[K_A]_q}\right\rangle\right)$$
$$\pm O\left(\frac{\sigma_{\max}^2}{N_A N_B d}\right)\rho_-\left\langle \overline{[K_A]_p}, \overline{[K_A]_q}\right\rangle \pm O\left(\frac{\sigma_{\max}^2}{N_A N_B d}\left(\tau_t^2 d^2 \kappa_0 + d\sqrt{\delta_{A/B}}\right)\right).$$

Then, by interchanging the roles of $p$ and $q$, we obtain

$$\left\langle \overline{[K_A]_p}, \frac{\mathrm{d}}{\mathrm{d}t}\overline{[K_A]_q}\right\rangle = \frac{2K}{1+K}\frac{\sigma_q^2}{N_A N_B d}\left(\left\langle \overline{[K_A]_p}, \overline{[K_B]_q}\right\rangle - \left\langle \overline{[K_A]_p}, \overline{[K_A]_q}\right\rangle\right)$$
$$\pm O\left(\frac{\sigma_{\max}^2}{N_A N_B d}\right)\rho_-\left\langle \overline{[K_A]_p}, \overline{[K_A]_q}\right\rangle \pm O\left(\frac{\sigma_{\max}^2}{N_A N_B d}\left(\tau_t^2 d^2 \kappa_0 + d\sqrt{\delta_{A/B}}\right)\right).$$

Add them together and we get

$$\frac{\mathrm{d}}{\mathrm{d}t}\left\langle \overline{[K_A]_p}, \overline{[K_A]_q}\right\rangle = \frac{2K}{1+K}\frac{\sigma_p^2}{N_A N_B d}\left(\left\langle \overline{[K_B]_p}, \overline{[K_A]_q}\right\rangle - \left\langle \overline{[K_A]_p}, \overline{[K_A]_q}\right\rangle\right)$$
$$+ \frac{2K}{1+K}\frac{\sigma_q^2}{N_A N_B d}\left(\left\langle \overline{[K_A]_p}, \overline{[K_B]_q}\right\rangle - \left\langle \overline{[K_A]_p}, \overline{[K_A]_q}\right\rangle\right)$$
$$\pm O\left(\frac{\sigma_{\max}^2}{N_A N_B d}\right)\rho_-\left\langle \overline{[K_A]_p}, \overline{[K_A]_q}\right\rangle \pm O\left(\frac{\sigma_{\max}^2}{N_A N_B d}\left(\tau_t^2 d^2 \kappa_0 + d\sqrt{\delta_{A/B}}\right)\right).$$

Interchange the roles of $p$ and $q$ and we obtain

$$\frac{\mathrm{d}}{\mathrm{d}t}\left\langle \overline{[K_B]_p}, \overline{[K_B]_q}\right\rangle = \frac{2K}{1+K}\frac{\sigma_p^2}{N_A N_B d}\left(\left\langle \overline{[K_A]_p}, \overline{[K_B]_q}\right\rangle - \left\langle \overline{[K_B]_p}, \overline{[K_B]_q}\right\rangle\right)$$
$$+ \frac{2K}{1+K}\frac{\sigma_q^2}{N_A N_B d}\left(\left\langle \overline{[K_B]_p}, \overline{[K_A]_q}\right\rangle - \left\langle \overline{[K_B]_p}, \overline{[K_B]_q}\right\rangle\right)$$
$$\pm O\left(\frac{\sigma_{\max}^2}{N_A N_B d}\right)\rho_-\left\langle \overline{[K_B]_p}, \overline{[K_B]_q}\right\rangle \pm O\left(\frac{\sigma_{\max}^2}{N_A N_B d}\left(\tau_t^2 d^2 \kappa_0 + d\sqrt{\delta_{A/B}}\right)\right).$$

For notational simplicity, define $Z_1 = \left\langle \overline{[K_A]_p}, \overline{[K_B]_q} \right\rangle$, $Z_2 = \left\langle \overline{[K_B]_p}, \overline{[K_A]_q} \right\rangle$, $Z_3 = \left\langle \overline{[K_A]_p}, \overline{[K_A]_q} \right\rangle$, and $Z_4 = \left\langle \overline{[K_B]_p}, \overline{[K_B]_q} \right\rangle$. Also define $G_p = \frac{2K}{1+K} \frac{\sigma_p^2}{N_A N_B d}$. Then, we can summarize the above equations as

$$
\frac{\mathrm{d}}{\mathrm{d}t}
\begin{bmatrix} Z_1 \\ Z_2 \\ Z_3 \\ Z_4 \end{bmatrix}
=
\begin{bmatrix}
-G_p - G_q & 0 & G_q & G_p \\
0 & -G_p - G_q & G_p & G_q \\
G_q & G_p & -G_p - G_q & 0 \\
G_p & G_q & 0 & -G_p - G_q
\end{bmatrix}
\begin{bmatrix} Z_1 \\ Z_2 \\ Z_3 \\ Z_4 \end{bmatrix}
$$
$$
\pm O\left( \frac{\sigma_{\max}^2}{N_A N_B d} \right) \rho_-
\begin{bmatrix} Z_1 \\ Z_2 \\ Z_3 \\ Z_4 \end{bmatrix}
\pm O\left( \frac{\sigma_{\max}^2}{N_A N_B d} \left( \tau_t^2 d^2 \kappa_0 + d\sqrt{\delta_{A/B}} \right) \right).
$$

Note that the eigenvalues of the first matrix is $-2G_p - 2G_q$, $-2G_p$, $-2G_q$ and $0$. Namely, it is negative semi-definite. Thus,

$$
\frac{\mathrm{d}}{\mathrm{d}t} \|Z\|^2 \leq O\left( \frac{\sigma_{\max}^2}{N_A N_B d} \right) \rho_- \|Z\|^2 + O\left( \frac{\sigma_{\max}^2}{N_A N_B d} \left( \tau_t^2 d^2 \kappa_0 + d\sqrt{\delta_{A/B}} \right) \right).
$$

□

*Proof of Lemma B.16.* By Corollary B.8,

$$
\left\langle \frac{\mathrm{d}}{\mathrm{d}t} \overline{[K_A]_p}, \overline{[K_{A,\xi}]_q} \right\rangle = \frac{2K}{1+K} \frac{\sigma_p^2}{N_A N_B d} \left( \left\langle \overline{[K_B]_p}, \overline{[K_{A,\xi}]_q} \right\rangle - \left\langle \overline{[K_A]_p}, \overline{[K_{A,\xi}]_q} \right\rangle \left\langle \overline{[K_A]_p}, \overline{[K_B]_p} \right\rangle \right)
$$
$$
\pm O\left( \frac{\sigma_p^2}{N_A N_B d} \left( \tau_t^2 d^2 \kappa_0 + d\sqrt{\delta_{A/B}} \right) \right),
$$

and

$$
\left\langle \overline{[K_A]_p}, \frac{\mathrm{d}}{\mathrm{d}t} \overline{[K_{A,\xi}]_q} \right\rangle = \pm O\left( \frac{\sigma_\xi^2}{N_A N_B d} \tau_t^2 d^3 \kappa_0 \delta_{\xi,\perp} \right).
$$

Therefore,

$$
\frac{\mathrm{d}}{\mathrm{d}t} \left\langle \overline{[K_A]_p}, \overline{[K_{A,\xi}]_q} \right\rangle = \frac{2K}{1+K} \frac{\sigma_p^2}{N_A N_B d} \left( \left\langle \overline{[K_B]_p}, \overline{[K_{A,\xi}]_q} \right\rangle - \left\langle \overline{[K_A]_p}, \overline{[K_{A,\xi}]_q} \right\rangle \right)
$$
$$
\pm O\left( \frac{\sigma_p^2}{N_A N_B d} \right) \rho_- \left\langle \overline{[K_A]_p}, \overline{[K_{A,\xi}]_q} \right\rangle \pm O\left( \frac{\sigma_{\max}^2}{N_A N_B d} \left( \tau_t^2 d^2 \kappa_0 + d\sqrt{\delta_{A/B}} \right) \right).
$$

Similarly, we also have

$$
\frac{\mathrm{d}}{\mathrm{d}t} \left\langle \overline{[K_B]_p}, \overline{[K_{A,\xi}]_q} \right\rangle = \frac{2K}{1+K} \frac{\sigma_p^2}{N_A N_B d} \left( \left\langle \overline{[K_A]_p}, \overline{[K_{A,\xi}]_q} \right\rangle - \left\langle \overline{[K_B]_p}, \overline{[K_{A,\xi}]_q} \right\rangle \right)
$$
$$
\pm O\left( \frac{\sigma_p^2}{N_A N_B d} \right) \rho_- \left\langle \overline{[K_B]_p}, \overline{[K_{A,\xi}]_q} \right\rangle \pm O\left( \frac{\sigma_{\max}^2}{N_A N_B d} \left( \tau_t^2 d^2 \kappa_0 + d\sqrt{\delta_{A/B}} \right) \right).
$$

Thus,

$$
\frac{\mathrm{d}}{\mathrm{d}t}\left(\left\langle[\boldsymbol{K_A}]_p,[\boldsymbol{K_{A,\xi}}]_q\right\rangle^2 + \left\langle[\boldsymbol{K_B}]_p,[\boldsymbol{K_{A,\xi}}]_q\right\rangle^2\right)
$$

$$
= -\frac{4K}{1+K}\frac{\sigma_p^2}{N_A N_B d}\left(\left\langle[\boldsymbol{K_A}]_p,[\boldsymbol{K_{A,\xi}}]_q\right\rangle^2 - \left\langle[\boldsymbol{K_B}]_p,[\boldsymbol{K_{A,\xi}}]_q\right\rangle^2\right)^2
$$

$$
\pm O\left(\frac{\sigma_p^2}{N_A N_B d}\right)\rho_-\left(\left\langle\overline{[\boldsymbol{K_A}]_p},\overline{[\boldsymbol{K_{A,\xi}}]_q}\right\rangle^2 + \left\langle\overline{[\boldsymbol{K_B}]_p},\overline{[\boldsymbol{K_{A,\xi}}]_q}\right\rangle^2\right)
$$

$$
\pm O\left(\frac{\sigma_{\max}^2}{N_A N_B d}\left(\tau_t^2 d^2 \kappa_0 + d\sqrt{\delta_{A/B}}\right)\right)
$$

$$
\leq O\left(\frac{\sigma_p^2}{N_A N_B d}\right)\rho_-\left(\left\langle\overline{[\boldsymbol{K_A}]_p},\overline{[\boldsymbol{K_{A,\xi}}]_q}\right\rangle^2 + \left\langle\overline{[\boldsymbol{K_B}]_p},\overline{[\boldsymbol{K_{A,\xi}}]_q}\right\rangle^2\right)
$$

$$
\pm O\left(\frac{\sigma_{\max}^2}{N_A N_B d}\left(\tau_t^2 d^2 \kappa_0 + d\sqrt{\delta_{A/B}}\right)\right).
$$

For the orthogonality between noises, by Corollary B.8, we have

$$
\left\langle\overline{[\boldsymbol{K_{A,\xi}}]_q},\frac{\mathrm{d}}{\mathrm{d}t}\overline{[\boldsymbol{K_{A,\xi}}]_s}\right\rangle = \pm O\left(\frac{\sigma_\xi^2}{N_A N_B d}\tau_t^2 d^3 \kappa_0 \delta_{\xi,\perp}\right)
$$

Clear that this bound holds for all other combinations. Thus,

$$
\frac{\mathrm{d}}{\mathrm{d}t}\hat{\delta}_{\perp,\xi,p,q} \leq O\left(\frac{\sigma_\xi^2}{N_A N_B d}\tau_t^2 d^3 \kappa_0 \delta_{\xi,\perp}\sqrt{\hat{\delta}_{\perp,\xi,p,q}}\right).
$$

$\square$

### B.4 PROOF OF THE MAIN LEMMA OF STAGE 1

*Proof of Lemma B.1.* First, we recap the estimations we have derived in previous subsections and introduce some notations. By Lemma B.10 and Lemma B.11, we have

$$
\frac{\mathrm{d}}{\mathrm{d}t}\rho_- \leq -\Omega(1)\frac{\sigma_{\min}^2}{N_A N_B d}\rho_- + O\left(\frac{\sigma_{\max}^2}{N_A N_B d}\left(\tau_t^2 d^2 \kappa_0 + \sqrt{\delta_{A/B}}\right)\right),
$$

$$
\frac{\mathrm{d}}{\mathrm{d}t}\hat{\rho}_{N/S} \leq -\Omega(1)\frac{\sigma_{\min}^2}{N_A N_B d}\hat{\rho}_{N/S} + O\left(\frac{\sigma_{\max}^2}{N_A N_B d}\left(d\tau_t^2 + \sqrt{\delta_{A/B}}\right)d\right).
$$

Define $\rho := \max\left\{\rho_-,\hat{\rho}_{N/S}\right\}$ to be the indicator of the progress we have made. We have

$$
\frac{\mathrm{d}}{\mathrm{d}t}\rho \leq -\Omega(1)\frac{\sigma_{\min}^2}{N_A N_B d}\rho + O\left(\frac{\sigma_{\max}^2}{N_A N_B d}\left(\tau_t^2 d^2 \kappa_0 + d\sqrt{\delta_{A/B}}\right).\right). \tag{11}
$$

Let $\hat{\kappa}_0 := \max_{p,q}\|[\boldsymbol{K_A}]_p\|^2 / \|[\boldsymbol{K_A}]_q\|^2$ be the condition number at time $t$. By Lemma B.12, we have

$$
\frac{\mathrm{d}}{\mathrm{d}t}\hat{\kappa}_0 \leq O(1)\frac{\sigma_{\max}^2}{N_A N_B d}\rho\hat{\kappa}_0. \tag{12}
$$

Now we consider the discretization errors, define

$$
\hat{\delta}_{A/B} = \max_{p\in[r],q\in[d-r]}\left\{\rho_{A/B,p} + \rho_{B/A,p} - 2, \rho_{A/B,\xi,q} + \rho_{B/A,\xi,q} - 2\right\}.
$$

Note that at time $t$, the first condition of (10) holds with $\delta_{A/B}$ replaced by $O(\hat{\delta}_{A/B})$. Meanwhile, by Lemma B.13, we have

$$
\frac{\mathrm{d}}{\mathrm{d}t}\hat{\delta}_{A/B} \leq O\left(\frac{\sigma_{\max}^2}{N_A N_B d}\left(\delta_{A/B}^2 + \tau_t^2 d\right)\right). \tag{13}
$$

Let $\hat{\delta}_{\xi,\kappa_0}(t)$ be the smallest number such that the second condition of (10) holds at time $t$. By Lemma B.14, we have

$$\frac{\mathrm{d}}{\mathrm{d}t}\hat{\delta}_{\xi,\kappa_0} \leq O\left(\frac{\sigma_\xi^2}{N_{\boldsymbol{A}}N_{\boldsymbol{B}}d}\left(d\tau_t^2 + \sqrt{\delta_{\boldsymbol{A}/\boldsymbol{B}}}\right)\right). \tag{14}$$

Then, define

$$\hat{\delta}_\perp := \max\left\{\hat{\delta}_{\perp,p,q}, \hat{\delta}_{\perp,\xi_{\boldsymbol{A}},p,k}, \hat{\delta}_{\perp,\xi_{\boldsymbol{B}},p,k}, \hat{\delta}_{\perp,\xi,k,l}, : p \neq q \in [d], k, l \in [d-r]\right\}$$

Clear that the last two conditions hold at time $t$ when $\delta_{\boldsymbol{AB},\perp}$ and $\delta_{\xi,\perp}$ are replaced by $\sqrt{\hat{\delta}_\perp(t)}$. By Lemma B.15 and Lemma B.16, we have

$$\frac{\mathrm{d}}{\mathrm{d}t}\hat{\delta}_\perp \leq O\left(\frac{\sigma_{\max}^2}{N_{\boldsymbol{A}}N_{\boldsymbol{B}}d}\right)\rho\hat{\delta}_\perp + O\left(\frac{\sigma_{\max}^2}{N_{\boldsymbol{A}}N_{\boldsymbol{B}}d}\left(\tau_t^2 d^2\kappa_0 + d\sqrt{\delta_{\boldsymbol{A}/\boldsymbol{B}}}\right)\right). \tag{15}$$

Now, we are ready to show that the errors do not blow up in Stage 1. Note that for all these $\delta$'s, we can make them arbitrarily inverse-polynomially small by choosing a sufficiently large $m$.

First, we consider $\hat{\delta}_{\boldsymbol{A}/\boldsymbol{B}}$. Note that the dependence of the RHS of (13) on $\hat{\delta}_{\boldsymbol{A}/\boldsymbol{B}}$ is quadratic. Hence, by making the initial value of $\hat{\delta}_{\boldsymbol{A}/\boldsymbol{B}}$, the RHS can be made to be dominated the $\tau_t^2$-related terms. Hence, $\hat{\delta}_{\boldsymbol{A}/\boldsymbol{B}} \leq O\left(\frac{\sigma_{\max}^2}{N_{\boldsymbol{A}}N_{\boldsymbol{B}}d}\tau_t^2 dT_1\right)$. As we will see later, $T_1 = \mathrm{poly}(d)$. Therefore, by choosing a sufficiently small $\tau_t^2$, we can make $\hat{\delta}_{\boldsymbol{A}/\boldsymbol{B}}$ remain small throughout Stage 1.

Then, we consider $\hat{\delta}_{\xi,\kappa_0}$. As we have argued earlier, the RHS of (14) can be made arbitrarily small, so that $\hat{\delta}_{\xi,\kappa_0}$ remains small in Stage 1.

Now, we consider the condition number $\hat{\kappa}_0$ and $\hat{\delta}_\perp$. For $\hat{\delta}_\perp$, by our previous argument, the second term of the RHS of (15) can be merged into the first term, by choosing a sufficiently large $m$ and a sufficiently small $\tau_t^2$. The same is also true for (11). Hence, for these quantities, we have

$$\frac{\mathrm{d}}{\mathrm{d}t}\rho \leq -\Omega(1)\frac{\sigma_{\min}^2}{N_{\boldsymbol{A}}N_{\boldsymbol{B}}d}\rho, \quad \frac{\mathrm{d}}{\mathrm{d}t}\hat{\kappa}_0 \leq O(1)\frac{\sigma_{\max}^2}{N_{\boldsymbol{A}}N_{\boldsymbol{B}}d}\rho\hat{\kappa}_0, \quad \frac{\mathrm{d}}{\mathrm{d}t}\hat{\delta}_\perp \leq O(1)\frac{\sigma_{\max}^2}{N_{\boldsymbol{A}}N_{\boldsymbol{B}}d}\rho\hat{\delta}_\perp.$$

Hence, by Lemma B.2, we have

$$\hat{\kappa}_0 \leq \hat{\kappa}_0(0)\exp\left(O\left(\frac{\sigma_{\max}^2}{\sigma_{\min}^2}\right)\rho(0)\right), \quad \hat{\delta}_\perp \leq \hat{\delta}_\perp(0)\exp\left(O\left(\frac{\sigma_{\max}^2}{\sigma_{\min}^2}\right)\rho(0)\right).$$

Note that $\rho_-(0) = O(1)$ and $\hat{\rho}_{N/S} \leq \frac{(d-r)\sigma_\xi^2}{r\sigma_{\min}^2}$. Therefore,

$$\exp\left(O\left(\frac{\sigma_{\max}^2}{\sigma_{\min}^2}\right)\rho(0)\right) \leq \exp\left(O(1)\frac{\sigma_{\max}^2}{\sigma_{\min}^2}\max\left\{1, \frac{(d-r)\sigma_\xi^2}{r\sigma_{\min}^2}\right\}\right) \leq \exp\left(\frac{1}{2}\log d\right) = \sqrt{d}.$$

In other words, both $\hat{\kappa}_0$ and $\hat{\delta}_\perp$ can at most grow $\sqrt{d}$ times.

Finally, we derive an upper bound on $T_1$ to complete the proof. Similar to the proof for the condition number, one can show that $\|[\boldsymbol{K}_{\boldsymbol{A}}]_p\|^2$ can at most grow $\sqrt{d}$ times in Stage 1. As a result, $1/(N_{\boldsymbol{A}}N_{\boldsymbol{B}}d)$ is lower bounded by some $1/\mathrm{poly}(d)$. Thus, by (11), $T_1 \leq \mathrm{poly}(d)$. $\qquad\square$

## B.5 NEGATIVE RESULTS

**Lemma B.17.** *There exists a $\boldsymbol{\sigma} \in \mathbb{R}^r$ satisfying the assumptions of Theorem 4.3 such that, at the end of Stage 1, the condition number of $\boldsymbol{K}_{\boldsymbol{A}}$ is $d^{\Omega(1)}$.*

*Proof.* We choose $d = r$ and $\sigma_1^2 = c\log d, \sigma_2^2 = \cdots = \sigma_d^2 = 1$. Clear that this satisfies the condition of Theorem 4.3. Note that it suffices to consider the infinite-width case, since, as we have proved earlier, the finite-width trajectory tracks the infinite-width one. By Lemma B.10, we have

$$\frac{\mathrm{d}}{\mathrm{d}t}\left\langle\overline{[\boldsymbol{K}_{\boldsymbol{A}}]_p}, \overline{[\boldsymbol{K}_{\boldsymbol{B}}]_p}\right\rangle \approx \frac{4K}{1+K}\frac{\sigma_p^2}{N_{\boldsymbol{A}}N_{\boldsymbol{B}}d}\left(1 + \left\langle\overline{[\boldsymbol{K}_{\boldsymbol{A}}]_p}, \overline{[\boldsymbol{K}_{\boldsymbol{B}}]_p}\right\rangle\right)\left(1 - \left\langle\overline{[\boldsymbol{K}_{\boldsymbol{A}}]_p}, \overline{[\boldsymbol{K}_{\boldsymbol{B}}]_p}\right\rangle\right).$$

By the proof of Lemma B.12, we have

$$\dot{\rho}_{p/q} \approx \frac{4K}{1+K} \frac{\sigma_p^2}{N_A N_B d} \left( \left\langle \overline{[K_A]_p}, \overline{[K_B]_p} \right\rangle - \frac{\langle K_A, K_B \rangle}{N_A N_B d} \right) \rho_{p/q}$$
$$- \frac{4K}{1+K} \frac{\sigma_q^2}{N_A N_B d} \left( \left\langle \overline{[K_A]_q}, \overline{[K_B]_q} \right\rangle - \frac{\langle K_A, K_B \rangle}{N_A N_B d} \right) \rho_{p/q}.$$

Note that, in the infinite-width case, we have

$$\left\langle \overline{[K_A]_1}, \overline{[K_B]_1} \right\rangle \geq \left\langle \overline{[K_A]_2}, \overline{[K_B]_2} \right\rangle = \cdots = \left\langle \overline{[K_A]_d}, \overline{[K_B]_d} \right\rangle.$$

Therefore, $\left\langle \overline{[K_A]_p}, \overline{[K_B]_p} \right\rangle - \frac{\langle K_A, K_B \rangle}{N_A N_B d} \geq 0$a and $\left\langle \overline{[K_A]_q}, \overline{[K_B]_q} \right\rangle - \frac{\langle K_A, K_B \rangle}{N_A N_B d} \leq 0$ for any $q \geq 2$. Hence,

$$\dot{\rho}_{1/2} \geq \frac{4K}{1+K} \frac{\sigma_1^2}{N_A N_B d} \left( \left\langle \overline{[K_A]_1}, \overline{[K_B]_1} \right\rangle - \frac{\langle K_A, K_B \rangle}{N_A N_B d} \right) \rho_{1/2}$$
$$\geq \frac{4K}{1+K} \frac{\sigma_1^2}{N_A N_B d} \left( \left( 1 - \frac{\kappa_1^2}{\|\kappa\|^2} \right) \left\langle \overline{[K_A]_1}, \overline{[K_B]_1} \right\rangle - \frac{(d-1)\kappa_1^2}{\|\kappa\|^2} \left\langle \overline{[K_A]_2}, \overline{[K_B]_2} \right\rangle \right) \rho_{1/2}.$$

For notational simplicity, define $X_1 = 1 - \left\langle \overline{[K_A]_1}, \overline{[K_B]_1} \right\rangle$, $X_2 = 1 - \left\langle \overline{[K_A]_2}, \overline{[K_B]_2} \right\rangle$, $Y = \rho_{1/2}$, $A = \frac{4K}{1+K} \frac{1}{N_A N_B d}$. Then we have

$$\dot{X}_1 \leq -\sigma_1^2 A X_1, \quad \dot{X}_2 \geq -2A X_2.$$

First, by Gronwall's lemma, we have $X_1(T) \leq \exp\left( -\sigma_1^2 \int_0^T A \right)$ and

$$X_2(T) \geq \exp\left( -2 \int_0^T A \right) \geq X_1^{2/\sigma_1^2}(T).$$

As a result, when $X_1$ reaches $1/d$, we have $X_2 = \Omega(1)$. Let $T_1$ be the time $X_1$ reaches $1/d$ and $T_2$ the time $X_2(T_2) = X_2(T_1)/2$. On $[T_1, T_2]$, we have

$$\dot{\rho}_{1/2} \geq \Omega(1)\sigma_1^2 A \rho_{1/2}.$$

By Gronwall's lemma, in order for $X_2$ to half, we need $\exp(-2 \int_{T_1}^{T_2} A) = 1/2$. Hence,

$$\rho_{1/2}(T_2) \geq \rho_{1/2}(T_1) \exp\left( \Omega(1)\sigma_1^2 \int_{T_1}^{T_2} A \right) \geq 2^{\Omega(\sigma_1^2)} = d^{\Omega(1)}.$$

$\square$

## C   STAGE 2

In this section, we show that, throughout Stage 2, the discretization error and the noise-signal ratio still remain small, and, at the end of Stage 2, the condition number is close to 1. Formally, we prove the following.

**Lemma C.1** (Stage 2). *Suppose that at the beginning of Stage 2, we have $\kappa_0 \leq \sqrt{d}$ and all errors mentioned in (16) are sufficiently small[5]. Let $c_{Target} > 1$ be a constant. Let $T_1$ be the earliest time that*

$$\frac{\|[K_A]_p\|^2}{\|[K_A]_q\|^2} \leq c_{Target}, \quad \forall p, q \in [r].$$

---

[5]By Lemma B.1, this condition indeed holds.

*We have $T_1 \leq \mathrm{poly}(d)$. Moreover, throughout Stage 2, we have*

$$\max\left\{1 - \left\langle \overline{[\boldsymbol{K_A}]_p}, \overline{[\boldsymbol{K_B}]_p} \right\rangle, \left| 1 - \frac{\|[\boldsymbol{K_A}]_p\|^2}{\|[\boldsymbol{K_B}]_p\|^2} \right| \right\} \leq \delta_- \qquad\qquad \forall p \in [r],$$

$$\max\left\{ \left| 1 - \frac{\|[\boldsymbol{K_{A,\xi}}]_p\|^2}{\|[\boldsymbol{K_{B,\xi}}]_p\|^2} \right|, \left| 1 - \frac{\|[\boldsymbol{K_{A,\xi}}]_p\|^2}{\|[\boldsymbol{K_{A,\xi}}]_q\|^2} \right| \right\} \leq \delta_- \qquad\qquad \forall p, q \in [d-r],$$

$$\max\left\{ \frac{\|[\boldsymbol{K_{C,\xi}}]_q\|}{\|[\boldsymbol{K_D}]_p\|} \ : \ \boldsymbol{C}, \boldsymbol{D} \in \{\boldsymbol{A}, \boldsymbol{B}\} \right\} \leq \delta_{N/S}, \qquad\qquad \forall p \in [r], q \in [d-r],$$

$$\max\left\{ \left| \left\langle \overline{[\boldsymbol{K_C}]_p}, \overline{[\boldsymbol{K_D}]_q} \right\rangle \right| \ : \ \boldsymbol{C}, \boldsymbol{D} \in \{\boldsymbol{A}, \boldsymbol{B}\} \right\} \leq \delta_{\boldsymbol{AB},\perp}, \qquad\qquad \forall p \neq q \in [r],$$

$$\max\left\{ \left| \left\langle \overline{[\boldsymbol{K_C}]_p}, \overline{[\boldsymbol{K_{D,\xi}}]_q} \right\rangle \right|, \left| \left\langle \overline{[\boldsymbol{K_{A,\xi}}]_s}, \overline{[\boldsymbol{K_{B,\xi}}]_q} \right\rangle \right| \ : \ \boldsymbol{C}, \boldsymbol{D} \in \{\boldsymbol{A}, \boldsymbol{B}\} \right\} \leq \delta_{\xi,\perp}, \quad \forall p \in [r], q, s \in [d-r],$$
$$\tag{16}$$

*where the $\delta$'s are some small $1/\mathrm{poly}(d)$ values.*

The rest of this section is organized as follows. We derive estimations for the $\boldsymbol{Q}$-matrices in Section C.1. In Section C.2, we maintain the last two conditions of (16). In Section C.3, we handle the first two conditions of (16). In Section C.4, we deal with the noise-signal ratio. We estimate the convergence rate in Section C.5. Finally, we prove Lemma C.1 in Section C.6.

### C.1 ESTIMATIONS FOR $Q$

As in Stage 1, we estimate the $\boldsymbol{Q}$-matrices in this subsection. The analysis here will be more complicated than the one in Section B.1 since now $\tau_t^2$ is no longer close to 0, and we cannot simply approximate $S_{\boldsymbol{A}}$ and $S_{\boldsymbol{B}}$ with $(1+K)^{-1}$. However, the idea is still fairly straightforward. We split all terms into the infinite-width part and the discretization error part. Then we Taylor expand the corresponding function around the infinite-width part to factor out the first-order error terms. Then, we evaluate and simplify these first-order terms with $\mathbb{E}_{\boldsymbol{z}^-}$ and $\mathbb{E}_{\boldsymbol{z}^\pm}$.

First, we need the following lemma which gives closed-form formulas for some expectations we will encounter later.

**Lemma C.2.** *Define $\langle \boldsymbol{z}^+, \boldsymbol{z}^- \rangle_{\hat{\boldsymbol{\kappa}}^2} := \sum_{k=1}^r \hat{\kappa}_k^2 z_k^+ z_k^-$ and $T_p := \tanh\left( \frac{\hat{\kappa}_p^2/d}{N_{\boldsymbol{A}} N_{\boldsymbol{B}}} \right)$, $p \in [r]$. For any $p \neq q \in [r]$, we have*

$$\mathbb{E}_{\boldsymbol{z}^-}\left\{ \exp\left( \frac{\langle \boldsymbol{z}^+, \boldsymbol{z}^- \rangle_{\hat{\boldsymbol{\kappa}}^2}}{N_{\boldsymbol{A}} N_{\boldsymbol{B}}} \right) \right\} = \prod_{k=1}^r \cosh\left( \frac{\hat{\kappa}_k^2/d}{N_{\boldsymbol{A}} N_{\boldsymbol{B}}} \right) =: Z_c,$$

$$\mathbb{E}_{\boldsymbol{z}^-}\left\{ \exp\left( \frac{\langle \boldsymbol{z}^+, \boldsymbol{z}^- \rangle_{\hat{\boldsymbol{\kappa}}^2}}{N_{\boldsymbol{A}} N_{\boldsymbol{B}}} \right) z_p^- \right\} = Z_c T_p z_p^+,$$

$$\mathbb{E}_{\boldsymbol{z}^-}\left\{ \exp\left( \frac{\langle \boldsymbol{z}^+, \boldsymbol{z}^- \rangle_{\hat{\boldsymbol{\kappa}}^2}}{N_{\boldsymbol{A}} N_{\boldsymbol{B}}} \right) z_p^- z_q^- \right\} = Z_c T_p T_q z_p^+ z_q^+.$$

Then, we derive estimations for $S_{\boldsymbol{A}}$ and $S_{\boldsymbol{B}}$. There are two types of errors we need to consider. The first one comes from the noises and the second one from the non-diagonalness of $\boldsymbol{K_A}^\top \boldsymbol{K_B}$. Similar to Lemma B.4, we deal with them separately. The next lemma handles the first type of error.

**Lemma C.3** (Estimations for $S$). *Define*

$$E_{+,-} := \exp\left( \frac{\langle \boldsymbol{K_A} \boldsymbol{z}^+, \boldsymbol{K_B} \boldsymbol{z}^- \rangle}{N_{\boldsymbol{A}} N_{\boldsymbol{B}}} \right), \quad \tilde{E}_{+,-} := \exp\left( \frac{\langle \boldsymbol{z}^+, \boldsymbol{z}^- \rangle_{\hat{\boldsymbol{\kappa}}^2}}{N_{\boldsymbol{A}} N_{\boldsymbol{B}}} \right),$$

$$\delta_{+,\xi_-} := \frac{\langle \boldsymbol{K_A} \boldsymbol{z}^+, \boldsymbol{K_{B,\xi}} \boldsymbol{\xi}_{\boldsymbol{B}}^- \rangle}{N_{\boldsymbol{A}} N_{\boldsymbol{B}}}, \quad \delta_{\xi+,T+} := \frac{\langle \boldsymbol{K_{A,\xi}} \boldsymbol{\xi}_{\boldsymbol{A}}^+, \boldsymbol{K_B} \mathrm{diag}([T_k]_{k \in [r]}) \boldsymbol{z}^+ \rangle}{N_{\boldsymbol{A}} N_{\boldsymbol{B}}},$$

$$Z_{\boldsymbol{A},c} := \mathbb{E}_{\boldsymbol{z}^-} E_{+,-}, \quad Z_{\boldsymbol{B},c} := \mathbb{E}_{\boldsymbol{z}^-} E_{-,+},$$

$$\tilde{S}_{\boldsymbol{A}} := \frac{E_{+,+}}{E_{+,+} + K Z_{\boldsymbol{A},c}}, \quad \tilde{S}_{\boldsymbol{B}} := \frac{E_{+,+}}{E_{+,+} + K Z_{\boldsymbol{B},c}}, \quad \tilde{S} := \frac{\tilde{E}_{+,+}}{\tilde{E}_{+,+} + K Z_c}.$$

*In Stage 2, we have*

$$S_{\boldsymbol{A}} = \tilde{S}_{\boldsymbol{A}} + \tilde{S}(1-\tilde{S})\left(\delta_{+,\xi+} + \delta_{\xi+,+} - \delta_{\xi+,T+}\right) \pm O\left(d^3\left(\delta_{\boldsymbol{AB},\perp} + \delta_{N/S}\right)\delta_{\xi,\perp}\delta_{N/S}\right),$$

$$S_{\boldsymbol{B}} = \tilde{S}_{\boldsymbol{B}} + \tilde{S}(1-\tilde{S})\left(\delta_{+,\boldsymbol{\xi}+} + \delta_{\boldsymbol{\xi}+,+} - \delta_{T+,\boldsymbol{\xi}+}\right) \pm O\left(d^3\left(\delta_{\boldsymbol{AB},\perp} + \delta_{N/S}\right)\delta_{\xi,\perp}\delta_{N/S}\right),$$

*and*

$$\frac{S_{\boldsymbol{A}}(\boldsymbol{x}_{\boldsymbol{A}}^+, \boldsymbol{x}_{\boldsymbol{B}}^+)\exp(\boldsymbol{f}_{\boldsymbol{A}}^+ \cdot \boldsymbol{f}_{\boldsymbol{B}}^-)}{\exp(\boldsymbol{f}_{\boldsymbol{A}}^+ \cdot \boldsymbol{f}_{\boldsymbol{B}}^+)} = \frac{\tilde{S}_{\boldsymbol{A}}E_{+,-}}{E_{+,+}} + \frac{\tilde{S}\tilde{E}_{+,-}}{\tilde{E}_{+,+}}\left(\delta_{+,\boldsymbol{\xi}-} + \delta_{\boldsymbol{\xi}+,-} - \tilde{S}\left(\delta_{+,\boldsymbol{\xi}+} + \delta_{\boldsymbol{\xi}+,+}\right) - (1-\tilde{S})\delta_{\boldsymbol{\xi}+,T+}\right)$$

$$\pm O\left(d^3\left(\delta_{\boldsymbol{AB},\perp} + \delta_{N/S}\right)\delta_{\xi,\perp}\delta_{N/S}\right),$$

$$\frac{S_{\boldsymbol{B}}(\boldsymbol{x}_{\boldsymbol{A}}^+, \boldsymbol{x}_{\boldsymbol{B}}^+)\exp(\boldsymbol{f}_{\boldsymbol{A}}^- \cdot \boldsymbol{f}_{\boldsymbol{B}}^+)}{\exp(\boldsymbol{f}_{\boldsymbol{A}}^+ \cdot \boldsymbol{f}_{\boldsymbol{B}}^+)} = \frac{\tilde{S}_{\boldsymbol{B}}E_{-,+}}{E_{+,+}} + \frac{\tilde{S}\tilde{E}_{-,+}}{\tilde{E}_{+,+}}\left(1 + \delta_{\boldsymbol{\xi}-,+} + \delta_{-,\boldsymbol{\xi}+} - \tilde{S}\left(\delta_{+,\boldsymbol{\xi}+} + \delta_{\boldsymbol{\xi}+,+}\right) - (1-\tilde{S})\delta_{T+,\boldsymbol{\xi}+}\right)$$

$$\pm O\left(d^3\left(\delta_{\boldsymbol{AB},\perp} + \delta_{N/S}\right)\delta_{\xi,\perp}\delta_{N/S}\right).$$

Then, we consider the error comes from the non-diagonalness of $\boldsymbol{K}_{\boldsymbol{A}}^\top\boldsymbol{K}_{\boldsymbol{B}}$.

**Lemma C.4** (Further estimations for $S$). *Define*

$$\tilde{\delta}_{+,-} = \sum_{i \neq j}\frac{\langle[\boldsymbol{K}_{\boldsymbol{A}}]_i, [\boldsymbol{K}_{\boldsymbol{B}}]_j\rangle}{N_{\boldsymbol{A}}N_{\boldsymbol{B}}}z_i^+ z_j^-, \quad \tilde{\delta}_{+,T+} = \sum_{i \neq j}\frac{\langle[\boldsymbol{K}_{\boldsymbol{A}}]_i, [\boldsymbol{K}_{\boldsymbol{B}}]_j\rangle}{N_{\boldsymbol{A}}N_{\boldsymbol{B}}}z_i^+ T_j z_j^+,$$

$$\tilde{E}_0 := \tilde{E}_{+,+} = \tilde{E}_{-,-} = \exp\left(\frac{\|\hat{\boldsymbol{\kappa}}\|^2}{N_{\boldsymbol{A}}N_{\boldsymbol{B}}}\right).$$

*In Stage 2, we have*

$$\tilde{S}_{\boldsymbol{A}}(\boldsymbol{z}^+) = \tilde{S}\left(1 + (1-\tilde{S})(\tilde{\delta}_{+,+} - \tilde{\delta}_{+,T+})\right) \pm O\left(d^2\delta_{\boldsymbol{AB},\perp}^2\right),$$

$$\tilde{S}_{\boldsymbol{B}}(\boldsymbol{z}^+) = \tilde{S}\left(1 + (1-\tilde{S})(\tilde{\delta}_{+,+} - \tilde{\delta}_{T+,+})\right) \pm O\left(d^2\delta_{\boldsymbol{AB},\perp}^2\right),$$

$$\frac{\tilde{S}_{\boldsymbol{A}}E_{+,-}}{E_{+,+}} = \frac{\tilde{S}\tilde{E}_{+,-}}{\tilde{E}_0}\left(1 - \tilde{S}\tilde{\delta}_{+,+} - (1-\tilde{S})\tilde{\delta}_{+,T+} + \tilde{\delta}_{+,-}\right) \pm O\left(d^2\delta_{\boldsymbol{AB},\perp}^2\right),$$

$$\frac{\tilde{S}_{\boldsymbol{B}}E_{-,+}}{E_{+,+}} = \frac{\tilde{S}\tilde{E}_{-,+}}{\tilde{E}_0}\left(1 - \tilde{S}\tilde{\delta}_{+,+} - (1-\tilde{S})\tilde{\delta}_{T+,+} + \tilde{\delta}_{-,+}\right) \pm O\left(d^2\delta_{\boldsymbol{AB},\perp}^2\right).$$

With the above two lemmas in hand, we can now derive estimations for the $\boldsymbol{Q}$-matrices.

**Lemma C.5** (Estimations for $\boldsymbol{Q}_1$). *Define $\boldsymbol{K}_{\boldsymbol{AB}} = \boldsymbol{K}_{\boldsymbol{A}}^\top\boldsymbol{K}_{\boldsymbol{B}}$ and $\boldsymbol{K}_{\boldsymbol{BA}} = \boldsymbol{K}_{\boldsymbol{B}}^\top\boldsymbol{K}_{\boldsymbol{A}}$. In Stage 2, for any $p \neq q \in [r]$, we have*

$$[\boldsymbol{Q}_1]_{p,p} = 2(1-\tilde{S})(1-T_p) \pm O\left(d^2\delta_{\boldsymbol{AB},\perp}^2\right),$$

$$[\boldsymbol{Q}_1]_{p,q} = -\tilde{S}(1-\tilde{S})\frac{[\boldsymbol{K}_{\boldsymbol{AB}}]_{p,q} + [\boldsymbol{K}_{\boldsymbol{BA}}]_{q,p}}{N_{\boldsymbol{A}}N_{\boldsymbol{B}}d}(2 - T_p - T_q)$$

$$- (1-\tilde{S})\frac{[\boldsymbol{K}_{\boldsymbol{AB}}]_{p,q}}{N_{\boldsymbol{A}}N_{\boldsymbol{B}}d}\tilde{S}\left(2T_pT_q - T_p - T_q\right)$$

$$- (1-\tilde{S})\frac{[\boldsymbol{K}_{\boldsymbol{AB}}]_{q,p}}{N_{\boldsymbol{A}}N_{\boldsymbol{B}}d}\left(2 - \tilde{S}(T_p + T_q) - (1-\tilde{S})(T_p^2 + T_q^2)\right) \pm O\left(d^2\delta_{\boldsymbol{AB},\perp}^2\right).$$

*In particular, we have*

$$|[\boldsymbol{Q}_1]_{p,q}| \leq O\left(\frac{\kappa_0\delta_{\boldsymbol{AB},\perp}}{d}\right) \leq O\left(\frac{\delta_{\boldsymbol{AB},\perp}}{\sqrt{d}}\right).$$

Note that the diagonal term has a zero-order term, i.e., it is not proportional to some error. That is the signal term. On the other hand, all off-diagonal terms depend on $[\boldsymbol{K}_{\boldsymbol{AB}}]_{p,q}$ ($p \neq q$). Recall that the dynamics of $\boldsymbol{K}_{\boldsymbol{A}}$ and $\boldsymbol{K}_{\boldsymbol{B}}$ can be described using these $\boldsymbol{Q}$-matrices. Therefore, for any $p \neq q$, we can have equations of form $\frac{d}{dt}[\boldsymbol{K}_{\boldsymbol{AB}}]_{p,q} \approx \boldsymbol{G}([\boldsymbol{K}_{\boldsymbol{AB}}]_{p,q}, [\boldsymbol{K}_{\boldsymbol{AB}}]_{q,p})$, where $\boldsymbol{G}$ is some complicated matrix. By carefully analyzing $\boldsymbol{G}$, we can then derive bounds for the off-diagonal entries using Gronwall's lemma.

Similar things also hold for $\boldsymbol{Q}_{1,\xi}$ and $\boldsymbol{Q}_2$. The difference here is that for these two matrices, we do not have signal terms.

**Lemma C.6** (Estimations for $\boldsymbol{Q}_{1,\xi}$). *In Stage 2, for any $p \in [d-r]$ and $q \in [r]$, we have*

$$[\boldsymbol{Q}_{1,\boldsymbol{\xi_A}}]_{p,q} = -(1-\tilde{S})\left(1+\tilde{S}+(1-\tilde{S})T_q^2\right)\frac{\langle[\boldsymbol{K_B}]_q, [\boldsymbol{K_{A,\xi}}]_p\rangle}{N_{\boldsymbol{A}}N_{\boldsymbol{B}}d} \pm O\left(d^3\left(\delta_{\boldsymbol{AB},\perp}+\delta_{N/S}\right)\delta_{\xi,\perp}\delta_{N/S}\right),$$

$$[\boldsymbol{Q}_{1,\boldsymbol{\xi_B}}]_{p,q} = -(1-\tilde{S})\left(1+\tilde{S}+(1-\tilde{S})T_q^2\right)\frac{\langle[\boldsymbol{K_A}]_q, [\boldsymbol{K_{B,\xi}}]_p\rangle}{N_{\boldsymbol{A}}N_{\boldsymbol{B}}d} \pm O\left(d^3\left(\delta_{\boldsymbol{AB},\perp}+\delta_{N/S}\right)\delta_{\xi,\perp}\delta_{N/S}\right).$$

*In particular, we have*

$$\max\left\{[\boldsymbol{Q}_{1,\boldsymbol{\xi_A}}]_{p,q}, [\boldsymbol{Q}_{1,\boldsymbol{\xi_B}}]_{p,q}\right\} \le O\left(\delta_{N/S}\delta_{\xi,\perp}\right).$$

**Lemma C.7** (Estimations for $\boldsymbol{Q}_2$). *In Stage 2, for any $p, q \in [d-r]$, we have*

$$[\boldsymbol{Q}_2]_{p,q} = \pm O\left(d^3\left(\delta_{\boldsymbol{AB},\perp}+\delta_{N/S}\right)\delta_{\xi,\perp}\delta_{N/S}\right).$$

**Lemma C.8** (Estimations for $Q_0$). *In Stage 2, we have*

$$Q_0 = -\sum_{k=1}^{r}\frac{\kappa_k^2}{\|\boldsymbol{\kappa}\|^2}[\boldsymbol{Q}_1]_{k,k} \pm O\left(d^2\delta_{\boldsymbol{AB},\perp}^2 + \delta_- + \kappa_0 d\delta_{N/S}\delta_{\xi,\perp}\right).$$

Finally, we use these estimations to simplify the formulas for the norms. We do not consider the tangent movement here since the situation is trickier there, and we will handle them in later subsections.

**Corollary C.9** (Dynamics of the norms). *In Stage 2, we have*

$$\frac{\mathrm{d}}{\mathrm{d}t}\|[\boldsymbol{K_A}]_p\|^2 = \frac{2\sigma_p^2\|[\boldsymbol{K_A}]_p\|\|[\boldsymbol{K_B}]_p\|}{N_{\boldsymbol{A}}N_{\boldsymbol{B}}d}\left\langle\overline{[\boldsymbol{K_A}]_p}, \overline{[\boldsymbol{K_B}]_p}\right\rangle[\boldsymbol{Q}_1]_{p,p} + 2\frac{\|[\boldsymbol{K_A}]_p\|^2}{N_{\boldsymbol{A}}^2 d}Q_0\sigma_p^2$$

$$\pm O\left(\frac{\sigma_p^2\kappa_p^2}{N_{\boldsymbol{A}}N_{\boldsymbol{B}}d}\kappa_0 d\delta_{\boldsymbol{AB},\perp}^2\right)$$

$$\frac{\mathrm{d}}{\mathrm{d}t}\|[\boldsymbol{K_{A,\xi}}]_q\|^2 = \frac{2\|[\boldsymbol{K_{A,\xi}}]_q\|^2}{N_{\boldsymbol{A}}^2 d}Q_0\sigma_\xi^2 \pm O\left(\frac{\sigma_\xi^2\|[\boldsymbol{K_{A,\xi}}]_q\|^2}{N_{\boldsymbol{A}}N_{\boldsymbol{B}}d}d\delta_{\xi,\perp}^2\right).$$

*The formulas for $\|[\boldsymbol{K_B}]_p\|$ and $\|[\boldsymbol{K_{B,\xi}}]_q\|$ can be obtained by interchanging the roles of $\boldsymbol{A}$ and $\boldsymbol{B}$.*

OMITTED PROOFS OF THIS SUBSECTION

*Proof of Lemma C.2.* First, we compute

$$\mathbb{E}_{\boldsymbol{z}^-}\left\{\exp\left(\frac{\langle\boldsymbol{z}^+, \boldsymbol{z}^-\rangle_{\hat{\boldsymbol{\kappa}}^2}}{N_{\boldsymbol{A}}N_{\boldsymbol{B}}}\right)\right\} = \prod_{k=1}^{r}\mathbb{E}_{z_k^-}\left\{\exp\left(\frac{\hat{\kappa}_k^2 z_k^+ z_k^-}{N_{\boldsymbol{A}}N_{\boldsymbol{B}}}\right)\right\} = \prod_{k=1}^{r}\frac{1}{2}\left(\exp\left(\frac{\hat{\kappa}_k^2/d}{N_{\boldsymbol{A}}N_{\boldsymbol{B}}}\right) + \exp\left(-\frac{\hat{\kappa}_k^2/d}{N_{\boldsymbol{A}}N_{\boldsymbol{B}}}\right)\right)$$

$$= \prod_{k=1}^{r}\cosh\left(\frac{\hat{\kappa}_k^2/d}{N_{\boldsymbol{A}}N_{\boldsymbol{B}}}\right).$$

Similarly, we also have

$$\mathbb{E}_{\boldsymbol{z}^-}\left\{\exp\left(\frac{\langle\boldsymbol{z}^+, \boldsymbol{z}^-\rangle_{\hat{\boldsymbol{\kappa}}^2}}{N_{\boldsymbol{A}}N_{\boldsymbol{B}}}\right)z_p^-\right\} = \mathbb{E}_{z_p^-}\left\{\exp\left(\frac{\hat{\kappa}_k^2 z_p^+ z_p^-}{N_{\boldsymbol{A}}N_{\boldsymbol{B}}}\right)z_p^-\right\}\prod_{k\neq p}\mathbb{E}_{z_k^-}\left\{\exp\left(\frac{\hat{\kappa}_k^2 z_k^+ z_k^-}{N_{\boldsymbol{A}}N_{\boldsymbol{B}}}\right)\right\}.$$

Each factor in $\prod_{k\neq p}$ is still $\cosh\left(\frac{\hat{\kappa}_k^2/d}{N_{\boldsymbol{A}}N_{\boldsymbol{B}}}\right)$. For the first term, we have

$$\mathbb{E}_{z_p^-}\left\{\exp\left(\frac{\hat{\kappa}_k^2 z_p^+ z_p^-}{N_{\boldsymbol{A}}N_{\boldsymbol{B}}}\right)z_p^-\right\} = \frac{1}{2\sqrt{d}}\mathbb{E}_{z_p^-}\left\{\exp\left(\frac{\hat{\kappa}_k^2 z_p^+/\sqrt{d}}{N_{\boldsymbol{A}}N_{\boldsymbol{B}}}\right)\right\} - \frac{1}{2\sqrt{d}}\mathbb{E}_{z_p^-}\left\{\exp\left(\frac{-\hat{\kappa}_k^2 z_p^+/\sqrt{d}}{N_{\boldsymbol{A}}N_{\boldsymbol{B}}}\right)z_p^-\right\}$$

$$= \frac{1}{\sqrt{d}}\sinh\left(\frac{\hat{\kappa}_k^2 z_p^+/\sqrt{d}}{N_{\boldsymbol{A}}N_{\boldsymbol{B}}}\right) = \frac{\mathrm{sgn}\, z_o^+}{\sqrt{d}}\sinh\left(\frac{\hat{\kappa}_k^2/d}{N_{\boldsymbol{A}}N_{\boldsymbol{B}}}\right) = \sinh\left(\frac{\hat{\kappa}_k^2/d}{N_{\boldsymbol{A}}N_{\boldsymbol{B}}}\right)z_p^+.$$

Therefore,

$$\mathbb{E}_{\boldsymbol{z}^-}\left\{\exp\left(\frac{\langle\boldsymbol{z}^+, \boldsymbol{z}^-\rangle_{\hat{\boldsymbol{\kappa}}^2}}{N_{\boldsymbol{A}}N_{\boldsymbol{B}}}\right)z_p^-\right\} = z_p^+\sinh\left(\frac{\hat{\kappa}_k^2/d}{N_{\boldsymbol{A}}N_{\boldsymbol{B}}}\right)\prod_{k\neq p}\cosh\left(\frac{\hat{\kappa}_k^2/d}{N_{\boldsymbol{A}}N_{\boldsymbol{B}}}\right) = Z_c T_p z_p^+.$$

The above calculation, *mutatis mutandis*, also yields the last identity. $\square$

*Proof of Lemma C.3.* First, we write

$$\left\langle \boldsymbol{f}_{\boldsymbol{A}}^{+}, \boldsymbol{f}_{\boldsymbol{B}}^{-}\right\rangle = \frac{\left\langle \boldsymbol{K}_{\boldsymbol{A}}\boldsymbol{z}^{+} + \boldsymbol{K}_{\boldsymbol{A},\xi}\boldsymbol{\xi}_{\boldsymbol{A}}^{+}, \boldsymbol{K}_{\boldsymbol{B}}\boldsymbol{z}^{-} + \boldsymbol{K}_{\boldsymbol{B},\xi}\boldsymbol{\xi}_{\boldsymbol{B}}^{-}\right\rangle}{N_{\boldsymbol{A}}N_{\boldsymbol{B}}}$$

$$= \frac{\left\langle \boldsymbol{K}_{\boldsymbol{A}}\boldsymbol{z}^{+}, \boldsymbol{K}_{\boldsymbol{B}}\boldsymbol{z}^{-}\right\rangle}{N_{\boldsymbol{A}}N_{\boldsymbol{B}}} + \frac{\left\langle \boldsymbol{K}_{\boldsymbol{A}}\boldsymbol{z}^{+}, \boldsymbol{K}_{\boldsymbol{B},\xi}\boldsymbol{\xi}_{\boldsymbol{B}}^{-}\right\rangle}{N_{\boldsymbol{A}}N_{\boldsymbol{B}}} + \frac{\left\langle \boldsymbol{K}_{\boldsymbol{A},\xi}\boldsymbol{\xi}_{\boldsymbol{A}}^{+}, \boldsymbol{K}_{\boldsymbol{B}}\boldsymbol{z}^{-}\right\rangle}{N_{\boldsymbol{A}}N_{\boldsymbol{B}}} \pm O\left(d^{2}\delta_{\xi,\perp}\delta_{N/S}^{2}\right).$$

Also note that the middle terms are $O(d^{2}\delta_{\xi,\perp}\delta_{N/S})$. Then, we compute

$$\exp\left(\boldsymbol{f}_{\boldsymbol{A}}^{+}\cdot\boldsymbol{f}_{\boldsymbol{B}}^{-}\right) = \exp\left(1 + \frac{\left\langle \boldsymbol{K}_{\boldsymbol{A}}\boldsymbol{z}^{+}, \boldsymbol{K}_{\boldsymbol{B},\xi}\boldsymbol{\xi}_{\boldsymbol{B}}^{-}\right\rangle}{N_{\boldsymbol{A}}N_{\boldsymbol{B}}} + \frac{\left\langle \boldsymbol{K}_{\boldsymbol{A},\xi}\boldsymbol{\xi}_{\boldsymbol{A}}^{+}, \boldsymbol{K}_{\boldsymbol{B}}\boldsymbol{z}^{-}\right\rangle}{N_{\boldsymbol{A}}N_{\boldsymbol{B}}}\right) \pm O\left(d^{2}\delta_{\xi,\perp}\delta_{N/S}^{2}\right).$$

Similar results also hold for other combinations of $\pm$. With the notations defined in this lemma, we can write these results as

$$\exp\left(\boldsymbol{f}_{\boldsymbol{A}}^{+}\cdot\boldsymbol{f}_{\boldsymbol{B}}^{-}\right) = E_{+,-}\left(1 + \delta_{+,\xi-} + \delta_{\xi+,-}\right) \pm O\left(d^{2}\delta_{\xi,\perp}\delta_{N/S}^{2}\right),$$

$$\exp\left(\boldsymbol{f}_{\boldsymbol{A}}^{-}\cdot\boldsymbol{f}_{\boldsymbol{B}}^{+}\right) = E_{-,+}\left(1 + \delta_{-,\xi+} + \delta_{\xi-,+}\right) \pm O\left(d^{2}\delta_{\xi,\perp}\delta_{N/S}^{2}\right),$$

$$\exp\left(\boldsymbol{f}_{\boldsymbol{A}}^{+}\cdot\boldsymbol{f}_{\boldsymbol{B}}^{+}\right) = E_{+,+}\left(1 + \delta_{+,\xi+} + \delta_{\xi+,+}\right) \pm O\left(d^{2}\delta_{\xi,\perp}\delta_{N/S}^{2}\right).$$

To compute $S_{\boldsymbol{A}}$ and $S_{\boldsymbol{B}}$, we then need to take expectations over the negative examples. Note that by taking expectation over $\boldsymbol{\xi}_{\boldsymbol{B}}^{-}$, the term $E_{+,-}\delta_{+,\xi-}$ becomes 0. Therefore, we have

$$\underset{\boldsymbol{x}_{\boldsymbol{B}}^{-}}{\mathbb{E}}\exp\left(\boldsymbol{f}_{\boldsymbol{A}}^{+}\cdot\boldsymbol{f}_{\boldsymbol{B}}^{-}\right) = \underset{\boldsymbol{x}_{\boldsymbol{B}}^{-}}{\mathbb{E}}\left\{E_{+,-}\left(1 + \delta_{\xi+,-}\right)\right\} \pm O\left(d^{2}\delta_{\xi,\perp}\delta_{N/S}^{2}\right).$$

Unfortunately, the same argument does not apply to $\delta_{\xi+,-}$ since both $E_{+,-}$ and $\delta_{\xi+,-}$ depend on $\boldsymbol{z}^{-}$. However, it is still possible to further simplify the expression. First, we write

$$\underset{\boldsymbol{z}^{-}}{\mathbb{E}}\left\{E_{+,-}\delta_{\xi+,-}\right\} = \underset{\boldsymbol{z}^{-}}{\mathbb{E}}\left\{\tilde{E}_{+,-}\left(1 \pm O(d\delta_{\boldsymbol{A}\boldsymbol{B},\perp})\right)\delta_{\xi+,-}\right\} = \underset{\boldsymbol{z}^{-}}{\mathbb{E}}\left\{\tilde{E}_{+,-}\delta_{\xi+,-}\right\} \pm O(d^{3}\delta_{\boldsymbol{A}\boldsymbol{B},\perp}\delta_{\xi,\perp}\delta_{N/S}).$$

Recall Lemma C.2. Then, we compute

$$\underset{\boldsymbol{z}^{-}}{\mathbb{E}}\left\{\tilde{E}_{+,-}\delta_{\xi+,-}\right\} = \underset{\boldsymbol{z}^{-}}{\mathbb{E}}\left\{\tilde{E}_{+,-}\frac{\left\langle \boldsymbol{K}_{\boldsymbol{A},\xi}\boldsymbol{\xi}_{\boldsymbol{A}}^{+}, \boldsymbol{K}_{\boldsymbol{B}}\boldsymbol{z}^{-}\right\rangle}{N_{\boldsymbol{A}}N_{\boldsymbol{B}}}\right\} = \frac{\left\langle \boldsymbol{K}_{\boldsymbol{A},\xi}\boldsymbol{\xi}_{\boldsymbol{A}}^{+}, \boldsymbol{K}_{\boldsymbol{B}}\underset{\boldsymbol{z}^{-}}{\mathbb{E}}\left\{\tilde{E}_{+,-}\boldsymbol{z}^{-}\right\}\right\rangle}{N_{\boldsymbol{A}}N_{\boldsymbol{B}}}$$

$$= Z_{c}\frac{\left\langle \boldsymbol{K}_{\boldsymbol{A},\xi}\boldsymbol{\xi}_{\boldsymbol{A}}^{+}, \boldsymbol{K}_{\boldsymbol{B}}\operatorname{diag}([T_{k}]_{k\in[r]})\boldsymbol{z}^{+}\right\rangle}{N_{\boldsymbol{A}}N_{\boldsymbol{B}}} = Z_{c}\delta_{\xi+,T+}.$$

Hence,

$$\underset{\boldsymbol{x}_{\boldsymbol{B}}^{-}}{\mathbb{E}}\exp\left(\boldsymbol{f}_{\boldsymbol{A}}^{+}\cdot\boldsymbol{f}_{\boldsymbol{B}}^{-}\right) = \underset{\boldsymbol{x}_{\boldsymbol{B}}^{-}}{\mathbb{E}}\left\{E_{+,-}\right\} + \underset{\boldsymbol{x}_{\boldsymbol{B}}^{-}}{\mathbb{E}}\left\{E_{+,-}\delta_{\xi+,-}\right\} \pm O\left(d^{2}\delta_{\xi,\perp}\delta_{N/S}^{2}\right)$$

$$= Z_{\boldsymbol{A},c} + Z_{c}\delta_{\xi+,T+} \pm O(d^{3}\delta_{\boldsymbol{A}\boldsymbol{B},\perp}\delta_{\xi,\perp}\delta_{N/S}) \pm O\left(d^{2}\delta_{\xi,\perp}\delta_{N/S}^{2}\right)$$

$$= Z_{\boldsymbol{A},c} + Z_{c}\delta_{\xi+,T+} \pm O\left(d^{3}\left(\delta_{\boldsymbol{A}\boldsymbol{B},\perp} + \delta_{N/S}\right)\delta_{\xi,\perp}\delta_{N/S}\right).$$

Similarly, we also have

$$\underset{\boldsymbol{x}_{\boldsymbol{A}}^{-}}{\mathbb{E}}\exp\left(\boldsymbol{f}_{\boldsymbol{A}}^{+}\cdot\boldsymbol{f}_{\boldsymbol{B}}^{-}\right) = Z_{\boldsymbol{B},c} + Z_{c}\delta_{T+,\xi+} \pm O\left(d^{3}\left(\delta_{\boldsymbol{A}\boldsymbol{B},\perp} + \delta_{N/S}\right)\delta_{\xi,\perp}\delta_{N/S}\right).$$

Recall that $\exp\left(\boldsymbol{f}_{\boldsymbol{A}}^{+}\cdot\boldsymbol{f}_{\boldsymbol{B}}^{+}\right) = E_{+,+}\left(1 + \delta_{+,\xi+} + \delta_{\xi+,+}\right) \pm O\left(d^{2}\delta_{\xi,\perp}\delta_{N/S}^{2}\right)$. Hence, we have

$$S_{\boldsymbol{A}}(\boldsymbol{x}_{\boldsymbol{A}}^{+}, \boldsymbol{x}_{\boldsymbol{B}}^{+}) = \frac{E_{+,+}(1 + \delta_{+,\xi+} + \delta_{\xi+,+})}{E_{+,+}(1 + \delta_{+,\xi+} + \delta_{\xi+,+}) + KZ_{\boldsymbol{A},c} + KZ_{c}\delta_{\xi+,T+}} \pm O\left(d^{3}\left(\delta_{\boldsymbol{A}\boldsymbol{B},\perp} + \delta_{N/S}\right)\delta_{\xi,\perp}\delta_{N/S}\right)$$

$$= \tilde{S}_{\boldsymbol{A}}\left(1 - \tilde{S}_{\boldsymbol{A}}(\delta_{+,\xi+} + \delta_{\xi+,+}) - (1 - \tilde{S}_{\boldsymbol{A}})\delta_{\xi+,T+}\right)$$

$$+ \tilde{S}_{\boldsymbol{A}}(\delta_{+,\xi+} + \delta_{\xi+,+}) \pm O\left(d^{3}\left(\delta_{\boldsymbol{A}\boldsymbol{B},\perp} + \delta_{N/S}\right)\delta_{\xi,\perp}\delta_{N/S}\right)$$

$$= \tilde{S}_{\boldsymbol{A}} + \tilde{S}(1 - \tilde{S})\left(\delta_{+,\xi+} + \delta_{\xi+,+} - \delta_{\xi+,T+}\right) \pm O\left(d^{3}\left(\delta_{\boldsymbol{A}\boldsymbol{B},\perp} + \delta_{N/S}\right)\delta_{\xi,\perp}\delta_{N/S}\right).$$

Similarly, we also have

$$S_B(\boldsymbol{x}_A^+, \boldsymbol{x}_B^+) = \tilde{S}_B + \tilde{S}(1 - \tilde{S})\left(\delta_{+,\boldsymbol{\xi}+} + \delta_{\boldsymbol{\xi}+,+} - \delta_{T+,\boldsymbol{\xi}+}\right) \pm O\left(d^3\left(\delta_{\boldsymbol{AB},\perp} + \delta_{N/S}\right)\delta_{\boldsymbol{\xi},\perp}\delta_{N/S}\right).$$

Then, we compute

$$\frac{S_A(\boldsymbol{x}_A^+, \boldsymbol{x}_B^+)\exp(\boldsymbol{f}_A^+ \cdot \boldsymbol{f}_B^-)}{\exp(\boldsymbol{f}_A^+ \cdot \boldsymbol{f}_B^+)} = \left(\tilde{S}_A + \tilde{S}(1 - \tilde{S})\left(\delta_{+,\boldsymbol{\xi}+} + \delta_{\boldsymbol{\xi}+,+} - \delta_{\boldsymbol{\xi}+,T+}\right)\right)\frac{E_{+,-}\left(1 + \delta_{+,\boldsymbol{\xi}-} + \delta_{\boldsymbol{\xi}+,-}\right)}{E_{+,+}\left(1 + \delta_{+,\boldsymbol{\xi}+} + \delta_{\boldsymbol{\xi}+,+}\right)}$$

$$\pm O\left(d^3\left(\delta_{\boldsymbol{AB},\perp} + \delta_{N/S}\right)\delta_{\boldsymbol{\xi},\perp}\delta_{N/S}\right)$$

$$= \frac{\tilde{S}_A E_{+,-}}{E_{+,+}}\left(1 + \delta_{+,\boldsymbol{\xi}-} + \delta_{\boldsymbol{\xi}+,-} - \tilde{S}\left(\delta_{+,\boldsymbol{\xi}+} + \delta_{\boldsymbol{\xi}+,+}\right) - (1 - \tilde{S})\delta_{\boldsymbol{\xi}+,T+}\right)$$

$$\pm O\left(d^3\left(\delta_{\boldsymbol{AB},\perp} + \delta_{N/S}\right)\delta_{\boldsymbol{\xi},\perp}\delta_{N/S}\right)$$

$$= \frac{\tilde{S}_A E_{+,-}}{E_{+,+}} + \frac{\tilde{S}\tilde{E}_{+,-}}{\tilde{E}}\left(\delta_{+,\boldsymbol{\xi}-} + \delta_{\boldsymbol{\xi}+,-} - \tilde{S}\left(\delta_{+,\boldsymbol{\xi}+} + \delta_{\boldsymbol{\xi}+,+}\right) - (1 - \tilde{S})\delta_{\boldsymbol{\xi}+,T+}\right)$$

$$\pm O\left(d^3\left(\delta_{\boldsymbol{AB},\perp} + \delta_{N/S}\right)\delta_{\boldsymbol{\xi},\perp}\delta_{N/S}\right),$$

and

$$\frac{S_B(\boldsymbol{x}_A^+, \boldsymbol{x}_B^+)\exp(\boldsymbol{f}_A^- \cdot \boldsymbol{f}_B^+)}{\exp(\boldsymbol{f}_A^+ \cdot \boldsymbol{f}_B^+)} = \frac{\tilde{S}_B E_{-,+}}{E_{+,+}} + \frac{\tilde{S}\tilde{E}_{-,+}}{\tilde{E}}\left(1 + \delta_{\boldsymbol{\xi}-,+} + \delta_{-,\boldsymbol{\xi}+} - \tilde{S}\left(\delta_{+,\boldsymbol{\xi}+} + \delta_{\boldsymbol{\xi}+,+}\right) - (1 - \tilde{S})\delta_{T+,\boldsymbol{\xi}+}\right)$$

$$\pm O\left(d^3\left(\delta_{\boldsymbol{AB},\perp} + \delta_{N/S}\right)\delta_{\boldsymbol{\xi},\perp}\delta_{N/S}\right). \qquad \square$$

*Proof of Lemma C.4.* We write

$$\frac{\langle \boldsymbol{K}_A \boldsymbol{z}^+, \boldsymbol{K}_B \boldsymbol{z}^-\rangle}{N_A N_B} = \sum_{k=1}^{r}\frac{\langle[\boldsymbol{K}_A]_k, [\boldsymbol{K}_B]_k\rangle}{N_A N_B}z_k^+ z_k^- + \sum_{i \neq j}\frac{\langle[\boldsymbol{K}_A]_i, [\boldsymbol{K}_B]_j\rangle}{N_A N_B}z_i^+ z_j^- =: I_{+,-} + \tilde{\delta}_{+,-}.$$

Note that, as a special case, we have $I_{+,+} = I_{-,-} = I_0$. In other words, $I_{+,+}$ and $I_{-,-}$ do not depend on the actual value of $\boldsymbol{z}^\pm$. Also note that $I_{+,-}$ is bounded by $O(d\delta_{\boldsymbol{AB},\perp})$. Then, we compute

$$E_{+,-} = \exp(I_{+,-})\left(1 + \tilde{\delta}_{+,-} \pm O\left(d^2 \delta_{\boldsymbol{AB},\perp}^2\right)\right) \quad \text{and} \quad E_{+,+} = \exp(I_0)\left(1 + \tilde{\delta}_{+,+} \pm O\left(d^2 \delta_{\boldsymbol{AB},\perp}^2\right)\right).$$

Take expectation over $\boldsymbol{z}^-$ and we obtain

$$\mathbb{E}_{\boldsymbol{z}^-} E_{+,-} = \mathbb{E}_{\boldsymbol{z}^-}\exp(I_{+,-}) + \mathbb{E}_{\boldsymbol{z}^-}\left\{\exp(I_{+,-})\tilde{\delta}_{+,-}\right\} \pm O\left(d^2 \delta_{\boldsymbol{AB},\perp}^2\right).$$

By Lemma C.2, the first term is $Z_c$. For the second term, we compute

$$\mathbb{E}_{\boldsymbol{z}^-}\left\{\exp(I_{+,-})\tilde{\delta}_{+,-}\right\} = \sum_{i \neq j}\frac{[\boldsymbol{K}_{\boldsymbol{AB}}]_{i,j}}{N_A N_B}\mathbb{E}_{\boldsymbol{z}^-}\left\{\exp(I_{+,-})z_j^-\right\}z_i^+$$

$$= Z_c\sum_{i \neq j}\frac{[\boldsymbol{K}_{\boldsymbol{AB}}]_{i,j}}{N_A N_B}T_j z_j^+ z_i^+$$

$$= Z_c\tilde{\delta}_{+,T+},$$

where the second line again comes from Lemma C.2. Hence, we have

$$\mathbb{E}_{\boldsymbol{z}^-} E_{+,-} = Z_c + Z_c\tilde{\delta}_{+,T+} \pm O\left(d^2 \delta_{\boldsymbol{AB},\perp}^2\right).$$

Then, for $\tilde{S}_A$, we have

$$\tilde{S}_A(\boldsymbol{z}^+) = \frac{E_{+,+}}{E_{+,+} + K\mathbb{E}_{\boldsymbol{z}^-} E_{+,-}} = \frac{\exp(I_0)\left(1 + \tilde{\delta}_{+,+}\right)}{\exp(I_0)\left(1 + \tilde{\delta}_{+,+}\right) + KZ_c + KZ_c\tilde{\delta}_{+,T+}} \pm O\left(d^2 \delta_{\boldsymbol{AB},\perp}^2\right)$$

$$= \tilde{S}\left(1 - \tilde{S}\tilde{\delta}_{+,+} - (1 - \tilde{S})\tilde{\delta}_{+,T+}\right) + \tilde{S}\tilde{\delta}_{+,+} \pm O\left(d^2 \delta_{\boldsymbol{AB},\perp}^2\right)$$

$$= \tilde{S}\left(1 + (1 - \tilde{S})(\tilde{\delta}_{+,+} - \tilde{\delta}_{+,T+})\right) \pm O\left(d^2 \delta_{\boldsymbol{AB},\perp}^2\right).$$

Similarly, we also have

$$\tilde{S}_{\boldsymbol{B}}(\boldsymbol{z}^+) = \tilde{S}\left(1 + (1-\tilde{S})(\tilde{\delta}_{+,+} - \tilde{\delta}_{T+,+})\right) \pm O\left(d^2 \delta^2_{\boldsymbol{AB},\perp}\right).$$

Then, we compute

$$\frac{\tilde{S}_{\boldsymbol{A}} E_{+,-}}{E_{+,+}} = \tilde{S}\left(1 + (1-\tilde{S})(\tilde{\delta}_{+,+} - \tilde{\delta}_{+,T+})\right) \frac{\exp(I_{+,-})\left(1 + \tilde{\delta}_{+,-}\right)}{\exp(I_{+,-})\left(1 + \tilde{\delta}_{+,+}\right)} \pm O\left(d^2 \delta^2_{\boldsymbol{AB},\perp}\right)$$

$$= \frac{\tilde{S}\exp(I_{+,-})}{\exp(I_0)}\left(1 - \tilde{S}\tilde{\delta}_{+,+} - (1-\tilde{S})\tilde{\delta}_{+,T+} + \tilde{\delta}_{+,-}\right) \pm O\left(d^2 \delta^2_{\boldsymbol{AB},\perp}\right).$$

Similarly, we also have

$$\frac{\tilde{S}_{\boldsymbol{B}} E_{-,+}}{E_{+,+}} = \frac{\tilde{S}\exp(I_{-,+})}{\exp(I_0)}\left(1 - \tilde{S}\tilde{\delta}_{+,+} - (1-\tilde{S})\tilde{\delta}_{T+,+} + \tilde{\delta}_{-,+}\right) \pm O\left(d^2 \delta^2_{\boldsymbol{AB},\perp}\right).$$

$\square$

*Proof of Lemma C.5.* Recall that

$$\boldsymbol{Q}_1 := \mathbb{E}\left\{\left(2 - S_{\boldsymbol{A}}(\boldsymbol{x}_{\boldsymbol{A}}^+, \boldsymbol{x}_{\boldsymbol{B}}^+) - S_{\boldsymbol{B}}(\boldsymbol{x}_{\boldsymbol{A}}^+, \boldsymbol{x}_{\boldsymbol{B}}^+)\right) \boldsymbol{z}^+(\boldsymbol{z}^+)^\top d\right\}$$
$$- K\mathbb{E}\left\{\frac{S_{\boldsymbol{A}}(\boldsymbol{x}_{\boldsymbol{A}}^+, \boldsymbol{x}_{\boldsymbol{B}}^+)\exp(\boldsymbol{f}_{\boldsymbol{A}}^+ \cdot \boldsymbol{f}_{\boldsymbol{B}}^-)}{\exp(\boldsymbol{f}_{\boldsymbol{A}}^+ \cdot \boldsymbol{f}_{\boldsymbol{B}}^+)} \boldsymbol{z}^-(\boldsymbol{z}^+)^\top d\right\} - K\mathbb{E}\left\{\frac{S_{\boldsymbol{B}}(\boldsymbol{x}_{\boldsymbol{A}}^+, \boldsymbol{x}_{\boldsymbol{B}}^+)\exp(\boldsymbol{f}_{\boldsymbol{A}}^- \cdot \boldsymbol{f}_{\boldsymbol{B}}^+)}{\exp(\boldsymbol{f}_{\boldsymbol{A}}^+ \cdot \boldsymbol{f}_{\boldsymbol{B}}^+)} \boldsymbol{z}^+(\boldsymbol{z}^-)^\top d\right\}.$$

Note that there is no $\boldsymbol{\xi}$ here other than the ones in the coefficient. As a result, all terms contain $\delta_{+,\boldsymbol{\xi}+}$ and alike are 0. Hence, we have

$$\boldsymbol{Q}_1 := \mathbb{E}\left\{\left(2 - \tilde{S}_{\boldsymbol{A}}(\boldsymbol{x}_{\boldsymbol{A}}^+, \boldsymbol{x}_{\boldsymbol{B}}^+) - \tilde{S}_{\boldsymbol{B}}(\boldsymbol{x}_{\boldsymbol{A}}^+, \boldsymbol{x}_{\boldsymbol{B}}^+)\right) \boldsymbol{z}^+(\boldsymbol{z}^+)^\top d\right\}$$
$$- K\mathbb{E}\left\{\frac{\tilde{S}_{\boldsymbol{A}} E_{+,-}}{E_{+,+}} \boldsymbol{z}^-(\boldsymbol{z}^+)^\top d\right\} - K\mathbb{E}\left\{\frac{\tilde{S}_{\boldsymbol{B}} E_{-,+}}{E_{+,+}} \boldsymbol{z}^+(\boldsymbol{z}^-)^\top d\right\}$$
$$=: \mathrm{T}_1(\boldsymbol{Q}_1) + \mathrm{T}_2(\boldsymbol{Q}_1) + \mathrm{T}_3(\boldsymbol{Q}_1).$$

Now we estimate each of these three terms separately. We also deal with the diagonal and off-diagonal terms separately. By Lemma C.4, we have

$$\mathrm{T}_1([\boldsymbol{Q}_1]_{p,p}) = \mathbb{E}\left\{2 - \tilde{S}_{\boldsymbol{A}}(\boldsymbol{x}_{\boldsymbol{A}}^+, \boldsymbol{x}_{\boldsymbol{B}}^+) - \tilde{S}_{\boldsymbol{B}}(\boldsymbol{x}_{\boldsymbol{A}}^+, \boldsymbol{x}_{\boldsymbol{B}}^+)\right\}$$
$$= \mathbb{E}\left\{2 - \tilde{S}\left(1 + (1-\tilde{S})(\tilde{\delta}_{+,+} - \tilde{\delta}_{+,T+})\right) - \tilde{S}\left(1 + (1-\tilde{S})(\tilde{\delta}_{+,+} - \tilde{\delta}_{T+,+})\right)\right\} \pm O\left(d^2 \delta^2_{\boldsymbol{AB},\perp}\right)$$
$$= 2(1-\tilde{S}) \pm O\left(d^2 \delta^2_{\boldsymbol{AB},\perp}\right).$$

Note that we use the fact that all these $\delta$'s have mean 0. Also by Lemma C.4, we have

$$\mathrm{T}_2([\boldsymbol{Q}_1]_{p,p}) = -K\mathbb{E}\left\{\left(\frac{\tilde{S}\tilde{E}_{+,-}}{\tilde{E}_0}\left(1 - \tilde{S}\tilde{\delta}_{+,+} - (1-\tilde{S})\tilde{\delta}_{+,T+} + \tilde{\delta}_{+,-}\right)\right) z_p^- z_p^+ d\right\} \pm O\left(d^2 \delta^2_{\boldsymbol{AB},\perp}\right)$$
$$= -\frac{\tilde{S}K}{\tilde{E}_0}\mathbb{E}\left\{\tilde{E}_{+,-} z_p^- z_p^+ d\right\}$$
$$- \frac{\tilde{S}K}{\tilde{E}_0}\mathbb{E}\left\{\left(\tilde{E}_{+,-}\left(-\tilde{S}\tilde{\delta}_{+,+} - (1-\tilde{S})\tilde{\delta}_{+,T+} + \tilde{\delta}_{+,-}\right)\right) z_p^- z_p^+ d\right\} \pm O\left(d^2 \delta^2_{\boldsymbol{AB},\perp}\right).$$

Note that $\tilde{\delta}_{+,+}$, $\tilde{\delta}_{+,-}$ and $\tilde{\delta}_{+,T+}$ only contain cross terms of form $z_i^+ z_j^\pm$ with $i \neq j$. Hence, the second term is 0. Meanwhile, by Lemma C.2, the first term is

$$-\frac{\tilde{S}K}{\tilde{E}_0}\mathbb{E}\left\{\tilde{E}_{+,-} z_p^- z_p^+ d\right\} = -\frac{\tilde{S}K Z_c T_p}{\tilde{E}_0} = -(1-\tilde{S})T_p.$$

As a result,

$$\mathrm{T}_2([\boldsymbol{Q}_1]_{p,p}) = -(1-\tilde{S})T_p \pm O\left(d^2 \delta^2_{\boldsymbol{AB},\perp}\right).$$

Similarly, one can show that $\mathrm{T}_3([\boldsymbol{Q}_1]_{p,p}) = -(1-\tilde{S})T_p \pm O\left(d^2\delta^2_{\boldsymbol{AB},\perp}\right)$ also holds. Thus,

$$[\boldsymbol{Q}_1]_{p,p} = 2(1-\tilde{S})(1-T_p) \pm O\left(d^2\delta^2_{\boldsymbol{AB},\perp}\right).$$

Now, we consider the off-diagonal terms. For notational simplicity, we define $\boldsymbol{K_{AB}} = \boldsymbol{K}_{\boldsymbol{A}}^\top \boldsymbol{K_B}$. For any $p \neq q$, we compute

$$
\begin{aligned}
\mathrm{T}_1([\boldsymbol{Q}_1]_{p,q}) &= -\mathbb{E}\left\{\left(\tilde{S}_{\boldsymbol{A}}(\boldsymbol{x}_{\boldsymbol{A}}^+, \boldsymbol{x}_{\boldsymbol{B}}^+) + \tilde{S}_{\boldsymbol{B}}(\boldsymbol{x}_{\boldsymbol{A}}^+, \boldsymbol{x}_{\boldsymbol{B}}^+)\right) z_p^+ z_q^+ d\right\} \\
&= -\tilde{S}(1-\tilde{S})\,\mathbb{E}\left\{\left(2\tilde{\delta}_{+,+} - \tilde{\delta}_{+,T+} - \tilde{\delta}_{T+,T}\right) z_p^+ z_q^+ d\right\} \pm O\left(d^2\delta^2_{\boldsymbol{AB},\perp}\right) \\
&= -\tilde{S}(1-\tilde{S})\sum_{i\neq j}\frac{[\boldsymbol{K_{AB}}]_{i,j}}{N_{\boldsymbol{A}}N_{\boldsymbol{B}}}(2 - T_i - T_j)\,\mathbb{E}\left\{z_i^+ z_j^+ z_p^+ z_q^+ d\right\} \pm O\left(d^2\delta^2_{\boldsymbol{AB},\perp}\right).
\end{aligned}
$$

Clear that the summand is nonzero only if $(i,j) = (p,q)$ or $(i,j) = (q,p)$. Hence,

$$\mathrm{T}_1([\boldsymbol{Q}_1]_{p,q}) = -\tilde{S}(1-\tilde{S})\frac{[\boldsymbol{K_{AB}}]_{p,q} + [\boldsymbol{K_{AB}}]_{q,p}}{N_{\boldsymbol{A}}N_{\boldsymbol{B}}d}(2 - T_p - T_q) \pm O\left(d^2\delta^2_{\boldsymbol{AB},\perp}\right).$$

Then, for $\mathrm{T}_2$, we compute

$$
\begin{aligned}
\mathrm{T}_2([\boldsymbol{Q}_1]_{p,q}) &= -K\,\mathbb{E}\left\{\frac{S_{\boldsymbol{A}}(\boldsymbol{x}_{\boldsymbol{A}}^+, \boldsymbol{x}_{\boldsymbol{B}}^+)\exp(\boldsymbol{f}_{\boldsymbol{A}}^+ \cdot \boldsymbol{f}_{\boldsymbol{B}}^-)}{\exp(\boldsymbol{f}_{\boldsymbol{A}}^+ \cdot \boldsymbol{f}_{\boldsymbol{B}}^+)}z_p^- z_q^+ d\right\} \\
&= -K\,\mathbb{E}\left\{\left(\frac{\tilde{S}\tilde{E}_{+,-}}{\tilde{E}_0}\left(1 - \tilde{S}\tilde{\delta}_{+,+} - (1-\tilde{S})\tilde{\delta}_{+,T+} + \tilde{\delta}_{+,-}\right)\right)z_p^- z_q^+ d\right\} \pm O\left(d^2\delta^2_{\boldsymbol{AB},\perp}\right) \\
&= -K\,\mathbb{E}\left\{\left(\frac{\tilde{S}\tilde{E}_{+,-}}{\tilde{E}_0}\left(-\tilde{S}\tilde{\delta}_{+,+} - (1-\tilde{S})\tilde{\delta}_{+,T+} + \tilde{\delta}_{+,-}\right)\right)z_p^- z_q^+ d\right\} \pm O\left(d^2\delta^2_{\boldsymbol{AB},\perp}\right) \\
&= -K\sum_{i\neq j}\frac{[\boldsymbol{K_{AB}}]_{i,j}}{N_{\boldsymbol{A}}N_{\boldsymbol{B}}}\,\mathbb{E}\left\{\left(\frac{\tilde{S}\tilde{E}_{+,-}}{\tilde{E}_0}\left(-\tilde{S}z_i^+ z_j^+ - (1-\tilde{S})z_i^+ z_j^+ T_j + z_i^+ z_j^-\right)\right)z_p^- z_q^+ d\right\} \\
&\quad \pm O\left(d^2\delta^2_{\boldsymbol{AB},\perp}\right).
\end{aligned}
$$

Again, the summand is nonzero only if $(i,j) = (p,q)$ or $(i,j) = (q,p)$. By Lemma C.2, we have

$$
\begin{aligned}
&\mathrm{T}_2([\boldsymbol{Q}_1]_{p,q}) \\
&= -\frac{\tilde{S}K}{\tilde{E}_0}\frac{[\boldsymbol{K_{AB}}]_{p,q}}{N_{\boldsymbol{A}}N_{\boldsymbol{B}}}\left(\left(-\tilde{S} - (1-\tilde{S})T_q\right)\mathbb{E}\left\{\exp(I_{+,-})z_p^+ z_p^-\right\} + \mathbb{E}\left\{\exp(I_{+,-})z_p^+ z_q^+ z_p^- z_q^- d\right\}\right) \\
&\quad - \frac{\tilde{S}K}{\tilde{E}_0}\frac{[\boldsymbol{K_{AB}}]_{q,p}}{N_{\boldsymbol{A}}N_{\boldsymbol{B}}}\left(\left(-\tilde{S} - (1-\tilde{S})T_p\right)\mathbb{E}\left\{\exp(I_{+,-})z_p^+ z_p^-\right\} + d^{-1}\mathbb{E}\left\{\exp(I_{+,-})\right\}\right) \pm O\left(d^2\delta^2_{\boldsymbol{AB},\perp}\right) \\
&= -\frac{\tilde{S}KZ_c}{\tilde{E}_0}\frac{[\boldsymbol{K_{AB}}]_{p,q}}{N_{\boldsymbol{A}}N_{\boldsymbol{B}}d}\left(\left(-\tilde{S} - (1-\tilde{S})T_q\right)T_p + T_pT_q\right) \\
&\quad - \frac{\tilde{S}KZ_c}{\tilde{E}_0}\frac{[\boldsymbol{K_{AB}}]_{q,p}}{N_{\boldsymbol{A}}N_{\boldsymbol{B}}d}\left(\left(-\tilde{S} - (1-\tilde{S})T_p\right)T_p + 1\right) \pm O\left(d^2\delta^2_{\boldsymbol{AB},\perp}\right) \\
&= -(1-\tilde{S})\frac{[\boldsymbol{K_{AB}}]_{p,q}}{N_{\boldsymbol{A}}N_{\boldsymbol{B}}d}\tilde{S}T_p(T_q - 1) - (1-\tilde{S})\frac{[\boldsymbol{K_{AB}}]_{q,p}}{N_{\boldsymbol{A}}N_{\boldsymbol{B}}d}\left(-\tilde{S}T_p - (1-\tilde{S})T_p^2 + 1\right) \pm O\left(d^2\delta^2_{\boldsymbol{AB},\perp}\right).
\end{aligned}
$$

Similarly, for $T_3$, we have

$$T_3([\boldsymbol{Q}_1]_{p,q})$$
$$= -K\,\mathbb{E}\left\{\frac{S_B(\boldsymbol{x}_A^+, \boldsymbol{x}_B^+)\exp(\boldsymbol{f}_A^- \cdot \boldsymbol{f}_B^+)}{\exp(\boldsymbol{f}_A^+ \cdot \boldsymbol{f}_B^+)}z_p^+ z_q^- d\right\}$$
$$= -K\,\mathbb{E}\left\{\frac{\tilde{S}\tilde{E}_{-,+}}{\tilde{E}_0}\left(1 - \tilde{S}\tilde{\delta}_{+,+} - (1-\tilde{S})\tilde{\delta}_{T+,+} + \tilde{\delta}_{-,+}\right)z_p^+ z_q^- d\right\} \pm O\left(d^2\delta_{\boldsymbol{AB},\perp}^2\right)$$
$$= -\frac{\tilde{S}K}{\tilde{E}_0}\sum_{i\neq j}\frac{[\boldsymbol{K}_{\boldsymbol{AB}}]_{i,j}}{N_A N_B}\mathbb{E}\left\{\tilde{E}_{-,+}\left(\left(-\tilde{S} - (1-\tilde{S})T_i\right)z_i^+ z_j^+ + z_i^- z_j^+\right)z_p^+ z_q^- d\right\} \pm O\left(d^2\delta_{\boldsymbol{AB},\perp}^2\right)$$
$$= -\frac{\tilde{S}K}{\tilde{E}_0}\frac{[\boldsymbol{K}_{\boldsymbol{AB}}]_{p,q}}{N_A N_B}\left(\left(-\tilde{S} - (1-\tilde{S})T_p\right)Z_c T_q/d + Z_c T_p T_q/d\right)$$
$$\quad -\frac{\tilde{S}K}{\tilde{E}_0}\frac{[\boldsymbol{K}_{\boldsymbol{AB}}]_{q,p}}{N_A N_B}\left(\left(-\tilde{S} - (1-\tilde{S})T_q\right)Z_c T_q/d + Z_c/d\right) \pm O\left(d^2\delta_{\boldsymbol{AB},\perp}^2\right)$$
$$= -(1-\tilde{S})\frac{[\boldsymbol{K}_{\boldsymbol{AB}}]_{p,q}}{N_A N_B d}\tilde{S}T_q(T_p - 1) - (1-\tilde{S})\frac{[\boldsymbol{K}_{\boldsymbol{AB}}]_{q,p}}{N_A N_B d}\left(1 - \tilde{S}T_q - (1-\tilde{S})T_q^2\right) \pm O\left(d^2\delta_{\boldsymbol{AB},\perp}^2\right).$$

Combine these together and we obtain

$$[\boldsymbol{Q}_1]_{p,q} = -\tilde{S}(1-\tilde{S})\frac{[\boldsymbol{K}_{\boldsymbol{AB}}]_{p,q} + [\boldsymbol{K}_{\boldsymbol{AB}}]_{q,p}}{N_A N_B d}(2 - T_p - T_q)$$
$$\quad - (1-\tilde{S})\frac{[\boldsymbol{K}_{\boldsymbol{AB}}]_{p,q}}{N_A N_B d}\tilde{S}\left(2T_p T_q - T_p - T_q\right)$$
$$\quad - (1-\tilde{S})\frac{[\boldsymbol{K}_{\boldsymbol{AB}}]_{q,p}}{N_A N_B d}\left(2 - \tilde{S}(T_p + T_q) - (1-\tilde{S})(T_p^2 + T_q^2)\right) \pm O\left(d^2\delta_{\boldsymbol{AB},\perp}^2\right).$$

$\square$

*Proof of Lemma C.6.* We write

$$\boldsymbol{Q}_{1,\xi_B} := \mathbb{E}\left\{\left(2 - S_A(\boldsymbol{x}_A^+, \boldsymbol{x}_B^+) - S_B(\boldsymbol{x}_A^+, \boldsymbol{x}_B^+)\right)\boldsymbol{\xi}_B^+(\boldsymbol{z}^+)^\top d\right\}$$
$$\quad - K\,\mathbb{E}\left\{\frac{S_A(\boldsymbol{x}_A^+, \boldsymbol{x}_B^+)\exp(\boldsymbol{f}_A^+ \cdot \boldsymbol{f}_B^-)}{\exp(\boldsymbol{f}_A^+ \cdot \boldsymbol{f}_B^+)}\boldsymbol{\xi}_B^-(\boldsymbol{z}^+)^\top d\right\} - K\,\mathbb{E}\left\{\frac{S_B(\boldsymbol{x}_A^+, \boldsymbol{x}_B^+)\exp(\boldsymbol{f}_A^- \cdot \boldsymbol{f}_B^+)}{\exp(\boldsymbol{f}_A^+ \cdot \boldsymbol{f}_B^+)}\boldsymbol{\xi}_B^+(\boldsymbol{z}^-)^\top d\right\}$$
$$=: T_1\left(\boldsymbol{Q}_{1,\xi_B}\right) + T_2\left(\boldsymbol{Q}_{1,\xi_B}\right) + T_3\left(\boldsymbol{Q}_{1,\xi_B}\right).$$

We will use the fact that if some quantity $X$ does not depend on $\boldsymbol{\xi}$, then $\mathbb{E}\{X\boldsymbol{\xi}\} = 0$ to simplify these terms. For $T_1$, by Lemma C.3, we have

$$T_1\left([\boldsymbol{Q}_{1,\xi_B}]_{p,q}\right) = -\mathbb{E}\left\{\left(S_A(\boldsymbol{x}_A^+, \boldsymbol{x}_B^+) + S_B(\boldsymbol{x}_A^+, \boldsymbol{x}_B^+)\right)[\boldsymbol{\xi}_B^+]_p z_q^+ d\right\}$$
$$= -\tilde{S}(1-\tilde{S})\,\mathbb{E}\left\{\left(2\delta_{+,\xi+} + 2\delta_{\xi+,+} - \delta_{\xi+,T+} - \delta_{T+,\xi+}\right)[\boldsymbol{\xi}_B^+]_p z_q^+ d\right\}$$
$$\quad \pm O\left(d^3\left(\delta_{\boldsymbol{AB},\perp} + \delta_{N/S}\right)\delta_{\xi,\perp}\delta_{N/S}\right)$$
$$= -\tilde{S}(1-\tilde{S})\,\mathbb{E}\left\{\left(2\delta_{+,\xi+} - \delta_{T+,\xi+}\right)[\boldsymbol{\xi}_B^+]_p z_q^+ d\right\} \pm O\left(d^3\left(\delta_{\boldsymbol{AB},\perp} + \delta_{N/S}\right)\delta_{\xi,\perp}\delta_{N/S}\right).$$

Recall that

$$\delta_{+,\xi+} = \sum_{i,j}\frac{[\boldsymbol{K}_A^\top \boldsymbol{K}_{\boldsymbol{B},\xi}]_{i,j}}{N_A N_B}z_i^+[\boldsymbol{\xi}_B^+]_j, \quad \delta_{T+,\xi+} = \sum_{i,j}\frac{[\boldsymbol{K}_A^\top \boldsymbol{K}_{\boldsymbol{B},\xi}]_{i,j}}{N_A N_B}T_i z_i^+[\boldsymbol{\xi}_B^+]_j.$$

Hence, we can further rewrite $T_1\left([\boldsymbol{Q}_{1,\xi_B}]_{p,q}\right)$ as

$$T_1\left([\boldsymbol{Q}_{1,\xi_B}]_{p,q}\right) = -\tilde{S}(1-\tilde{S})\sum_{i,j}\frac{[\boldsymbol{K}_A^\top \boldsymbol{K}_{\boldsymbol{B},\xi}]_{i,j}}{N_A N_B}(2 - T_i)\,\mathbb{E}\left\{z_i^+[\boldsymbol{\xi}_B^+]_j[\boldsymbol{\xi}_B^+]_p z_q^+ d\right\}$$
$$\quad \pm O\left(d^3\left(\delta_{\boldsymbol{AB},\perp} + \delta_{N/S}\right)\delta_{\xi,\perp}\delta_{N/S}\right).$$

Note that the summand is nonzero only if $i = q$ and $j = p$. Thus,

$$\mathrm{T}_1\left([\boldsymbol{Q}_{1,\xi_B}]_{p,q}\right) = -\tilde{S}(1-\tilde{S})\left(2 - T_q\right)\frac{\langle[\boldsymbol{K_A}]_q, [\boldsymbol{K_{B,\xi}}]_p\rangle}{N_A N_B d} \pm O\left(d^3\left(\delta_{\boldsymbol{AB},\perp} + \delta_{N/S}\right)\delta_{\xi,\perp}\delta_{N/S}\right).$$

Similarly, we also have

$$\mathrm{T}_2\left([\boldsymbol{Q}_{1,\xi_B}]_{p,q}\right) = -\frac{K\tilde{S}}{\tilde{E}}\mathbb{E}\left\{\tilde{E}_{+,-}\delta_{+,\xi-}[\boldsymbol{\xi_B^-}]_p z_q^+ d\right\} \pm O\left(d^3\left(\delta_{\boldsymbol{AB},\perp} + \delta_{N/S}\right)\delta_{\xi,\perp}\delta_{N/S}\right),$$

$$\mathrm{T}_3\left([\boldsymbol{Q}_{1,\xi_B}]_{p,q}\right) = -\frac{K\tilde{S}}{\tilde{E}}\mathbb{E}\left\{\tilde{E}_{-,+}\left(\delta_{-,\xi+} - \tilde{S}\delta_{+,\xi+} - (1-\tilde{S})\delta_{T+,\xi+}\right)[\boldsymbol{\xi_B^-}]_p z_q^+ d\right\}$$
$$\pm O\left(d^3\left(\delta_{\boldsymbol{AB},\perp} + \delta_{N/S}\right)\delta_{\xi,\perp}\delta_{N/S}\right).$$

Then, we write

$$\mathbb{E}\left\{\tilde{E}_{+,-}\delta_{+,\xi-}[\boldsymbol{\xi_B^-}]_p z_q^+ d\right\} = \sum_{i,j}\frac{[\boldsymbol{K_A^\top K_{B,\xi}}]_{i,j}}{N_A N_B}\mathbb{E}\left\{\tilde{E}_{+,-}z_i^+[\boldsymbol{\xi_B^-}]_j[\boldsymbol{\xi_B^-}]_p z_q^+ d\right\}.$$

If $j \neq p$, clear that that summand is 0. If $i \neq q$, then by flipping the sign of $z_q^\pm$ simultaneously, we flip the sign of $\tilde{E}_{+,-}z_i^+ z_q^+$. Therefore, when $i \neq q$, the summand is also 0. Thus,

$$\mathrm{T}_2\left([\boldsymbol{Q}_{1,\xi_B}]_{p,q}\right) = -\frac{\tilde{S}K Z_c}{\tilde{E}}\frac{\langle[\boldsymbol{K_A}]_q, [\boldsymbol{K_{B,\xi}}]_p\rangle}{N_A N_B d} \pm O\left(d^3\left(\delta_{\boldsymbol{AB},\perp} + \delta_{N/S}\right)\delta_{\xi,\perp}\delta_{N/S}\right)$$
$$= -(1-\tilde{S})\frac{\langle[\boldsymbol{K_A}]_q, [\boldsymbol{K_{B,\xi}}]_p\rangle}{N_A N_B d} \pm O\left(d^3\left(\delta_{\boldsymbol{AB},\perp} + \delta_{N/S}\right)\delta_{\xi,\perp}\delta_{N/S}\right).$$

Similarly, for $\mathrm{T}_3$, we compute

$$\mathbb{E}\left\{\tilde{E}_{-,+}\left(\delta_{-,\xi+} - \tilde{S}\delta_{+,\xi+} - (1-\tilde{S})\delta_{T+,\xi+}\right)[\boldsymbol{\xi_B^-}]_p z_q^+ d\right\}$$
$$= \sum_{i,j}\frac{[\boldsymbol{K_A^\top K_{B,\xi}}]_{i,j}}{N_A N_B}\mathbb{E}\left\{\tilde{E}_{-,+}\left(z_i^-[\boldsymbol{\xi_B^-}]_j - \tilde{S}z_i^+[\boldsymbol{\xi_B^-}]_j - (1-\tilde{S})(1-T_i)z_i^-[\boldsymbol{\xi_B^-}]_j\right)[\boldsymbol{\xi_B^-}]_p z_q^+ d\right\}$$
$$= \frac{[\boldsymbol{K_A^\top K_{B,\xi}}]_{q,p}}{N_A N_B}\mathbb{E}\left\{\tilde{E}_{-,+}\left(z_q^- - \tilde{S}z_q^+ - (1-\tilde{S})(1-T_i)z_q^-\right)z_q^+\right\}$$
$$= Z_c\frac{[\boldsymbol{K_A^\top K_{B,\xi}}]_{q,p}}{N_A N_B d}\left(\tilde{S}(T_q - 1 - T_q^2) + T_q^2\right).$$

Thus,

$$\mathrm{T}_3\left([\boldsymbol{Q}_{1,\xi_B}]_{p,q}\right) = -(1-\tilde{S})\frac{\langle[\boldsymbol{K_A}]_q, [\boldsymbol{K_{B,\xi}}]_p\rangle}{N_A N_B d}\left(\tilde{S}(T_q - 1 - T_q^2) + T_q^2\right)$$
$$\pm O\left(d^3\left(\delta_{\boldsymbol{AB},\perp} + \delta_{N/S}\right)\delta_{\xi,\perp}\delta_{N/S}\right).$$

Combine these together, rearrange terms and we obtain

$$[\boldsymbol{Q}_{1,\xi_B}]_{p,q} = -(1-\tilde{S})\left(1 + \tilde{S} + (1-\tilde{S})T_q^2\right)\frac{\langle[\boldsymbol{K_A}]_q, [\boldsymbol{K_{B,\xi}}]_p\rangle}{N_A N_B d} \pm O\left(d^3\left(\delta_{\boldsymbol{AB},\perp} + \delta_{N/S}\right)\delta_{\xi,\perp}\delta_{N/S}\right).$$

To obtain the formula for $\boldsymbol{Q}_{1,\xi_A}$, it suffices to interchange the roles of $\boldsymbol{A}$ and $\boldsymbol{B}$. $\qquad\square$

*Proof of Lemma C.7.* Recall that

$$\boldsymbol{Q}_2 := \mathbb{E}\left\{\left(2 - S_A(\boldsymbol{x}_A^+, \boldsymbol{x}_B^+) - S_B(\boldsymbol{x}_A^+, \boldsymbol{x}_B^+)\right)\boldsymbol{\xi_B^+}(\boldsymbol{\xi_A^+})^\top d\right\}$$
$$- K\mathbb{E}\left\{\frac{S_A(\boldsymbol{x}_A^+, \boldsymbol{x}_B^+)\exp(\boldsymbol{f_A^+} \cdot \boldsymbol{f_B^-})}{\exp(\boldsymbol{f_A^+} \cdot \boldsymbol{f_B^+})}\boldsymbol{\xi_B^-}(\boldsymbol{\xi_A^+})^\top d\right\} - K\mathbb{E}\left\{\frac{S_B(\boldsymbol{x}_A^+, \boldsymbol{x}_B^+)\exp(\boldsymbol{f_A^-} \cdot \boldsymbol{f_B^+})}{\exp(\boldsymbol{f_A^+} \cdot \boldsymbol{f_B^+})}\boldsymbol{\xi_B^+}(\boldsymbol{\xi_A^-})^\top d\right\}$$
$$=: \mathrm{T}_1(\boldsymbol{Q}_2) + \mathrm{T}_2(\boldsymbol{Q}_2) + \mathrm{T}_3(\boldsymbol{Q}_2).$$

We now estimate each of these three terms. Again, the strategy is to leverage the symmetry of the distribution of $\boldsymbol{\xi}$ to argue that some part of the expectation is 0. For $T_1$, we have

$$\mathrm{T}_1([\boldsymbol{Q}_2]_{p,q}) = -\tilde{S}(1-\tilde{S})\,\mathbb{E}\left\{(2\delta_{+,\boldsymbol{\xi}+} + 2\delta_{\boldsymbol{\xi}+,+} - \delta_{\boldsymbol{\xi}+,T+} - \delta_{T+,\boldsymbol{\xi}+})\,[\boldsymbol{\xi}_B^+]_p[\boldsymbol{\xi}_A^+]_q d\right\}$$
$$\pm O\left(d^3\left(\delta_{\boldsymbol{AB},\perp} + \delta_{N/S}\right)\delta_{\xi,\perp}\delta_{N/S}\right).$$

Note that none of these $\delta$'s depends on both $\boldsymbol{\xi}_A^+$ and $\boldsymbol{\xi}_B^+$. Therefore, the first term is 0. Similarly, for $T_2$, we have

$$\mathrm{T}_2([\boldsymbol{Q}_2]_{p,q}) = -\frac{K\tilde{S}}{\tilde{E}}\,\mathbb{E}\left\{\left(\tilde{E}_{+,-}\left(\delta_{+,\boldsymbol{\xi}-} + \delta_{\boldsymbol{\xi}+,-} - \tilde{S}\left(\delta_{+,\boldsymbol{\xi}+} + \delta_{\boldsymbol{\xi}+,+}\right) - (1-\tilde{S})\delta_{\boldsymbol{\xi}+,T+}\right)\right)[\boldsymbol{\xi}_B^-]_p[\boldsymbol{\xi}_A^+]_q d\right\}$$
$$\pm O\left(d^3\left(\delta_{\boldsymbol{AB},\perp} + \delta_{N/S}\right)\delta_{\xi,\perp}\delta_{N/S}\right)$$
$$= \pm O\left(d^3\left(\delta_{\boldsymbol{AB},\perp} + \delta_{N/S}\right)\delta_{\xi,\perp}\delta_{N/S}\right).$$

The same is also true for $T_3$. Thus,

$$[\boldsymbol{Q}_2]_{p,q} = \pm O\left(d^3\left(\delta_{\boldsymbol{AB},\perp} + \delta_{N/S}\right)\delta_{\xi,\perp}\delta_{N/S}\right).$$

$\square$

*Proof of Lemma C.8.* Recall from Corollary A.4 that $Q_0$ is defined as

$$Q_0 := -\mathop{\mathbb{E}}_{\boldsymbol{x}_A^+,\boldsymbol{x}_B^+}\left\{\left(2 - S_A(\boldsymbol{x}_A^+,\boldsymbol{x}_B^+) - S_B(\boldsymbol{x}_A^+,\boldsymbol{x}_B^+)\right)\langle\boldsymbol{f}_A^+,\boldsymbol{f}_B^+\rangle\right\}$$
$$+ K\mathop{\mathbb{E}}_{\boldsymbol{x}_A^+,\boldsymbol{x}_B^{\pm}}\left\{\frac{S_A(\boldsymbol{x}_A^+,\boldsymbol{x}_B^+)\exp(\tau_t^2\boldsymbol{f}_A^+\cdot\boldsymbol{f}_B^-)}{\exp(\tau_t^2\boldsymbol{f}_A^+\cdot\boldsymbol{f}_B^+)}\langle\boldsymbol{f}_A^+,\boldsymbol{f}_B^-\rangle\right\}$$
$$+ K\mathop{\mathbb{E}}_{\boldsymbol{x}_A^{\pm},\boldsymbol{x}_B^+}\left\{\frac{S_B(\boldsymbol{x}_A^+,\boldsymbol{x}_B^+)\exp(\tau_t^2\boldsymbol{f}_A^-\cdot\boldsymbol{f}_B^+)}{\exp(\tau_t^2\boldsymbol{f}_A^+\cdot\boldsymbol{f}_B^+)}\langle\boldsymbol{f}_A^-,\boldsymbol{f}_B^+\rangle\right\}.$$

We write

$$\langle\boldsymbol{f}_A^+,\boldsymbol{f}_B^-\rangle = \frac{\langle\boldsymbol{K}_A\boldsymbol{z}^+,\boldsymbol{K}_B\boldsymbol{z}^-\rangle}{N_A N_B} \pm O\left(\kappa_0 d\delta_{N/S}\delta_{\xi,\perp}\right)$$
$$= \sum_{i,j\in[r]}\frac{[\boldsymbol{K}_{AB}]_{i,j}}{N_A N_B}z_i^+z_j^- \pm O\left(\kappa_0 d\delta_{N/S}\delta_{\xi,\perp}\right).$$

Therefore, we have

$$Q_0 = -\sum_{i,j\in[r]}\frac{[\boldsymbol{K}_{AB}]_{i,j}}{N_A N_B}[\boldsymbol{Q}_1]_{i,j} \pm O\left(\kappa_0 d\delta_{N/S}\delta_{\xi,\perp}\right)$$
$$= -\sum_{k=1}^{r}\frac{\kappa_k^2}{\|\boldsymbol{\kappa}\|^2}[\boldsymbol{Q}_1]_{k,k} \pm O\left(d^2\delta_{\boldsymbol{AB},\perp}^2 + \delta_- + \kappa_0 d\delta_{N/S}\delta_{\xi,\perp}\right).$$

$\square$

*Proof of Corollary C.9.* Recall from Lemma A.5 that

$$\frac{\mathrm{d}}{\mathrm{d}t}\|[\boldsymbol{K}_A]_p\|^2 = 2\frac{\langle[\boldsymbol{K}_A]_p,[\boldsymbol{K}_B\boldsymbol{Q}_1]_p\rangle}{N_A N_B d}\sigma_p^2 + 2\frac{\langle[\boldsymbol{K}_A]_p,[\boldsymbol{K}_{B,\boldsymbol{\xi}}\boldsymbol{Q}_{1,\boldsymbol{\xi}_B}]_p\rangle}{N_A N_B d}\sigma_p^2 + 2\frac{\|[\boldsymbol{K}_A]_p\|^2}{N_A^2 d}Q_0\sigma_p^2$$
$$=: \sum_{i=1}^{3}\mathrm{T}_i\left(\frac{\mathrm{d}}{\mathrm{d}t}\|[\boldsymbol{K}_A]_p\|^2\right).$$

By Lemma C.5, we have

$$\mathrm{T}_1\left(\frac{\mathrm{d}}{\mathrm{d}t}\|[\boldsymbol{K}_A]_p\|^2\right) = 2\sum_{k=1}^{r}\frac{\|[\boldsymbol{K}_A]_p\|\,\|[\boldsymbol{K}_B]_k\|}{N_A N_B d}\left\langle\overline{[\boldsymbol{K}_A]_p},\overline{[\boldsymbol{K}_B]_k}\right\rangle[\boldsymbol{Q}_1]_{k,p}\sigma_p^2$$
$$= \frac{2\sigma_p^2\|[\boldsymbol{K}_A]_p\|\,\|[\boldsymbol{K}_B]_p\|}{N_A N_B d}\left\langle\overline{[\boldsymbol{K}_A]_p},\overline{[\boldsymbol{K}_B]_p}\right\rangle[\boldsymbol{Q}_1]_{p,p}$$
$$\pm O\left(\frac{\sigma_p^2\kappa_p^2}{N_A N_B d}\kappa_0 d\delta_{\boldsymbol{AB},\perp}^2\right).$$

By Lemma C.6, we have

$$
\begin{aligned}
\mathrm{T}_2\left(\frac{\mathrm{d}}{\mathrm{d}t}\left\|[\boldsymbol{K_A}]_p\right\|^2\right) &= 2\sum_{k=1}^{d-r}\frac{\left\|[\boldsymbol{K_A}]_p\right\|\left\|[\boldsymbol{K_{B,\xi}}]_k\right\|}{N_A N_B d}\left\langle\overline{[\boldsymbol{K_A}]_p}, \overline{[\boldsymbol{K_{B,\xi}}]_k}\right\rangle[\boldsymbol{Q}_{1,\xi_B}]_{k,p}\sigma_p^2 \\
&= O\left(\frac{\sigma_p^2\kappa_p^2}{N_A N_B d}d\delta_{N/S}^2\delta_{\xi,\perp}^2\right).
\end{aligned}
$$

Combine these together and we obtain

$$
\begin{aligned}
\frac{\mathrm{d}}{\mathrm{d}t}\left\|[\boldsymbol{K_A}]_p\right\|^2 &= \frac{2\sigma_p^2\left\|[\boldsymbol{K_A}]_p\right\|\left\|[\boldsymbol{K_B}]_p\right\|}{N_A N_B d}\left\langle\overline{[\boldsymbol{K_A}]_p}, \overline{[\boldsymbol{K_B}]_p}\right\rangle[\boldsymbol{Q}_1]_{p,p} + 2\frac{\left\|[\boldsymbol{K_A}]_p\right\|^2}{N_A^2 d}Q_0\sigma_p^2 \\
&\pm O\left(\frac{\sigma_p^2\kappa_p^2}{N_A N_B d}\kappa_0 d\delta_{\boldsymbol{AB},\perp}^2\right).
\end{aligned}
$$

To get the formula for $\frac{\mathrm{d}}{\mathrm{d}t}\left\|[\boldsymbol{K_B}]_p\right\|^2$, it suffices to interchange the roles of $\boldsymbol{A}$ and $\boldsymbol{B}$. Similarly, for the noises, we write

$$
\begin{aligned}
\frac{\mathrm{d}}{\mathrm{d}t}\left\|[\boldsymbol{K_{A,\xi}}]_q\right\|^2 &= \frac{2\left\langle[\boldsymbol{K_{A,\xi}}]_q, [\boldsymbol{K_B Q}_{1,\xi_A}^\top]_q\right\rangle}{N_A N_B d}\sigma_\xi^2 + \frac{2\left\langle[\boldsymbol{K_{A,\xi}}]_q, [\boldsymbol{K_{B,\xi}Q}_2]_q\right\rangle}{N_A N_B d}\sigma_\xi^2 + \frac{2\left\|[\boldsymbol{K_{A,\xi}}]_q\right\|_F^2}{N_A^2 d}Q_0\sigma_\xi^2 \\
&= \sum_{i=1}^3 \mathrm{T}_i\left(\frac{\mathrm{d}}{\mathrm{d}t}\left\|[\boldsymbol{K_{A,\xi}}]_q\right\|^2\right).
\end{aligned}
$$

By Lemma C.6, we have

$$
\begin{aligned}
\mathrm{T}_1\left(\frac{\mathrm{d}}{\mathrm{d}t}\left\|[\boldsymbol{K_{A,\xi}}]_q\right\|^2\right) &= 2\sum_{k=1}^{r}\frac{\left\|[\boldsymbol{K_{A,\xi}}]_q\right\|\left\|[\boldsymbol{K_B}]_k\right\|}{N_A N_B d}\left\langle\overline{[\boldsymbol{K_{A,\xi}}]_q}, \overline{[\boldsymbol{K_B}]_k}\right\rangle[\boldsymbol{Q}_{1,\xi_A}]_{q,k}\sigma_\xi^2 \\
&= O\left(\frac{\sigma_\xi^2\left\|[\boldsymbol{K_{A,\xi}}]_q\right\|^2}{N_A N_B d}d\delta_{\xi,\perp}^2\right).
\end{aligned}
$$

By Lemma C.7, we have

$$
\begin{aligned}
\mathrm{T}_1\left(\frac{\mathrm{d}}{\mathrm{d}t}\left\|[\boldsymbol{K_{A,\xi}}]_q\right\|^2\right) &= 2\sum_{k=1}^{d-r}\frac{\left\|[\boldsymbol{K_{A,\xi}}]_q\right\|\left\|[\boldsymbol{K_{B,\xi}}]_k\right\|}{N_A N_B d}\left\langle\overline{[\boldsymbol{K_{A,\xi}}]_q}, \overline{[\boldsymbol{K_{B,\xi}}]_k}\right\rangle[\boldsymbol{Q}_2]_{k,q}\sigma_\xi^2 \\
&= O\left(\frac{\sigma_\xi^2\left\|[\boldsymbol{K_{A,\xi}}]_q\right\|^2}{N_A N_B d}d^4\left(\delta_{\boldsymbol{AB},\perp} + \delta_{N/S}\right)\delta_{\xi,\perp}^2\delta_{N/S}\right).
\end{aligned}
$$

Combine these together and we get

$$
\frac{\mathrm{d}}{\mathrm{d}t}\left\|[\boldsymbol{K_{A,\xi}}]_q\right\|^2 = \frac{2\left\|[\boldsymbol{K_{A,\xi}}]_q\right\|^2}{N_A^2 d}Q_0\sigma_\xi^2 \pm O\left(\frac{\sigma_\xi^2\left\|[\boldsymbol{K_{A,\xi}}]_q\right\|^2}{N_A N_B d}d\delta_{\xi,\perp}^2\right).
$$

$\square$

## C.2 MAINTAINING THE ORTHOGONALITY

In this subsection, we control the growth of $\delta_{\boldsymbol{AB},\perp}$ and $\delta_{\xi,\perp}$. First, we derive the equations that govern the evolution of the off-diagonal terms.

**Lemma C.10.** *In Stage 2, for any $p \neq q \in [r]$, we have*

$$\frac{\mathrm{d}}{\mathrm{d}t} \left\langle \overline{[\boldsymbol{K_A}]_p}, \overline{[\boldsymbol{K_B}]_q} \right\rangle = [\boldsymbol{Q}_1]_{q,p} \frac{\kappa_q/\kappa_p \sigma_p^2 + \kappa_p/\kappa_q \sigma_q^2}{N_{\boldsymbol{A}} N_{\boldsymbol{B}} d}$$
$$- \left\langle \overline{[\boldsymbol{K_A}]_p}, \overline{[\boldsymbol{K_B}]_q} \right\rangle \left( \frac{[\boldsymbol{Q}_1]_{p,p} \sigma_p^2}{N_{\boldsymbol{A}} N_{\boldsymbol{B}} d} + \frac{[\boldsymbol{Q}_1]_{q,q} \sigma_q^2}{N_{\boldsymbol{A}} N_{\boldsymbol{B}} d} \right)$$
$$+ \left\langle \overline{[\boldsymbol{K_A}]_p}, \overline{[\boldsymbol{K_A}]_q} \right\rangle \frac{[\boldsymbol{Q}_1]_{q,q} \sigma_q^2}{N_{\boldsymbol{A}} N_{\boldsymbol{B}} d} + \left\langle \overline{[\boldsymbol{K_B}]_p}, \overline{[\boldsymbol{K_B}]_q} \right\rangle \frac{[\boldsymbol{Q}_1]_{p,p} \sigma_p^2}{N_{\boldsymbol{A}} N_{\boldsymbol{B}} d}$$
$$\pm O \left( \frac{\sigma_{\max}^2}{N_{\boldsymbol{A}} N_{\boldsymbol{B}} d} d \kappa_0 \delta_{\boldsymbol{AB},\perp} \left( \sqrt{\delta_-} + \delta_{\boldsymbol{AB},\perp} \right) \right).$$

**Lemma C.11.** *In Stage 2, for any $p \neq q \in [r]$, we have*

$$\frac{\mathrm{d}}{\mathrm{d}t} \left\langle \overline{[\boldsymbol{K_A}]_p}, \overline{[\boldsymbol{K_A}]_q} \right\rangle = [\boldsymbol{Q}_1]_{q,p} \frac{\kappa_q/\kappa_p \sigma_p^2}{N_{\boldsymbol{A}} N_{\boldsymbol{B}} d} + [\boldsymbol{Q}_1]_{p,q} \frac{\kappa_p/\kappa_q \sigma_q^2}{N_{\boldsymbol{A}} N_{\boldsymbol{B}} d}$$
$$- \left\langle \overline{[\boldsymbol{K_A}]_p}, \overline{[\boldsymbol{K_A}]_q} \right\rangle \left( \frac{[\boldsymbol{Q}_1]_{p,p} \sigma_p^2}{N_{\boldsymbol{A}} N_{\boldsymbol{B}} d} + \frac{[\boldsymbol{Q}_1]_{q,q} \sigma_q^2}{N_{\boldsymbol{A}} N_{\boldsymbol{B}} d} \right)$$
$$+ \left\langle \overline{[\boldsymbol{K_A}]_p}, \overline{[\boldsymbol{K_B}]_q} \right\rangle \frac{[\boldsymbol{Q}_1]_{q,q} \sigma_q^2}{N_{\boldsymbol{A}} N_{\boldsymbol{B}} d} + \left\langle \overline{[\boldsymbol{K_B}]_p}, \overline{[\boldsymbol{K_A}]_q} \right\rangle \frac{[\boldsymbol{Q}_1]_{p,p} \sigma_p^2}{N_{\boldsymbol{A}} N_{\boldsymbol{B}} d}$$
$$\pm \left( \frac{\sigma_{\max}^2}{N_{\boldsymbol{A}} N_{\boldsymbol{B}} d} d \kappa_0 \delta_{\boldsymbol{AB},\perp} \left( \sqrt{\delta_-} + \delta_{\boldsymbol{AB},\perp} \right) \right),$$

*and*

$$\frac{\mathrm{d}}{\mathrm{d}t} \left\langle \overline{[\boldsymbol{K_B}]_p}, \overline{[\boldsymbol{K_B}]_q} \right\rangle = [\boldsymbol{Q}_1]_{p,q} \frac{\kappa_q/\kappa_p \sigma_p^2}{N_{\boldsymbol{A}} N_{\boldsymbol{B}} d} + [\boldsymbol{Q}_1]_{q,p} \frac{\kappa_p/\kappa_q \sigma_q^2}{N_{\boldsymbol{A}} N_{\boldsymbol{B}} d}$$
$$- \left\langle \overline{[\boldsymbol{K_B}]_p}, \overline{[\boldsymbol{K_B}]_q} \right\rangle \left( \frac{[\boldsymbol{Q}_1]_{p,p} \sigma_p^2}{N_{\boldsymbol{A}} N_{\boldsymbol{B}} d} + \frac{[\boldsymbol{Q}_1]_{q,q} \sigma_q^2}{N_{\boldsymbol{A}} N_{\boldsymbol{B}} d} \right)$$
$$+ \left\langle \overline{[\boldsymbol{K_B}]_p}, \overline{[\boldsymbol{K_A}]_q} \right\rangle \frac{[\boldsymbol{Q}_1]_{q,q} \sigma_q^2}{N_{\boldsymbol{A}} N_{\boldsymbol{B}} d} + \left\langle \overline{[\boldsymbol{K_A}]_p}, \overline{[\boldsymbol{K_B}]_q} \right\rangle \frac{[\boldsymbol{Q}_1]_{p,p} \sigma_p^2}{N_{\boldsymbol{A}} N_{\boldsymbol{B}} d}$$
$$\pm \left( \frac{\sigma_{\max}^2}{N_{\boldsymbol{A}} N_{\boldsymbol{B}} d} d \kappa_0 \delta_{\boldsymbol{AB},\perp} \left( \sqrt{\delta_-} + \delta_{\boldsymbol{AB},\perp} \right) \right).$$

Now, we are ready to control the off-diagonalness. The proof is similar to the one of Lemma B.15. To provide intuitions, we first consider an idealized case. That is, we assume here the noise-signal ratio is 0 and $\boldsymbol{K_A} = \boldsymbol{K_B}$, and explain how to control $\left\langle \overline{[\boldsymbol{K_A}]_p}, \overline{[\boldsymbol{K_B}]_q} \right\rangle$ for $p \neq q \in [r]$ under these assumptions. In this case, we have

$$\frac{\mathrm{d}}{\mathrm{d}t} \left\langle \overline{[\boldsymbol{K_A}]_p}, \overline{[\boldsymbol{K_B}]_q} \right\rangle \approx [\boldsymbol{Q}_1]_{q,p} \frac{\kappa_q/\kappa_p \sigma_p^2 + \kappa_p/\kappa_q \sigma_q^2}{N_{\boldsymbol{A}} N_{\boldsymbol{B}} d},$$

and

$$[\boldsymbol{Q}_1]_{p,q} \approx -(1 - \tilde{S}) \left( 2 - T_p^2 - T_q^2 + \tilde{S}(2 - T_p - T_q)^2 \right) \frac{[\boldsymbol{K_{AB}}]_{p,q}}{N_{\boldsymbol{A}} N_{\boldsymbol{B}} d}.$$

Note that the coefficient is negative. As a result, $\left\langle \overline{[\boldsymbol{K_A}]_p}, \overline{[\boldsymbol{K_B}]_q} \right\rangle$ will move towards 0.

When $\boldsymbol{K_A} = \boldsymbol{K_B}$ is not exactly true, the situation is trickier because we need to deal with the middle four terms of the RHS of $\frac{\mathrm{d}}{\mathrm{d}t} \left\langle \overline{[\boldsymbol{K_A}]_p}, \overline{[\boldsymbol{K_B}]_q} \right\rangle$. The idea is to view all these off-diagonal errors as a whole and show that their sum is non-increasing, up to some higher-order terms.

**Lemma C.12** (Orthogonality between signals). *For any $p \neq q$, define*

$$\hat{\delta}_{\perp,p,q} := \left\langle \overline{[\boldsymbol{K_A}]_p}, \overline{[\boldsymbol{K_B}]_q} \right\rangle^2 + \left\langle \overline{[\boldsymbol{K_B}]_p}, \overline{[\boldsymbol{K_A}]_q} \right\rangle^2 + \left\langle \overline{[\boldsymbol{K_A}]_p}, \overline{[\boldsymbol{K_A}]_q} \right\rangle^2 + \left\langle \overline{[\boldsymbol{K_B}]_p}, \overline{[\boldsymbol{K_B}]_q} \right\rangle^2.$$

*In Stage 2, we have*

$$\frac{\mathrm{d}}{\mathrm{d}t}\hat{\delta}_{\perp,p,q} \leq O\left(\frac{\sigma_{\max}^2}{N_A N_B d}\kappa_0\left(d^2\delta_{\boldsymbol{AB},\perp}^2 + \delta_{\boldsymbol{AB},\perp}\sqrt{\delta_-} + \delta_-\right)\sqrt{\hat{\delta}_{\perp,p,q}}\right).$$

Note that the LHS is of order $\delta_{\boldsymbol{AB},\perp}^2$ and for the RHS, the only term that can potentially have the same order is the $\left(\delta_-\sqrt{\hat{\delta}_{\perp,p,q}}\right)$-related term. We will show later that $\delta_- = o(\delta_{\boldsymbol{AB},\perp})$, whence this is also a higher order term.

Then, we consider the orthogonality between signals and noises.

**Lemma C.13** (Orthogonality between signals and noises)**.** *For any $p \in [r]$ and $q \in [d-r]$, define*

$$\hat{\delta}_{\xi,\perp,p,q} := \left\langle\overline{[\boldsymbol{K_A}]_p}, \overline{[\boldsymbol{K_{A,\xi}}]_q}\right\rangle^2 + \left\langle\overline{[\boldsymbol{K_B}]_p}, \overline{[\boldsymbol{K_{A,\xi}}]_q}\right\rangle^2$$
$$+ \left\langle\overline{[\boldsymbol{K_A}]_p}, \overline{[\boldsymbol{K_{B,\xi}}]_q}\right\rangle^2 + \left\langle\overline{[\boldsymbol{K_B}]_p}, \overline{[\boldsymbol{K_{B,\xi}}]_q}\right\rangle^2.$$

*In Stage 2, we have*

$$\frac{\mathrm{d}}{\mathrm{d}t}\hat{\delta}_{\xi,\perp,p,q} \leq O\left(\frac{\sigma_{\max}^2}{N_A N_B d}d\kappa_0\delta_{\xi,\perp}^2\left(\delta_{\xi,\perp} + \sqrt{\delta_-}\right)\right).$$

Finally, we deal with the orthogonality between the noises.

**Lemma C.14** (Orthogonality between noises)**.** *In Stage 2, for any $p, q \in [d-r]$, we have*

$$\frac{\mathrm{d}}{\mathrm{d}t}\left\langle\overline{[\boldsymbol{K_{A,\xi}}]_p}, \overline{[\boldsymbol{K_{B,\xi}}]_q}\right\rangle = \pm O\left(\frac{\sigma_\xi^2}{N_A N_B d}d\delta_{\xi,\perp}^2\right).$$

OMITTED PROOF OF THIS SUBSECTION

*Proof of Lemma C.10.* Recall that

$$\frac{\mathrm{d}}{\mathrm{d}t}\overline{[\boldsymbol{K_A}]_p} = \left(\boldsymbol{I} - \overline{[\boldsymbol{K_A}]_p}\left(\overline{[\boldsymbol{K_A}]_p}\right)^\top\right)\left(\frac{[\boldsymbol{K_B Q_1}]_p}{\|[\boldsymbol{K_A}]_p\|} + \frac{[\boldsymbol{K_{B,\xi} Q_{1,\xi_B}}]_p}{\|[\boldsymbol{K_A}]_p\|}\right)\frac{\sigma_p^2}{N_A N_B d},$$

$$\frac{\mathrm{d}}{\mathrm{d}t}\overline{[\boldsymbol{K_B}]_q} = \left(\boldsymbol{I} - \overline{[\boldsymbol{K_B}]_q}\left(\overline{[\boldsymbol{K_B}]_q}\right)^\top\right)\left(\frac{[\boldsymbol{K_A Q_1^\top}]_q}{\|[\boldsymbol{K_B}]_q\|} + \frac{[\boldsymbol{K_{A,\xi} Q_{1,\xi_A}}]_q}{\|[\boldsymbol{K_B}]_q\|}\right)\frac{\sigma_q^2}{N_A N_B d}.$$

Then, we write

$$\left\langle\overline{[\boldsymbol{K_A}]_p}, \frac{\mathrm{d}}{\mathrm{d}t}\overline{[\boldsymbol{K_B}]_q}\right\rangle = \overline{[\boldsymbol{K_A}]_p}^\top\left(\boldsymbol{I} - \overline{[\boldsymbol{K_B}]_q}\left(\overline{[\boldsymbol{K_B}]_q}\right)^\top\right)\frac{[\boldsymbol{K_A Q_1^\top}]_q}{\|[\boldsymbol{K_B}]_q\|}\frac{\sigma_q^2}{N_A N_B d}$$

$$+ \overline{[\boldsymbol{K_A}]_p}^\top\left(\boldsymbol{I} - \overline{[\boldsymbol{K_B}]_q}\left(\overline{[\boldsymbol{K_B}]_q}\right)^\top\right)\frac{[\boldsymbol{K_{A,\xi} Q_{1,\xi_A}}]_q}{\|[\boldsymbol{K_B}]_q\|}\frac{\sigma_q^2}{N_A N_B d}.$$

For the second term, by Lemma C.6, we have

$$\overline{[\boldsymbol{K_A}]_p}^\top\left(\boldsymbol{I} - \overline{[\boldsymbol{K_B}]_q}\left(\overline{[\boldsymbol{K_B}]_q}\right)^\top\right)\frac{[\boldsymbol{K_{A,\xi} Q_{1,\xi_A}}]_q}{\|[\boldsymbol{K_B}]_q\|}$$

$$= \sum_{k=1}^{d-r}\left(\left\langle\overline{[\boldsymbol{K_A}]_p}, \overline{[\boldsymbol{K_{A,\xi}}]_k}\right\rangle - \left\langle\overline{[\boldsymbol{K_A}]_p}, \overline{[\boldsymbol{K_B}]_q}\right\rangle\left\langle\overline{[\boldsymbol{K_B}]_q}, \overline{[\boldsymbol{K_{A,\xi}}]_k}\right\rangle\right)\frac{\|[\boldsymbol{K_{A,\xi}}]_k\|\,[\boldsymbol{Q_{1,\xi_A}}]_{k,q}}{\|[\boldsymbol{K_B}]_q\|}$$

$$= \pm O\left(d\delta_{\xi,\perp}^2\delta_{N/S}^2\right).$$

Hence,

$$\left\langle\overline{[\boldsymbol{K_A}]_p}, \frac{\mathrm{d}}{\mathrm{d}t}\overline{[\boldsymbol{K_B}]_q}\right\rangle = \overline{[\boldsymbol{K_A}]_p}^\top\left(\boldsymbol{I} - \overline{[\boldsymbol{K_B}]_q}\left(\overline{[\boldsymbol{K_B}]_q}\right)^\top\right)\frac{[\boldsymbol{K_A Q_1^\top}]_q}{\|[\boldsymbol{K_B}]_q\|}\frac{\sigma_q^2}{N_A N_B d}$$

$$\pm O\left(\frac{\sigma_{\max}^2}{N_A N_B d}d\delta_{\xi,\perp}^2\delta_{N/S}^2\right).$$

For the first term, we have

$$\overline{[K_A]_p}^\top \left(I - \overline{[K_B]_q}\left(\overline{[K_B]_q}\right)^\top\right) \frac{[K_A Q_1^\top]_q}{\|[K_B]_q\|}$$

$$= \sum_{k=1}^{r} \left(\left\langle \overline{[K_A]_p}, \overline{[K_A]_k}\right\rangle - \left\langle \overline{[K_A]_p}, \overline{[K_B]_q}\right\rangle \left\langle \overline{[K_B]_q}, \overline{[K_A]_k}\right\rangle\right) \frac{\|[K_A]_k\| [Q_1]_{q,k}}{\|[K_B]_q\|}.$$

When $k \notin \{p,q\}$, we have $\left|\left\langle \overline{[K_A]_p}, \overline{[K_A]_k}\right\rangle\right| \le O(\delta_{AB,\perp})$, $\left|\left\langle \overline{[K_B]_q}, \overline{[K_A]_k}\right\rangle\right| \le O(\delta_{AB,\perp})$, and $\|[K_A]_k\| / \|[K_B]_q\| \le \kappa_0$. Meanwhile, by Lemma C.5, we also have $|[Q_1]_{q,k}| \le O(\delta_{AB,\perp})$. Hence,

$$\left| \sum_{k \notin \{p,q\}} \left(\left\langle \overline{[K_A]_p}, \overline{[K_A]_k}\right\rangle - \left\langle \overline{[K_A]_p}, \overline{[K_B]_q}\right\rangle \left\langle \overline{[K_B]_q}, \overline{[K_A]_k}\right\rangle\right) \frac{\|[K_A]_k\| [Q_1]_{q,k}}{\|[K_B]_q\|} \right|$$
$$\le O\left(d\kappa_0 \delta_{AB,\perp}^2\right).$$

Therefore,

$$\left\langle \overline{[K_A]_p}, \frac{\mathrm{d}}{\mathrm{d}t}\overline{[K_B]_q}\right\rangle \left(\frac{\sigma_q^2}{N_A N_B d}\right)^{-1}$$

$$= \left(1 - \left\langle \overline{[K_A]_p}, \overline{[K_B]_q}\right\rangle \left\langle \overline{[K_B]_q}, \overline{[K_A]_p}\right\rangle\right) \frac{\|[K_A]_p\| [Q_1]_{q,p}}{\|[K_B]_q\|}$$

$$+ \left(\left\langle \overline{[K_A]_p}, \overline{[K_A]_q}\right\rangle - \left\langle \overline{[K_A]_p}, \overline{[K_B]_q}\right\rangle \left\langle \overline{[K_B]_q}, \overline{[K_A]_q}\right\rangle\right) \frac{\|[K_A]_q\| [Q_1]_{q,q}}{\|[K_B]_q\|}$$

$$\pm O\left(d\kappa_0 \delta_{AB,\perp}^2\right) \pm O\left(d\delta_{\xi,\perp}^2 \delta_{N/S}^2\right).$$

For the first term, we have

$$\left(1 - \left\langle \overline{[K_A]_p}, \overline{[K_B]_q}\right\rangle \left\langle \overline{[K_B]_q}, \overline{[K_A]_p}\right\rangle\right) \frac{\|[K_A]_p\| [Q_1]_{q,p}}{\|[K_B]_q\|}$$

$$= \left(1 \pm \delta_{AB,\perp}^2\right) \frac{\kappa_p [Q_1]_{q,p}}{\kappa_q} \left(1 \pm \sqrt{\delta_-}\right)$$

$$= \frac{\kappa_p [Q_1]_{q,p}}{\kappa_q} \pm O\left(\kappa_0 \delta_{AB,\perp} \sqrt{\delta_-}\right) \pm O\left(\kappa_0 \delta_{AB,\perp}^3\right).$$

For the second term, we have

$$\left(\left\langle \overline{[K_A]_p}, \overline{[K_A]_q}\right\rangle - \left\langle \overline{[K_A]_p}, \overline{[K_B]_q}\right\rangle \left\langle \overline{[K_B]_q}, \overline{[K_A]_q}\right\rangle\right) \frac{\|[K_A]_q\| [Q_1]_{q,q}}{\|[K_B]_q\|}$$

$$= \left(\left\langle \overline{[K_A]_p}, \overline{[K_A]_q}\right\rangle - \left\langle \overline{[K_A]_p}, \overline{[K_B]_q}\right\rangle (1 \pm \delta_-)\right) [Q_1]_{q,q} \left(1 \pm \sqrt{\delta_-}\right)$$

$$= \left(\left\langle \overline{[K_A]_p}, \overline{[K_A]_q}\right\rangle - \left\langle \overline{[K_A]_p}, \overline{[K_B]_q}\right\rangle\right) [Q_1]_{q,q} \pm O\left(\delta_\perp \sqrt{\delta_-}\right).$$

Thus,

$$\left\langle \overline{[K_A]_p}, \frac{\mathrm{d}}{\mathrm{d}t}\overline{[K_B]_q}\right\rangle \left(\frac{\sigma_q^2}{N_A N_B d}\right)^{-1}$$

$$= \frac{\kappa_p [Q_1]_{q,p}}{\kappa_q} \pm O\left(\kappa_0 \delta_{AB,\perp} \sqrt{\delta_-}\right) \pm O\left(\kappa_0 \delta_{AB,\perp}^3\right)$$

$$+ \left(\left\langle \overline{[K_A]_p}, \overline{[K_A]_q}\right\rangle - \left\langle \overline{[K_A]_p}, \overline{[K_B]_q}\right\rangle\right) [Q_1]_{q,q} \pm O\left(\delta_\perp \sqrt{\delta_-}\right)$$

$$\pm O\left(d\kappa_0 \delta_{AB,\perp}^2\right) \pm O\left(d\delta_{\xi,\perp}^2 \delta_{N/S}^2\right)$$

$$= \frac{\kappa_p [Q_1]_{q,p}}{\kappa_q} + \left(\left\langle \overline{[K_A]_p}, \overline{[K_A]_q}\right\rangle - \left\langle \overline{[K_A]_p}, \overline{[K_B]_q}\right\rangle\right) [Q_1]_{q,q}$$

$$\pm O\left(d\kappa_0 \delta_{AB,\perp}\left(\sqrt{\delta_-} + \delta_{AB,\perp}\right)\right).$$

Hence,

$$
\left\langle \overline{[K_A]_p}, \frac{\mathrm{d}}{\mathrm{d}t}\overline{[K_B]_q} \right\rangle = \frac{\kappa_p [Q_1]_{q,p}}{\kappa_q} \frac{\sigma_q^2}{N_A N_B d} + \left( \left\langle \overline{[K_A]_p}, \overline{[K_A]_q} \right\rangle - \left\langle \overline{[K_A]_p}, \overline{[K_B]_q} \right\rangle \right) \frac{[Q_1]_{q,q}\sigma_q^2}{N_A N_B d}
$$
$$
\pm O\left( \frac{\sigma_{\max}^2}{N_A N_B d} d\kappa_0 \delta_{AB,\perp} \left( \sqrt{\delta_-} + \delta_{AB,\perp} \right) \right).
$$

Interchange the roles of $A$, $B$, $p$, $q$, replace $Q_1$ with $Q_1^\top$, and we obtain

$$
\left\langle \frac{\mathrm{d}}{\mathrm{d}t}\overline{[K_A]_p}, \overline{[K_B]_q} \right\rangle = \frac{\kappa_q [Q_1]_{q,p}}{\kappa_p} \frac{\sigma_p^2}{N_A N_B d} + \left( \left\langle \overline{[K_B]_p}, \overline{[K_B]_q} \right\rangle - \left\langle \overline{[K_A]_p}, \overline{[K_B]_q} \right\rangle \right) \frac{[Q_1]_{p,p}\sigma_p^2}{N_A N_B d}
$$
$$
\pm O\left( \frac{\sigma_{\max}^2}{N_A N_B d} d\kappa_0 \delta_{AB,\perp} \left( \sqrt{\delta_-} + \delta_{AB,\perp} \right) \right).
$$

Combine these together and we complete the proof. $\qquad\square$

*Proof of Lemma C.11.* Similar to the proof of the previous lemma, we compute

$$
\left\langle \frac{\mathrm{d}}{\mathrm{d}t}\overline{[K_A]_p}, \overline{[K_A]_q} \right\rangle = \overline{[K_A]_q}^\top \left( I - \overline{[K_A]_p}\left(\overline{[K_A]_p}\right)^\top \right) \left( \frac{[K_B Q_1]_p}{\|[K_A]_p\|} + \frac{[K_{B,\xi} Q_{1,\xi_B}]_p}{\|[K_A]_p\|} \right) \frac{\sigma_p^2}{N_A N_B d}
$$
$$
= \overline{[K_A]_q}^\top \left( I - \overline{[K_A]_p}\left(\overline{[K_A]_p}\right)^\top \right) \frac{[K_B Q_1]_p}{\|[K_A]_p\|} \frac{\sigma_p^2}{N_A N_B d} \pm O\left( \frac{\sigma_{\max}^2}{N_A N_B d} d\delta_{\xi,\perp}^2 \delta_{N/S}^2 \right).
$$

Again, we have

$$
\overline{[K_A]_q}^\top \left( I - \overline{[K_A]_p}\left(\overline{[K_A]_p}\right)^\top \right) \frac{[K_B Q_1]_p}{\|[K_A]_p\|}
$$
$$
= \overline{[K_A]_q}^\top \left( I - \overline{[K_A]_p}\left(\overline{[K_A]_p}\right)^\top \right) \overline{[K_B]_p} \frac{\|[K_B]_p\|\,[Q_1]_{p,p}}{\|[K_A]_p\|}
$$
$$
\quad + \overline{[K_A]_q}^\top \left( I - \overline{[K_A]_p}\left(\overline{[K_A]_p}\right)^\top \right) \overline{[K_B]_q} \frac{\|[K_B]_q\|\,[Q_1]_{q,p}}{\|[K_A]_p\|}
$$
$$
\quad + \sum_{k \notin \{p,q\}} \overline{[K_A]_q}^\top \left( I - \overline{[K_A]_p}\left(\overline{[K_A]_p}\right)^\top \right) \overline{[K_B]_k} \frac{\|[K_B]_k\|\,[Q_1]_{k,p}}{\|[K_A]_p\|}
$$
$$
= \frac{\kappa_q [Q_1]_{q,p}}{\kappa_p} + \left( \left\langle \overline{[K_B]_p}, \overline{[K_A]_q} \right\rangle - \left\langle \overline{[K_A]_p}, \overline{[K_A]_q} \right\rangle \right) [Q_1]_{p,p} \pm \left( d\kappa_0 \delta_{AB,\perp} \left( \sqrt{\delta_-} + \delta_{AB,\perp} \right) \right).
$$

Therefore,

$$
\left\langle \frac{\mathrm{d}}{\mathrm{d}t}\overline{[K_A]_p}, \overline{[K_A]_q} \right\rangle = [Q_1]_{q,p} \frac{\kappa_q/\kappa_p \sigma_p^2}{N_A N_B d} + \left( \left\langle \overline{[K_B]_p}, \overline{[K_A]_q} \right\rangle - \left\langle \overline{[K_A]_p}, \overline{[K_A]_q} \right\rangle \right) \frac{[Q_1]_{p,p}\sigma_p^2}{N_A N_B d}
$$
$$
\pm \left( \frac{\sigma_{\max}^2}{N_A N_B d} d\kappa_0 \delta_{AB,\perp} \left( \sqrt{\delta_-} + \delta_{AB,\perp} \right) \right).
$$

Interchange the roles of $p$, $q$ and we obtain

$$
\left\langle \overline{[K_A]_p}, \frac{\mathrm{d}}{\mathrm{d}t}\overline{[K_A]_q} \right\rangle = [Q_1]_{p,q} \frac{\kappa_p/\kappa_q \sigma_q^2}{N_A N_B d} + \left( \left\langle \overline{[K_B]_q}, \overline{[K_A]_p} \right\rangle - \left\langle \overline{[K_A]_p}, \overline{[K_A]_q} \right\rangle \right) \frac{[Q_1]_{q,q}\sigma_q^2}{N_A N_B d}
$$
$$
\pm \left( \frac{\sigma_{\max}^2}{N_A N_B d} d\kappa_0 \delta_{AB,\perp} \left( \sqrt{\delta_-} + \delta_{AB,\perp} \right) \right).
$$

Combine these together and we get

$$
\begin{aligned}
\frac{\mathrm{d}}{\mathrm{d}t}\left\langle \overline{[\boldsymbol{K_A}]_p}, \overline{[\boldsymbol{K_A}]_q} \right\rangle &= [\boldsymbol{Q}_1]_{q,p}\frac{\kappa_q/\kappa_p\sigma_p^2}{N_A N_B d} + [\boldsymbol{Q}_1]_{p,q}\frac{\kappa_p/\kappa_q\sigma_q^2}{N_A N_B d} \\
&\quad - \left\langle \overline{[\boldsymbol{K_A}]_p}, \overline{[\boldsymbol{K_A}]_q} \right\rangle \left( \frac{[\boldsymbol{Q}_1]_{p,p}\sigma_p^2}{N_A N_B d} + \frac{[\boldsymbol{Q}_1]_{q,q}\sigma_q^2}{N_A N_B d} \right) \\
&\quad + \left\langle \overline{[\boldsymbol{K_A}]_p}, \overline{[\boldsymbol{K_B}]_q} \right\rangle \frac{[\boldsymbol{Q}_1]_{q,q}\sigma_q^2}{N_A N_B d} + \left\langle \overline{[\boldsymbol{K_B}]_p}, \overline{[\boldsymbol{K_A}]_q} \right\rangle \frac{[\boldsymbol{Q}_1]_{p,p}\sigma_p^2}{N_A N_B d} \\
&\quad \pm \left( \frac{\sigma_{\max}^2}{N_A N_B d} d\kappa_0 \delta_{\boldsymbol{AB},\perp}\left( \sqrt{\delta_-} + \delta_{\boldsymbol{AB},\perp} \right) \right).
\end{aligned}
$$

Interchange the roles of $\boldsymbol{A}, \boldsymbol{B}$, replace $\boldsymbol{Q}_1$ with $\boldsymbol{Q}_1^\top$, and we obtain the formula for $\frac{\mathrm{d}}{\mathrm{d}t}\left\langle \overline{[\boldsymbol{K_B}]_p}, \overline{[\boldsymbol{K_B}]_q} \right\rangle.$ □

*Proof of Lemma C.12.* First, we consider the $[\boldsymbol{Q}_1]_{p,q}$-related terms, by Lemma C.5, we have

$$
\begin{aligned}
[\boldsymbol{Q}_1]_{p,q} &= -\tilde{S}(1-\tilde{S})\frac{[\boldsymbol{K_{AB}}]_{p,q} + [\boldsymbol{K_{BA}}]_{q,p}}{N_A N_B d}(2 - T_p - T_q) \\
&\quad - (1-\tilde{S})\frac{[\boldsymbol{K_{AB}}]_{p,q}}{N_A N_B d}\tilde{S}\left( 2T_p T_q - T_p - T_q \right) \\
&\quad - (1-\tilde{S})\frac{[\boldsymbol{K_{AB}}]_{q,p}}{N_A N_B d}\left( 2 - \tilde{S}(T_p + T_q) - (1-\tilde{S})(T_p^2 + T_q^2) \right) \pm O\left( d^2 \delta_{\boldsymbol{AB},\perp}^2 \right) \\
&= O\left( \frac{\kappa_p \kappa_q \delta_{\boldsymbol{AB},\perp}}{N_A N_B d} \right).
\end{aligned}
$$

Hence,

$$
[\boldsymbol{Q}_1]_{p,q}\frac{\kappa_q/\kappa_p\sigma_p^2}{N_A N_B d} = O\left( \frac{\kappa_q^2}{N_A N_B d}\frac{\sigma_p^2}{N_A N_B d}\delta_{\boldsymbol{AB},\perp} \right) = O\left( \frac{1}{\sqrt{d}}\frac{\sigma_{\max}^2}{N_A N_B d}\delta_{\boldsymbol{AB},\perp} \right).
$$

The same bound also hold for other $[\boldsymbol{Q}_1]_{p,q}$-related terms. The important thing here is that we have and additional $1/\sqrt{d}$ factor.

Now, we are ready to prove the result. For notational simplicity, define $Z_1 = \left\langle \overline{[\boldsymbol{K_A}]_p}, \overline{[\boldsymbol{K_B}]_q} \right\rangle$, $Z_2 = \left\langle \overline{[\boldsymbol{K_B}]_p}, \overline{[\boldsymbol{K_A}]_q} \right\rangle$, $Z_3 = \left\langle \overline{[\boldsymbol{K_A}]_p}, \overline{[\boldsymbol{K_A}]_q} \right\rangle$, and $Z_4 = \left\langle \overline{[\boldsymbol{K_B}]_p}, \overline{[\boldsymbol{K_B}]_q} \right\rangle$. Also define $G_p := [\boldsymbol{Q}_1]_{p,p}\sigma_p^2/(N_A N_B d)$ and similarly for $G_q$. Then, we can write the results of Lemma C.10 and Lemma C.11 as

$$
\begin{aligned}
\frac{\mathrm{d}}{\mathrm{d}t}\begin{bmatrix} Z_1 \\ Z_2 \\ Z_3 \\ Z_4 \end{bmatrix} &= \begin{bmatrix} -G_p - G_q & 0 & G_q & G_p \\ 0 & -G_p - G_q & G_p & G_q \\ G_q & G_p & -G_p - G_q & 0 \\ G_p & G_q & 0 & -G_p - G_q \end{bmatrix}\begin{bmatrix} Z_1 \\ Z_2 \\ Z_3 \\ Z_4 \end{bmatrix} \\
&\quad \pm O\left( \frac{1}{\sqrt{d}}\frac{\sigma_{\max}^2}{N_A N_B d}\delta_{\boldsymbol{AB},\perp} \right) \pm O\left( \frac{\sigma_{\max}^2}{N_A N_B d}d\kappa_0\delta_{\boldsymbol{AB},\perp}\left( \sqrt{\delta_-} + \delta_{\boldsymbol{AB},\perp} \right) \right).
\end{aligned}
$$

The eigenvalues of the first matrix are $-2G_p, -2G_q, -2G_p - 2G_q$ and $0$. For the first three eigenvalues, note that

$$
G_p = \frac{[\boldsymbol{Q}_1]_{p,p}\sigma_p^2}{N_A N_B d} \geq \frac{1}{\sqrt{d}}\frac{\sigma_{\max}^2}{N_A N_B d}.
$$

Hence, the signals will dominate the noises, in particular, the first term on the second line, and push $\|\boldsymbol{Z}\|$ toward 0. Now we consider the eigen-pair $(0, (1,1,1,1))$, for which we will use the actual

form of $[\boldsymbol{Q}_1]_{p,q}$ we obtained in Lemma C.5. We have

$$\frac{\mathrm{d}}{\mathrm{d}t}\sum_{i=1}^{4}Z_i = 2\left([\boldsymbol{Q}_1]_{p,q}+[\boldsymbol{Q}_1]_{q,p}\right)\frac{\kappa_q/\kappa_p\sigma_p^2+\kappa_p/\kappa_q\sigma_q^2}{N_A N_B d}$$

$$\pm O\left(\frac{\sigma_{\max}^2}{N_A N_B d}d\kappa_0\delta_{\boldsymbol{AB},\perp}\left(\sqrt{\delta_-}+\delta_{\boldsymbol{AB},\perp}\right)\right).$$

By Lemma C.5, we have

$$[\boldsymbol{Q}_1]_{p,q}+[\boldsymbol{Q}_1]_{q,p}$$
$$= -(1-\tilde{S})\frac{[\boldsymbol{K_{AB}}]_{p,q}+[\boldsymbol{K_{BA}}]_{p,q}}{N_A N_B d}\left(2-T_p^2-T_q^2+\tilde{S}(2-T_p-T_q)^2\right)\pm O\left(d^2\delta_{\boldsymbol{AB},\perp}^2\right).$$

Then, we write

$$[\boldsymbol{K_{AB}}]_{p,q}+[\boldsymbol{K_{BA}}]_{p,q} = \|[\boldsymbol{K_A}]_p\|\,\|[\boldsymbol{K_B}]_q\|\,Z_1 + \|[\boldsymbol{K_B}]_p\|\,\|[\boldsymbol{K_A}]_q\|\,Z_2$$
$$= (Z_1+Z_2)\kappa_p\kappa_q(1\pm\sqrt{\delta_-}).$$

Hence,

$$[\boldsymbol{Q}_1]_{p,q}+[\boldsymbol{Q}_1]_{q,p} = -(1-\tilde{S})\frac{\kappa_p\kappa_q}{N_A N_B d}(Z_1+Z_2)\left(2-T_p^2-T_q^2+\tilde{S}(2-T_p-T_q)^2\right)$$
$$\pm O\left(d^2\delta_{\boldsymbol{AB},\perp}^2+\delta_{\boldsymbol{AB},\perp}\sqrt{\delta_-}\right).$$

To convert $Z_1+Z_2$ to $\sum_{i=1}^{4}Z_i$. Note that we have

$$Z_1+Z_2-Z_3-Z_4 = \left\langle\overline{[\boldsymbol{K_A}]_p}-\overline{[\boldsymbol{K_B}]_p},\overline{[\boldsymbol{K_A}]_q}-\overline{[\boldsymbol{K_B}]_q}\right\rangle \le \delta_-.$$

Therefore,

$$[\boldsymbol{Q}_1]_{p,q}+[\boldsymbol{Q}_1]_{q,p} = -\frac{1-\tilde{S}}{2}\frac{\kappa_p\kappa_q}{N_A N_B d}\left(2-T_p^2-T_q^2+\tilde{S}(2-T_p-T_q)^2\right)\sum_{i=1}^{4}Z_i$$
$$\pm O\left(d^2\delta_{\boldsymbol{AB},\perp}^2+\delta_{\boldsymbol{AB},\perp}\sqrt{\delta_-}+\delta_-\right).$$

Thus,

$$\frac{\mathrm{d}}{\mathrm{d}t}\sum_{i=1}^{4}Z_i = -(1-\tilde{S})\frac{\kappa_q/\kappa_p\sigma_p^2+\kappa_p/\kappa_q\sigma_q^2}{N_A N_B d}\frac{\kappa_p\kappa_q}{N_A N_B d}\left(2-T_p^2-T_q^2+\tilde{S}(2-T_p-T_q)^2\right)\sum_{i=1}^{4}Z_i$$
$$\pm O\left(\frac{\sigma_{\max}^2}{N_A N_B d}\kappa_0\left(d^2\delta_{\boldsymbol{AB},\perp}^2+\delta_{\boldsymbol{AB},\perp}\sqrt{\delta_-}+\delta_-\right)\right).$$

Note that the coefficient of the first term is negative. Combine this with the previous bound for $\|\boldsymbol{Z}\|$, and we complete the proof. $\qquad\square$

*Proof of Lemma C.13.* Recall that

$$\frac{\mathrm{d}}{\mathrm{d}t}\overline{[\boldsymbol{K_A}]_p} = \left(\boldsymbol{I}-\overline{[\boldsymbol{K_A}]_p}\left(\overline{[\boldsymbol{K_A}]_p}\right)^{\top}\right)\left(\frac{[\boldsymbol{K_B}\boldsymbol{Q}_1]_p}{\|[\boldsymbol{K_A}]_p\|}+\frac{[\boldsymbol{K_{B,\xi}}\boldsymbol{Q}_{1,\xi_B}]_p}{\|[\boldsymbol{K_A}]_p\|}\right)\frac{\sigma_p^2}{N_A N_B d},$$

$$\frac{\mathrm{d}}{\mathrm{d}t}\overline{[\boldsymbol{K_{A,\xi}}]_q} = \left(\boldsymbol{I}-\overline{[\boldsymbol{K_{A,\xi}}]_q}\left(\overline{[\boldsymbol{K_{A,\xi}}]_q}\right)^{\top}\right)\left(\frac{[\boldsymbol{K_B}\boldsymbol{Q}_{1,\xi_A}^{\top}]_q}{\|[\boldsymbol{K_{A,\xi}}]_q\|}+\frac{[\boldsymbol{K_{B,\xi}}\boldsymbol{Q}_2]_q}{\|[\boldsymbol{K_{A,\xi}}]_q\|}\right)\frac{\sigma_{\xi}^2}{N_A N_B d}.$$

We now compute $\left\langle\frac{\mathrm{d}}{\mathrm{d}t}\overline{[\boldsymbol{K_A}]_p},\overline{[\boldsymbol{K_{A,\xi}}]_q}\right\rangle$ and $\left\langle\overline{[\boldsymbol{K_A}]_p},\frac{\mathrm{d}}{\mathrm{d}t}\overline{[\boldsymbol{K_{A,\xi}}]_q}\right\rangle$ separately. First, we write

$$\left\langle\frac{\mathrm{d}}{\mathrm{d}t}\overline{[\boldsymbol{K_A}]_p},\overline{[\boldsymbol{K_{A,\xi}}]_q}\right\rangle = \overline{[\boldsymbol{K_{A,\xi}}]_q}^{\top}\left(\boldsymbol{I}-\overline{[\boldsymbol{K_A}]_p}\left(\overline{[\boldsymbol{K_A}]_p}\right)^{\top}\right)\frac{[\boldsymbol{K_B}\boldsymbol{Q}_1]_p}{\|[\boldsymbol{K_A}]_p\|}\frac{\sigma_p^2}{N_A N_B d}$$

$$+\overline{[\boldsymbol{K_{A,\xi}}]_q}^{\top}\left(\boldsymbol{I}-\overline{[\boldsymbol{K_A}]_p}\left(\overline{[\boldsymbol{K_A}]_p}\right)^{\top}\right)\frac{[\boldsymbol{K_{B,\xi}}\boldsymbol{Q}_{1,\xi_B}]_p}{\|[\boldsymbol{K_A}]_p\|}\frac{\sigma_p^2}{N_A N_B d}$$

$$=: \mathrm{T}_1\left(\left\langle\frac{\mathrm{d}}{\mathrm{d}t}\overline{[\boldsymbol{K_A}]_p},\overline{[\boldsymbol{K_{A,\xi}}]_q}\right\rangle\right)+\mathrm{T}_2\left(\left\langle\frac{\mathrm{d}}{\mathrm{d}t}\overline{[\boldsymbol{K_A}]_p},\overline{[\boldsymbol{K_{A,\xi}}]_q}\right\rangle\right).$$

Then, we compute

$$\mathrm{T}_1\left(\left\langle \frac{\mathrm{d}}{\mathrm{d}t}\overline{[\boldsymbol{K_A}]_p}, \overline{[\boldsymbol{K_{A,\xi}}]_q}\right\rangle\right)$$

$$= \left(\left\langle \overline{[\boldsymbol{K_{A,\xi}}]_q}, \overline{[\boldsymbol{K_B}]_p}\right\rangle - \left\langle \overline{[\boldsymbol{K_{A,\xi}}]_q}, \overline{[\boldsymbol{K_A}]_p}\right\rangle \left\langle \overline{[\boldsymbol{K_A}]_p}, \overline{[\boldsymbol{K_B}]_p}\right\rangle\right)\frac{\|[\boldsymbol{K_B}]_p\|\,[\boldsymbol{Q_1}]_{p,p}}{\|[\boldsymbol{K_A}]_p\|}\frac{\sigma_p^2}{N_{\boldsymbol{A}}N_{\boldsymbol{B}}d}$$

$$+ \sum_{k\neq p}\left(\left\langle \overline{[\boldsymbol{K_{A,\xi}}]_q}, \overline{[\boldsymbol{K_B}]_k}\right\rangle - \left\langle \overline{[\boldsymbol{K_{A,\xi}}]_q}, \overline{[\boldsymbol{K_A}]_p}\right\rangle \left\langle \overline{[\boldsymbol{K_A}]_p}, \overline{[\boldsymbol{K_B}]_k}\right\rangle\right)\frac{\|[\boldsymbol{K_B}]_k\|\,[\boldsymbol{Q_1}]_{k,p}}{\|[\boldsymbol{K_A}]_p\|}\frac{\sigma_p^2}{N_{\boldsymbol{A}}N_{\boldsymbol{B}}d}$$

$$= \left(\left\langle \overline{[\boldsymbol{K_{A,\xi}}]_q}, \overline{[\boldsymbol{K_B}]_p}\right\rangle - \left\langle \overline{[\boldsymbol{K_{A,\xi}}]_q}, \overline{[\boldsymbol{K_A}]_p}\right\rangle\right)\frac{[\boldsymbol{Q_1}]_{p,p}\sigma_p^2}{N_{\boldsymbol{A}}N_{\boldsymbol{B}}d} \pm O\left(\frac{\sigma_{\max}^2}{N_{\boldsymbol{A}}N_{\boldsymbol{B}}d}d\kappa_0\delta_{\xi,\perp}\left(\delta_{\xi,\perp}+\sqrt{\delta_-}\right)\right),$$

and, by Lemma C.6, we have

$$\mathrm{T}_2\left(\left\langle \frac{\mathrm{d}}{\mathrm{d}t}\overline{[\boldsymbol{K_A}]_p}, \overline{[\boldsymbol{K_{A,\xi}}]_q}\right\rangle\right) = \sum_{k=1}^{d-r}\overline{[\boldsymbol{K_{A,\xi}}]_q}^{\top}\left(\boldsymbol{I}-\overline{[\boldsymbol{K_A}]_p}\left(\overline{[\boldsymbol{K_A}]_p}\right)^{\top}\right)\frac{[\boldsymbol{K_{B,\xi}}]_k[\boldsymbol{Q_{1,\xi_B}}]_{k,p}}{\|[\boldsymbol{K_A}]_p\|}\frac{\sigma_p^2}{N_{\boldsymbol{A}}N_{\boldsymbol{B}}d}$$

$$= \pm O\left(\frac{\sigma_{\max}^2}{N_{\boldsymbol{A}}N_{\boldsymbol{B}}d}d\delta_{N/S}^2\delta_{\xi,\perp}^2\right).$$

Combine these together and we get

$$\left\langle \frac{\mathrm{d}}{\mathrm{d}t}\overline{[\boldsymbol{K_A}]_p}, \overline{[\boldsymbol{K_{A,\xi}}]_q}\right\rangle = \left(\left\langle \overline{[\boldsymbol{K_{A,\xi}}]_q}, \overline{[\boldsymbol{K_B}]_p}\right\rangle - \left\langle \overline{[\boldsymbol{K_{A,\xi}}]_q}, \overline{[\boldsymbol{K_A}]_p}\right\rangle\right)\frac{[\boldsymbol{Q_1}]_{p,p}\sigma_p^2}{N_{\boldsymbol{A}}N_{\boldsymbol{B}}d}$$

$$\pm O\left(\frac{\sigma_{\max}^2}{N_{\boldsymbol{A}}N_{\boldsymbol{B}}d}d\kappa_0\delta_{\xi,\perp}\left(\delta_{\xi,\perp}+\sqrt{\delta_-}\right)\right).$$

Then, we compute $\left\langle \overline{[\boldsymbol{K_A}]_p}, \frac{\mathrm{d}}{\mathrm{d}t}\overline{[\boldsymbol{K_{A,\xi}}]_q}\right\rangle$. We write

$$\left\langle \overline{[\boldsymbol{K_A}]_p}, \frac{\mathrm{d}}{\mathrm{d}t}\overline{[\boldsymbol{K_{A,\xi}}]_q}\right\rangle = \sum_{k=1}^{r}\overline{[\boldsymbol{K_A}]_p}^{\top}\left(\boldsymbol{I}-\overline{[\boldsymbol{K_{A,\xi}}]_q}\left(\overline{[\boldsymbol{K_{A,\xi}}]_q}\right)^{\top}\right)\frac{[\boldsymbol{K_B}]_k[\boldsymbol{Q_{1,\xi_A}}]_{q,k}}{\|[\boldsymbol{K_{A,\xi}}]_q\|}\frac{\sigma_\xi^2}{N_{\boldsymbol{A}}N_{\boldsymbol{B}}d}$$

$$+ \sum_{k=1}^{d-r}\overline{[\boldsymbol{K_A}]_p}^{\top}\left(\boldsymbol{I}-\overline{[\boldsymbol{K_{A,\xi}}]_q}\left(\overline{[\boldsymbol{K_{A,\xi}}]_q}\right)^{\top}\right)\frac{[\boldsymbol{K_{B,\xi}}]_k[\boldsymbol{Q_2}]_{k,q}}{\|[\boldsymbol{K_{A,\xi}}]_q\|}\frac{\sigma_\xi^2}{N_{\boldsymbol{A}}N_{\boldsymbol{B}}d}$$

$$=: \mathrm{T}_1\left(\left\langle \overline{[\boldsymbol{K_A}]_p}, \frac{\mathrm{d}}{\mathrm{d}t}\overline{[\boldsymbol{K_{A,\xi}}]_q}\right\rangle\right) + \mathrm{T}_2\left(\left\langle \overline{[\boldsymbol{K_A}]_p}, \frac{\mathrm{d}}{\mathrm{d}t}\overline{[\boldsymbol{K_{A,\xi}}]_q}\right\rangle\right).$$

For $\mathrm{T}_1$, when $k = p$, by Lemma C.6, we have

$$\overline{[\boldsymbol{K_A}]_p}^{\top}\left(\boldsymbol{I}-\overline{[\boldsymbol{K_{A,\xi}}]_q}\left(\overline{[\boldsymbol{K_{A,\xi}}]_q}\right)^{\top}\right)\frac{[\boldsymbol{K_B}]_k[\boldsymbol{Q_{1,\xi_A}}]_{q,k}}{\|[\boldsymbol{K_{A,\xi}}]_q\|}$$

$$= \left(\left\langle \overline{[\boldsymbol{K_A}]_p}, \overline{[\boldsymbol{K_B}]_p}\right\rangle - \left\langle \overline{[\boldsymbol{K_A}]_p}, \overline{[\boldsymbol{K_{A,\xi}}]_q}\right\rangle \left\langle \overline{[\boldsymbol{K_{A,\xi}}]_q}, \overline{[\boldsymbol{K_B}]_p}\right\rangle\right)\frac{\|[\boldsymbol{K_B}]_p\|}{\|[\boldsymbol{K_{A,\xi}}]_q\|}[\boldsymbol{Q_{1,\xi_A}}]_{q,p}$$

$$= -(1-\tilde{S})\left(1+\tilde{S}+(1-\tilde{S})T_p^2\right)\left(1\pm O\left(\delta_-+\delta_{\xi,\perp}^2\right)\right)\frac{\|[\boldsymbol{K_B}]_p\|^2}{N_{\boldsymbol{A}}N_{\boldsymbol{B}}d}\left\langle \overline{[\boldsymbol{K_B}]_p}, \overline{[\boldsymbol{K_{A,\xi}}]_q}\right\rangle$$

$$\pm O\left(d^3\left(\delta_{\boldsymbol{AB},\perp}+\delta_{N/S}\right)\delta_{\xi,\perp}\delta_{N/S}\right)$$

$$= -\Theta\left(\frac{\|[\boldsymbol{K_B}]_p\|^2}{N_{\boldsymbol{A}}N_{\boldsymbol{B}}d}\right)\left\langle \overline{[\boldsymbol{K_B}]_p}, \overline{[\boldsymbol{K_{A,\xi}}]_q}\right\rangle \pm O\left(d^3\left(\delta_{\boldsymbol{AB},\perp}+\delta_{N/S}\right)\delta_{\xi,\perp}\delta_{N/S}\right).$$

When $k \neq p$, we have

$$\overline{[\boldsymbol{K_A}]_p}^{\top}\left(\boldsymbol{I}-\overline{[\boldsymbol{K_{A,\xi}}]_q}\left(\overline{[\boldsymbol{K_{A,\xi}}]_q}\right)^{\top}\right)\frac{[\boldsymbol{K_B}]_k[\boldsymbol{Q_{1,\xi_A}}]_{q,k}}{\|[\boldsymbol{K_{A,\xi}}]_q\|} = O\left(\kappa_0\delta_{\boldsymbol{AB},\perp}\delta_{\xi,\perp}\right).$$

Hence,

$$\mathrm{T}_1\left(\left\langle\overline{[K_A]_p}, \frac{\mathrm{d}}{\mathrm{d}t}\overline{[K_{A,\xi}]_q}\right\rangle\right) = -\Theta\left(\frac{\|[K_B]_p\|^2}{N_A N_B d}\right)\frac{\sigma_\xi^2}{N_A N_B d}\left\langle\overline{[K_B]_p}, \overline{[K_{A,\xi}]_q}\right\rangle$$

$$\pm O\left(\frac{\sigma_{\max}^2}{N_A N_B d}d^3\kappa_0\left(\delta_{AB,\perp} + \delta_{N/S}\right)\delta_{\xi,\perp}\delta_{N/S}\right).$$

Now we consider $\mathrm{T}_2$. By Lemma C.7, we have

$$\mathrm{T}_2\left(\left\langle\overline{[K_A]_p}, \frac{\mathrm{d}}{\mathrm{d}t}\overline{[K_{A,\xi}]_q}\right\rangle\right) = \pm O\left(\frac{\sigma_\xi^2}{N_A N_B d}d^4\delta_{\xi,\perp}^2\delta_{N/S}\left(\delta_{AB,\perp} + \delta_{N/S}\right)\right).$$

Therefore,

$$\left\langle\overline{[K_A]_p}, \frac{\mathrm{d}}{\mathrm{d}t}\overline{[K_{A,\xi}]_q}\right\rangle = -\Theta\left(\frac{\|[K_B]_p\|^2}{N_A N_B d}\right)\frac{\sigma_\xi^2}{N_A N_B d}\left\langle\overline{[K_B]_p}, \overline{[K_{A,\xi}]_q}\right\rangle$$

$$\pm O\left(\frac{\sigma_{\max}^2}{N_A N_B d}d^3\kappa_0\left(\delta_{AB,\perp} + \delta_{N/S}\right)\delta_{\xi,\perp}\delta_{N/S}\right).$$

Thus,

$$\frac{\mathrm{d}}{\mathrm{d}t}\left\langle\overline{[K_A]_p}, \overline{[K_{A,\xi}]_q}\right\rangle = \left(\left\langle\overline{[K_{A,\xi}]_q}, \overline{[K_B]_p}\right\rangle - \left\langle\overline{[K_{A,\xi}]_q}, \overline{[K_A]_p}\right\rangle\right)\frac{[Q_1]_{p,p}\sigma_p^2}{N_A N_B d}$$

$$- \Theta\left(\frac{\|[K_B]_p\|^2}{N_A N_B d}\right)\frac{\sigma_\xi^2}{N_A N_B d}\left\langle\overline{[K_B]_p}, \overline{[K_{A,\xi}]_q}\right\rangle$$

$$\pm O\left(\frac{\sigma_{\max}^2}{N_A N_B d}d\kappa_0\delta_{\xi,\perp}\left(\delta_{\xi,\perp} + \sqrt{\delta_-}\right)\right).$$

Similarly, one can show that

$$\frac{\mathrm{d}}{\mathrm{d}t}\left\langle\overline{[K_B]_p}, \overline{[K_{A,\xi}]_q}\right\rangle = \left(\left\langle\overline{[K_{A,\xi}]_q}, \overline{[K_A]_p}\right\rangle - \left\langle\overline{[K_{A,\xi}]_q}, \overline{[K_B]_p}\right\rangle\right)\frac{[Q_1]_{p,p}\sigma_p^2}{N_A N_B d}$$

$$- \Theta\left(\frac{\|[K_A]_p\|^2}{N_A N_B d}\right)\frac{\sigma_\xi^2}{N_A N_B d}\left\langle\overline{[K_A]_p}, \overline{[K_{A,\xi}]_q}\right\rangle$$

$$\pm O\left(\frac{\sigma_{\max}^2}{N_A N_B d}d\kappa_0\delta_{\xi,\perp}\left(\delta_{\xi,\perp} + \sqrt{\delta_-}\right)\right).$$

For notational simplicity, define

$$X = \left\langle\overline{[K_A]_p}, \overline{[K_{A,\xi}]_q}\right\rangle, \quad Y = \left\langle\overline{[K_B]_p}, \overline{[K_{A,\xi}]_q}\right\rangle, \quad C_1 = \frac{[Q_1]_{p,p}\sigma_p^2}{N_A N_B d}, \quad C_2 = \frac{\kappa_p^2}{N_A N_B d}\frac{\sigma_\xi^2}{N_A N_B d}.$$

Then, we can write

$$\frac{\mathrm{d}}{\mathrm{d}t}\begin{bmatrix}X\\Y\end{bmatrix} = \begin{bmatrix}-C_1 & C_1 - \Theta(C_2)\\C_1 - \Theta(C_2) & -C_1\end{bmatrix}\begin{bmatrix}X\\Y\end{bmatrix} \pm O\left(\frac{\sigma_{\max}^2}{N_A N_B d}d\kappa_0\delta_{\xi,\perp}\left(\delta_{\xi,\perp} + \sqrt{\delta_-}\right)\right).$$

The first matrix is negative semi-definite, whence

$$\frac{\mathrm{d}}{\mathrm{d}t}\left(X^2 + Y^2\right) \le O\left(\frac{\sigma_{\max}^2}{N_A N_B d}d\kappa_0\delta_{\xi,\perp}^2\left(\delta_{\xi,\perp} + \sqrt{\delta_-}\right)\right).$$

Since the roles of $K_{A,\xi}$ and $K_{B,\xi}$ are interchangeable, the same bound also holds for $\left\langle\overline{[K_A]_p}, \overline{[K_{B,\xi}]_q}\right\rangle$ and $\left\langle\overline{[K_A]_p}, \overline{[K_{B,\xi}]_q}\right\rangle$. $\qquad\square$

*Proof of Lemma C.14.* We write

$$
\left\langle \overline{[\boldsymbol{K}_{A,\boldsymbol{\xi}}]_p}, \frac{\mathrm{d}}{\mathrm{d}t}\overline{[\boldsymbol{K}_{B,\boldsymbol{\xi}}]_q} \right\rangle = \overline{[\boldsymbol{K}_{A,\boldsymbol{\xi}}]_p}^\top \left( \boldsymbol{I} - \overline{[\boldsymbol{K}_{B,\boldsymbol{\xi}}]_q} \left( \overline{[\boldsymbol{K}_{B,\boldsymbol{\xi}}]_q} \right)^\top \right) \left( \frac{[\boldsymbol{K}_A\boldsymbol{Q}_{1,\boldsymbol{\xi}_B}^\top]_q}{\|[\boldsymbol{K}_{B,\boldsymbol{\xi}}]_q\|} + \frac{[\boldsymbol{K}_{A,\boldsymbol{\xi}}\boldsymbol{Q}_2^\top]_q}{\|[\boldsymbol{K}_{B,\boldsymbol{\xi}}]_q\|} \right) \frac{\sigma_\xi^2}{N_A N_B d}
$$

$$
= \overline{[\boldsymbol{K}_{A,\boldsymbol{\xi}}]_p}^\top \left( \boldsymbol{I} - \overline{[\boldsymbol{K}_{B,\boldsymbol{\xi}}]_q} \left( \overline{[\boldsymbol{K}_{B,\boldsymbol{\xi}}]_q} \right)^\top \right) \frac{[\boldsymbol{K}_A\boldsymbol{Q}_{1,\boldsymbol{\xi}_B}^\top]_q}{\|[\boldsymbol{K}_{B,\boldsymbol{\xi}}]_q\|} \frac{\sigma_\xi^2}{N_A N_B d}
$$

$$
+ \overline{[\boldsymbol{K}_{A,\boldsymbol{\xi}}]_p}^\top \left( \boldsymbol{I} - \overline{[\boldsymbol{K}_{B,\boldsymbol{\xi}}]_q} \left( \overline{[\boldsymbol{K}_{B,\boldsymbol{\xi}}]_q} \right)^\top \right) \frac{[\boldsymbol{K}_{A,\boldsymbol{\xi}}\boldsymbol{Q}_2^\top]_q}{\|[\boldsymbol{K}_{B,\boldsymbol{\xi}}]_q\|} \frac{\sigma_\xi^2}{N_A N_B d}
$$

$$
=: \mathrm{T}_1 \left( \left\langle \overline{[\boldsymbol{K}_{A,\boldsymbol{\xi}}]_p}, \frac{\mathrm{d}}{\mathrm{d}t}\overline{[\boldsymbol{K}_{B,\boldsymbol{\xi}}]_q} \right\rangle \right) + \mathrm{T}_2 \left( \left\langle \overline{[\boldsymbol{K}_{A,\boldsymbol{\xi}}]_p}, \frac{\mathrm{d}}{\mathrm{d}t}\overline{[\boldsymbol{K}_{B,\boldsymbol{\xi}}]_q} \right\rangle \right).
$$

Then, we compute

$$
\mathrm{T}_1 \left( \left\langle \overline{[\boldsymbol{K}_{A,\boldsymbol{\xi}}]_p}, \frac{\mathrm{d}}{\mathrm{d}t}\overline{[\boldsymbol{K}_{B,\boldsymbol{\xi}}]_q} \right\rangle \right)
$$

$$
= \sum_{k=1}^{r} \left( \left\langle \overline{[\boldsymbol{K}_{A,\boldsymbol{\xi}}]_p}, \overline{[\boldsymbol{K}_A]_k} \right\rangle - \left\langle \overline{[\boldsymbol{K}_{A,\boldsymbol{\xi}}]_p}, \overline{[\boldsymbol{K}_{B,\boldsymbol{\xi}}]_q} \right\rangle \left\langle \overline{[\boldsymbol{K}_{B,\boldsymbol{\xi}}]_q}, \overline{[\boldsymbol{K}_A]_k} \right\rangle \right) \frac{\|[\boldsymbol{K}_A]_k\| [\boldsymbol{Q}_{1,\boldsymbol{\xi}_B}]_{q,k}}{\|[\boldsymbol{K}_{B,\boldsymbol{\xi}}]_q\|} \frac{\sigma_\xi^2}{N_A N_B d}.
$$

Note that, the first part of each summand is bounded by $O(\delta_{\xi,\perp})$. For the second part, by Lemma C.6 we also have

$$
\frac{\|[\boldsymbol{K}_A]_k\| [\boldsymbol{Q}_{1,\boldsymbol{\xi}_B}]_{q,k}}{\|[\boldsymbol{K}_{B,\boldsymbol{\xi}}]_q\|} = \pm \frac{\|[\boldsymbol{K}_A]_k\|}{\|[\boldsymbol{K}_{B,\boldsymbol{\xi}}]_q\|} \left( O(1) \frac{\langle [\boldsymbol{K}_B]_p, [\boldsymbol{K}_{A,\boldsymbol{\xi}}]_q \rangle}{N_A N_B d} \pm O \left( d^3 \left( \delta_{AB,\perp} + \delta_{N/S} \right) \delta_{\xi,\perp} \delta_{N/S} \right) \right)
$$

$$
= \pm O(\delta_{\xi,\perp}).
$$

Hence,

$$
\mathrm{T}_1 \left( \left\langle \overline{[\boldsymbol{K}_{A,\boldsymbol{\xi}}]_p}, \frac{\mathrm{d}}{\mathrm{d}t}\overline{[\boldsymbol{K}_{B,\boldsymbol{\xi}}]_q} \right\rangle \right) = O \left( \frac{\sigma_\xi^2}{N_A N_B d} d\delta_{\xi,\perp}^2 \right).
$$

Now we consider $\mathrm{T}_2$. By Lemma C.7, We have

$$
\mathrm{T}_2 \left( \left\langle \overline{[\boldsymbol{K}_{A,\boldsymbol{\xi}}]_p}, \frac{\mathrm{d}}{\mathrm{d}t}\overline{[\boldsymbol{K}_{B,\boldsymbol{\xi}}]_q} \right\rangle \right)
$$

$$
= \sum_{k=1}^{d-r} \overline{[\boldsymbol{K}_{A,\boldsymbol{\xi}}]_p}^\top \left( \boldsymbol{I} - \overline{[\boldsymbol{K}_{B,\boldsymbol{\xi}}]_q} \left( \overline{[\boldsymbol{K}_{B,\boldsymbol{\xi}}]_q} \right)^\top \right) \overline{[\boldsymbol{K}_{A,\boldsymbol{\xi}}]_k} \frac{\|[\boldsymbol{K}_{A,\boldsymbol{\xi}}]_k\| [\boldsymbol{Q}_2]_{q,k}}{\|[\boldsymbol{K}_{B,\boldsymbol{\xi}}]_q\|} \frac{\sigma_\xi^2}{N_A N_B d}
$$

$$
= \pm \sum_{k=1}^{d-r} O(1)[\boldsymbol{Q}_2]_{q,k} \frac{\sigma_\xi^2}{N_A N_B d}
$$

$$
= \pm O \left( \frac{\sigma_{\max}^2}{N_A N_B d} d^4 \left( \delta_{AB,\perp} + \delta_{N/S} \right) \delta_{\xi,\perp} \delta_{N/S} \right).
$$

Combine this together and we obtain

$$
\left\langle \overline{[\boldsymbol{K}_{A,\boldsymbol{\xi}}]_p}, \frac{\mathrm{d}}{\mathrm{d}t}\overline{[\boldsymbol{K}_{B,\boldsymbol{\xi}}]_q} \right\rangle = \pm O \left( \frac{\sigma_\xi^2}{N_A N_B d} d\delta_{\xi,\perp}^2 \right).
$$

Similarly, one can derive the same bound for $\left\langle \frac{\mathrm{d}}{\mathrm{d}t}\overline{[\boldsymbol{K}_{A,\boldsymbol{\xi}}]_p}, \overline{[\boldsymbol{K}_{B,\boldsymbol{\xi}}]_q} \right\rangle$ and complete the proof. □

## C.3 MAINTAINING $\mathbf{K_A} \approx \mathbf{K_B}$

In this subsection, we show that $\left\langle \overline{[\boldsymbol{K}_A]_p}, \overline{[\boldsymbol{K}_B]_p} \right\rangle \approx 1$, $\|[\boldsymbol{K}_A]_p\| \approx \|[\boldsymbol{K}_B]_p\|$, and also $\|[\boldsymbol{K}_{A,\boldsymbol{\xi}}]_q\| \approx \|[\boldsymbol{K}_{B,\boldsymbol{\xi}}]_q\|$ throughout Stage 2. The proofs are similar to the corresponding ones in Stage 1.

**Lemma C.15.** *In Stage 2, we have*

$$\frac{\mathrm{d}}{\mathrm{d}t} \left\langle \overline{[K_A]_p}, \overline{[K_B]_p} \right\rangle = \Omega \left( \frac{\sigma_p^2 [Q_1]_{p,p}}{N_A N_B d} \right) \left( 1 - \left\langle \overline{[K_A]_p}, \overline{[K_B]_p} \right\rangle \right) \pm O \left( \frac{\sigma_{\max}^2}{N_A N_B d} d\kappa_0 \delta_{AB,\perp}^2 \right).$$

**Lemma C.16.** *Define* $\rho_{A/B,p} := \|[K_A]_p\|^2 / \|[K_B]_p\|^2$ *and* $\rho_{B/A,p} := \|[K_B]_p\|^2 / \|[K_A]_p\|^2$.
*In Stage 2, we have*

$$\frac{\mathrm{d}}{\mathrm{d}t} \left( \rho_{A/B,p} + \rho_{B/A,p} \right) = \frac{2[Q_1]_{p,p} \sigma_p^2}{N_A N_B d} \left\langle \overline{[K_A]_p}, \overline{[K_B]_p} \right\rangle \left( 2 - \rho_{A/B,p} - \rho_{B/A,p} \right)$$

$$\pm O \left( \frac{\sigma_{\max}^2}{N_A N_B d} \left( d\kappa_0 \delta_{AB,\perp}^2 + \delta_-^2 \right) \right).$$

**Lemma C.17.** *For any* $p, q \in [d - r]$, *define* $\rho_{\xi,A/A,p/q} = \|[K_{A,\xi}]_p\|^2 / \|[K_{A,\xi}]_q\|^2$ *and* $\rho_{\xi,A/B,p/q} = \|[K_{A,\xi}]_p\|^2 / \|[K_{B,\xi}]_q\|^2$. *In Stage 2, we have*

$$\frac{\mathrm{d}}{\mathrm{d}t} \rho_{\xi,A/A,p/q} = \pm O \left( \frac{\sigma_\xi^2}{N_A N_B d} d\delta_{\xi,\perp}^2 \right),$$

$$\frac{\mathrm{d}}{\mathrm{d}t} \left( \rho_{\xi,A/B,p/q} + \rho_{\xi,B/A,q/p} \right) = \pm O \left( \frac{\sigma_\xi^2}{N_A N_B d} \left( d\delta_{\xi,\perp}^2 + \delta_-^2 \right) \right).$$

OMITTED PROOF OF THIS SUBSECTION

*Proof of Lemma C.15.* First, we write

$$\left\langle \frac{\mathrm{d}}{\mathrm{d}t} \overline{[K_A]_p}, \overline{[K_B]_p} \right\rangle = \overline{[K_B]_p}^\top \left( I - \overline{[K_A]_p} \left( \overline{[K_A]_p} \right)^\top \right) \frac{[K_B Q_1]_p}{\|[K_A]_p\|} \frac{\sigma_p^2}{N_A N_B d}$$

$$+ \overline{[K_B]_p}^\top \left( I - \overline{[K_A]_p} \left( \overline{[K_A]_p} \right)^\top \right) \frac{[K_{B,\xi} Q_{1,\xi_B}]_p}{\|[K_A]_p\|} \frac{\sigma_p^2}{N_A N_B d}$$

$$=: \mathrm{T}_1 \left( \left\langle \frac{\mathrm{d}}{\mathrm{d}t} \overline{[K_A]_p}, \overline{[K_B]_p} \right\rangle \right) + \mathrm{T}_2 \left( \left\langle \frac{\mathrm{d}}{\mathrm{d}t} \overline{[K_A]_p}, \overline{[K_B]_p} \right\rangle \right).$$

For $\mathrm{T}_1$, we compute

$$\mathrm{T}_1 \left( \left\langle \frac{\mathrm{d}}{\mathrm{d}t} \overline{[K_A]_p}, \overline{[K_B]_p} \right\rangle \right) = \overline{[K_B]_p}^\top \left( I - \overline{[K_A]_p} \left( \overline{[K_A]_p} \right)^\top \right) \overline{[K_B]_p} \frac{\|[K_B]_p\| [Q_1]_{p,p}}{\|[K_A]_p\|} \frac{\sigma_p^2}{N_A N_B d}$$

$$+ \sum_{k \neq p} \overline{[K_B]_p}^\top \left( I - \overline{[K_A]_p} \left( \overline{[K_A]_p} \right)^\top \right) \overline{[K_B]_k} \frac{\|[K_B]_k\| [Q_1]_{k,p}}{\|[K_A]_p\|} \frac{\sigma_p^2}{N_A N_B d}$$

$$= \left( 1 - \left\langle \overline{[K_A]_p}, \overline{[K_B]_p} \right\rangle^2 \right) \frac{\|[K_B]_p\| [Q_1]_{p,p}}{\|[K_A]_p\|} \frac{\sigma_p^2}{N_A N_B d}$$

$$\pm O \left( \frac{\sigma_{\max}^2}{N_A N_B d} d\kappa_0 \delta_{AB,\perp}^2 \right).$$

For $\mathrm{T}_2$, we compute

$$\mathrm{T}_2 \left( \left\langle \frac{\mathrm{d}}{\mathrm{d}t} \overline{[K_A]_p}, \overline{[K_B]_p} \right\rangle \right) = \sum_{k=1}^{d-r} \overline{[K_B]_p}^\top \left( I - \overline{[K_A]_p} \left( \overline{[K_A]_p} \right)^\top \right) \frac{[K_{B,\xi}]_k [Q_{1,\xi_B}]_{k,p}}{\|[K_A]_p\|} \frac{\sigma_p^2}{N_A N_B d}$$

$$= \pm O \left( \frac{\sigma_{\max}^2}{N_A N_B d} d\delta_{N/S}^2 \delta_{\xi,\perp}^2 \right).$$

Therefore,

$$\left\langle \frac{\mathrm{d}}{\mathrm{d}t} \overline{[K_A]_p}, \overline{[K_B]_p} \right\rangle = \left( 1 - \left\langle \overline{[K_A]_p}, \overline{[K_B]_p} \right\rangle^2 \right) \frac{\|[K_B]_p\| [Q_1]_{p,p}}{\|[K_A]_p\|} \frac{\sigma_p^2}{N_A N_B d} \pm O \left( \frac{\sigma_{\max}^2}{N_A N_B d} d\kappa_0 \delta_{AB,\perp}^2 \right).$$

Then, by symmetry, we have

$$
\frac{\mathrm{d}}{\mathrm{d}t}\left\langle \overline{[\boldsymbol{K_A}]_p}, \overline{[\boldsymbol{K_B}]_p} \right\rangle = \left(1 - \left\langle \overline{[\boldsymbol{K_A}]_p}, \overline{[\boldsymbol{K_B}]_p} \right\rangle^2 \right)[\boldsymbol{Q}_1]_{p,p}\left(\frac{\|[\boldsymbol{K_B}]_p\|}{\|[\boldsymbol{K_A}]_p\|} + \frac{\|[\boldsymbol{K_A}]_p\|}{\|[\boldsymbol{K_B}]_p\|}\right)\frac{\sigma_p^2}{N_A N_B d}
$$
$$
\pm O\left(\frac{\sigma_{\max}^2}{N_A N_B d}d\kappa_0\delta_{\boldsymbol{AB},\perp}^2\right)
$$
$$
= \Omega\left(\frac{\sigma_p^2[\boldsymbol{Q}_1]_{p,p}}{N_A N_B d}\right)\left(1 - \left\langle \overline{[\boldsymbol{K_A}]_p}, \overline{[\boldsymbol{K_B}]_p} \right\rangle\right) \pm O\left(\frac{\sigma_{\max}^2}{N_A N_B d}d\kappa_0\delta_{\boldsymbol{AB},\perp}^2\right).
$$

$\square$

*Proof of Lemma C.16.* Similar to Stage 1, we define $\rho_{\boldsymbol{A}/\boldsymbol{B},p} = \|[\boldsymbol{K_A}]_p\|^2 / \|[\boldsymbol{K_B}]_p\|^2$. By Lemma A.5, we have

$$
\frac{\mathrm{d}}{\mathrm{d}t}\rho_{\boldsymbol{A}/\boldsymbol{B},p} = \frac{\frac{\mathrm{d}}{\mathrm{d}t}\|[\boldsymbol{K_A}]_p\|^2}{\|[\boldsymbol{K_B}]_p\|^2} - \rho_{\boldsymbol{A}/\boldsymbol{B},p}\frac{\frac{\mathrm{d}}{\mathrm{d}t}\|[\boldsymbol{K_B}]_p\|^2}{\|[\boldsymbol{K_B}]_p\|^2}
$$
$$
= 2\frac{\langle[\boldsymbol{K_A}]_p, [\boldsymbol{K_B Q}_1]_p\rangle}{N_A N_B d\|[\boldsymbol{K_B}]_p\|^2}\sigma_p^2 + 2\frac{\langle[\boldsymbol{K_A}]_p, [\boldsymbol{K_{B,\xi}Q}_{1,\xi_B}]_p\rangle}{N_A N_B d\|[\boldsymbol{K_B}]_p\|^2}\sigma_p^2 + 2\frac{\rho_{\boldsymbol{A}/\boldsymbol{B},p}}{N_A^2 d}Q_0\sigma_p^2
$$
$$
- \rho_{\boldsymbol{A}/\boldsymbol{B},p}\left(2\frac{\langle[\boldsymbol{K_B}]_p, [\boldsymbol{K_A Q}_1^\top]_p\rangle}{N_A N_B d\|[\boldsymbol{K_B}]_p\|^2}\sigma_p^2 + 2\frac{\langle[\boldsymbol{K_B}]_p, [\boldsymbol{K_{A,\xi}Q}_{1,\xi_A}]_p\rangle}{N_A N_B d\|[\boldsymbol{K_B}]_p\|^2}\sigma_p^2 + 2\frac{1}{N_B^2 d}Q_0\sigma_p^2\right)
$$
$$
= 2\frac{\langle[\boldsymbol{K_A}]_p, [\boldsymbol{K_B Q}_1]_p\rangle}{N_A N_B d\|[\boldsymbol{K_B}]_p\|^2}\sigma_p^2 + 2\frac{\langle[\boldsymbol{K_A}]_p, [\boldsymbol{K_{B,\xi}Q}_{1,\xi_B}]_p\rangle}{N_A N_B d\|[\boldsymbol{K_B}]_p\|^2}\sigma_p^2
$$
$$
- \rho_{\boldsymbol{A}/\boldsymbol{B},p}\left(2\frac{\langle[\boldsymbol{K_B}]_p, [\boldsymbol{K_A Q}_1^\top]_p\rangle}{N_A N_B d\|[\boldsymbol{K_B}]_p\|^2}\sigma_p^2 + 2\frac{\langle[\boldsymbol{K_B}]_p, [\boldsymbol{K_{A,\xi}Q}_{1,\xi_A}]_p\rangle}{N_A N_B d\|[\boldsymbol{K_B}]_p\|^2}\sigma_p^2\right)
$$
$$
+ 2Q_0\sigma_p^2\left(\frac{1}{N_A^2 d} - \frac{1}{N_B^2 d}\right)\rho_{\boldsymbol{A}/\boldsymbol{B},p}.
$$

Then, by Lemma C.5 and Lemma C.6, we have

$$
\langle[\boldsymbol{K_A}]_p, [\boldsymbol{K_B Q}_1]_p\rangle = \langle[\boldsymbol{K_A}]_p, [\boldsymbol{K_B}]_p\rangle[\boldsymbol{Q}_1]_{p,p} \pm O\left(d\kappa_p^2\kappa_0\delta_{\boldsymbol{AB},\perp}^2\right),
$$
$$
\langle[\boldsymbol{K_B}]_p, [\boldsymbol{K_A Q}_1^\top]_p\rangle = \langle[\boldsymbol{K_A}]_p, [\boldsymbol{K_B}]_p\rangle[\boldsymbol{Q}_1]_{p,p} \pm O\left(d\kappa_p^2\kappa_0\delta_{\boldsymbol{AB},\perp}^2\right),
$$
$$
\langle[\boldsymbol{K_A}]_p, [\boldsymbol{K_{B,\xi}Q}_{1,\xi_B}]_p\rangle = \pm O\left(d\kappa_p^2\delta_{N/S}^2\delta_{\xi,\perp}\right),
$$
$$
\langle[\boldsymbol{K_B}]_p, [\boldsymbol{K_{A,\xi}Q}_{1,\xi_A}]_p\rangle = \pm O\left(d\kappa_p^2\delta_{N/S}^2\delta_{\xi,\perp}\right).
$$

Thus,

$$
\frac{\mathrm{d}}{\mathrm{d}t}\rho_{\boldsymbol{A}/\boldsymbol{B},p} = 2\frac{[\boldsymbol{Q}_1]_{p,p}\sigma_p^2}{N_A N_B d}\frac{\langle[\boldsymbol{K_A}]_p, [\boldsymbol{K_B}]_p\rangle}{\|[\boldsymbol{K_B}]_p\|^2}\left(1 - \rho_{\boldsymbol{A}/\boldsymbol{B},p}\right)
$$
$$
+ 2Q_0\sigma_p^2\left(\frac{1}{N_A^2 d} - \frac{1}{N_B^2 d}\right)\rho_{\boldsymbol{A}/\boldsymbol{B},p} \pm O\left(\frac{\sigma_{\max}^2}{N_A N_B d}d\kappa_0\delta_{\boldsymbol{AB},\perp}^2\right).
$$

Interchange the roles of $\boldsymbol{A}, \boldsymbol{B}$ and we get

$$
\frac{\mathrm{d}}{\mathrm{d}t}\rho_{\boldsymbol{B}/\boldsymbol{A},p} = 2\frac{[\boldsymbol{Q}_1]_{p,p}\sigma_p^2}{N_A N_B d}\frac{\langle[\boldsymbol{K_A}]_p, [\boldsymbol{K_B}]_p\rangle}{\|[\boldsymbol{K_A}]_p\|^2}\left(1 - \rho_{\boldsymbol{B}/\boldsymbol{A},p}\right)
$$
$$
+ 2Q_0\sigma_p^2\left(\frac{1}{N_B^2 d} - \frac{1}{N_A^2 d}\right)\rho_{\boldsymbol{B}/\boldsymbol{A},p} \pm O\left(\frac{\sigma_{\max}^2}{N_A N_B d}d\kappa_0\delta_{\boldsymbol{AB},\perp}^2\right).
$$

Hence,

$$
\begin{aligned}
\frac{\mathrm{d}}{\mathrm{d}t}\left(\rho_{A/B,p}+\rho_{B/A,p}\right) &= 2\frac{[\boldsymbol{Q}_1]_{p,p}\sigma_p^2}{N_A N_B d}\left\langle[\boldsymbol{K}_A]_p,[\boldsymbol{K}_B]_p\right\rangle\left(\frac{1-\rho_{A/B,p}}{\|[\boldsymbol{K}_B]_p\|^2}+\frac{1-\rho_{B/A,p}}{\|[\boldsymbol{K}_A]_p\|^2}\right) \\
&\quad + 2Q_0\sigma_p^2\left(\frac{1}{N_A^2 d}-\frac{1}{N_B^2 d}\right)\left(\rho_{A/B,p}-\rho_{B/A,p}\right)\pm O\left(\frac{\sigma_{\max}^2}{N_A N_B d}d\kappa_0\delta_{AB,\perp}^2\right) \\
&= \frac{2[\boldsymbol{Q}_1]_{p,p}\sigma_p^2}{N_A N_B d}\left\langle\overline{[\boldsymbol{K}_A]}_p,\overline{[\boldsymbol{K}_B]}_p\right\rangle\left(2-\rho_{A/B,p}-\rho_{B/A,p}\right) \\
&\quad \pm O\left(\frac{\sigma_{\max}^2}{N_A N_B d}\left(d\kappa_0\delta_{AB,\perp}^2+\delta_-^2\right)\right).
\end{aligned}
$$

$\square$

*Proof of Lemma C.17.* For notational simplicity, we drop the subscript $\xi$ in the proof. Recall from Corollary C.9 that

$$
\frac{\mathrm{d}}{\mathrm{d}t}\|[\boldsymbol{K}_{A,\xi}]_q\|^2 = \frac{2\|[\boldsymbol{K}_{A,\xi}]_q\|^2}{N_A^2 d}Q_0\sigma_\xi^2\pm O\left(\frac{\sigma_\xi^2\|[\boldsymbol{K}_{A,\xi}]_q\|^2}{N_A N_B d}d\delta_{\xi,\perp}^2\right).
$$

Hence, for any $p,q\in[d-r]$, we have

$$
\begin{aligned}
\frac{\mathrm{d}}{\mathrm{d}t}\rho_{A/A,p/q} &= \frac{\frac{\mathrm{d}}{\mathrm{d}t}\|[\boldsymbol{K}_{A,\xi}]_p\|^2}{\|[\boldsymbol{K}_{A,\xi}]_q\|^2}-\rho_{A/A,p/q}\frac{\frac{\mathrm{d}}{\mathrm{d}t}\|[\boldsymbol{K}_{A,\xi}]_q\|^2}{\|[\boldsymbol{K}_{A,\xi}]_q\|^2} \\
&= \frac{2}{N_A^2 d}Q_0\sigma_\xi^2\rho_{A/A,p/q}\pm O\left(\frac{\sigma_\xi^2}{N_A N_B d}d\delta_{\xi,\perp}^2\right) \\
&\quad -\rho_{A/A,p/q}\left(\frac{2}{N_A^2 d}Q_0\sigma_\xi^2\pm O\left(\frac{\sigma_\xi^2}{N_A N_B d}d\delta_{\xi,\perp}^2\right)\right) \\
&= \pm O\left(\frac{\sigma_\xi^2}{N_A N_B d}d\delta_{\xi,\perp}^2\right).
\end{aligned}
$$

Similarly, we have

$$
\begin{aligned}
\frac{\mathrm{d}}{\mathrm{d}t}\rho_{A/B,p/q} &= \frac{\frac{\mathrm{d}}{\mathrm{d}t}\|[\boldsymbol{K}_{A,\xi}]_p\|^2}{\|[\boldsymbol{K}_{B,\xi}]_q\|^2}-\rho_{A/B,p/q}\frac{\frac{\mathrm{d}}{\mathrm{d}t}\|[\boldsymbol{K}_{B,\xi}]_q\|^2}{\|[\boldsymbol{K}_{B,\xi}]_q\|^2} \\
&= \frac{2}{N_A^2 d}Q_0\sigma_\xi^2\rho_{A/B,p/q}\pm O\left(\frac{\sigma_\xi^2}{N_A N_B d}d\delta_{\xi,\perp}^2\right) \\
&\quad -\rho_{A/B,p/q}\left(\frac{2}{N_B^2 d}Q_0\sigma_\xi^2\pm O\left(\frac{\sigma_\xi^2}{N_A N_B d}d\delta_{\xi,\perp}^2\right)\right) \\
&= \left(\frac{2}{N_A^2 d}-\frac{2}{N_B^2 d}\right)Q_0\sigma_\xi^2\rho_{A/B,p/q}\pm O\left(\frac{\sigma_\xi^2}{N_A N_B d}d\delta_{\xi,\perp}^2\right).
\end{aligned}
$$

By symmetry, we also have

$$
\frac{\mathrm{d}}{\mathrm{d}t}\rho_{B/A,q/p} = \left(\frac{2}{N_B^2 d}-\frac{2}{N_A^2 d}\right)Q_0\sigma_\xi^2\rho_{B/A,q/p}\pm O\left(\frac{\sigma_\xi^2}{N_A N_B d}d\delta_{\xi,\perp}^2\right).
$$

Hence,

$$
\begin{aligned}
\frac{\mathrm{d}}{\mathrm{d}t}\left(\rho_{A/B,p/q}+\rho_{B/A,q/p}\right) &= \left(\frac{2}{N_A^2 d}-\frac{2}{N_B^2 d}\right)Q_0\sigma_\xi^2\left(\rho_{A/B,p/q}-\rho_{B/A,q/p}\right)\pm O\left(\frac{\sigma_\xi^2}{N_A N_B d}d\delta_{\xi,\perp}^2\right) \\
&= \pm O\left(\frac{\sigma_\xi^2}{N_A N_B d}\left(d\delta_{\xi,\perp}^2+\delta_-^2\right)\right).
\end{aligned}
$$

$\square$

## C.4 Controlling the noise-signal ratio

In this subsection, we show that the noise-signal ratio remains small throughout Stage 2.

**Lemma C.18.** *Let $\|[K_A]_p\|$ be the smallest among all $\{\|[K_A]_k\|\}_{k\in[r]}$. For any $q \in [r]$, in Stage 2, we have*

$$\frac{\mathrm{d}}{\mathrm{d}t} \frac{\|[K_{A,\xi}]_q\|^2}{\|[K_A]_p\|^2} \leq O\left(\frac{\sigma_\xi^2}{N_A N_B d}\left(d\delta_{\xi,\perp}^2 + d^2\delta_{AB,\perp}^2 + \delta_- + d^2\delta_{N/S}\delta_{\xi,\perp}\right)\right)\frac{\|[K_{A,\xi}]_q\|^2}{\|[K_A]_p\|^2}.$$

*Proof.* Recall from Corollary C.9 that

$$\frac{\mathrm{d}}{\mathrm{d}t}\|[K_A]_p\|^2 = \frac{2\sigma_p^2\|[K_A]_p\|\|[K_B]_p\|}{N_A N_B d}\left\langle\overline{[K_A]_p}, \overline{[K_B]_p}\right\rangle[Q_1]_{p,p} + 2\frac{\|[K_A]_p\|^2}{N_A^2 d}Q_0\sigma_p^2$$

$$\pm O\left(\frac{\sigma_p^2\kappa_p^2}{N_A N_B d}\kappa_0 d\delta_{AB,\perp}^2\right)$$

$$\frac{\mathrm{d}}{\mathrm{d}t}\|[K_{A,\xi}]_q\|^2 = \frac{2\|[K_{A,\xi}]_q\|^2}{N_A^2 d}Q_0\sigma_\xi^2 \pm O\left(\frac{\sigma_\xi^2\|[K_{A,\xi}]_q\|^2}{N_A N_B d}d\delta_{\xi,\perp}^2\right).$$

Since the condition number of $K_A$ is bounded by $\sqrt{d}$, it suffices to consider the smallest $\|[K_A]_p\|$, for which we have

$$\frac{\mathrm{d}}{\mathrm{d}t}\|[K_A]_p\|^2 = \frac{2\sigma_p^2}{N_A N_B d}[Q_1]_{p,p}\|[K_A]_p\|^2 + \frac{2\sigma_p^2}{N_A^2 d}Q_0\|[K_A]_p\|^2$$

$$\pm O\left(\frac{\sigma_p^2\kappa_p^2}{N_A N_B d}\left(\kappa_0 d\delta_{AB,\perp}^2 + \delta_-\right)\right)$$

$$= \frac{2\sigma_p^2}{N_A N_B d}[Q_1]_{p,p}\|[K_A]_p\|^2 - \frac{2\sigma_p^2}{N_A^2 d}\left(\sum_{k=1}^r \frac{\kappa_k^2}{\|\kappa\|^2}[Q_1]_{k,k}\right)\|[K_A]_p\|^2$$

$$\pm O\left(\frac{\sigma_p^2\kappa_p^2}{N_A N_B d}\left(d^2\delta_{AB,\perp}^2 + d^2\delta_{N/S}\delta_{\xi,\perp} + \delta_-\right)\right),$$

where the last line comes from Lemma C.8. By Lemma B.4, $[Q_1]_{p,p}$ is negative correlated with $\kappa_p^2$. As a result, we have

$$\frac{\mathrm{d}}{\mathrm{d}t}\|[K_A]_p\|^2 \geq -O\left(\frac{\sigma_p^2\kappa_p^2}{N_A N_B d}\left(d^2\delta_{AB,\perp}^2 + d^2\delta_{N/S}\delta_{\xi,\perp} + \delta_-\right)\right).$$

For $K_{A,\xi}$, we simply have

$$\frac{\mathrm{d}}{\mathrm{d}t}\|[K_{A,\xi}]_q\|^2 \leq O\left(\frac{\sigma_\xi^2}{N_A N_B d}\left(d\delta_{\xi,\perp}^2 + d^2\delta_{AB,\perp}^2 + \delta_- + d^2\delta_{N/S}\delta_{\xi,\perp}\right)\right)\|[K_{A,\xi}]_q\|^2.$$

Thus,

$$\frac{\mathrm{d}}{\mathrm{d}t}\frac{\|[K_{A,\xi}]_q\|^2}{\|[K_A]_p\|^2} = \frac{\frac{\mathrm{d}}{\mathrm{d}t}\|[K_{A,\xi}]_q\|^2}{\|[K_A]_p\|^2} - \frac{\|[K_{A,\xi}]_q\|^2}{\|[K_A]_p\|^2}\frac{\frac{\mathrm{d}}{\mathrm{d}t}\|[K_A]_p\|^2}{\|[K_A]_p\|^2}$$

$$\leq O\left(\frac{\sigma_\xi^2}{N_A N_B d}\left(d\delta_{\xi,\perp}^2 + d^2\delta_{AB,\perp}^2 + \delta_- + d^2\delta_{N/S}\delta_{\xi,\perp}\right)\right)\frac{\|[K_{A,\xi}]_q\|^2}{\|[K_A]_p\|^2}.$$

$\square$

## C.5 ESTIMATING THE CONVERGENCE RATE

In this subsection, we estimate how fast the condition number will become close to 1.

**Lemma C.19.** *Suppose that* $\|[\boldsymbol{K_A}]_p\|$ *is the largest and* $\|[\boldsymbol{K_A}]_q\|$ *the smallest among all* $\{\|[\boldsymbol{K_A}]_k\|\}_{k \in [r]}$*. In Stage 2, we have*

$$\frac{\mathrm{d}}{\mathrm{d}t} \frac{\|[\boldsymbol{K_A}]_p\|^2}{\|[\boldsymbol{K_A}]_q\|^2} \leq -\frac{4(1-\tilde{S})\sigma_{\min}^2}{N_{\boldsymbol{A}} N_{\boldsymbol{B}} d} (T_p - T_q) \frac{\|[\boldsymbol{K_A}]_p\|^2}{\|[\boldsymbol{K_A}]_q\|^2}$$
$$\pm O\left( \frac{\sigma_p^2}{N_{\boldsymbol{A}} N_{\boldsymbol{B}} d} \left( d^2 \delta_{\boldsymbol{AB},\perp}^2 + \delta_- + d^2 \delta_{N/S} \delta_{\xi,\perp} \right) \kappa_0^2 \right).$$

**Corollary C.20** (Convergence rate)**.** *Suppose that* $\|[\boldsymbol{K_A}]_p\|$ *is the largest and* $\|[\boldsymbol{K_A}]_q\|$ *the smallest among all* $\{\|[\boldsymbol{K_A}]_k\|\}_{k \in [r]}$*. For any constant* $c > 1$*, it takes at most* $\mathrm{poly}(d)$ *amount of time for* $\|[\boldsymbol{K_A}]_p\|^2 / \|[\boldsymbol{K_A}]_q\|^2$ *to become smaller than c.*

OMITTED PROOF OF THIS SUBSECTION

*Proof of Lemma C.19.* By Corollary C.9, Lemma C.8 and Lemma C.5, we have

$$\frac{\mathrm{d}}{\mathrm{d}t} \|[\boldsymbol{K_A}]_p\|^2 = \frac{2\sigma_p^2}{N_{\boldsymbol{A}} N_{\boldsymbol{B}} d} [\boldsymbol{Q}_1]_{p,p} \|[\boldsymbol{K_A}]_p\|^2 + \frac{2\sigma_p^2}{N_{\boldsymbol{A}}^2 d} Q_0 \|[\boldsymbol{K_A}]_p\|^2$$
$$\pm O\left( \frac{\sigma_p^2 \kappa_p^2}{N_{\boldsymbol{A}} N_{\boldsymbol{B}} d} \left( \kappa_0 d \delta_{\boldsymbol{AB},\perp}^2 + \delta_- \right) \right)$$
$$= \frac{4(1-\tilde{S})\sigma_p^2}{N_{\boldsymbol{A}} N_{\boldsymbol{B}} d}(1-T_p) \|[\boldsymbol{K_A}]_p\|^2 + \frac{4(1-\tilde{S})\sigma_p^2}{N_{\boldsymbol{A}}^2 d} \left( \sum_{k=1}^r \frac{\kappa_k^2}{\|\boldsymbol{\kappa}\|^2}(1-T_p) \right) \|[\boldsymbol{K_A}]_p\|^2$$
$$\pm O\left( \frac{\sigma_p^2 \kappa_p^2}{N_{\boldsymbol{A}} N_{\boldsymbol{B}} d} \left( d^2 \delta_{\boldsymbol{AB},\perp}^2 + \delta_- + d^2 \delta_{N/S} \delta_{\xi,\perp} \right) \right)$$
$$= -\frac{4(1-\tilde{S})\sigma_p^2}{N_{\boldsymbol{A}} N_{\boldsymbol{B}} d} \left( T_p - \tilde{T} \right) \|[\boldsymbol{K_A}]_p\|^2 \pm O\left( \frac{\sigma_p^2 \kappa_p^2}{N_{\boldsymbol{A}} N_{\boldsymbol{B}} d} \left( d^2 \delta_{\boldsymbol{AB},\perp}^2 + \delta_- + d^2 \delta_{N/S} \delta_{\xi,\perp} \right) \right),$$

where $\tilde{T} = \sum_{k=1}^r \frac{\kappa_k^2}{\|\boldsymbol{\kappa}\|^2} T_p$. Then, we compute

$$\frac{\mathrm{d}}{\mathrm{d}t} \frac{\|[\boldsymbol{K_A}]_p\|^2}{\|[\boldsymbol{K_A}]_q\|^2} = \frac{\frac{\mathrm{d}}{\mathrm{d}t}\|[\boldsymbol{K_A}]_p\|^2}{\|[\boldsymbol{K_A}]_q\|^2} - \frac{\|[\boldsymbol{K_A}]_p\|^2}{\|[\boldsymbol{K_A}]_q\|^2} \frac{\frac{\mathrm{d}}{\mathrm{d}t}\|[\boldsymbol{K_A}]_q\|^2}{\|[\boldsymbol{K_A}]_q\|^2}$$
$$= -\frac{4(1-\tilde{S})}{N_{\boldsymbol{A}} N_{\boldsymbol{B}} d} \left( \sigma_p^2 \left( T_p - \tilde{T} \right) - \sigma_q^2 \left( T_q - \tilde{T} \right) \right) \frac{\|[\boldsymbol{K_A}]_p\|^2}{\|[\boldsymbol{K_A}]_q\|^2}$$
$$\pm O\left( \frac{\sigma_p^2}{N_{\boldsymbol{A}} N_{\boldsymbol{B}} d} \kappa_0^2 \left( d^2 \delta_{\boldsymbol{AB},\perp}^2 + \delta_- + d^2 \delta_{N/S} \delta_{\xi,\perp} \right) \right).$$

Since $\|[\boldsymbol{K_A}]_p\|$ is the largest, $\|[\boldsymbol{K_A}]_q\|$ is the smallest, $T_p$ is positively correlated with $\|[\boldsymbol{K_A}]_p\|$, and $\tilde{T}$ is a weighted average of $T_p$, we have

$$\sigma_p^2 \left( T_p - \tilde{T} \right) - \sigma_q^2 \left( T_q - \tilde{T} \right) \geq \sigma_q^2 \left( T_p - T_q \right).$$

Thus,

$$\frac{\mathrm{d}}{\mathrm{d}t} \frac{\|[\boldsymbol{K_A}]_p\|^2}{\|[\boldsymbol{K_A}]_q\|^2} \leq -\frac{4(1-\tilde{S})\sigma_{\min}^2}{N_{\boldsymbol{A}} N_{\boldsymbol{B}} d}(T_p - T_q) \frac{\|[\boldsymbol{K_A}]_p\|^2}{\|[\boldsymbol{K_A}]_q\|^2}$$
$$\pm O\left( \frac{\sigma_p^2}{N_{\boldsymbol{A}} N_{\boldsymbol{B}} d} \left( d^2 \delta_{\boldsymbol{AB},\perp}^2 + \delta_- + d^2 \delta_{N/S} \delta_{\xi,\perp} \right) \kappa_0^2 \right).$$

$\square$

*Proof of Corollary C.20.* Recall that

$$T_p = \tanh\left(\frac{\hat{\kappa}_p^2}{N_A N_B d}\right) = \tanh\left(\frac{\|[\boldsymbol{K_A}]_p\|^2}{\|\boldsymbol{K_A}\|_F^2}\left(1 \pm O(\delta_- + \delta_{N/S})\right)\right).$$

Since $\kappa_0 \le \sqrt{d}$, we have $\|[\boldsymbol{K_A}]_p\|^2 / \|[\boldsymbol{K_A}]_F\|^2 \le 1/2$. Note that $\tanh'(z) = 1 - \tanh^2(z) = \Omega(1)$ for any $z \le 1.1/2$. Therefore,

$$T_p - T_q \ge \Omega\left(\frac{\|[\boldsymbol{K_A}]_p\|^2 - \|[\boldsymbol{K_A}]_q\|^2}{\|\boldsymbol{K_A}\|_F^2}\right) \pm O(\delta_- + \delta_{N/S}) \ge \Omega\left(\frac{1}{d}\right).$$

Then, by Lemma C.19, we have

$$\frac{\mathrm{d}}{\mathrm{d}t}\frac{\|[\boldsymbol{K_A}]_p\|^2}{\|[\boldsymbol{K_A}]_q\|^2} \le -\frac{4(1-\tilde{S})\sigma_{\min}^2}{N_A N_B d}(T_p - T_q)\frac{\|[\boldsymbol{K_A}]_p\|^2}{\|[\boldsymbol{K_A}]_q\|^2}$$

$$\pm O\left(\frac{\sigma_p^2}{N_A N_B d}\left(d^2\delta_{\boldsymbol{AB},\perp}^2 + \delta_- + d^2\delta_{N/S}\delta_{\xi,\perp}\right)\kappa_0^2\right)$$

$$\le -\Omega\left(\frac{\sigma_{\min}^2}{N_A N_B d}\right)\frac{\|[\boldsymbol{K_A}]_p\|^2}{\|[\boldsymbol{K_A}]_q\|^2}.$$

By the proof of Lemma C.19, the largest $\|[\boldsymbol{K_A}]_p\|^2$ is non-increasing. Hence, $N_A N_B d$ is upper bounded by some $\mathrm{poly}(d)$. Thus, it takes at most $\mathrm{poly}(d)$ for $\|[\boldsymbol{K_A}]_p^2\| / \|[\boldsymbol{K_A}]_q\|$ to become smaller than $c$. □

## C.6 PROOF OF THE MAIN LEMMA OF STAGE 2

*Proof of Lemma C.1.* The polynomial bound on the convergence time has been proved in Corollary C.20. For the errors, recall from Lemma C.12, Lemma C.13 and Lemma C.14 that

$$\frac{\mathrm{d}}{\mathrm{d}t}\hat{\delta}_{\perp,p,q} \le O\left(\frac{\sigma_{\max}^2}{N_A N_B d}\kappa_0\left(d^2\delta_{\boldsymbol{AB},\perp}^2 + \delta_{\boldsymbol{AB},\perp}\sqrt{\delta_-} + \delta_-\right)\sqrt{\hat{\delta}_{\perp,p,q}}\right),$$

$$\frac{\mathrm{d}}{\mathrm{d}t}\hat{\delta}_{\xi,\perp,p,q} \le O\left(\frac{\sigma_{\max}^2}{N_A N_B d}d\kappa_0\delta_{\xi,\perp}^2\left(\delta_{\xi,\perp} + \sqrt{\delta_-}\right)\right),$$

$$\frac{\mathrm{d}}{\mathrm{d}t}\left\langle\overline{[\boldsymbol{K_{A,\xi}}]_p}, \overline{[\boldsymbol{K_{B,\xi}}]_q}\right\rangle = \pm O\left(\frac{\sigma_\xi^2}{N_A N_B d}d\delta_{\xi,\perp}^2\right).$$

Recall from Lemma C.15, Lemma C.16, and Lemma C.17 that

$$\frac{\mathrm{d}}{\mathrm{d}t}\left\langle\overline{[\boldsymbol{K_A}]_p}, \overline{[\boldsymbol{K_B}]_p}\right\rangle = \Omega\left(\frac{\sigma_p^2[\boldsymbol{Q_1}]_{p,p}}{N_A N_B d}\right)\left(1 - \left\langle\overline{[\boldsymbol{K_A}]_p}, \overline{[\boldsymbol{K_B}]_p}\right\rangle\right) \pm O\left(\frac{\sigma_{\max}^2}{N_A N_B d}d\kappa_0\delta_{\boldsymbol{AB},\perp}^2\right),$$

$$\frac{\mathrm{d}}{\mathrm{d}t}\left(\rho_{\boldsymbol{A/B},p} + \rho_{\boldsymbol{B/A},p}\right) \le O\left(\frac{\sigma_{\max}^2}{N_A N_B d}\left(d\kappa_0\delta_{\boldsymbol{AB},\perp}^2 + \delta_-^2\right)\right),$$

$$\frac{\mathrm{d}}{\mathrm{d}t}\rho_{\xi,\boldsymbol{A/A},p/q} \le O\left(\frac{\sigma_\xi^2}{N_A N_B d}d\delta_{\xi,\perp}^2\right),$$

$$\frac{\mathrm{d}}{\mathrm{d}t}\left(\rho_{\xi,\boldsymbol{A/B},p/q} + \rho_{\xi,\boldsymbol{B/A},q/p}\right) \le O\left(\frac{\sigma_\xi^2}{N_A N_B d}\left(d\delta_{\xi,\perp}^2 + \delta_-^2\right)\right).$$

(17)

By Lemma C.18, we have

$$\frac{\mathrm{d}}{\mathrm{d}t}\frac{\|[\boldsymbol{K_{A,\xi}}]_q\|^2}{\|[\boldsymbol{K_A}]_p\|^2} \le O\left(\frac{\sigma_\xi^2}{N_A N_B d}\left(d\delta_{\xi,\perp}^2 + d^2\delta_{\boldsymbol{AB},\perp}^2 + \delta_- + d^2\delta_{N/S}\delta_{\xi,\perp}\right)\right)\frac{\|[\boldsymbol{K_{A,\xi}}]_q\|^2}{\|[\boldsymbol{K_A}]_p\|^2}.$$

Note that on the RHS of these equations, the only terms whose order may potentially be smaller than or equal to LHS are the $\delta_-$-related terms. However, (17), we can make sure $\delta_-$ is at most $\delta_{\boldsymbol{AB},\perp}^{1.5}$. As a result, the orders of the RHS are all greater than the orders of the LHS, which implies that these errors can at most double within $\mathrm{poly}(d)$ time if they are sufficiently small at the beginning of Stage 2. □

## D   FROM GRADIENT FLOW TO GRADIENT DESCENT

Converting the above gradient flow argument to a gradient descent one is standard. All our estimations can tolerate an inverse polynomially large error. Since the all quantities of interest here are polynomially and inverse polynomially bounded, at each step of gradient descent, one can always make the GF-to-GD discretization error sufficiently (inverse polynomially) small by choose a sufficiently (inverse polynomially) small learning rate and generating sufficiently (polynomially) many samples. Since the times need for Stage 1 and Stage 2 are both polynomial, this also implies a polynomial sample complexity.

