# OpenReview forum: "On the Importance of Contrastive Loss in Multimodal Learning"
_ICLR.cc/2023/Conference — Submitted to ICLR 2023_

### Official Review · Reviewer_KDMd · 2022-10-23

**Confidence:** 3
**Correctness:** 3
**Technical Novelty And Significance:** 2
**Empirical Novelty And Significance:** Not applicable
**Recommendation:** 5

**Clarity, Quality, Novelty And Reproducibility:**

The paper is well written. All theorems/lemmas are provided with solid assumptions and proof.
The simulation does not provide any detail.

**Strength And Weaknesses:**

Strength:
1. The authors considered the normalization step in the contrastive/non-contrastive loss when conducting the analysis of their dynamics. This is a big improvement over previous theoretical works (Tian et al. (2021), Wen & Li (2021), Jing et al. (2022)) on this topic.
2. The phenomenon of the two-stage training dynamics is novel. Most previous theoretical work ignores training dynamics and only focuses on generalization bounds that are unable to explain such phenomena.


Weaknesses:
1. The non-contrastive setting is known to collapse. In fact, all successful non-contrastive methods like BYOL/DINO rely on asymmetric architecture. This is already proven by Tian et al. (2021). The authors derive an obvious conclusion (align->1, balance->0) that does not provide any insight into the problem.
2. The overall claim on contrastive loss is shallow. The authors show that under perfect conditions, align-> one and balance->same rank as input. For linear settings, this is obvious that the optimal solution exists. However, the analysis seems impossible to scale to nonlinear settings as the definition of `balance` becomes trivial for nonlinear networks.
3. There is no discussion on possible follow-up based on this work. For example, based on theoretical discovery, Tian et al. (2021) and Jing et al. (2022) propose the corresponding solution. It is unclear how this theory can boost empirical advancement.


**Summary Of The Paper:**

This paper theoretically studied the joint-embedding training dynamics in a linear setting - which may be suitable for a multimodal environment. They studied both contrastive and non-contrastive cases and showed that contrastive negative pairs are essential for preventing representations from becoming a rank-one solution. The analysis largely follows Tian et al. (2021), Wen & Li (2021), and Jing et al. (2022) using gradient flows. Lastly, the authors provide a numerical simulation that verifies their claims.

**Summary Of The Review:**

This is a well-written theory paper studying contrastive learning dynamics but only limited to the small scope and seems not to provide real insight.

---

> ### Author Response · Authors · 2022-11-18
> **Response to Reviewer KDMd**
>
> Thank you for your reviews. We'd like to address your concerns as follows.
>
> * It is actually not obvious that non-contrastive methods will always collapse in the multimodal setting because the models for different modules are naturally asymmetric.
> * First, we want to emphasize our definition of balance also works for nonlinear networks as it considers the output embeddings instead of the weights. In the revised version, we also add experiments in more realistic settings and similar phenomena can also be observed there.
>
>   Moreover, the existence of optimal solutions does not imply that gradient descent can find that solution, especially when the loss is highly nonconvex, and the theoretical analysis here is far from trivial. Besides, our results say more than the learned representations will have full rank. We prove that eventually, the condition number will become close to $1$, despite that input data itself may have a much larger condition number. This is a big improvement upon previous theoretical results, all of which essentially assume that the input data is isotropic. In that case, it is somewhat reasonable to expect the model, with or without contrastive pairs, not to collapse because all directions will grow at the same rate. On the other hand, the situation is largely unclear when the condition number is larger than $1$. The model may simply collapse to the direction corresponding to the largest singular value, similar to the situation in tensor power methods.
>
> * The focus of our results is on the theoretical side. However, it can still potentially lead to empirical advancement.
>   Our two-stage analysis and its relationship with the temperature may offer intuitions on how to choose and adjust the temperature. Our theoretical analysis and the newly-added experiments suggest that we should continue training the model even after its training performance seems to plateau, as this will gradually improve the balance and the downstream performance.

---

> > ### Comment · Reviewer_KDMd · 2022-12-12
> > **Thanks for the response**
> >
> > I thank the authors for their detailed response. I thin my 2nd and 3rd questions are well addressed. I also appreciate the extra experiments on real data (COCO) in the revised paper. I will increase my score.
> >
> > However, I am not convinced by "It is actually not obvious that non-contrastive methods will always collapse in the multimodal setting because the models for different modules are naturally asymmetric."
> > In fact, in image-only setting, due to augmentation, the two modules are also naturally asymmetric. To support the claim in this paper, the authors are supposed to show with special settings (like BYOL/DINO), non-contrastive methods will collapse. Otherwise, there are limited insight of the theory compares to existing theory in single-modality setting.
> >
> > Please correct me if I am wrong. I am happy to increase my score further if this question is addressed.

---

> > > ### Author Response · Authors · 2022-12-13
> > > **On the non-contrastive results**
> > >
> > > Thank you for your response! First, we'd like to emphasize that the main contribution is to go from condition number = $1$ to condition number = $\omega(1)$ instead of single-modal to multi-modal. This is not covered by any previous single-modal result, and the main challenge here is to show the positive contrastive learning result.
> > >
> > > In the single-modal setting, the data-generating model used in the previous theoretical result [1] is $x = Mz + \xi$, where $\xi$ represents the random augmentation. In this case, what the model needs to do is essentially "de-noising" the inputs. on the other hand, in the multi-modal setting where the $M$ matrices for different modules are different, the model needs first to match the signals of the inputs. In some sense, this creates an asymmetry similar to the one used in BYOL, where the model needs to match the online network with the target network.
> > >
> > > We admit that it would be good to have an analysis for more practical non-contrastive methods such as BYOL/DINO. This is mentioned in the discussion section as possible future directions.
> > >
> > > [1] Zixin Wen, Yuanzhi Li. Toward Understanding the Feature Learning Process of Self-supervised Contrastive Learning. 2021.

---

### Official Review · Reviewer_kjwZ · 2022-10-27

**Confidence:** 2
**Correctness:** 3
**Technical Novelty And Significance:** 3
**Empirical Novelty And Significance:** Not applicable
**Recommendation:** 6

**Clarity, Quality, Novelty And Reproducibility:**

In terms of clarity, I believe the paper would benefit from more summarizations and takeaway comments describing each key formula or theorem.  It is virtually impossible to completely check the math thoroughly in this work in the timespan allotted to the reviewers.  As such, it is quite difficult to truly gauge the correctness of this work.

There are also numerous typos and grammatical errors (e.g. Now, we consider the contrastive The main” on page 8, etc.) which should be cleaned up.

I believe that the topic this paper tackles is quite novel, and I believe the experiments listed in the work have no issues with reproducibility.

**Strength And Weaknesses:**

The strength of the paper lies in its supposed mathematical rigor and derivations.  A weakness of this paper is that it is limited to a linear data-generating model, and therefore the conclusions are unable to directly translate to understanding multimodal contrastive losses used in practice.

I enjoyed how the authors included both contrastive and non-contrastive loss formulations.  The authors analyze the non-contrastive loss (such as BYOL) in comparison with the contrastive loss to demonstrate why contrastive losses are better (in terms of going beyond alignment only).

**Summary Of The Paper:**

In this paper, the authors examine the role of contrastive loss across modalities (such as in CLIP).  They demonstrate that the contrastive loss promotes the learning of “aligned” and “balanced” representations.  The authors investigate the learning dynamics of the optimization process (under specific listed conditions) and show that the dynamics can be interpreted in the form of two stages.

**Summary Of The Review:**

Given the novelty of the work, as a comprehensive step towards understanding multimodal contrastive losses, I am inclined to support this paper for acceptance.  This is predicated on the correctness of the listed mathematics - I willingly confess that a large caveat of my review is that I was not able to carefully check all the math or the claims in this paper.  Therefore, I will provide a marginal acceptance and look forward to extended discussions with the AC.

---

> ### Author Response · Authors · 2022-11-18
> **Response to Reviewer kjwZ**
>
> Thank you for your reviews! We fixed the typos in the revision and also added experimental results on nonlinear models. Unfortunately, due to the page limit, we are not able to write more detailed proof outlines in the main text. As compensation, we add more explanations in the appendix in the revision.

---

### Official Review · Reviewer_SHU1 · 2022-10-27

**Confidence:** 2
**Correctness:** 3
**Technical Novelty And Significance:** 2
**Empirical Novelty And Significance:** 2
**Recommendation:** 6

**Clarity, Quality, Novelty And Reproducibility:**

This paper is well-written and original, the method is theory-grounded and novel.

**Strength And Weaknesses:**

Strengths:
(a) The paper provided a fresh view that the alignment and balance of representation play a important role in the training dynamics of contrastive learning, and the training can be decoupled into two stages.
(b) The paper is backed by a series of theory-grounded definitions and proof.
Weaknesses:
(a) The experiment part only covers simulation results, not real data, the model only cover linear models.

**Summary Of The Paper:**

The paper set out to investigate the learning dynamics of contrastive learning (e.g.CLIP) in, investigate how contrastive learning learn to align the representations from different views efficiently.

**Summary Of The Review:**

This paper is well-written and original, it provides an interesting view in investigating the training dynamic of contrastive learning with alignment and balance of feature representations, however, the experiment seems only done on simulated data and linear models.

---

> ### Author Response · Authors · 2022-11-18
> **Response to Reviewer SHU1**
>
> Thank you for your reviews! We added new experiments on the MSCOCO dataset using more practical models in the revised version (cf. Fig. 2). One can see that even when the data and models are nonlinear, we can still observe similar dynamics: the alignment score quickly reaches near $100\%$, and both the balance score and downstream zero-shot accuracy gradually increase during training. This matches our theoretical analysis, which is the central part of the paper.

---

### Decision · Program_Chairs · 2023-01-20

**Decision:**

Reject

**Justification For Why Not Higher Score:**

Reviewers appreciated the mathematical rigor and found the topic to be relatively unexplored and important. However, concerns were raised about the depth of the results (how generally do they hold) and the gap between theory and practice. The AC and the most critical reviewer both considered the rebuttal and appreciated the clarifications but were unconvinced that the paper is quite at the ICLR level. The AC also appreciates the MSCOCO experiment but was not able to discern the two phases described by the theory.

**Justification For Why Not Lower Score:**

N/A

**Metareview: Summary, Strengths And Weaknesses:**

Summary:
This paper studies the training dynamics of contrastive learning using a simplified model (linear data model, 1-layer net, etc). The main finding is that there are two phases of learning: first the model aligns positive pairs, second the model spreads out negative pairs. The mathematical analysis is supported by simulations and experiments on MSCOCO.

Strengths:
* Learning dynamics of contrastive learning are relatively unstudied
* Simulation results corroborate the theory

Weaknesses:
* Gap between theory and practice may be large (assumptions of the theory are not realistic and MSCOCO experiments don't show two clear phases)
* Similar results have been shown before (e.g., collapse of noncontrastive methods, relationship of contrastive learning to alignment and uniformity/balance)
* The analysis does not lead to algorithmic improvements